# Permafrost Variability over the Northern Hemisphere Based on the MERRA-2 Reanalysis

Jing Tao[1,a], Randal D. Koster[2], Rolf H. Reichle[2], Barton A. Forman[3], Yuan Xue[3,b], Richard H. Chen[4], Mahta Moghaddam[4]

[1]Earth System Science Interdisciplinary Center, University of Maryland, College Park, Maryland, USA.

[2]Global Modelling and Assimilation Office, NASA Goddard Space Flight Center, Greenbelt, Maryland, USA.

[3]Department of Civil and Environmental Engineering, University of Maryland, College Park, Maryland, USA.

[4]Department of Electrical Engineering, University of Southern California, Los Angeles, California, USA.

[a]Now at Climate and Ecosystem Sciences Division, Lawrence Berkeley National Laboratory, Berkeley, California, USA; and Department of Civil and Environmental Engineering, University of Washington, Seattle, Washington, USA.

[b]Now at George Mason University, Fairfax, Virginia, USA.

*Correspondence to*: Jing Tao (jingtao@lbl.gov)

**Abstract.** This study introduces and evaluates a comprehensive, model-generated dataset of Northern Hemisphere permafrost conditions at 81-km$^2$ resolution. Surface meteorological forcing fields from the Modern-Era Retrospective Analysis for Research and Applications-2 (MERRA-2) reanalysis were used to drive an improved version of the land component of MERRA-2 in middle-to-high northern latitudes from 1980 to 2017. The resulting simulated permafrost distribution across the Northern Hemisphere mostly captures the observed extent of continuous and discontinuous permafrost but misses the ecosystem-protected permafrost zones in western Siberia. Noticeable discrepancies also appear along the southern edge of the permafrost regions where sporadic and isolated permafrost types dominate. The evaluation of the simulated active layer thickness (ALT) against in-situ measurements demonstrates reasonable skill except in Mongolia. The RMSE (bias) of climatological ALT is 1.22 m (-0.48 m) across all sites and 0.33 m (-0.04 m) without the Mongolia sites. In northern Alaska,

both ALT retrievals from airborne remote sensing for 2015 and the corresponding simulated ALT exhibit limited skill versus in-situ measurements at the model scale. In addition, the simulated ALT has larger spatial variability than the remotely sensed ALT, although it agrees well with the retrievals when considering measurements uncertainty. Controls on the spatial variability of ALT are examined with idealized numerical experiments focusing on northern Alaska; meteorological forcing and soil types

5 are found to have dominant impacts on the spatial variability of ALT, with vegetation also playing a role through its modulation of snow accumulation. A correlation analysis further reveals that accumulated above-freezing air temperature and maximum snow water equivalent explain most of the year-to-year variability of ALT nearly everywhere over the model-simulated permafrost regions.

## 1 Introduction

Permafrost is an important component of the climate system, and its variations can have significant impacts on climate and society. Of deep concern is a potential positive feedback loop by which carbon stored within permafrost regions is released through global warming, thereby adding greenhouse gases to the atmosphere that accelerate the warming further (Dorrepaal et al., 2009; Schuur et al., 2009; MacDougall et al., 2012; Schuur et al., 2015). Communities and infrastructure in ice-rich

15 permafrost regions are particularly vulnerable to land subsidence and infrastructure damage caused by permafrost thaw (Nelson et al., 2001; Liu et al., 2010; Guo and Sun, 2015).

Permafrost variations, including pronounced permafrost degradation due to a warming climate, have been reported for many regions, including Alaska (Nicholas and Hinkel, 1996; Osterkamp and Romanovsky, 1996; Jorgenson et al., 2001; Hinkel and

20 Nelson, 2003; Jafarov et al., 2012; Liu et al., 2012; Jones et al., 2016; Batir et al., 2017), Canada (Chen et al., 2003; James et al., 2013), Norway (Gisnas et al., 2013), Sweden (Pannetier and Frampton, 2016), Russia (Romanovsky et al., 2007; Romanovsky et al., 2010), Mongolia (Sharkhuu and Sharkhuu, 2012), and the Qinghai–Tibet Plateau (Zhou et al., 2013; Wang et al., 2016a; Lu et al., 2017; Ran et al., 2018). For the entire Northern Hemisphere, rapidly accelerated permafrost degradation in recent years has been reported by Luo et al. (2016) based on in-situ measurements at a point-scale or at a spatially-aggregated

scale (up to 1000m×1000m) from the Circumpolar Active Layer Monitoring (CALM) network. However, the current state and evolution of global permafrost (including permafrost temperature, ice content, and degradation rates) are still largely unknown across much of the Northern latitudes.

The impact of a changing climate on permafrost dynamics must depend on local site characteristics. Subsurface heat transfer processes and active layer thickness (ALT; the maximum thaw depth at the end of the thawing season) are influenced by more than surface meteorological forcing – they are also influenced by vegetation type, surface organic layer characteristics, soil properties and soil moisture (Stieglitz et al., 2003; Shur and Jorgenson, 2007; Yi et al., 2007; Luetschg et al., 2008; Dankers et al., 2011; Johnson et al., 2013; Jean and Payette, 2014; Yi et al., 2015; Fisher et al., 2016; Matyshak et al., 2017; Tao et al.,
2017). Understanding the contributions from the different controls on ALT (and permafrost conditions in general) is crucial for assessing permafrost behaviour and its resilience to a warming climate.

Physically-based numerical model simulations are potentially useful for quantifying and understanding these dynamics at large spatial scales; they can also provide insights into associated impacts on the global carbon cycle. Permafrost dynamics can be
modelled, for example, by driving a land surface model (LSM) offline (i.e., uncoupled from an atmospheric model) with meteorological forcing data (including air temperature, radiation, precipitation, etc.) from some credible source. LSMs that have been used to quantify large-scale permafrost patterns (i.e., distributions and thermal states) and their interactions with a warming climate include, for example, the Joint UK Land Environment Simulator (JULES, Dankers et al., 2011), the ORganizing Carbon and Hydrology in Dynamic EcosystEms (ORCHIDEE) - aMeliorated Interactions between Carbon and
Temperature (ORCHIDEE-MICT, Guimberteau et al., 2018), the Catchment Land Surface Model (CLSM, Tao et al., 2017), and the Community Land Model (Alexeev et al., 2007; Nicolsky et al., 2007a; Yi et al., 2007; Lawrence and Slater, 2008; Lawrence et al., 2008; Lawrence et al., 2012; Koven et al., 2013; Chadburn et al., 2017; Guo and Wang, 2017). Most of these land models were run at coarse spatial resolutions, e.g., ranging from $0.5° \times 0.5°$ to $1.8° \times 3.6°$ for LSMs participating in the Permafrost Carbon Network (PCN) (Wang et al., 2016a) and from $0.188° \times 0.188°$ to $4.10° \times 5°$ for the models participating

in the Coupled Model Intercomparison Project phase 5 (CMIP5) (Koven et al., 2013; https://portal.enes.org/data/enes-model-data/cmip5/resolution).

Differences in the permafrost behaviour simulated with these models reflect model-specific process representations as well as
biases associated with different meteorological forcing datasets (Barman and Jain, 2016; Wang et al., 2016a; Wang et al., 2016b; Guo et al., 2017; Guimberteau et al., 2018). Such forcing biases are difficult to avoid given the sparsity of direct observations of meteorological variables in most parts of the high latitudes. Even reanalyses, which assimilate a variety of global observations, inevitably have biases in high latitudes due to observation sparsity in cold regions combined with the many challenges of physical process modelling. Nevertheless, despite these issues, permafrost behaviour simulated with LSMs
driven offline by reanalysis forcing fields can still be useful for understanding the impacts of climate variability on permafrost. The present paper utilizes this approach. Specifically, we generate here a dataset of Northern Hemisphere permafrost conditions by driving an updated version of NASA's Catchment Land Surface Model (CLSM) with Modern-Era Retrospective Analysis for Research and Applications-2 (MERRA-2; Gelaro et al., 2017) surface meteorological forcing fields for the middle-to-high latitudes across the Northern Hemisphere over the period 1980-2017. We perform the simulations at 81 km$^2$
resolution encompassing permafrost areas in the middle-to-high latitudes of the Northern Hemisphere. This resolution is high relative to most existing modelling studies at the global scale; published simulations at higher resolution are limited to plot scales (e.g., CALM-site scale in Shiklomanov et al. (2010)), landscape scales (e.g., polygonal tundra landscape scale in Kumar et al. (2016)), or regional scales (e.g., 4 km$^2$ in Jafarov et al. (2012) covering Alaska; 1 km$^2$ in Gisnas et al. (2013) covering Norway).

Due to the sparsity of in-situ measurements at the regional to global scale, evaluating the spatial pattern of ALT produced by any such simulation remains challenging. Indeed, it is difficult to compare the simulated values at model resolutions with in-situ observations taken at the point scale unless the measurement point is uniformly representative of the area covered by the model grid cell or the representation errors associated with the point-to-grid comparison are well defined. Remotely sensed
permafrost products, which provide a unique source of spatially distributed ALT at the landscape-scale, may provide help in

this regard. Existing remote sensing ALT products have been retrieved from ground-based Ground Penetrating Radar (GPR) (Chen et al., 2016a; Jafarov et al., 2017), airborne polarimetric Synthetic Aperture Radar (SAR), and spaceborne interferometric SAR (Liu et al., 2012; Li et al., 2015; Schaefer et al., 2015). These ALT products are available at the landscape-scale and can complement our modelling analysis. In this study, we use remote sensing information from the NASA Airborne

5    Microwave Observatory of Subcanopy and Subsurface (AirMOSS) mission. In 2015, AirMOSS acquired P-band (420-440 MHz) SAR observations over portions of northern Alaska from which Chen et al. (2019) retrieved regional estimates of ALT and soil layer dielectric properties that are related to soil moisture and freeze/thaw states. In their study, Chen et al. (2019) mainly focus on the development and improvement of the ALT retrieval algorithm, whereas the present study uses the ALT retrievals in combination with in-situ measurements to aid in assessing the (fully independent) ALT simulations.

In the present paper we evaluate our simulated permafrost extent and ALTs against an observations-based permafrost distribution map and against multi-year in-situ observations. We also compare the skill of our model estimates to that of the AirMOSS ALT retrievals. In these comparisons, we account for uncertainty to the extent possible. Overall, we pursue three scientific objectives: 1) evaluate the relative importance of the factors that determine the spatial variability of ALT, 2) evaluate

15   CLSM-simulated ALT and permafrost extent against observations, and 3) quantify and assess the large-scale characteristics of ALT (in terms of means and interannual variability) in Northern Hemisphere permafrost regions from 1980 through 2017. As a side benefit, the side-by-side comparison of modelled and remotely sensed ALT estimates is an important first step toward combining this information effectively in future model-data fusion efforts. Section 2 below describes the model and datasets used in this study, Section 3 describes methods, and Section 4 provides results. Our findings are summarized and discussed

20   in Section 5.

## 2 Model and data sets

### 2.1 NASA Catchment Land Surface Model (CLSM)

CLSM is the land model component of NASA's Goddard Earth Observing System (GEOS) Earth system model and was part of the model configuration underlying the MERRA-2 reanalysis product (Reichle et al., 2017a; Gelaro et al., 2017). CLSM explicitly accounts for sub-grid heterogeneity in soil moisture characteristics with a statistical approach (Koster et al., 2000; Ducharne et al., 2000). The land fraction within each computational unit (or grid cell) is partitioned into three soil moisture regimes, namely the wilting (i.e., non-transpiring), unsaturated, and saturated area fractions. Over each of the three moisture regimes, a distinct parameterization is applied to estimate the relevant physical processes (e.g., runoff and evapotranspiration). This version of CLSM includes a three-layer snow model that estimates the evolution of snow water equivalent (SWE), snow depth, and snow heat content (Stieglitz et al., 2001) in response to the forcing data. The snow model accounts for key physical mechanisms that contribute to the growth and ablation of the snowpack, including snow accumulation, aging, melting, and refreezing. The model also includes the insulation of the ground from the atmosphere by the snowpack. The CLSM subsurface heat transfer module uses an explicit finite difference scheme to solve the heat diffusion equation for six soil layers (0-0.1m, 0.1-0.3m, 0.3-0.7m, 0.7-1.4m, 1.4-3m, and 3-13m). The soil layer thicknesses increase with depth following a geometric series for consistency with the linear heat diffusion calculation (Koster et al., 2000). A no-heat-flux condition is employed at 13m depth.

The updated version of CLSM used here includes modifications aimed at improving permafrost simulation. It accounts, for example, for the impact of soil carbon on the soil thermal properties with soil porosity, thermal conductivity, and specific heat capacity calculated separately for mineral soil and soil carbon, after which the two are averaged using a carbon-weighting scheme. Higher (lower) soil carbon content, therefore, results in lower (higher) soil thermal conductivity. The updated version produces more realistic subsurface thermodynamics in cold regions than does the original scheme (Tao et al., 2017). This version of CLSM, however, does not include dynamic soil carbon pools.

Particularly relevant to the present analysis is our calculation of ALT from CLSM simulation output. We compute ALT from the simulated soil temperature profile and the ice content within the soil layer that contains the thawed-to-frozen transition. Precisely, the thawed-to-frozen depth is calculated as:

$$z_{bottom}(l) - f_{ice}(l, t) \times \Delta z(l), \tag{1}$$

where layer $l$ is the deepest layer that is fully or partially thawed, $z_{bottom}(l)$ represents the depth at the bottom of layer $l$, $f_{ice}(l, t)$ is the fraction of ice in layer $l$ at time t (i.e., $f_{ice}(l, t) \in [0\ 1]$), and $\Delta z(l)$ is the thickness of layer $l$. To identify layer $l$, we use a 0°C degree temperature threshold. Specifically, T > 0°C degree indicates that a layer is fully thawed, T = 0°C degree indicates that a layer is partially thawed, and T < 0°C degree indicates that a layer is fully frozen. That is, layer $l$ is the deepest layer that satisfies $T(l) \geq 0$°C. Equation (1) then expresses that the thawed-to-frozen depth is equal to the bottom depth of the layer $l$ but adjusted upward according to the ice fraction within the partially thawed layer $l$. This upward adjustment, by the way, allows the thawed-to-frozen depth to be a continuous variable; it is not quantized to the imposed layer depths. We search for the deepest $l$ if multiple thawed-to-frozen transitions are present (e.g., if a seasonal frost at the surface is separated from the permafrost below by a thawed soil layer). The annual ALT for a given year, then, is defined as the deepest depth at which a thawed-to-frozen transition occurs within that year. Note that the calculation of equation (1) is made at the scale of a model grid cell, and thus features such as talik are not represented if they occur at sub-grid cell scale.

We drive the improved CLSM version of Tao et al. (2017) in a land-only (offline) configuration across permafrost areas in the Northern Hemisphere. The simulation domain, shown in Figure 1a, covers the major permafrost regions of the Northern Hemisphere middle-to-high latitudes for which soil carbon data are available from the Northern Circumpolar Soil Carbon Database version 2 (NCSCDv2, https://bolin.su.se/data/ncscd/) (Hugelius et al., 2013a; Hugelius et al., 2013b). The NCSCDv2 data are used to calculate the CLSM soil thermal properties used in the simulations (Tao et al., 2017). The model simulation covered the period from 1980 to 2017 and was performed at a 81-km$^2$ spatial resolution on the 9-km Equal Area Scalable Earth grid, version 2 (Brodzik et al., 2012).

Surface meteorological forcing were extracted from the MERRA-2 reanalysis data, which are provided at a resolution of 0.5° latitude × 0.625° longitude (Global Modeling and Assimilation Office (GMAO), 2015a, b). At latitudes south of 62.5°N within our simulation domain, the MERRA-2 precipitation forcing used here is informed by gauge measurements from the daily 0.5° global Climate Prediction Center Unified gauge product (Chen et al., 2008) as described in (Reichle et al., 2017b). We further

rescaled the precipitation to the long-term, seasonally varying climatology of the Global Precipitation Climatology Project version 2.2 product (Huffman et al., 2009). Further details regarding model parameters and forcing inputs are found in Tao et al. (2017).

The model was spun-up for 180 years by looping five successive times through the 36-year period of MERRA-2 forcing from

1 January 1980 to 1 January 2016 in order to achieve a quasi-equilibrium state. The spatial terrestrial state variables at the end of the fifth loop were used to initialize the model for the final simulation experiment from 1980 to 2017.

**2.2 Remotely Sensed ALT from AirMOSS**

Radar backscatter measurements are sensitive to changes in the soil dielectric constant (or relative permittivity) which in turn

are associated with changes in soil moisture and the soil freeze-thaw state. Based on this relationship, Chen et al. (2019) used the AirMOSS airborne P-band (420-440 MHz) synthetic aperture radar (SAR) observations collected during two campaigns in 2015 to estimate ALT in northern Alaska. As shown in Figure 2a, the AirMOSS flights originated from Fairbanks International Airport and headed west toward the Seward Peninsula (HUS, KYK, COC), then turned back east (KGR) prior to heading north towards the Arctic coast overpassing Ambler (AMB), Ivotuk (IVO), and Atqasuk (ATQ). From there, the flights

turned south again, flying over Barrow (BRW), Deadhorse (DHO), and Coldfoot (CFT) en route to Fairbanks. In the present paper, the remotely-sensed ALT retrievals are compared with in-situ observations and CLSM-simulated ALT.

Chen et al. (2019) used AirMOSS P-band SAR observations at two different times to retrieve active layer properties: (1) acquisitions on 29 August 2015 when the downward thawing process approximately reached its deepest depth (i.e., the bottom

of the active layer), and (2) acquisitions on 1 October 2015 when the active layer started to refreeze from the surface while the bottom of the active layer remained thawed. ALT was assumed constant from late August to early October because over this period changes in thawing depth are found typically negligible (Carey and Woo, 2005; Chen et al., 2016b; Zona et al., 2016). Strictly speaking, the radar retrievals represent the approximate thaw depth of the thawed-to-frozen boundary on 29 August

2015 and 1 October 2015. The unknown, true ALT for 2015 might occur later if the thawing continued and the maximum thaw depth occurred after the October flight time. Based on an analysis of in-situ observations (not shown), however, it is rare that this occurs, and the subsequent impact on the estimated ALT value would be relatively small in any case. We therefore equate the retrieved thaw depth with ALT.

In the retrieval algorithm,  used a three-layer dielectric structure to represent the active layer and underlying permafrost. In their algorithm, the two uppermost layers together constitute the active layer that account for a top, unsaturated zone and an underlying, saturated zone. The bottommost (third) layer of the retrieval model structure represents the permafrost. Because the soil moisture at saturation only depends on the porosity of the soil medium, the dielectric constant of the saturated zone in the active layer is assumed constant over the time window. An iterative forward-model inversion scheme was used to

simultaneously retrieve the dielectric constants and layer thicknesses of the three-layer dielectric structure from the SAR observations collected on 29 August 2015 and 1 October 2015. Note that the retrieved ALT cannot exceed the radar sensing depth of about 60 cm. This is the depth below which the AirMOSS radar is expected to lose sensitivity to subsurface features, and it is calculated based on the radar system noise floor and calibration accuracy. Therefore, any retrieved ALT larger than 60 cm is expected to have large uncertainties, and the error is further expected to grow linearly as the retrieved values of ALT

essentially "saturate." This limitation may also lead to underestimates of the actual thaw depth.

In this study, we focus on the retrievals of four flight lines across the Alaska North Slope, including IVO (Ivotuk), ATQ (Atqasuk), BRW (Barrow), and DHO (Deadhorse) as shown in Figure 2a. These four transects cover areas with light to moderate vegetation. Since the radar scattering model is only applicable to bare surfaces or lightly vegetated tundra areas

(Chen et al., 2019), the ALT estimates derived for IVO, ATQ, BRW, and DHO are considered more accurate than ALT

retrievals for the remaining transects, which include more vegetated areas. Moreover, some of the southern transects cover discontinuous permafrost where the ALT often exceeds the P-band radar sensing depth of about 60 cm and thus the retrievals have large uncertainty (Chen et al., 2019).

**2.3 Circum-Arctic Permafrost Conditions and In-situ Observations of ALT**

The permafrost distribution simulated by CLSM is evaluated against the observations-based Circum-Arctic Map of Permafrost and Ground-Ice Conditions (Brown et al., 2002) shown in Figure 1b. The map is based on the distribution and character of permafrost and ground ice using a physiographic approach. Permafrost conditions are categorized into four classes: continuous (90-100%), discontinuous (50-90%), sporadic (10-50%), and isolated (0-10%), where the numbers in parentheses indicate the
area fraction of permafrost extent.

In-situ observations of ALT obtained by the CALM network (https://www2.gwu.edu/~calm/; Brown et al., 2000) were used to evaluate both the AirMOSS ALT retrievals and CLSM-simulated ALT results. The CALM network provides observations from 1990 to 2017, but few sites have records in the early 1990s. We did not use measurements that were flagged as having
been taken too early in the season or under unusual conditions (e.g., after the site was burned or covered with lava, which occurred at sites R30A and R30B in Kamchatka). In total there are 220 sites located within the CLSM simulation domain (Figure 1b), and we use 213 sites to evaluate results. Thaw depth measurements are usually made at the end of the thawing season. Most of the CALM sites (129 out of the 213 sites used here) employ a spatially-distributed mechanical probing method to measure thaw depths along a transect or across a rectangular grid ranging in size from 10m×10m to 1000m×1000m. At 20
sites thaw tubes or boreholes are used to measure the thaw depth. At 63 sites, ground temperature measurements from boreholes are used to infer thaw depth. For the remaining site, no information about the measurement method is available. Only point-scale measurements are available from the thaw tube/borehole and ground temperature sites (including, e.g., the sites in Mongolia).

In addition, daily in-situ observations of soil temperature profiles at ten Alaskan sites from the Permafrost Laboratory at the University of Alaska Fairbanks (UAF) (http://permafrost.gi.alaska.edu/sites_map; Romanovsky et al., 2009) were used to infer thawed-to-frozen depth using the 0ºC degree threshold and to complement the CALM ALT observations in Alaska. Table 1 provides the coordinates and measuring methods of the UAF in-situ sites. The UAF measurements were used along with the CALM data to evaluate the ALT estimates derived from the CLSM simulation and the AirMOSS radar observations for the North Slope of Alaska in section 4.1.

## 3 Methods

### 3.1 Comparing ALT from In-situ Observations, AirMOSS Retrievals, and CLSM Results in Alaska

First, we compare AirMOSS radar retrievals and CLSM simulation results of ALT for 2015 against each other and against in-situ observations: (i) we compare the spatial patterns of the AirMOSS retrievals with those of the model-simulated ALT over northern Alaska; and (ii) we evaluate the simulated ALT against both the AirMOSS retrievals and in-situ observations from the CALM and UAF networks. We rely on several metrics to evaluate the model and radar-retrieval performance, including bias, root mean square error (RMSE), and correlation coefficient (R). The results are discussed in section 4.1.

We conducted the intercomparison at the model scale. The radar retrievals were provided at 2-arcsec $\times$ 2-arcsec (roughly 20 m x 60 m in the Arctic) resolution whereas the CLSM-simulated ALTs are at 81 km$^2$. We thus aggregated the AirMOSS retrievals to the CLSM model grid by averaging all the retrieval data points within each 81 km$^2$ model grid cell. Only model grid cells that were at least 30% covered by radar retrievals were used in the comparison. The AirMOSS transects cover several different regions with different climatologic regimes, topography, vegetation and soil type (Figure 2). Note that although the vegetation class used in the model (Figure 2b) suggests the presence of dwarf trees over the Alaska North Slope, the actual satellite-based LAI, vegetation height, greenness fraction and albedo will still instruct the model that the tree cover there is extremely sparse. The data sources for these vegetation-related boundary conditions can be found in Table 1 of Tao et al.

(2017). Overall, the variability of ALT along these transects encompasses the influence of a variety of factors at the regional scale.

The daily UAF in-situ soil temperature profile observations on the AirMOSS flight date (29 August 2015) were used to calculate the thawed-to-frozen depth (i.e., approximated ALT). The ALT measurements at all of the 13 CALM sites covered by the AirMOSS transects were obtained in August of 2015 (Table 1). Among them, eight CALM sites obtained ALT measurements slightly earlier than the overflight date (within at most 18 days from 29 August 2015). Nevertheless, we assume that these earlier measurements still represent the thaw depth at the end of August reasonably well. Prior to comparison with the model results and the aggregated radar retrievals, the distributed measurements for a given CALM site (see sampling methods in Table 1) were averaged into a single value. If multiple CALM or UAF sites lay within a single CLSM grid cell , a single "spatially-averaged" observed value was computed for the cell.

We employed the strategy of Schaefer et al. (2015) to handle the uncertainty propagation, i.e., adding in quadrature the uncertainty components from each scale/level involved (see the supplementary file for a detailed description). For AirMOSS retrievals, the sampling uncertainty of mean ALT at the 81 km$^2$ model grid-cell scale is negligible given the large sampling size and the fact that the retrieval uncertainty dominates the overall uncertainty (see supplementary file). Here, we use a nominal estimate of 0.15 m to represent the AirMOSS uncertainty (i.e., the average of the lower and upper bound of the actual retrieval uncertainty for individual radar pixels as discussed by Chen et al. (2019)).

When comparing in-situ measurements with model results at the 81 km$^2$ scale (i.e., a point-to-grid comparison), the ultimate measurement uncertainty propagated from the point-scale measurements to the 81 km$^2$ scale is, for all intents and purposes, unknown due to a lack of sufficient measurements over the 81 km$^2$ scale to compute upscaling errors (see supplementary file). We thus show instead the standard deviation of CALM measurements to illustrate, in a highly approximate way, the spatial representativeness error of the in-situ measurements – a small (large) standard deviation represents a homogeneous (heterogeneous) area in terms of ALT, meaning that the in-situ mean likely can (cannot) represent an average over a larger

scale, assuming the site-scale heterogeneity is somewhat transferable to the larger scale. Such transferability might only apply to the largest in-situ site scales (e.g., 1000 m × 1000 m) to the model grid-scale (81 km$^2$) and is thus, in general, questionable. We thus make no claim here that the standard deviations shown represent true uncertainty levels.

## 3.2 Idealized Experiments

After comparing the spatial patterns of the AirMOSS retrievals with the CLSM-simulated ALT results, we then investigate the factors that affect the spatial variability of ALT through a series of idealized experiments. Specifically, we repeated the simulation along the AirMOSS transects multiple times, each time removing the spatial variation in some aspect of the model forcing or parameters and then quantifying the resulting impact on ALT variability.

For these supplemental simulations, we first identified a grid cell within the IVO transect (shown in Figure 2a) that represents roughly average (typical) conditions across the ten different transects. In the first idealized experiment, we then modified the baseline configuration by applying the surface meteorological forcing data from the selected representative grid cell within the IVO transect to all grid cells along all AirMOSS transects. Thus, in this modified simulation (HomF, for homogenized forcing), spatial variability in meteorological forcing is artificially removed. All model parameters related to soil type and vegetation, however, remain spatially variable, matching those in the baseline simulation. In the next idealized experiment (HomF&Veg), we further replaced the vegetation-related parameters (including vegetation class, vegetation height, and time-variable Leaf Area Index (LAI) and greenness) along the AirMOSS transects using the corresponding parameters from the representative grid cell, which is characterized by dwarf tree vegetation cover. Thus, in this simulation, spatial variability in both forcing and vegetation is artificially removed.

In a third idealized experiment (HomF&Veg&Soil), spatial variability in soil type and topography-related model parameters is removed along with that of the forcing and vegetation. The homogenized parameters include soil organic carbon content, porosity, saturated hydraulic conductivity, Clapp-Hornberger parameters, wilting point, soil class, sand and clay fraction,

vertical decay factor for transmissivity, baseflow parameters, area partitioning parameters, and timescale parameters for moisture transfer (Ducharne et al., 2000; Koster et al., 2000). Here we use an intermediate soil carbon content value (i.e., 40 kg/m$^2$) for the homogenization; recall that the carbon content impacts the soil thermal properties (see section 2.1). Our investigation reveals that the model sensitivity to soil carbon content is much larger for lower soil organic carbon content

(SOC) than for higher SOC, and easily gets saturated for high SOC (i.e., larger than ~100 kg/m$^2$) (not shown). Thus, we trust that 40 kg/m$^2$ is an appropriate value representing an intermediate SOC condition. All other soil parameters are homogenized to those at the representative grid cell.

Finally, we investigate potential nonlinearities by conducting two additional experiments: one in which we homogenized both

the vegetation and soil parameters (HomVeg&Soil) and another in which we homogenized both forcing and soil parameters (HomF&Soil). Put differently, in experiment HomVeg&Soil only the forcing varies along the transects, whereas in experiment HomF&Soil, only the vegetation parameters varies along the transects. Combined with the experiment HomF&Veg (in which only soil properties vary along the transects), these three experiments show in a different way how each individual factor (forcing, vegetation, or soil) can contribute to ALT variability. Table 2 provides a summary of these idealized experiments.

Taken together, the six experiments (including the baseline) allow us to identify the individual contribution of each factor to the ALT variability along the AirMOSS transects. The results are discussed in section 4.2.

### 3.3 Quantifying ALT Spatiotemporal Characteristics

In section 4.3 we quantify the large-scale characteristics of ALT over the Northern Hemisphere for the current climate (1980

- 2017) as determined by the response of the land model to 38 years of MERRA-2 forcing (section 2.1). Output from this multi-decadal, offline simulation allows the characterization of permafrost dynamics at each grid cell. In particular, we can compute a number of relevant ALT statistics, including mean, standard deviation, and skewness, from the diagnosed yearly values at each cell, and we can examine how these statistics relate to those of MERRA-2 forcing data (particularly the mean annual air temperature, MAAT) over the last 38 years.

Besides MAAT statistics, we also consider the evolution of the air temperature during the warm season in terms of the energy it could provide to the land surface and thus to the determination of ALT. A simple surrogate for the total warm-season energy in year N can be computed from daily-averaged air temperature, $T_{air}(t)$, and the freezing temperature, $T_f$ (0ºC degree), as follows:

$$T_{cum}(N) = \sum_{t=1}^{t=M} T_{pos}(t) , \tag{2}$$

where

$$T_{pos}(t) = \begin{cases} T_{air}(t) - T_f & if\ T_{air}(t) > T_f \\ 0 & if\ T_{air}(t) \leq T_f \end{cases}, \tag{3}$$

The index t in equation (2) for year N starts with a value of 1 on 1 September of year (N-1) and ends with a value of M on 31 August of year N. The number of days M is 365 or 366 depending on the presence of a leap year. Note the air temperature throughout this study means the near-surface air temperature (i.e., 2 m above the displacement height) derived from MERRA-2.

We first computed the correlation coefficient (R) between the annual time series of ALT and $\sqrt{T_{cum}}$ and between the annual time series of ALT and maximum SWE ($SWE_{max}$) to quantify the degree to which variations of ALT can be explained solely by air temperature or by snow mass. Then, to quantify the joint contributions of $\sqrt{T_{cum}}$ and $SWE_{max}$, we performed a multiple linear regression analysis by fitting the equation

$$ALT = a_0 + a_1\sqrt{T_{cum}} + a_2 SWE_{max} , \tag{4}$$

to the available data. The correlation coefficient relating ALT to $\sqrt{T_{cum}}$ and $SWE_{max}$ is the square root of the coefficient of multiple determination ($R^2$) obtained through fitting Equation (4). This equation is similar in form to the common degree-day model for predicting ALT from accumulated degree days of thaw based on the Stefan solution (e.g., Shiklomanov and Nelson, 2002; Zhang et al., 2005; Riseborough et al., 2008; Shiklomanov et al., 2010). Here, however, we constructed equation (4) for a different purpose: to explore how much of the temporal variability of ALT can be jointly explained by snow mass and

above-freezing air temperature. Before calculating these correlation coefficients, we removed the linear trend within ALT, $T_{cum}$, and $\text{SWE}_{max}$ to avoid potentially exaggerating the correlation due to an underlying trend. The results are discussed in section 4.3.

**3.4 Evaluating Simulated Northern Hemisphere Permafrost Extent and ALT**

We first evaluated the simulated permafrost extent against the observation-based permafrost map (Brown et al., 2002 as shown in Figure 1b). Note the model's description of permafrost is binary – either permafrost exists across a grid cell or it is completely absent. We cannot then expect an exact comparison to a specification of isolated permafrost (0-10% of area by definition) or even, to a lesser extent, sporadic permafrost (10-50% of area by definition). Therefore, we compared our

simulated permafrost area with that of the total area of continuous, discontinuous, and sporadic permafrost area together from Brown et al. (2002) and computed the percentage error relative to the observation-based area (i.e., the total area of continuous, discontinuous and sporadic permafrost regions). We also compared our simulated permafrost area against the total area of only continuous and discontinuous permafrost regions.

Further, the CALM network of in-situ ALT measurements (section 2.3) allows a quantitative evaluation of the simulated ALTs for the grid cells containing measurement sites. Our comparisons here focus on both multi-year annual ALTs and the average (climatological) ALT at the 81 km$^2$ scale of CLSM data. To ensure a consistent comparison, we average the simulated ALTs only over the years for which observations are available. As noted in section 3.1 and in the supplementary file, the uncertainty of the CALM ALT measurements in the context of evaluating grid cell-scale model results theoretically involves uncertainty

derived from probing point measurement uncertainty, site-scale mean uncertainty, and upscaling errors in going from the site-scale to the model-scale. This latter uncertainty in particular is unknown. In our figures (in section 4.4) we show the standard deviation of the observed ALT as a very crude surrogate for the spatial representativeness error associated with the point-to-grid comparison. As before, we make no claim here that the standard deviations shown represent the relevant statistical uncertainty. The results are discussed in section 4.4.

## 4 Results

### 4.1 Simulated ALT versus In-situ Measurements and AirMOSS Retrievals in Alaska

In this section, we compare the simulated ALT and the AirMOSS ALT retrievals at the 81-km$^2$ model resolution. Note that Chen et al. (2019) provide maps of the AirMOSS retrievals and an evaluation versus in-situ measurements at the native (20 m by 60 m) scale of the retrievals.

Figure 3 compares the spatial pattern of AirMOSS ALT retrievals and CLSM-simulated results. Generally, the patterns of the AirMOSS retrievals and CLSM results are quite different. For example, the AirMOSS-retrieved ALT is greater in the northern portion of the DHO transect than in the southern portion (Figure 3a), whereas this pattern is largely reversed in the simulated ALT for DHO (Figure 3b). Across all transects, there are portions where the AirMOSS ALT is less than the CLSM-simulated ALT and portions where the AirMOSS ALT is greater (Figure 3c), though it should be noted that the differences in Figure 3c are generally less than the assumed uncertainty of 0.l5 m (see section 3.1). Generally, the CLSM-simulated ALT shows relatively larger spatial variability (0.35 - 0.85 m) than the AirMOSS retrievals (0.4 – 0.6 m). The AirMOSS ALT exhibits some spatial variability at the native resolution (see Chen et al., 2019), but much of this variability averages out during the aggregation to the coarse model grid (Figure 3a). Variations of the simulated ALT within a single transect (Figure 3a) are predominantly induced by changes in soil type (indicated in Figure 2c and 2d). In essence, the higher the organic carbon content within the soil, the smaller the simulated ALT due to slower heat transfer associated with lower thermal conductivity, higher porosity, heat capacity, etc. (Tao et al., 2017). See also section 4.2 for a discussion of the influence of soil texture on the spatial pattern of ALT.

Next, we compare the simulated ALT in 2015 with in-situ observations from the CALM and UAF sites that are collocated with the AirMOSS transects (section 3.1). Figures 4a and 4b show that the CLSM-simulated ALTs agree with the in-situ observations with an overall mean bias of -0.05 m and a RMSE of 0.17 m. The most significant discrepancies between the CLSM-simulated ALT and in-situ measurements are at U6, U31, FB1&FBD&FBW (Figure 4a), where the simulated ALT

underestimates the in-situ measurements by 0.25-0.28 m, and at U28 where the simulated ALT overestimates the in-situ ALT by 0.27 m. Nevertheless, the scatter in Figure 4b is large, and the corresponding correlation coefficient is quite weak (0.27).

The AirMOSS ALT radar retrievals, for their part, again averaged to the 81-km$^2$ model resolution (section 2.2), show less spatial variability than the observations (Figure 4a). The largest error for the AirMOSS retrievals at the model scale is also at FB1&FBD&FBW, where the retrievals significantly underestimate the observed in-situ ALT by 0.38m. Note that radar retrievals at the 81 km$^2$ scale are not available at some sites because of our imposed 30% filling restriction. Although the AirMOSS ALT retrievals generally underestimate the in-situ ALT measurements (as shown in Figure 4a), the retrievals tend to be more consistent with the observations when the in-situ measurements are within the ~60 cm sensing depth of the P-band radar data, as indicated in Table 3. Specifically, excluding the sites with in-situ ALT measurements that exceed the AirMOSS sensing depth of ~60 cm, the overall mean bias for the AirMOSS retrievals at the 81 km$^2$ scale drops to -0.01 m, and the correlation coefficient increases to 0.64. In contrast, the CLSM simulation results show a bias of 0.01 m and a zero correlation coefficient at these sites.

Nevertheless, as noted in section 3.1, given that the upscaling errors in going from the CALM site-scale to the model-scale is unknown and the fact that the standard deviation of these measurements (as shown by error bars in Figure 4a and 4b) indicates large representativeness errors of the in-situ measurements, the point-to-grid comparison result is hard to quantify. In this regard, the AirMOSS retrievals aggregated to the same scale as model results provide a comparable counterpart for evaluation. Figures 4c further shows that the CLSM-simulated ALT agrees well with the AirMOSS ALT retrievals to within the measurement uncertainty of 0.15 m at all the site-located model grid cells. Indeed as Figure 3c illustrated, the differences between simulated ALT and the AirMOSS retrievals over all the transects examined here are generally below the measurement uncertainty of 0.15 m.

**4.2 Sources of ALT Spatial Variability:  Results from Idealized Experiments**

Here we investigate the specific factors that drive ALT spatial variability along all ten of the AirMOSS transects (Figure 2a). For this analysis, the simulated ALT estimates were aggregated across the width of the radar swath (compare Figure 3).  Figure 5a illustrates that the simulated ALT captures the spatial variability exhibited by the in-situ measurements. This conclusion is, however, very tentative given the limited number of in-situ ALT observations.

The simulated ALT is shallowest in the northern transects (ATQ, BRW, and DHO) and deepest in the southeastern transects (KYK, COC, KGR, and AMB).  This pattern correlates somewhat (R = 0.46) with that of the mean screen-level (2-meter) air temperature (Tair) for the preceding 12-month period (i.e., from 1 September 2014 to 31 August 2015) from MERRA-2 (green line in Figure 5a).  The soil carbon content, by contrast, appears anti-correlated (R = -0.59) with the simulated ALT, as exemplified by the transect portions within the red box (Figure 5a and 5b). Such a correlation presumably reflects the fact that soil with high organic carbon content has low thermal conductivity, which hinders heat transfer from the surface to the deeper soil in the summertime, thus resulting in a relatively smaller ALT.  In addition, heat transfer is slowed by a higher effective heat capacity associated with higher organic carbon content – not from the carbon itself, but from the extra water that can be held in the soil due to the increased porosity.  The maximum snow depth (Figure 5c) displays a positive correlation with ALT (R=0.47), reflecting, at least in part, the fact that subsurface soil temperatures remain relatively insulated under thick and persistent snow cover, which reduces heat transfer out of the soil column during the wintertime and hence facilitates a deeper thawing during the summer and thus a deeper ALT.

The correlations in Figure 5 suggest (without proving causality) that for the model, surface meteorological forcing (including air temperature and precipitation) as well as soil type are important drivers of ALT variability along the AirMOSS transects. However, the relatively low values of the correlations indicate that a simple linear relationship cannot explain the mutual control that these variables exert on ALT spatial variability.  In the remainder of this section, we use a series of idealized model simulations (as described in section 3.2) to better quantify the relative impacts of these driving factors along the AirMOSS transects.

The results of the idealized experiments are shown in Figure 6. The above-mentioned, large-scale spatial variation of ALT in the baseline simulation, with larger values in the southeastern transects (KYK, COC, and KGR) and lower values in the northern transects (ATQ, BRW, and DHO), is absent after homogenizing the meteorological forcing (HomF; Figure 6a). Experiment HomF correspondingly has much less spatial variation in the temperature of the top soil layer than does the baseline simulation (Figure 6b). In addition, homogenizing the forcing (which includes snowfall) significantly reduces the variability in maximum snow depth along the AirMOSS transects (Figure 6c). These results indicate that in the model, meteorological forcing exerts the dominant control over the spatial patterns of ALT, the temperature in the top soil layer, and snow depth at the regional scale, as expected.

Homogenizing the vegetation attributes in addition to the forcing (HomF&Veg) results in ALT differences (relative to HomF) primarily along the northern transects (ATQ, BRW, and DHO). Along these transects, homogenizing the vegetation parameters (including LAI and tree height) to those of the representative grid cell within the IVO transect results in generally shallower ALT. This is because the generally lower albedo of the taller and leafier trees (representative of the IVO transect) during the snow season resulted in increased snowmelt and thus reduced snowpack during the snow season (compare the green and red curves in Figure 6c), thereby reducing the thermal insulation of the wintertime ground. With reduced insulation, cold season ground temperatures dropped, making it more difficult for temperatures to recover during summer(Tao et al., 2017).

As might be expected, the simulation in which soil properties are homogenized in conjunction with forcing and vegetation (i.e., HomF&Veg&Soil) essentially eliminates all remaining spatial variability in ALT, snow depth, and soil temperature. Owing to the strong control of soil type-related parameters (see section 3.2 and Table 2) on soil moisture, spatial variability in soil moisture remains high in HomF and HomF&Veg and is only eliminated once these soil type-related parameters are homogenized (Figure 6d), which explains the abrupt changes shown in Figure 3c as mentioned in section 3.1. (Note that to maintain consistency with the hardwired scaling factors for snow-free albedo within the model (Mahanama et al., 2015), we

still used the original, vegetation-related parameters to calculate surface albedo during snow-free conditions along the transects. This is likely the cause of the few tiny bumps seen in the Figure 6a for HomF&Veg&Soil.)

An alternative view of these results is provided in Figure 7a, which shows the (spatial) standard deviation of ALT along the AirMOSS transects for each of the above experiments. Homogenizing the meteorological forcing data results in a significant reduction of the ALT standard deviation (from 0.16 to 0.10). Additionally homogenizing the vegetation only reduces the ALT standard deviation slightly (from 0.10 to 0.09). The remaining ALT variability is eliminated through the additional homogenization of the soil type-related parameters (HomF&Veg&Soil), which emerge as another important driver of ALT variability along the AirMOSS transects. Note that the ALT variability associated with soil type is generally realized at smaller spatial scales than that associated with the meteorological forcing discussed earlier regarding Figure 6a. The impact of potential nonlinearities are examined in Figure 7b, which shows the individual impact of vegetation, soil, and forcing heterogeneity on the ALT standard deviation along the transects, with the other inputs having been homogenized. The graphic confirms that the meteorological forcing is the dominant driver of ALT spatial variability in our modelling system, followed by the soil type-related parameters and the vegetation parameters.

Note that in Figure 6a, the soil impact on ALT (difference between HomF&Veg&Soil in black and HomF&Veg in red) appears smaller than that of the vegetation (difference between HomF in green and HomF&Veg in red) over the northern transects (ATQ, BRW and DHO). Even so, Figure 7b shows that, in terms of the integrated impact along all the transects, the soil influence clearly outweighs the influence of vegetation – at several other transects, including HUS, KYK, COC, AMB, IVO and the first half of ATQ (where vegetation conditions might be similar to those used for homogenizing), the changes in vegetation parameters do not have much impact.

**4.3 Spatiotemporal Characteristics of ALT across the Northern Hemisphere**

Figure 8a shows the distribution of mean ALT over the modelling domain, and Figure 8b shows the ALT standard deviation in time over the 38-year period. As might be expected, ALT tends to increase with distance from the pole, with the largest values found in Mongolia and near the southern portion of Hudson Bay, though there are areas (e.g., just north of 60ºN at

~120ºE) with local minima that break this pattern.  The largest ALT standard deviations (red color in Figure 8b) are found mainly in discontinuous and sporadic permafrost regions (see Figure 1b) where ALTs are deeper on average than that in continuous permafrost region.  Figure 8c provides the skewness of the temporal distribution. Though there are some exceptions, by and large the skewness is positive in most permafrost regions, suggesting that the largest positive ALT anomalies tend to be of greater magnitude than the largest negative anomalies.

Figure 8d displays the average of annual mean 2-meter air temperature as derived from MERRA-2. The observed continuous and discontinuous permafrost areas shown in Figure 1b are well confined within the cold side of the 0°C (273.15K) isotherm in the mean air temperature map (Figure 8d). For the most part, the observed sporadic and isolated permafrost regions of Figure 1b also lie on the cold side of the 0°C isotherm. The consistency with this isotherm, however, is not as clearly present in the

simulated permafrost extent (i.e., the extent of the non-grey and non-white areas in Figure 8a).

The relationship between the spatiotemporal characteristics of simulated ALT and air temperature forcing has been investigated before in many studies at the site to landscape scale (e.g., Klene et al., 2001; Shiklomanov and Nelson, 2002; Zhang et al., 2005; Juliussen and Humlum, 2007) and at the regional scale (e.g., Anisimov et al., 2007). Here we simply

analyze the correlation coefficient between ALT and two variables: the proxy of total energy input into the ground (i.e., $\sqrt{T_{cum}}$, see section 3.3) and the maximum SWE.  Our goal is to explore how much of the spatiotemporal variability of ALT across the globe can be jointly explained by these two variables.

Figure 9a shows a map of the correlation coefficient between the 37-year time series (i.e., from September 1980 through

August 2017) of $\sqrt{T_{cum}}$ and the corresponding time series of simulated ALT. The areas with p values larger than 0.05, which

indicate correlations that are not statistically different from zero at the 95% confidence level, are shown as green. Figure 9a demonstrates that most permafrost regions indeed have significant positive correlations (red colours) between ALT and $\sqrt{T_{cum}}$. Clearly, in these regions, air temperature exerts a dominant control on year-to-year ALT variability.

However, not all regions exhibit a significant correlation; other variable(s) must also be exerting control on interannual ALT variability. One reasonable candidate variable is snowpack. As noted above, snow acts as a thermal insulator -- regions with thicker snowpack are better able to insulate the ground from becoming too cold during winter, thereby supporting higher subsurface temperatures during non-winter months. Variable, but often thick, snowpack is in fact common in the areas of Figure 9a that show a low (green) or negative (blue) correlation between ALT and $\sqrt{T_{cum}}$ – areas such as Central Siberia, the

Southern part of eastern Siberia, and a vast region in Canada surrounding the Hudson Bay, as well as other small areas that appear in high mountains or on the windward side of the mountains (e.g., locations B, C and D in Figure 1a).

       In Figure 9b we show the correlation coefficient between the time series of ALT and the maximum SWE ($SWE_{max}$) during the preceding winter. A positive correlation is seen in many areas, most notably in areas with a poor or negative correlation

between ALT and $\sqrt{T_{cum}}$ (Figure 9a) – for example, just west of Hudson Bay and along a zonal band at 60°N in Russia. Apparently, in these areas, the impacts of snow physics on ALT outweigh the impacts of lumped energy input ($\sqrt{T_{cum}}$). In some other areas ALT correlates positively with both $\sqrt{T_{cum}}$ and $SWE_{max}$. Figure 9c shows how the resulting coefficient of multiple correlation varies in space. High correlations largely blanket the modelled area. That is, over most of the area examined, a substantial portion of the year-to-year variability of ALT can be explained by joint variations in $\sqrt{T_{cum}}$ and

$SWE_{max}$. Even so, a few limited areas still exhibit low correlations ($p>0.05$, green colour in Figure 9c). Some of these areas are in mountainous regions, for instance the Eastern Siberian (Ostsibirisches) Bergland, where more complex environmental controls might be playing a dominant role. In addition, MERRA-2 snow forcing might be severely erroneous in these regions.

**4.4 Evaluation of Simulated Permafrost Extent and ALT across the Northern Hemisphere**

Qualitatively, the simulated permafrost extent (Figure 8a) generally shows reasonable agreement with the observation-based permafrost map in Figure 1b, especially for the continuous permafrost regions. This is shown explicitly in Figure 10a. The main deficiency in the simulation results is the failure to capture a large area of permafrost in western Siberia (labelled as A

in Figure 1a). The reasons for this particular deficiency are unclear. One possible reason is that the permafrost in western Siberia is characterized as an ecosystem-protected permafrost zone (Shur and Jorgenson, 2007) where a thick moss-organic layer (i.e., moss-dominated mires (Anisimov and Reneva, 2006; Anisimov, 2007; Peregon et al., 2009)) protects the permafrost below from thawing under a warm air temperature. This is mainly attributed to the low thermal conductivity of the organic layer in summer, which strongly insulates the permafrost from the warm atmosphere, and the high thermal conductivity of the

frozen organic layer in winter, which allows cold temperature penetration from above, provided the snowpack is not too thick (Nicolsky et al., 2007b; Jafarov and Schaefer, 2016). This mechanism is lacking in the current version of CLSM (Tao et al., 2017). Thus, improving the model through a better representation of thermal processes in an organic layer above the soil column in combination with initializing the simulation with a sufficiently cold soil temperature should improve the simulation results. This work is reserved for a future study.

Another possible reason for the poor skill in western Siberia is that the model initial conditions there were too warm, although MERRA-2 appears to underestimate summer air temperatures in this region (Draper et al., 2018; their Figure 7e). Note that some other global models, such as CLM3 and the Community Climate System Model version 3 (CCSM3) as reported in Lawrence et al. (2012), also missed this area of permafrost and that updated versions of these models (i.e., CLM4 and CCSM4)

showed improved performance in this regard (Lawrence et al., 2012). Guo et al. (2017) reported underestimated permafrost extent simulated in western Siberia using CLM4.5 driven by three different reanalysis forcings (i.e., CFSR, ERA-I and MERRA), and they showed an improved simulation of permafrost extent in this area when using another reanalysis forcing, the CRUNCEP (Climatic Research Unit - NCEP) (Guo and Wang, 2017). Guimberteau et al. (2018) found similar improvements stemming from the use of CRUNCEP forcing. We leave for further study whether the MERRA-2 forcing data

is responsible for the western Siberia deficiency seen in our own results.

The disagreements between the simulated and observed permafrost extents (covering about a few degrees latitude) toward the south in Figure 10a (green and blue areas at the southern edge of permafrost regions) are less of a concern, since the comparison in such areas is muddied by the interpretation of "isolated" permafrost in the observational map (Figure 1b). The specific areas of each type shown in Figure 10a are listed in Table 4. The simulated permafrost extent covers 81.3% of the observation-based area (i.e., the total area of continuous, discontinuous and sporadic permafrost regions), and misses 18.7% of the observed permafrost area. When comparing simulated permafrost extent with only continuous and discontinuous types, these metrics change to 87.7% and 12.3%, respectively. Meanwhile, the permafrost extent is overestimated by $3.2 \times 10^6$ km$^2$.

To produce Figure 10b, multi-year averages of CLSM-simulated ALT values were spatially averaged over each of the four permafrost types outlined in Figure 1b. (As is appropriate, permafrost is only occasionally simulated over the fourth, "isolated", permafrost type. The ALT average shown for this type is thus based on a particularly limited number of grid cells.) The average ALT is smallest in the continuous permafrost zone, higher in the discontinuous zone, and higher still in the sporadic permafrost zone; it is highest in areas of isolated permafrost. The progression, of course, is in qualitative agreement with expectations – larger breaks in permafrost coverage imply a greater amount of available energy, which should also act to increase ALT.

The observed and CLSM-simulated annual ALT and multi-year ALT averages are compared in Figure 11. Generally, the simulated annual ALT and the averages agree reasonably well with observations for shallow permafrost regions, that is, for smaller ALT. A large bias, however, is found for most of the Mongolia sites; in Mongolia, the observed annual ALT and the climatological ALTs tend to be much larger than the simulated ALTs (light purple dots in Figure 11). Overall, the RMSE, bias and R are all significantly improved when the Mongolian sites are excluded from consideration. Specifically for the climatological ALTs, the RMSE (and bias) of simulated ALT climatological means is 1.22 m (and -0.48 m), and it drops to 0.33 m (and -0.04 m) if the Mongolia sites are excluded (Figure 11d). Given simplifications in the model, uncertainties in boundary conditions (e.g., vegetation types, soil properties, etc.), and upscaling issues stemming from the coarse-scale nature

of the forcing data relative to the point-scale and plot-scale nature of the observations (i.e., the representative errors as indicated by the large standard deviation shown in Figure 11a), these results seem encouraging. The correlation coefficient metric (R), however, is somewhat less encouraging, amounting to only 0.5 when considering all sites. The correlation coefficient is in fact lower (0.3) when the Mongolian sites are excluded; the correlation coefficient is 0.39 for the Mongolian sites considered

in isolation. Note that the existing literature on simulated ALT fields (e.g., Dankers et al. (2011), Lawrence et al. (2012) and Guo et al. (2017)) reveals a general tendency for models to overestimate ALT climatology at the global scale. In light of this, our results suggest that the CLSM-simulated ALT fields are perhaps among the better simulation products, especially for shallow permafrost.

Comparing the observed and simulated spatial distributions of the ALT averages provides a further test of the accuracy of the simulation results (as shown in Figure 12). The model successfully simulates the large-scale spatial patterns in ALT, capturing, for example, the variations in Siberia, Svalbard, northern Canada, and northern Alaska (see Figure 12a, b). Figure 12c, d show the differences between the observed and estimated values in middle latitudes ($45°N$ to $60°N$) and high latitudes ($60°N$ to $90°N$), respectively; in agreement with Figure 11a, the model clearly performs better in high-latitude regions, i.e., outside of

Mongolia. Many of the sites north of $60°N$ (Figure 12d) are coloured grey, indicating a small error in the simulation of ALT at these sites – the errors at these sites range from only -0.10m to 0.10m.

The significant underestimation of ALT in Mongolia might result from errors in the meteorological forcing provided by MERRA-2. However, a comparison (not shown) of MERRA-2 air temperatures with measurements at six weather stations

collocated with CALM sites in Mongolia calls this explanation into question. While MERRA-2 summer temperatures are indeed too low at four of the weather stations examined, they are too high at the other two weather stations. An additional reason for the underestimation of ALT in Mongolia might be a mismatch between the land surface parameter values used in the model and the actual conditions at each site. For instance, detailed soil information (https://www2.gwu.edu/~calm/data/webforms/mg_f.html) indicate that some Mongolian sites have special "rocky" soil types

including limestones (e.g., M04), slatestones (e.g., M05), gravelly sand (e.g., M06 and M08), etc. that are not well represented

in the model. As another example, sites on south-facing slopes presumably have much deeper ALT than those on slopes with less exposure to the sun, which is not captured by CLSM. The large representative errors of Mongolian sites are clearly illustrated by the standard deviation (although computed only with 3 to 5 measurements) as shown by the error bars in Figure 11a.

## 5 Conclusion and Discussion

We produced a dataset (effectively a derivative of MERRA-2) of permafrost variations in space and time across middle-to-high latitudes. This dataset can be considered unique in terms of its daily temporal resolution combined with a relatively high spatial resolution at the global scale (i.e., 81 km$^2$). The dataset, which is derived from a state-of-the-art reanalysis (MERRA-

10 2), shows reasonable skill in capturing permafrost extent (87.7% of the total area of continuous and discontinuous types, according to one validation dataset) and in adequately estimating ALT climatology (with a RMSE of 0.33m and a mean bias of -0.04m), excluding Mongolian sites. We note that our MERRA-2-driven permafrost simulation results, while potentially better than those we might have obtained with MERRA forcing, are still lacking (e.g., in western Siberia). Still, with its resolution and available variables (ALT, subsurface temperature at different depths), the dataset could prove valuable to many

future permafrost analyses.

This work also provides a first comparison between two highly complementary approaches to estimating permafrost: model simulation and remote sensing. In northern Alaska, excluding sites that have ALT measurements exceeding the radar sensing depth (~ 60cm), the evaluation metrics for ALT retrievals against in-situ measurements are better than those for simulated

ALT at the 81 km$^2$ scale. However, the remotely sensed ALT estimates generally show lower levels of spatial variability relative to the simulated ALT estimates (and relative to the in-situ observations), and the spatial patterns of the simulated and retrieved values differ considerably. The remote sensing approach is still relatively new, with many aspects still requiring development. It is important, though, to begin considering the modeling and remote sensing approaches side by side, as both should play important roles in permafrost quantification in the years to come. Indeed, once the science fully develops, joint

use of modeling and remote sensing (e.g., through the application of downscaling methods) should allow the generation of more accurate permafrost products at higher resolution.

It is important to note that the retrieved ALT was determined by the dielectric transition from thawed to frozen conditions whereas the modelled ALT and the ALT for some of the in-situ measurements was based on a freezing temperature of 0ºC (see sections 2.1 and 2.3). Depending on local conditions, soil does not typically freeze at 0ºC but rather at slightly lower temperatures (e.g., around -1ºC) due to the presence of dissolved compounds that depress the freezing point (Watanabe and Wake, 2009). The sharp drop in conductivity and dielectric constant is much more accurately tied to a frozen state than to a temperature threshold. These and other differences in the various ALT measurement methods (section 2.3) introduce considerable uncertainty into our comparisons. The use of the 0ºC degree threshold in CLSM for determining the thawed or frozen layer may explain in part the model's underestimation of ALT, as may the lack of an explicit treatment of local aspect, errors in assigned model parameters, and so on.

Analysis of the CLSM-simulated data, along with data produced in idealized experiments with specific homogenized controls, show how the statistics of permafrost variability in space are controlled by forcing variability and by variability in the imposed surface boundary conditions. In the idealized experiments, we employ successive homogenization of controls to quantify how meteorological forcing, soil type, and vegetation cover affect the underground thermodynamic processes associated with the variability of ALT along the AirMOSS flight paths in Alaska. Meteorological forcing and soil type are found to be the two dominant factors controlling ALT variability along these transects. Vegetation plays a smaller role by modulating the accumulation of snow. A multiple regression analysis relating yearly ALT jointly to accumulated air temperature and maximum SWE shows that time variations in these two latter quantities explain most of the time variability of ALT in the CLSM-identified permafrost regions.

Many aspects of the modelling framework may contribute to the noted errors in the simulated ALTs. For example, the observed climatological ALTs at the Mongolia sites are all larger than 3m. This depth falls well within the 6th soil layer of the model,

which has a thickness of 10m; the subsurface vertical resolution in the CLSM may be too coarse to capture these deeper ALTs. Test simulations (not shown) with alternative model configurations indicate that increasing the number of soil layers may act to decrease somewhat the simulated ALT, suggesting that our values may be a little overestimated; however, based on results from a new study by Sapriza-Azuri et al.(2018), our use of a no-heat-flux condition at the bottom boundary rather than a

dynamic geothermal flux may lead to underestimates of ALT. Such uncertainties should naturally be kept in mind when interpreting our results. Our supplemental simulations (not shown) also suggest that increasing the total modelled soil depth has only a small impact on simulated ALT. Uncertainty in our description of soil organic carbon, i.e., both soil carbon content and vertical carbon distribution, leads to corresponding uncertainty in our ALT simulations. We indeed find a significant improvement in simulated ALT at several Mongolian sites when we arbitrarily impose less total soil carbon content and

concentrate less soil carbon in top layers (not shown). Besides the vertical distribution of soil carbon, the vertical variation in other soil hydrological properties (e.g. soil texture and porosity) should also play a significant role since they all affect soil thermal conductivity and heat capacity. In addition, the lack of a necessary organic layer on top of soil column and the related thermal processes is also a major deficiency for the model especially in ecosystem-protected performant regions.

Another issue affecting our ALT comparisons is the climatological representation of vegetation parameters such as LAI used in CLSM. An additional investigation (not shown) revealed large differences between the LAI climatology used in CLSM and more realistic, time-varying, satellite-based LAI products at several Mongolian sites. In addition, while we did exclude from our analyses any measurements that were affected by notable disturbance (e.g., wildfire), the impacts of other potential land changes on ALT, including overgrazing in Mongolia (Sharkhuu and Sharkhuu, 2012; Liu et al., 2013), were not explicitly

treated in the model. The model also lacks the vertical advective transport of heat in the subsurface due to downward flowing liquid water, which can significantly affect permafrost thawing (Kane et al., 2001; Rowland et al., 2011; Kurylyk et al., 2014). Also relevant are potential errors in the MERRA-2 forcing. The MERRA-2 reanalysis is known to have problems capturing trends in high latitudes (Simmons et al., 2017).

Such modelling deficiencies must always be kept in mind when evaluating a product like the one examined here. That said, as long as appropriate caution is employed, the product could have significant value for further analyses of permafrost. The product features daily subsurface temperatures and depth-to-freezing estimates over middle-to-high latitudes in the Northern Hemisphere at an 81 km$^2$ resolution, covering the period 1980-2017. It is, in a sense, a value-added derivative product of the

MERRA-2 reanalysis and will be available via the National Snow and Ice Data Center (NSIDC). The comparisons against observations discussed above, along with the intuitively sensible connections shown between permafrost variability, forcing variability, and boundary condition variability, gives confidence that this dataset contains useful information. These data can potentially contribute, for example, to ecological studies focused on the dynamics of microbial activity and soil respiration in cold regions, on vegetation migration/adaptation in response to climate change, and so on.

**Acknowledgments**

Funding for this work was provided by the NASA Interdisciplinary Science program (NNX14AO23G). We thank Qing Liu at GMAO/GSFC/NASA for providing us corrected MERRA-2 precipitation. We acknowledge the University of Maryland supercomputing resources (http://www.it.umd.edu/hpcc) made available for the research reported in this paper.

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

Table 1 – In-situ permafrost measurement sites covered by the AirMOSS transects in 2015.

| AirMOSS flight (Official full name) | Permafrost Site (CALM or UAF)* | Latitude (degree) | Longitude (degree) | Sampling Method@ | Measurement Date |
|---|---|---|---|---|---|
| COC (Council) | U27 (CALM) | 64.8333 | -163.7000 | 4 | 8/30/2015 |
| | U28 (CALM) | 65.4500 | -164.6167 | 4 | 8/29/2015 |
| IVO (Ivotuk) | IV4 (UAF) | 68.4803 | -155.7437 | 1# | 8/29/2015 |
| ATQ (Atqasuk) | U3 (CALM) | 70.4500 | -157.4000 | 4 | 8/25/2015 |
| BRW (Barrow) | U1 (CALM) | 71.3167 | -156.6000 | 4 | 8/21/2015 |
| | U2 (CALM) | 71.3167 | -156.5833 | 2 | 8/24/2015 |
| | BR2 (UAF) | 71.3090 | -156.6615 | 1 | 8/29/2015 |
| DHO (Deadhorse) | U4 (CALM) | 70.3667 | -148.5500 | 3 | 8/25/2015 |
| | U5 (CALM) | 70.3667 | -148.5667 | 4 | 8/11/2015 |
| | U6 (CALM) | 70.1667 | -148.4667 | 3 | 8/26/2015 |
| | U31 (CALM) | 69.6969 | -148.6821 | 3 | 8/15/2015 |
| | U8 (CALM) | 69.6833 | -148.7167 | 3 | 8/27/2015 |
| | U32A (CALM) | 69.4410 | -148.6703 | 3 | 8/16/2015 |
| | U32B (CALM) | 69.4010 | -148.8056 | 3 | 8/16/2015 |
| | U9A (CALM) | 69.1667 | -148.8333 | 3 | 8/25/2015 |
| | WD1 & WDN (UAF) | 70.3745 | -148.5522 | 1 | 8/29/2015 |
| | DH2 (UAF) | 70.1613 | -148.4653 | 1 | 8/29/2015 |
| | FB1 (UAF) | 69.6739 | -148.7219 | 1 | 8/29/2015 |
| | FBD (UAF) | 69.6741 | -148.7208 | 1% | 8/29/2015 |

| | | | | | |
|---|---|---|---|---|---|
| | FBW (UAF) | 69.6746 | -148.7196 | 1 | 8/29/2015 |
| | SG1 (UAF) | 69.4330 | -148.6738 | 1 | 8/29/2015 |
| | SG2 (UAF) | 69.4283 | -148.7001 | 1 | 8/29/2015 |
| | HV1 (UAF) | 69.1466 | -148.8483 | 1[%] | 8/29/2015 |

* CALM: sites from the Circumpolar Active Layer Monitoring (CALM) network; UAF: sites from the Permafrost Laboratory at the University of Alaska Fairbanks (UAF).

@Sampling method: 1. Single point; 2. 320 random sampling points within 10m × 10m area; 3. 100m × 100m grid with a 10m sampling interval; 4. 1000m × 1000m grid with a 100m sampling interval.

5   # Two sensors are installed at IV4.

%Observations were taken from two conditions, including a frost-boil and an inter-boil area.

Table 2 – List of idealized simulation experiments along the AirMOSS transects.

| Experiment Name | Meteorological forcing | Vegetation | Soil parameters* |
|---|---|---|---|
| Baseline | Original | Original | Original |
| HomF | Homogenized | Original | Original |
| HomF&Veg | Homogenized | Homogenized | Original |
| HomF&Veg&Soil | Homogenized | Homogenized | Homogenized |
| HomVeg&Soil | Original | Homogenized | Homogenized |
| HomF&Soil | Homogenized | Original | Homogenized |

*CLSM soil parameters include soil organic carbon content, porosity, saturated hydraulic conductivity, Clapp-Hornberger parameters, wilting point, soil class, sand and clay fraction, vertical decay factor for transmissivity, baseflow parameters, area partitioning parameters, and time scale parameters for moisture transfer (Koster et al., 2000; Ducharne et al., 2000; Tao et al., 2017).

Table 3 – Evaluation metrics for model-simulated ALT and AirMOSS retrievals for 2015.

| Metric | All sites | | Sites with ALT measurements within AirMOSS sensing depth (~60 cm) | |
|---|---|---|---|---|
| | CLSM-simulated ALT | AirMOSS ALT retrievals | CLSM-Simulated ALT | AirMOSS ALT retrievals |
| RMSE (m) | 0.17 | 0.17 | 0.12 | 0.06 |
| Bias (m) | -0.05 | -0.12 | 0.01 | -0.01 |
| R | 0.27 | 0.61 | -0.00 | 0.64 |

Table 4 – Evaluation results for simulated permafrost extent against the permafrost map by Brown et al. (2002). The calculation was based on the comparison between simulated permafrost area and the total area of continuous, discontinuous and sporadic permafrost regions from Brown's map. The number in the brackets was calculated against the total area of continuous and discontinuous permafrost regions.

| Case | CLSM | Obs. | Simulated Area ($\times 10^6\,\mathrm{km}^2$) | Percentage Relative to Observation |
|------|------|------|-----------------------------------------------|------------------------------------|
| 4 | No | No | 48.8 | - |
| 3 | Yes | No | 1.9 | - |
| 2 | No | Yes | 3.2 (1.7) | 18.7% (12.3%) |
| 1 | Yes | Yes | 13.8 (12.3) | 81.3 % (87.7%) |

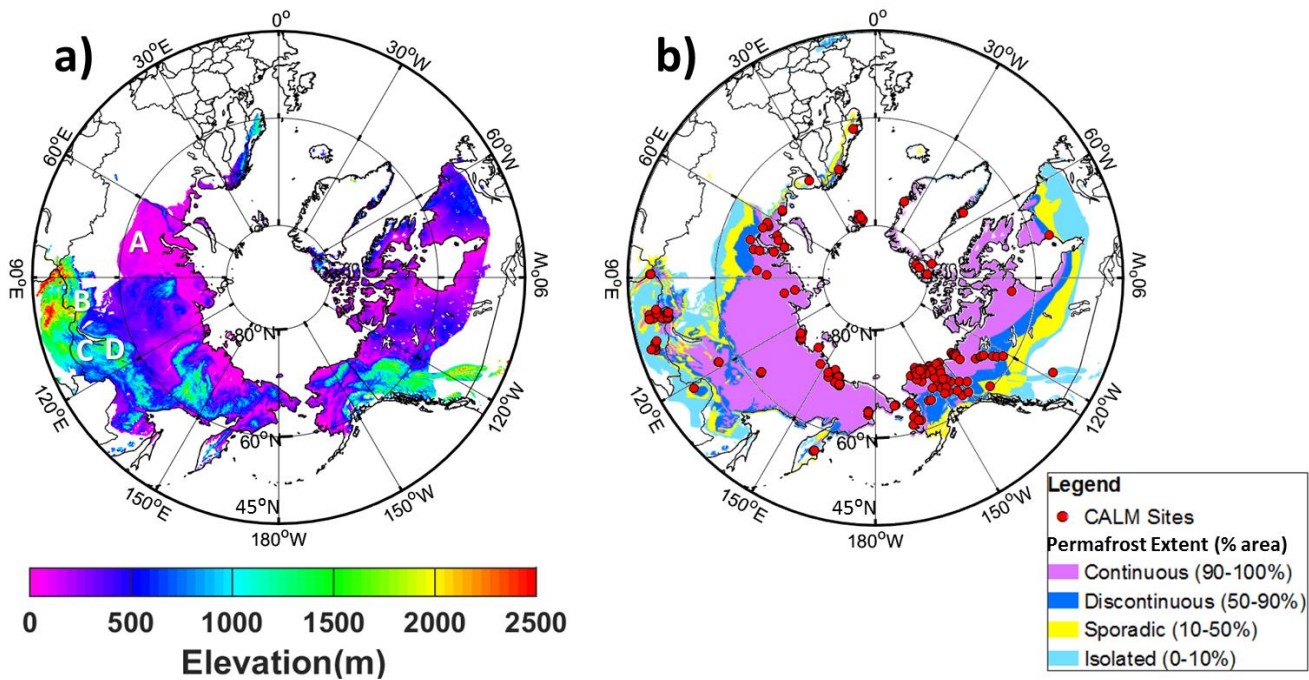

Figure 1: a) Elevation above mean sea level in the simulation domain, which is defined by the area for which NCSCDv2 data are available. Regions A, B, C, and D are discussed in the text. b) Permafrost and ground ice conditions adapted from (Brown et al., 2002). Red dots represent CALM sites.

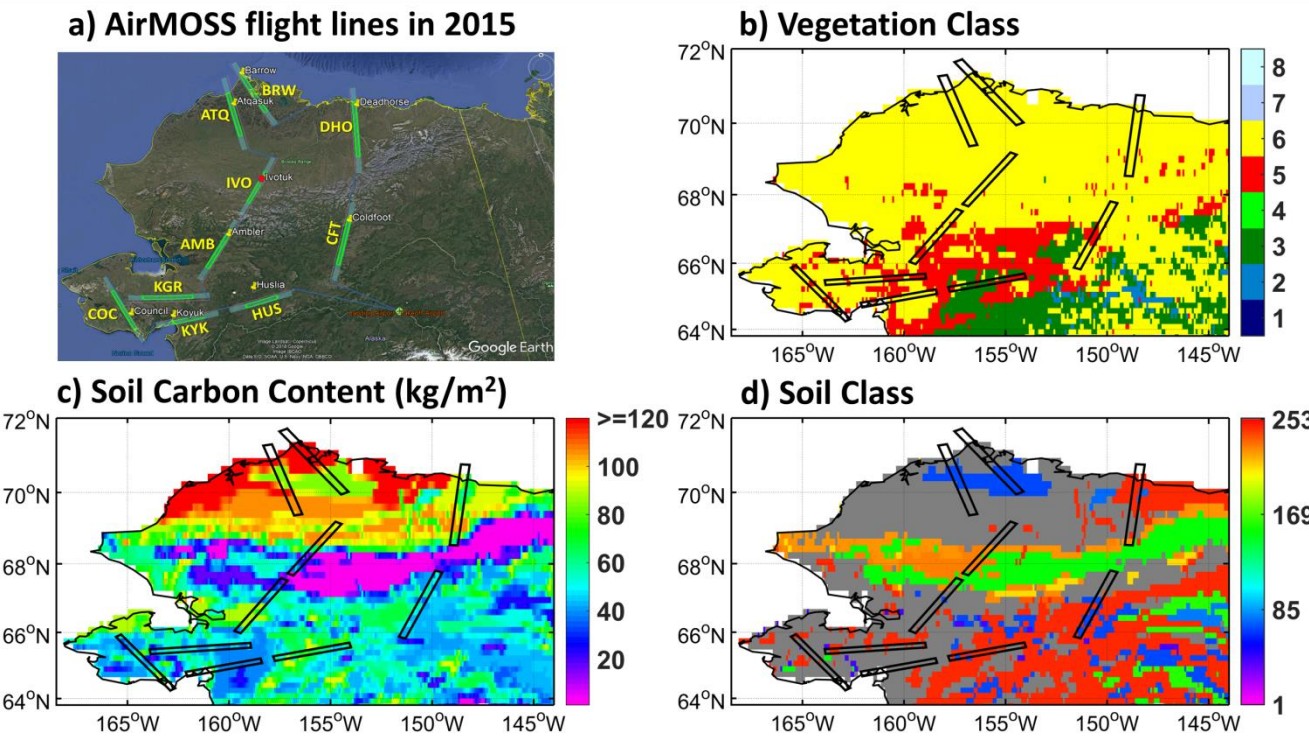

**Figure 2: a) Ten transects of AirMOSS flights conducted in Alaska on 29 August 2015 and 1 October 2015, including HUS (Huslia), KYK (Koyuk), COC (Council), KGR (Kougarok), AMB (Ambler), IVO (Ivotuk), ATQ (Atqasuk), BRW (Barrow), DHO (Deadhorse), and CFT (Coldfoot). Each flight swath width is approximately 15 km. The red dot on IVO illustrates the location of the representative grid cell used and discussed in section 3.2. b) Vegetation class, c) soil organic carbon content, and d) soil class used in CLSM. The eight vegetation classes are 1-broadleaf evergreen trees, 2-broadleaf deciduous trees, 3-needleleaf trees, 4-grassland, 5-broadleaf shrubs, 6-dwarf trees, 7-bare soil, and 8-desert soil, respectively. The 253 soil classes include one "peat" class (#253), which is shown in dark grey, and 252 mineral soil classes (De Lannoy et al., 2014).**

**Figure 3: a) Radar retrievals of ALT derived from P-band radar observations on 29 August 2015 and 01 October 2015 for IVO, ATQ, BRW, and DHO, aggregated to 81 km² model grid cells. b) CLSM-simulated ALT. c) Difference between the aggregated ALT retrievals and the CLSM-simulated results. Magenta squares represent CALM sites covered by the flight swath whereas black circles represent UAF sites.**

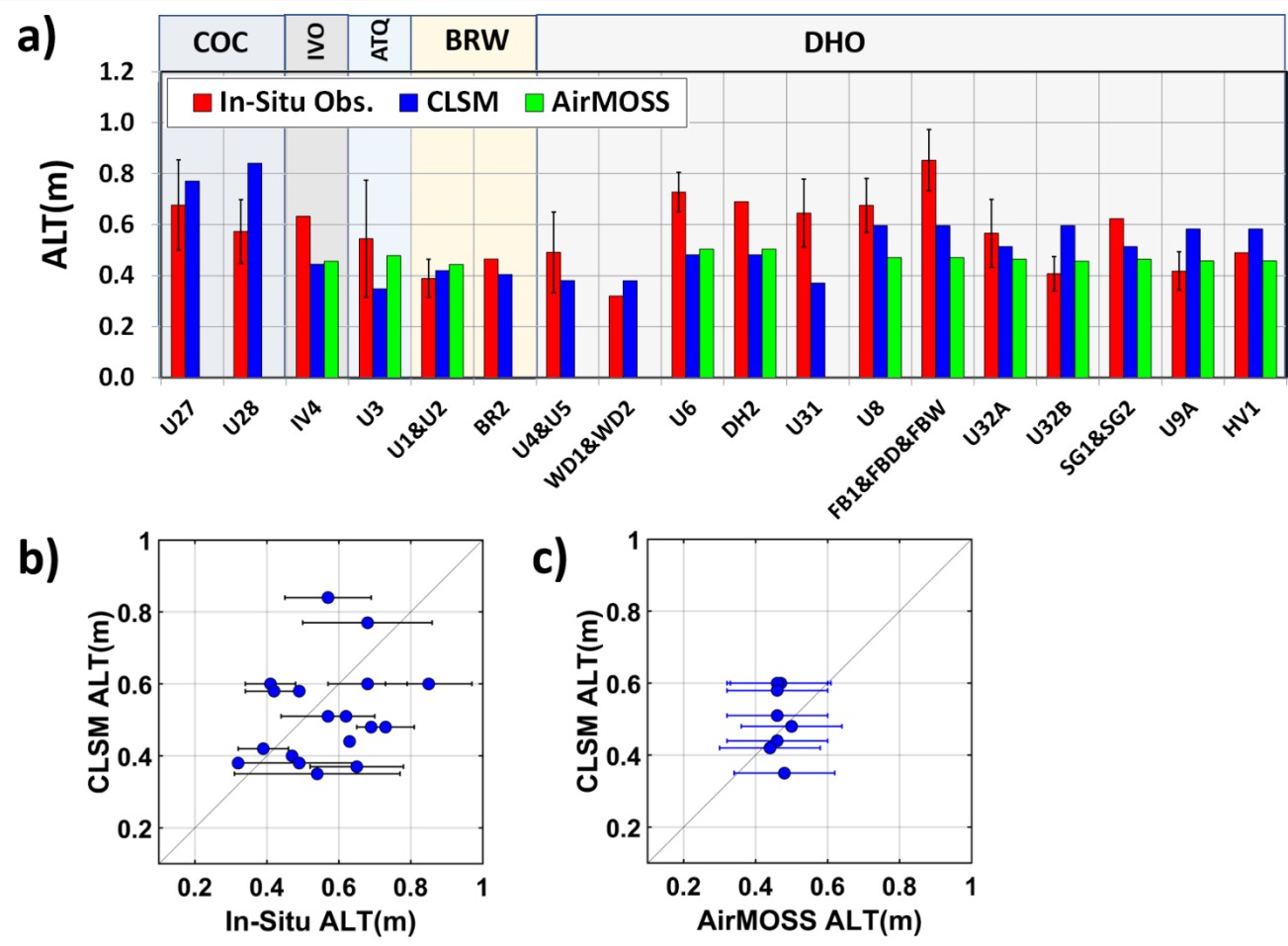

**Figure 4: a)** ALT observations (red) for 2015 from CALM and UAF sites covered by AirMOSS swaths and from radar retrievals aggregated to 81 km$^2$ grid cells (green), and CLSM-simulated ALT at 81 km$^2$ (blue). The short name of the corresponding covering swath is shown on the top (see also Figure 2a). Error bars represent the standard deviation for multiple observations at in-situ sites. No standard deviations are provided for UAF sites since single-point measurements were deployed. Averaged values were provided if multiple sites appear within a same model grid cell (e.g., U1&U2, U4&U5, WD1&WD2, FB1&FBD&FBW, and SG1&SG2). The sites are arranged aligning with the flight direction. **b)** CLSM estimates of ALT for 2015 versus in-situ measurements with error bars indicating the standard deviation as in a). **c)** Same as b) but versus aggregated AirMOSS ALT at model scale. The error bars here represent the uncertainty for radar retrievals at the 81 km$^2$ scale as explained in section 3.1. Corresponding estimates of CLSM uncertainty, which are presumably large, are not shown in the figure.

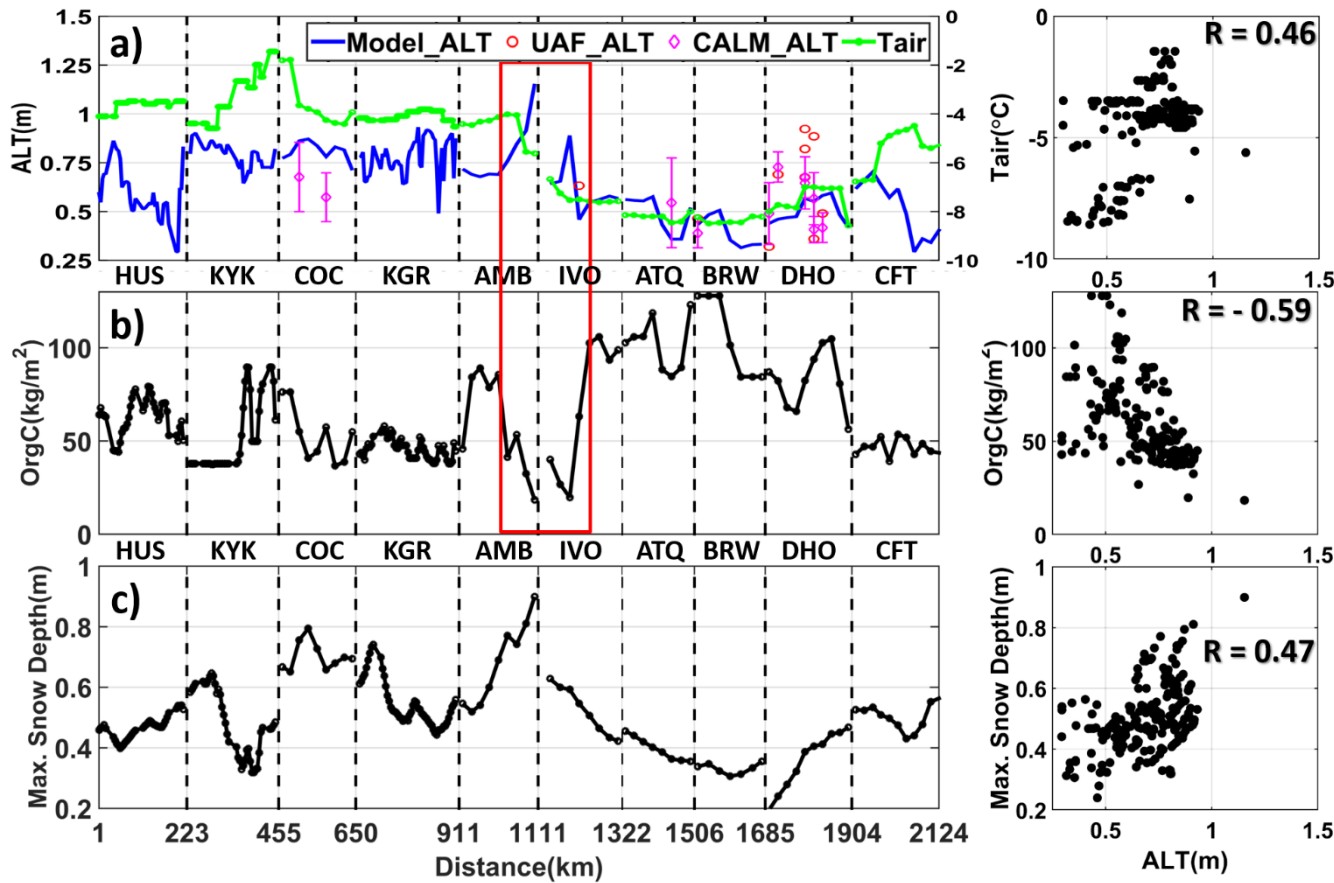

**Figure 5: a) CLSM-simulated ALT (thawed-to-frozen depth) on 29 August 2015 along the AirMOSS flight transects. In-situ ALT observations from UAF and CALM are shown as red circles and magenta diamonds, respectively. Averaged air temperature at 2 m (Tair) from the preceding annual period (i.e., 01 September 2014 to 31 August 2015) is shown in green with the scale on the right ordinate. b) Organic carbon content and c) maximum snow depth during the preceding annual period (again from 01 September 2014 to 31 August 2015). The red rectangle across a) and b) highlights a portion of the domain that shows anti-correlation between organic carbon content and modelled ALT (see Section 4.2). The abscissa in c) provides cumulative distances in units of km along the transects.**

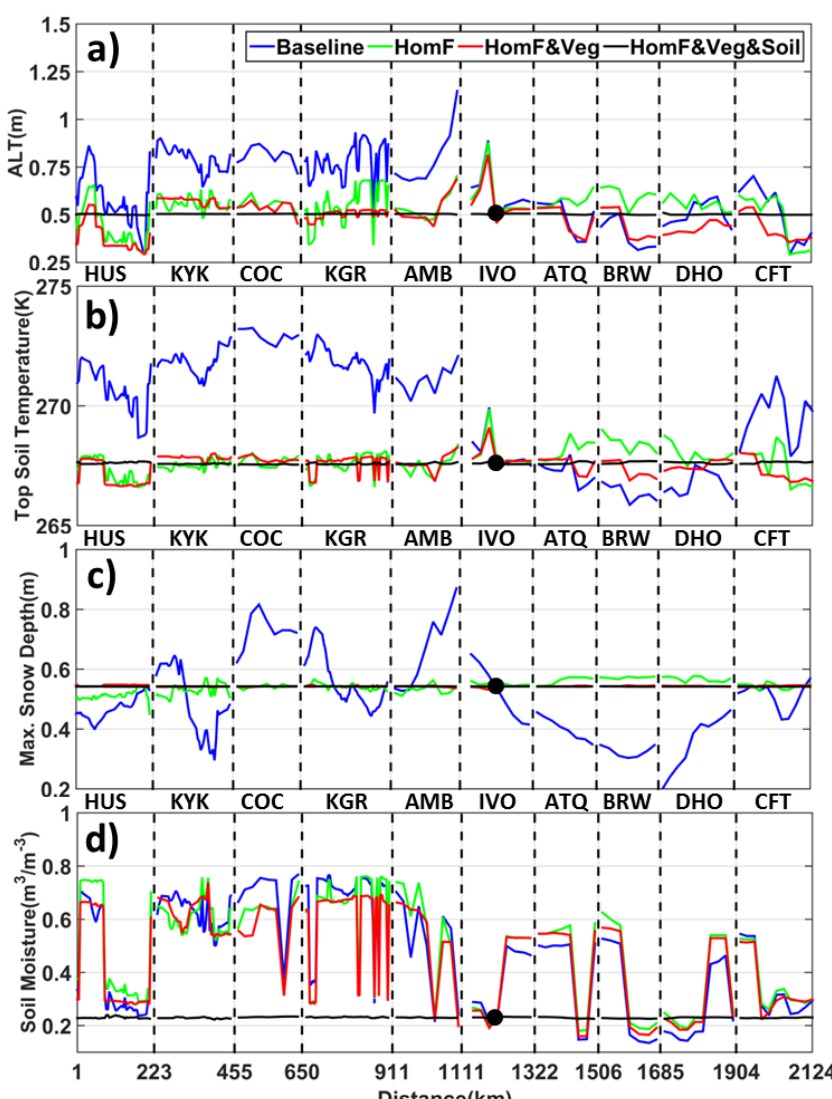

**Figure 6: a) CLSM-simulated ALT (thawed-to-frozen depth) on the flight date (i.e., 29 August 2015) from the top four experiments listed in Table 2; b) simulated top layer soil temperature on the flight date, c) maximum snow depth the during the preceding annual period (i.e., from 01 September 2014 to 31 August 2015), and d) soil moisture within the soil profile on the flight date along the connected transects for the four experiments. The black dot indicates the representative location within the IVO transect from which the forcing, vegetation and/or soil data are used to homogenize the inputs in the idealized experiments. By construction, all simulations provide identical results at this representative location.**

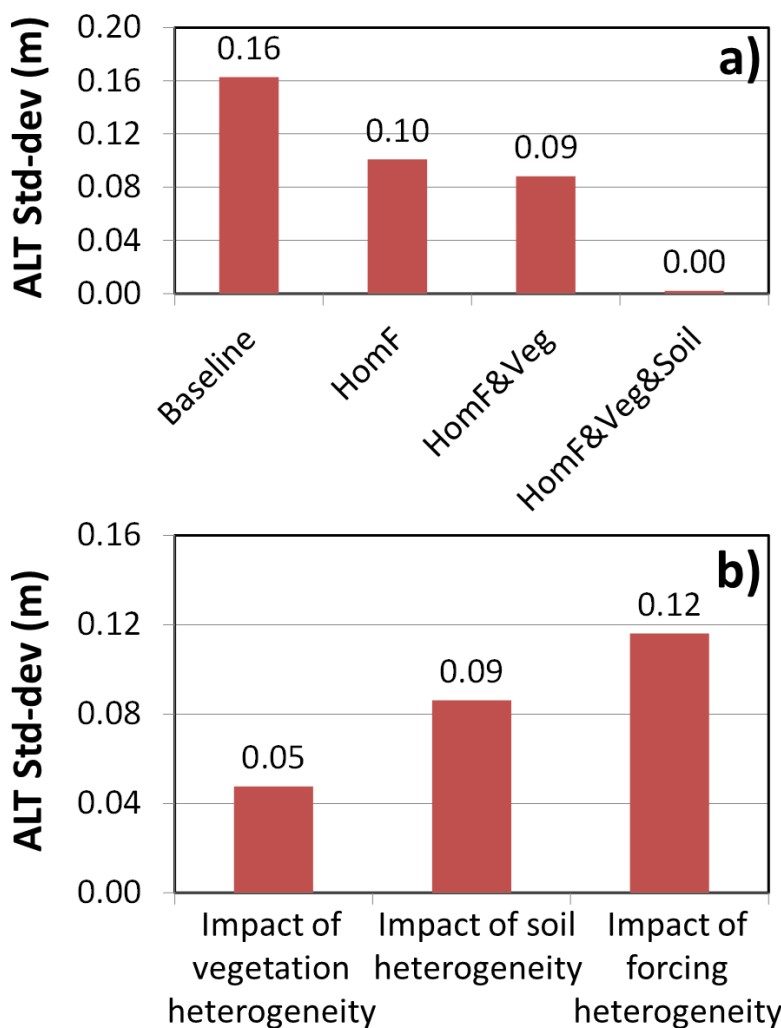

Figure 7: a) Standard deviation of ALT along the AirMOSS transects from the top four experiments listed in Table 2. b) The individual impact (or contribution) from heterogeneous vegetation, soil type and meteorological forcing, respectively. For instance, the impact of vegetation (or soil, or forcing) heterogeneity is the ALT standard deviation along the transects from HomF&Soil (or HomF&Veg, or HomVeg&Soil).

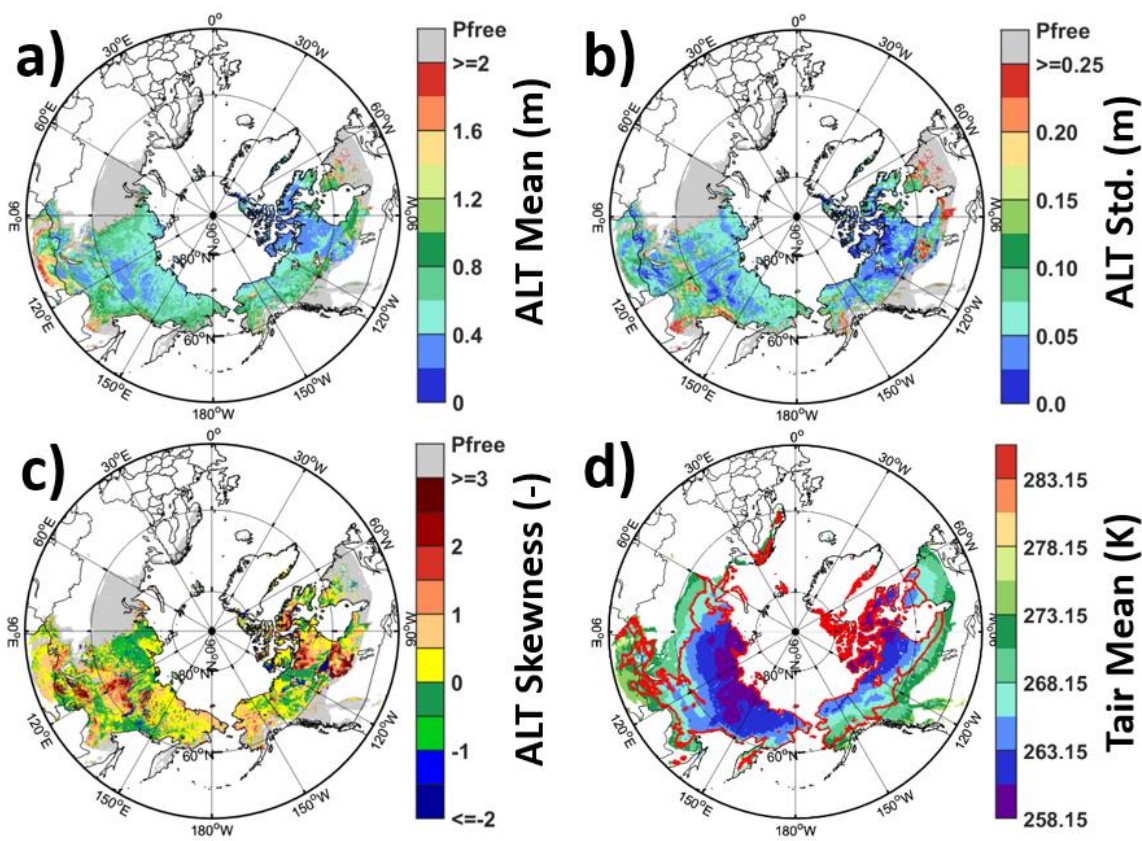

**Figure 8: a) Mean, b) standard deviation, and c) skewness of CLSM-simulated ALT over the 38 years (1980 - 2017). Grey indicates permafrost-free (Pfree) areas in the simulation. d) 38-year averaged MERRA-2 annual atmospheric temperature at 2 m above displacement height (Tair). The red boundary outlines the continuous and discontinuous permafrost regions according to Brown et al. (2002).**

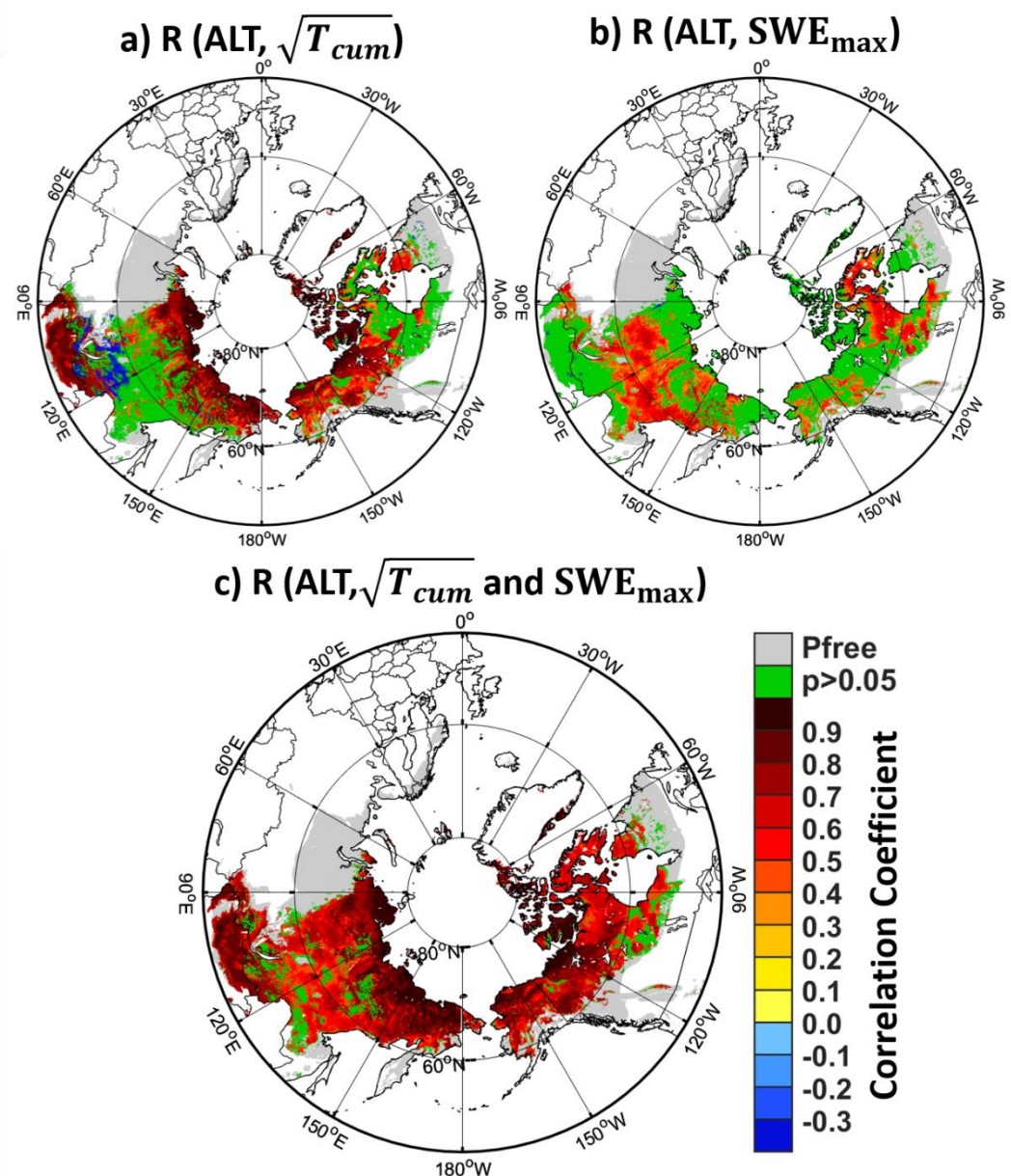

**Figure 9: Correlation coefficient between a) ALT and square root of the effective accumulated air temperature ($\sqrt{T_{cum}}$) and b) ALT and maximum SWE ($SWE_{max}$) from the preceding September to the present August over the period 1980-2017. c) Multi-variable coefficient of correlation for a fitted multiple linear regression model between ALT and $\sqrt{T_{cum}}$ and $SWE_{max}$. Areas that have a p-value larger than 0.05 (i.e., statistically insignificant correlation) are masked in green. Grey indicates permafrost-free (Pfree) areas in the simulation.**

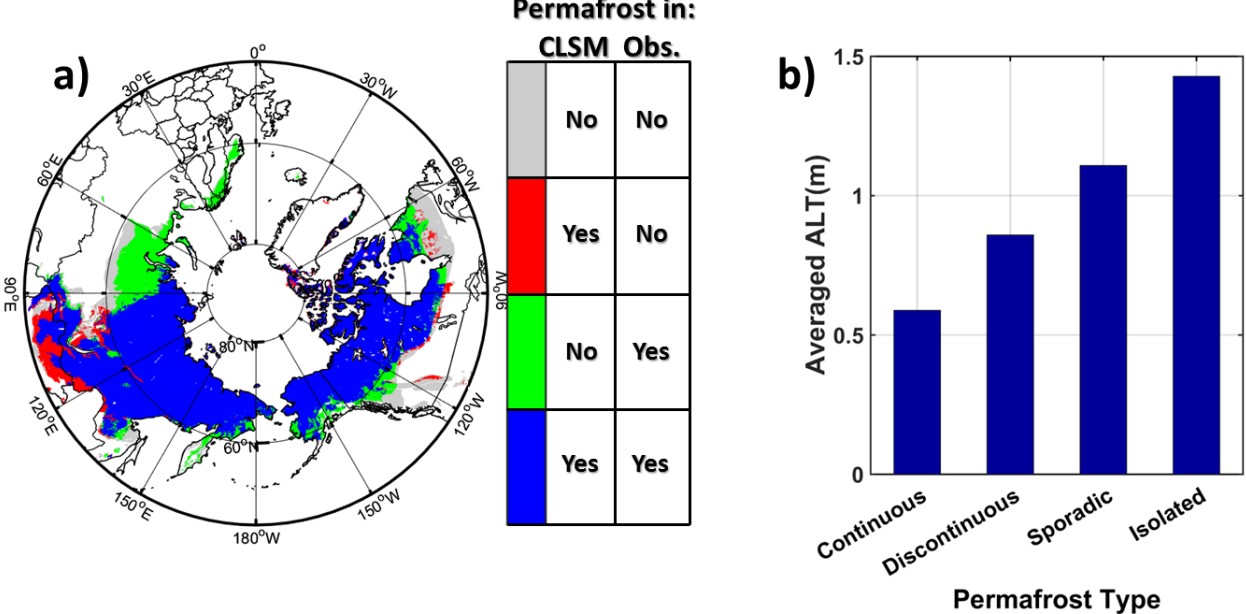

**Figure 10: a) Four comparison categories include: 1) blue - CLSM collocates permafrost with the observation-based permafrost map of Brown et al. (2002) as either continuous, discontinuous, or sporadic permafrost; 2) green - CLSM has no permafrost, but the observation-based permafrost map does as either continuous, discontinuous, or sporadic types; 3) red - CLSM does have permafrost, but the observation-based permafrost map does not or contains isolated permafrost; and 4) grey - CLSM has no permafrost and neither does the observation-based permafrost map (except for isolated permafrost). b) area-weighted average of ALT as simulated by CLSM for the four different permafrost types.**

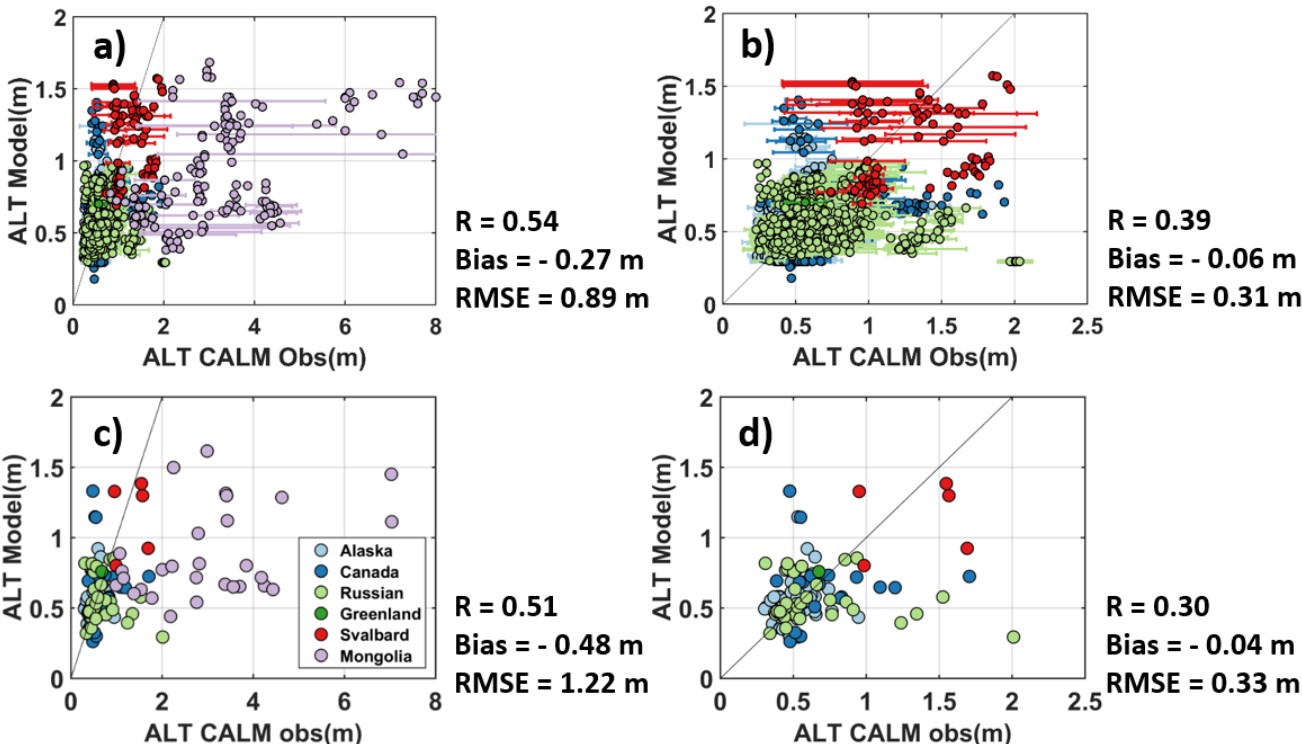

**Figure 11: a) Annual ALT from CLSM simulation vs. CALM observations with horizontal error bars indicating standard deviations of measurements within the model grid cell. Error bar is absent if the number of measurements within a 81 km² grid cell is less than three. b) As in a) but excluding the Mongolia sites. c) 38-yr average ALT for the period 1980-2017 from CLSM simulation vs. CALM observations. d) As in c) but without the Mongolia sites. The correlation coefficient (R), bias, and root mean squared error (RMSE) are provided next to each subplot.**

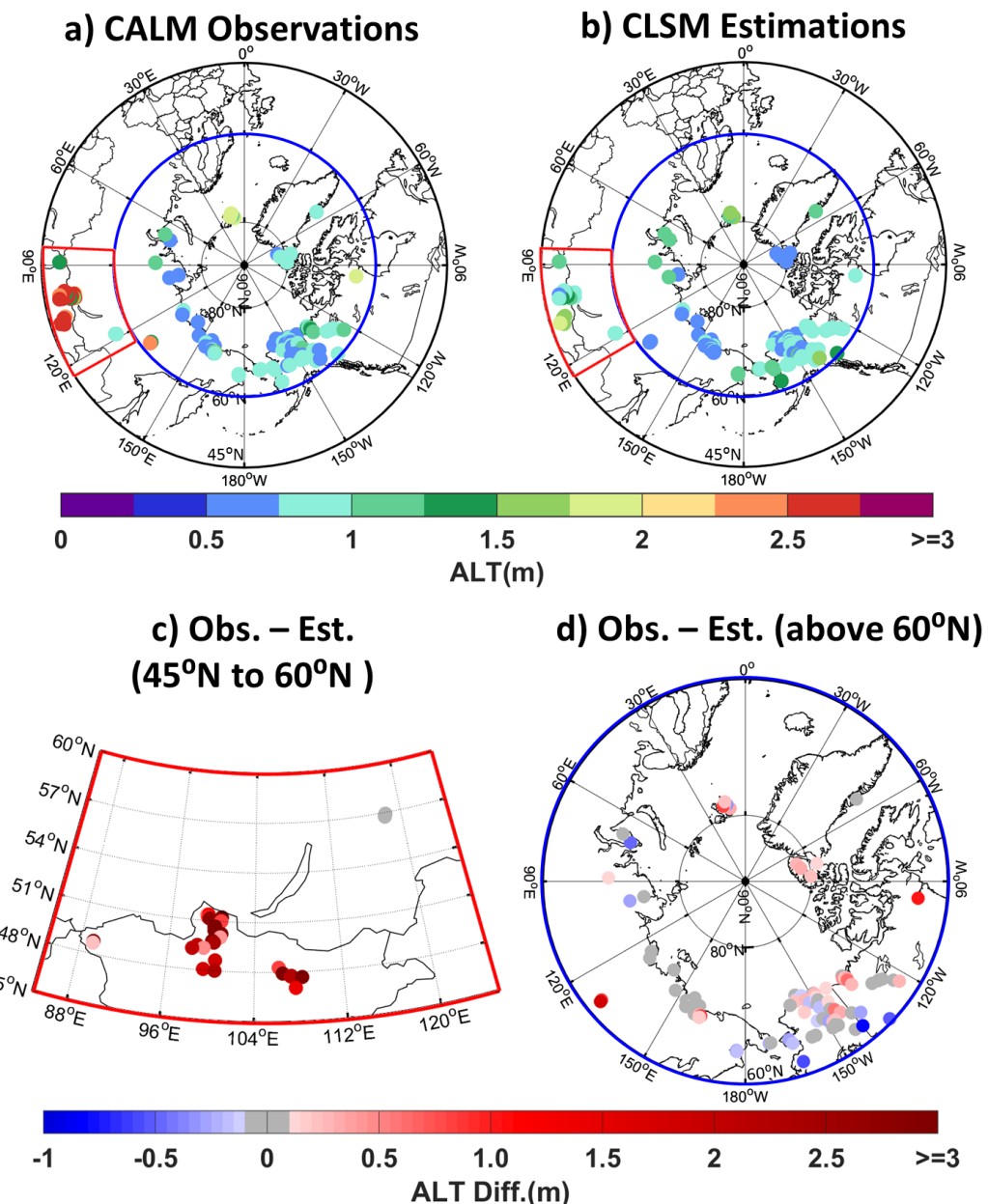

**Figure 12: Multi-year average ALT at CALM site locations for a) CALM observations and b) CLSM results. c) ALT difference between observations and model results for locations within 45ºN- 60ºN latitude and 85ºE-125ºE longitude. d) Same as c) but for locations poleward of 60ºN latitude. In c) and d) grey indicates absolute ALT differences less than 0.10 m.**

# ~~Permafrost~~Permafrost Variability over the Northern Hemisphere Based on the MERRA-2 Reanalysis

Jing Tao[1,a], Randal D. Koster[2], Rolf H. Reichle[2], Barton A. Forman[3], Yuan Xue[3,ab], Richard H. Chen[4], Mahta Moghaddam[4]

[1]Earth System Science Interdisciplinary Center, University of Maryland, College Park, Maryland, USA.

[2]Global Modelling and Assimilation Office, NASA Goddard Space Flight Center, Greenbelt, Maryland, USA.

[3]Department of Civil and Environmental Engineering, University of Maryland, College Park, Maryland, USA.

[4]Department of Electrical Engineering, University of Southern California, Los Angeles, California, USA.

[a]Now at Climate and Ecosystem Sciences Division, Lawrence Berkeley National Laboratory, Berkeley, California, USA; and Department of Civil and Environmental Engineering, University of Washington, Seattle, Washington, USA.

[b]Now at George Mason University, Fairfax, Virginia, USA.

*Correspondence to*: Jing Tao (jingtao@~~umd.edu~~lbl.gov)

**Abstract.** This study introduces and evaluates a comprehensive, model-generated dataset of Northern Hemisphere permafrost conditions at ~~high~~81-km$^2$ resolution. Surface meteorological forcing fields from the Modern-Era Retrospective Analysis for Research and Applications-2 (MERRA-2) reanalysis were used to drive an improved version of the land component of MERRA-2 in middle-to-high northern latitudes from 1980 to 2017. The resulting simulated permafrost distribution across the Northern Hemisphere mostly captures ~~well~~ the observed extent of continuous and discontinuous permafrost ~~except~~but misses the ecosystem-protected permafrost zones in western Siberia~~, which is permafrost free in the simulation.~~. Noticeable discrepancies also appear along the southern edge of the permafrost ~~region~~regions where sporadic and isolated permafrost types dominate. The evaluation of the simulated active layer thickness (ALT) ~~climatology~~against in-situ measurements demonstrates reasonable skill except in Mongolia. ~~Specifically, the~~The RMSE (~~and~~bias) of climatological ALT is 1.22 m (and

(-0.48 m) across all sites and 0.33 m (and (-0.04 m) without the Mongolia sites. In northern Alaska, both ALT retrievals from airborne remote sensing for 2015 and the corresponding simulated ALT exhibit limited skill versus in-situ measurements at the model scale. In addition, the ~~remotely sensed~~ simulated ALT ~~retrievals generally demonstrate lower levels of~~has ~~less~~larger spatial variability than ~~both~~ the ~~observed and~~ remotely sensed ~~simulated~~ ALT, although it agrees well with the retrievals when considering measurements uncertainty. Controls on the spatial variability of ALT are examined with idealized numerical experiments focusing on northern Alaska; meteorological forcing and soil ~~type~~types are found to have dominant impacts on the spatial variability of ALT, with vegetation also playing a role through its modulation of snow accumulation. A correlation analysis further reveals that accumulated above-freezing air temperature and maximum snow water equivalent explain most of the year-to-year variability of ALT nearly everywhere over the model-simulated permafrost regions. ~~Simulated ALT trends from 1980 to 2017 indicate that some permafrost areas are experiencing significant degradation, with ALT increasing up to 0.5 cm/year. It is difficult, however, to adequately assess the accuracy of the simulated ALT trends given the limited availability and relatively short records of in-situ measurements.~~

## 1 Introduction

Permafrost is an important component of the climate system, and its variations can have significant impacts on climate and society. Of deep concern is a potential positive feedback loop by which carbon stored within permafrost regions is released through global warming, thereby adding greenhouse gases to the atmosphere that accelerate the warming further (Dorrepaal et al., 2009; Schuur et al., 2009; MacDougall et al., 2012; Schuur et al., 2015). Communities and infrastructure in ice-rich permafrost regions are particularly vulnerable to land subsidence and infrastructure damage caused by permafrost thaw (Nelson et al., 2001; Liu et al., 2010; Guo and Sun, 2015).

Permafrost variations, including pronounced permafrost degradation due to a warming climate, have been reported for many regions, including Alaska (Nicholas and Hinkel, 1996; Osterkamp and Romanovsky, 1996; Jorgenson et al., 2001; Hinkel and Nelson, 2003; Jafarov et al., 2012; Liu et al., 2012; Jones et al., 2016; Batir et al., 2017), Canada (Chen et al., 2003; James et

al., 2013), Norway (Gisnas et al., 2013), Sweden (Pannetier and Frampton, 2016), Russia (Romanovsky et al., 2007; Romanovsky et al., 2010), Mongolia (Sharkhuu and Sharkhuu, 2012), ~~Norway (Gisnas et al., 2013),~~ and the Qinghai–Tibet Plateau (Zhou et al., 2013; Wang et al., 2016a; Lu et al., 2017; Ran et al., 2018~~, Russia (Romanovsky et al., 2010; Romanovsky et al., 2007) and Sweden (Pannetier and Frampton, 2016)~~. ~~Some of these findings are based on in-situ measurements at a point scale or at a spatially-aggregated scale (up to 1000m×1000m), such as through the Circumpolar Active Layer Monitoring (CALM) network~~(e.g., Luo et al., 2016). For the entire Northern Hemisphere, rapidly accelerated permafrost degradation in recent years has been reported by Luo et al. (2016) based on in-situ measurements at a point-scale or at a spatially-aggregated scale (up to 1000m×1000m) from the Circumpolar Active Layer Monitoring (CALM) network~~and in the Arctic Report Card~~. However, the current state and evolution of global permafrost (including permafrost temperature, ice content, and degradation rates) are still largely unknown across much of the Northern latitudes.

The impact of a changing climate on permafrost dynamics must depend on local site characteristics. Subsurface heat transfer processes and active layer thickness (ALT; the maximum thaw depth at the end of the thawing season) are influenced by more than surface meteorological forcing – they are also influenced by vegetation type, surface organic layer characteristics, soil properties and soil moisture (Stieglitz et al., 2003; Shur and Jorgenson, 2007; Yi et al., 2007; Luetschg et al., 2008; Dankers et al., 2011; Johnson et al., 2013; Jean and Payette, 2014; Yi et al., 2015; Fisher et al., 2016; Matyshak et al., 2017; Tao et al., 2017). Understanding the contributions from the different controls on ALT (and permafrost conditions in general) is crucial for assessing permafrost behaviour and its resilience to a warming climate.

Physically-based numerical model simulations are potentially useful for quantifying and understanding these dynamics at large spatial scales; they can also provide insights into associated impacts on the global carbon cycle. Permafrost dynamics can be modelled, for example, by driving a land surface model (LSM) offline (i.e., uncoupled from an atmospheric model) with meteorological forcing data (including air temperature, radiation, precipitation, etc.) from some credible source. ~~Permafrost variations, including pronounced permafrost degradation due to a warming climate, have been reported for many regions, including Alaska~~ (Jorgenson et al., 2001; Liu et al., 2012; Nicholas and Hinkel, 1996; Batir et al., 2017; Osterkamp and

Romanovsky, 1996; Hinkel and Nelson, 2003; Jafarov et al., 2012; Jones et al., 2016), Canada (Chen et al., 2003; James et al., 2013), Mongolia (Sharkhuu and Sharkhuu, 2012), Norway (Gisnas et al., 2013), the Qinghai–Tibet Plateau (Zhou et al., 2013; Lu et al., 2017; Wang et al., 2016a), Russia (Romanovsky et al., 2010; Romanovsky et al., 2007) and Sweden (Pannetier and Frampton, 2016). Some of these findings are based on in-situ measurements at a point-scale or at a spatially-aggregated scale

(up to 1000m×1000m), such as through the Circumpolar Active Layer Monitoring (CALM) network. In particular, rapidly accelerated permafrost degradation in recent years has already been reported at CALM in situ sites over the Northern Hemisphere (Luo et al., 2016). In addition, given the apparent climate warming seen in recent years (exemplified by the fact that the average Arctic air temperature in 2017 (ending in September) was the second warmest on record since 1900 (Arctic Report Card; http://www.arctic.noaa.gov/Report-Card/Report-Card-2017) and that 2017 was the warmest year on record for

global ocean temperatures (Cheng and Zhu, 2018)), important reductions in permafrost might be occurring as well. However, current global permafrost thermal states (i.e., permafrost temperature, ice content and degradation rates across much of Northern latitudes) are still largely unknown. Monitoring permafrost degradation in a timely manner is particularly critical for ecosystem management and for various policy decisions.

For large spatial scales, numerical model simulations are potentially useful. Simulations and/or predictions with a variety of land surface models (LSMs) have been used to quantify large-scale permafrost patterns (i.e., distributions and thermal states) and their interactions with a warming climate. LSMs utilized for this include, for example, the Joint UK Land Environment Simulator (JULES, Dankers et al., 2011), the ORganizing Carbon and Hydrology in Dynamic EcosystEms (ORCHIDEE) - aMeliorated Interactions between Carbon and Temperature (ORCHIDEE MICT, Guimberteau et al., 2018), the Catchment

Land Surface Model (CLSM, Tao et al., 2017), and the Community Land Model (Lawrence and Slater, 2005; Alexeev et al., 2007; Nicolsky et al., 2007a; Yi et al., 2007; Lawrence and Slater, 2008; Lawrence et al., 2008; Lawrence et al., 2012; Koven et al., 2013; Chadburn et al., 2017; Guo and Wang, 2017). Most of these land models were run at coarse spatial resolutions, e.g., ranging from $0.5° × 0.5°$ to $1.8° × 3.6°$ for LSMs participating in the Permafrost Carbon Network (PCN) (Wang et al., 2016a) and from $0.188° × 0.188°$ to $4.10° × 5°$ for the models participating in the Coupled Model Intercomparison Project

phase 5 (CMIP5) (Koven et al., 2013; https://portal.enes.org/data/enes-model-data/cmip5/resolution). As a result, it is difficult

to compare the simulated values with in-situ observations taken at the point scale. Other types of numerical models have been run at relatively higher resolution, but not globally; such simulation domains were limited to regional scales (e.g., 2 km × 2 km in Jafarov et al. (2012) covering Alaska;1 km × 1 km in Gisnas et al. (2013) covering Norway) as necessitated by the availability of ancillary data and the heavy computational burden. As discussed further below, one of the unique contributions

of the present work is a global simulation of permafrost at a somewhat higher resolution than earlier global-scale studies.

The impact of a changing climate on permafrost dynamics must depend on local site characteristics. Subsurface heat transfer processes and active layer thickness (ALT; the maximum thaw depth at the end of the thawing season) are influenced by more than surface meteorological forcing – they are also influenced by vegetation type, surface organic layer characteristics, soil

properties and soil moisture (Yi et al., 2007; Fisher et al., 2016; Shur and Jorgenson, 2007; Tao et al., 2017; Johnson et al., 2013; Jean and Payette, 2014; Yi et al., 2015; Stieglitz et al., 2003; Luetschg et al., 2008; Matyshak et al., 2017; Dankers et al., 2011). Understanding the contributions from the different controls on ALT (and permafrost conditions in general) is crucial for assessing permafrost behaviour and its resilience to a warming climate.

Again, such understanding can potentially be derived from models. Permafrost dynamics can be modelled, for example, by driving a land surface model offline (i.e., uncoupled from an atmospheric model) with meteorological forcing data (including air temperature, radiation, precipitation, etc.) from some credible source. LSMs that have been used to quantify large-scale permafrost patterns (i.e., distributions and thermal states) and their interactions with a warming climate include, for example, the Joint UK Land Environment Simulator (JULES, Dankers et al., 2011), the ORganizing Carbon and Hydrology in Dynamic

EcosystEms (ORCHIDEE) - aMeliorated Interactions between Carbon and Temperature (ORCHIDEE-MICT, Guimberteau et al., 2018), the Catchment Land Surface Model (CLSM, Tao et al., 2017), and the Community Land Model (Alexeev et al., 2007; Nicolsky et al., 2007a; Yi et al., 2007; Lawrence and Slater, 2008; Lawrence et al., 2008; Lawrence et al., 2012; Koven et al., 2013; Chadburn et al., 2017; Guo and Wang, 2017). Most of these land models were run at coarse spatial resolutions, e.g., ranging from 0.5° × 0.5° to 1.8° × 3.6° for LSMs participating in the Permafrost Carbon Network (PCN) During the

course of the simulation, the model produces estimates of ALT and permafrost thermal characteristics. A wide range of

~~simulated permafrost behaviour has been reported in the literature, with differences reflecting model-specific process representations and~~(Wang et al., 2016a) and from 0.188° × 0.188° to 4.10° × 5° for the models participating in the Coupled Model Intercomparison Project phase 5 (CMIP5) (Koven et al., 2013; https://portal.enes.org/data/enes-model-data/cmip5/resolution).

Differences in the permafrost behaviour simulated with these models reflect model-specific process representations as well as biases associated with different meteorological forcing datasets (Barman and Jain, 2016; Wang et al., 2016a; Wang et al., 2016b; Guo et al., 2017; Guimberteau et al., 2018) ~~(Barman and Jain, 2016; Slater and Lawrence, 2013; Guimberteau et al., 2018; Guo et al., 2017; Wang et al., 2016a; Wang et al., 2016b). The latter source of bias is particularly~~. Such forcing biases

are difficult to ~~reconcile~~avoid given ~~that~~the sparsity of direct observations of meteorological variables in most parts of the high latitudes ~~are sparse. In addition, reanalysis datasets that~~. Even reanalyses, which assimilate a variety of global observations ~~provide global coverage but still~~, inevitably have biases in high latitudes due to ~~this~~ observation sparsity in cold regions combined with the many challenges of physical process modelling.

~~Despite~~ Nevertheless, despite these issues, permafrost behaviour simulated with LSMs driven offline by reanalysis forcing fields can still be useful for understanding the impacts of climate variability on permafrost~~, and, in turn, to evaluate the performance of the reanalysis data. (Consider the example of the LSMs participating in the Permafrost Carbon Network.) The present paper utilizes this approach in the context of the Modern-Era Retrospective Analysis for Research and Applications-2 (MERRA-2), an atmospheric reanalysis system that assimilates a wide range of conventional and satellite observations (Gelaro

et al., 2017). We~~. The present paper utilizes this approach. Specifically, we generate here a dataset of Northern Hemisphere permafrost conditions by driving an updated version of NASA's Catchment Land Surface Model (CLSM) with ~~MERRA-2~~Modern-Era Retrospective Analysis for Research and Applications-2 (MERRA-2; Gelaro et al., 2017) ~~(MERRA-2; Gelaro et al., 2017)~~ surface meteorological forcing fields for the middle-to-high latitudes across the Northern Hemisphere over the period 1980-2017. ~~Note that MERRA-2 has been found to be skilful in its simulation of near-surface atmospheric conditions~~

(Reichle et al., 2017a; Reichle et al., 2017b; Bosilovich et al., 2015; Bosilovich et al., 2017) ~~and to show improvements in the~~

representation of cryospheric processes compared with its predecessor MERRA (Gelaro et al., 2017). In particular, MERRA-2 assimilates substantially more satellite observations and employs more physically reasonable hydrology representations for glaciated land surfaces compared to MERRA, and it also uses observation-based, seasonally-varying sea ice albedo as opposed to MERRA's fixed value of 0.6 (Gelaro et al., 2017). A recent study shows that permafrost and ALT simulation results obtained with forcing data from the original MERRA reanalysis are inferior to those driven by other reanalysis-based forcing data sets, particularly those from the NOAA Climate Forecast System Reanalysis (CFSR) and European Centre for Medium Range Weather Forecasts Re-Analysis Interim (ERA-I) (Guo et al., 2017). The superiority of MERRA-2 forcing compared to MERRA forcing in the context of permafrost simulation is presumed here (given its general improvements in the cryosphere), though a side-by-side test of the two forcing datasets in this regard has not been performed. We perform the simulations at 81 km$^2$ resolution encompassing all permafrost areas in the middle-to-high latitudes of the Northern Hemisphere. This resolution is high relative to most existing modelling studies at the global scale; published simulations at higher resolution are limited to plot scales (e.g., CALM-site scale in Shiklomanov et al. (2010)Shiklomanov et al. (2010)), landscape scales (e.g., polygonal tundra landscape scale in Kumar et al. (2016)Kumar et al. (2016)), or regional scales (e.g., 4 km$^2$ in Jafarov et al. (2012)Jafarov et al. (2012) covering Alaska; 1 km$^2$ in Gisnas et al. (2013)Gisnas et al. (2013) covering Norway).

Detailed observations are another obvious source of understanding, and here, to complement our modelling analysis, we also make use of remote sensing information from the NASA Airborne Microwave Observatory of Subcanopy and Subsurface (AirMOSS) mission. In 2015, AirMOSS acquired P-band (420-440 MHz) radar observations over portions of northern Alaska from which Chen et al. (2019a) retrieved regional estimates of ALT and soil layer dielectric properties that are related to soil moisture and freeze/thaw states. In their study, Chen et al. (2019a) mainly focus on the development and improvement of the ALT retrieval algorithm, whereas the present study emphasizes using the ALT retrievals to assess the (fully independent) ALT simulations.

Due to the sparsity of in-situin-situ measurements at the regional to global scale, evaluating the spatial pattern of ALT produced by any such simulation remains challenging. Indeed, it is difficult to compare the simulated values at model resolutions with

in situin-situ observations taken at the point scale unless the measurement point is uniformly representative of the area covered by the model grid cell or the upscaling (representation) errors associated with the point-to-grid comparison are well defined. Remotely sensed permafrost products, which provide a unique source of spatially distributed ALT at the landscape-scale, may provide help in this regard. Existing remote sensing ALT products have been retrieved from ground-based Ground Penetrating

Radar (GPR) (Chen et al., 2016a; Jafarov et al., 2017), airborne polarimetric Synthetic Aperture Radar (SAR) (Chen et al., 2019a), and spaceborne interferometric SAR (Liu et al., 2012; Li et al., 2015; Schaefer et al., 2015). These ALT products are available at the landscape-scale and can complement our modelling analysis. In this study, we use remote sensing information from the NASA Airborne Microwave Observatory of Subcanopy and Subsurface (AirMOSS) mission.  In 2015, AirMOSS acquired P-band (420-440 MHz) SAR observations over portions of northern Alaska from which Chen et al. (2019a) (2019)

retrieved regional estimates of ALT and soil layer dielectric properties that are related to soil moisture and freeze/thaw states. In their study, Chen et al. (2019b)Chen et al. (2019a) mainly focus on the development and improvement of the ALT retrieval algorithm, whereas the present study uses the ALT retrievals in combination with in situin-situ measurements to aid in assessing the (fully independent) ALT simulations.

In the present paper we evaluate our simulated permafrost extent ALT and ALTs permafrost extent against an observations-based permafrost distribution map, and against multi-year in situin-situ observations from CALM, and against . We also compare the skill of our model estimates to that of the AirMOSS ALT retrievals derived from AirMOSS..  In these comparisons, we account for uncertainty to the extent possible.  Overall, we pursue three scientific objectives: 1) evaluate the relative importance of the factors that determine the spatial variability of ALT, 2) evaluate CLSM-simulated ALT climatology

and permafrost extent against observations, and 3) quantify and assess the large-scale characteristics of ALT (in terms of means, and interannual variability and trend) in Northern Hemisphere permafrost regions from 1980 through 2017.  As a side benefit, the side-by-side comparison of modelled and remotely sensed ALT estimates is an important first step toward combining this information effectively in future model-data fusion efforts.  Section 2 below describes the model and datasets used in this study, Section 3 describes methods, and Section 4 provides results.  Our findings are summarized and discussed

in Section 5.

**Formatted:** Line spacing: Double

**2 Model and data sets**

**2.1 NASA Catchment Land Surface Model (CLSM)**

CLSM is the land model component of NASA's Goddard Earth Observing System (GEOS) Earth system model and was part of the model configuration underlying the MERRA-2 reanalysis product (Reichle et al., 2017a; Gelaro et al., 2017). CLSM
explicitly accounts for sub-grid heterogeneity in soil moisture characteristics with a statistical approach (Koster et al., 2000; Ducharne et al., 2000). The land fraction within each computational unit (or grid cell) is partitioned into three soil moisture regimes, namely the wilting (i.e., non-transpiring), unsaturated, and saturated area fractions. Over each of the three moisture regimes, a distinct parameterization is applied to estimate the relevant physical processes (e.g., runoff and evapotranspiration). ~~CLSM also includes~~ This version of CLSM ~~does not include dynamic soil carbon pools, but it does~~ includes a three-layer snow
model that estimates the evolution of snow water equivalent (SWE), snow depth, and snow heat content ~~(Stieglitz et al., 2001)~~ ~~in response to the forcing data.~~(Stieglitz et al., 2001) in response to the forcing data. The snow model accounts for key physical mechanisms that contribute to the growth and ablation of the snowpack, including snow accumulation, aging, melting, and refreezing. The model also includes the insulation of the ground from the atmosphere by the snowpack. The CLSM subsurface heat transfer module uses an explicit finite difference scheme to solve the heat diffusion equation for six soil layers (0-0.1m,
0.1-0.3m, 0.3-0.7m, 0.7-1.4m, 1.4-3m, and 3-13m). ~~A no-heat-flux condition is employed at the bottom of the model's soil column~~The soil layer thicknesses increase with depth following a geometric series for consistency with the linear heat diffusion calculation (Koster et al., 2000). A no-heat-flux condition is employed at 13m depth.

The updated version of CLSM used here ~~(Tao et al., 2017)~~ includes modifications aimed at improving permafrost simulation.
It accounts, for example, for the impact of soil carbon on the soil thermal properties with soil porosity, thermal conductivity, and specific heat capacity calculated separately for mineral soil and soil carbon, after which the two are averaged using a carbon-weighting scheme. Higher (lower) soil carbon content, therefore, results in lower (higher) soil thermal conductivity. The updated version produces more realistic subsurface thermodynamics in cold regions than does the original scheme ~~(Tao et al., 2017)~~.(Tao et al., 2017). This version of CLSM, however, does not include dynamic soil carbon pools.

Particularly relevant to the present analysis is our calculation of ALT from CLSM simulation output. We compute ALT from the simulated soil temperature profile and the ice content within the soil layer that contains the thawed-to-frozen transition. Precisely, the thawed-to-frozen depth is calculated as:

$$z_{bottom}(l) - f_{ice}(l, t) \times \Delta z(l), \tag{1}$$

where layer $l$ is the deepest layer that is fully or partially thawed, $z_{bottom}(l)$ represents the depth at the bottom of layer $l$, $f_{ice}(l, t)$ is the fraction of ice in layer $l$ at time t (i.e., $f_{ice}(l, t) \in [0\ 1]$), and $\Delta z(l)$ is the thickness of layer $l$. To identify layer $l$, we use a 0°C degree temperature threshold. Specifically, T > 0°C degree indicates that a layer is fully thawed, T = 0°C degree indicates that a layer is partially thawed, and T < 0°C degree indicates that a layer is fully frozen. That is, layer $l$ is the deepest layer that satisfies $T(l) \geq$ 0°C. Equation (1) then expresses that the thawed-to-frozen depth is equal to the bottom depth of the layer $l$ but adjusted upward according to the ice fraction within the partially thawed layer $l$. ~~The annual ALT for a given year, then, is defined as the maximum thawed-to-frozen depth within that year.~~ This upward adjustment, by the way, allows the thawed-to-frozen depth to be a continuous variable; it is not quantized to the imposed layer depths. We search for the deepest $l$ if multiple thawed-to-frozen transitions are present (e.g., if a seasonal frost at the surface is separated from the permafrost below by a thawed soil layer). The annual ALT for a given year, then, is defined as the deepest depth at which a thawed-to-frozen transition occurs within that year. Note that the calculation of equation (1) is made at the scale of a model grid cell, and thus features such as talik are not represented if they occur at sub-grid cell scale.

~~We drive the improved CLSM version of Tao et al. (2017) in a land-only (offline) configuration across permafrost areas in the Northern Hemisphere. The simulation domain, shown in~~

We drive the improved CLSM version of Tao et al. (2017) in a land-only (offline) configuration across permafrost areas in the Northern Hemisphere. The simulation domain, shown in Figure 1~~Figure 1~~a, covers the major permafrost regions of the Northern Hemisphere middle-to-high latitudes for which soil carbon data are available from the Northern Circumpolar Soil Carbon Database version 2 (NCSCDv2, https://bolin.su.se/data/ncscd/) (Hugelius et al., 2013a; Hugelius et al., 2013b)~~. The~~

NCSCDv2 data are used to calculate the CLSM soil thermal properties used in the simulations (Tao et al., 2017). The model simulation covered the period from 1980 to 2017 and was performed at a 81-km² spatial resolution on the 9-km Equal Area Scalable Earth grid, version 2 (Brodzik et al., 2012).

Surface meteorological forcing were extracted from the MERRA-2 reanalysis data, which are provided at a resolution of 0.5° latitude × 0.625° longitude (Global Modeling and Assimilation Office (GMAO), 2015a, b). At latitudes south of 62.5°N within our simulation domain, the MERRA-2 precipitation forcing used here is informed by gauge measurements from the daily 0.5° global Climate Prediction Center Unified gauge product (Chen et al., 2008) as described in (Reichle et al., 2017b). We further rescaled the precipitation to the long-term, seasonally varying climatology of the Global Precipitation Climatology Project
version 2.2 product (Huffman et al., 2009). Further details regarding model parameters and forcing inputs are found in Tao et al. (2017).

The model was spun-up for 180 years by looping five successive times through the 36-year period of MERRA-2 forcing from 1 January 1980 to 1 January 2016 in order to achieve a quasi-equilibrium state. The spatial terrestrial state variables at the end
of the fifth loop were used to initialize the model for the final simulation experiment from 1980 to 2017. The details of the spin-up procedure employed here admittedly impact our trend analysis (section 4.5); the approach makes use of the warmer conditions during the last few decades and thus should produce warmer 1980 initial conditions than would be produced with realistic historical forcing over hundreds of years (e.g., Sapriza-Azuri et al., 2018). The NCSCDv2 data are used to calculate the CLSM soil thermal properties used in the simulations (Tao et al., 2017). The model simulation covered the period from
1980 to 2017 and was performed at a 81-km² spatial resolution on the 9-km Equal Area Scalable Earth grid, version 2 (Brodzik et al., 2012).

Surface meteorological forcing were extracted from the MERRA-2 reanalysis data, which are provided at a resolution of 0.5° latitude × 0.625° longitude (Global Modeling and Assimilation Office (GMAO), 2015a, b). At latitudes south of 62.5°N within
our simulation domain, the MERRA-2 precipitation forcing used here is informed by gauge measurements from the daily 0.5°

global Climate Prediction Center Unified gauge product (Chen et al., 2008) as described in (Reichle et al., 2017b). We further rescaled the precipitation to the long-term, seasonally varying climatology of the Global Precipitation Climatology Project version 2.2 product (Huffman et al., 2009). Further details regarding model parameters and forcing inputs are found in Tao et al. (2017).

The model was spun-up for 180 years by looping five successive times through the 36-year period of MERRA-2 forcing from 1 January 1980 to 1 January 2016 in order to achieve a quasi-equilibrium state. The spatial terrestrial state variables at the end of the fifth loop were used to initialize the model for the final simulation experiment from 1980 to 2017.

## 2.2 Remotely Sensed ALT from AirMOSS

Radar backscatter measurements are sensitive to changes in the soil dielectric constant (or relative permittivity) which in turn are associated with changes in soil moisture and the soil freeze-thaw state. Based on this relationship, Chen et al. (2019)(2019a) used the AirMOSS airborne P-band (420-440 MHz) synthetic aperture radar (SAR) observations collected during two campaigns in 2015 to estimate ALT in northern Alaska. As shown in Figure 2Figure 2a, the AirMOSS flights originated from Fairbanks International Airport and headed west toward the Seward Peninsula (HUS, KYK, COC), then turned back east (KGR) prior to heading north towards the Arctic coast overpassing Ambler (AMB), Ivotuk (IVO), and Atqasuk (ATQ). From there, the flights turned south again, flying over Barrow (BRW), Deadhorse (DHO), and Coldfoot (CFT) en route to Fairbanks. In the present paper, the remotely-sensed ALT retrievals are compared with in situin-situ observations and CLSM-simulated ALT.

Chen et al. (2019b)Chen et al. (2019a) used AirMOSS P-band SAR observations at two different times to retrieve active layer properties: (1) acquisitions on 29 August 2015 when the downward thawing process approximately reached its deepest depth (i.e., the bottom of the active layer), and (2) acquisitions on 1 October 2015 when the active layer started to refreeze from the surface while the bottom of the active layer remained thawed. ALT was assumed constant from late August to early October

because over this period changes in thawing depth are found typically negligible (Carey and Woo, 2005; Chen et al., 2016b; Zona et al., 2016).  Strictly speaking, the radar retrievals represent the approximate thaw depth of the thawed-to-frozen boundary on 29 August 2015 and 1 October 2015.  The unknown, true ALT for 2015 might occur later if the thawing continued and the maximum thaw depth occurred after the October flight time. Based on an analysis of in situ~~in-situ~~ observations (not shown), however, it is rare that this occurs, and the subsequent impact on the estimated ALT value would be relatively small in any case. We therefore equate the retrieved thaw depth with ALT.

In the retrieval algorithm, (Chen et al., 2019a) used a three-layer dielectric structure to represent the active layer and underlying permafrost. In their algorithm, the two uppermost layers together constitute the active layer that account for a top, unsaturated zone and an underlying, saturated zone. The bottommost (third) layer of the retrieval model structure represents the permafrost. Because the soil moisture at saturation only depends on the porosity of the soil medium, the dielectric constant of the saturated zone in the active layer is assumed constant over the time window. An iterative forward-model inversion scheme was used to simultaneously retrieve the dielectric constants and layer thicknesses of the three-layer dielectric structure from the SAR observations collected on 29 August 2015 and 1 October 2015. Note that the retrieved ALT cannot exceed the radar sensing depth of about 60 cm. This is the depth below which the AirMOSS radar is expected to lose sensitivity to subsurface features, and it is calculated based on the radar system noise floor and calibration accuracy. Therefore, any retrieved ALT larger than 60 cm is expected to have large uncertainties, and the error is further expected to grow linearly as the retrieved values of ALT essentially "saturate."  This limitation may also lead to underestimates of the actual thaw depth.

In this study, we focus on the retrievals of four flight lines across the Alaska North Slope, including IVO (Ivotuk), ATQ (Atqasuk), BRW (Barrow), and DHO (Deadhorse) as shown in Figure 2~~Figure 2~~a. These four transects cover areas with light to moderate vegetation. Since the radar scattering model is only applicable to bare surfaces or lightly vegetated tundra areas (Chen et al., 2019b), the ALT estimates derived for IVO, ATQ, BRW, and DHO are considered more accurate than ALT retrievals for the remaining transects, which include more vegetated areas. Moreover, some of the southern transects cover

**Formatted:** Font: (Asian) Times New Roman

discontinuous permafrost where the ALT often exceeds the P-band radar sensing depth of about 60 cm and thus ~~cannot be retrieved from AirMOSS observations~~the retrievals have large uncertainty (Chen et al., 2019b). ~~here.~~

**2.3 Circum-Arctic Permafrost Conditions and ~~In-situ~~In-situ Observations of ALT**

The permafrost distribution simulated by CLSM is evaluated against the observations-based Circum-Arctic Map of Permafrost and Ground-Ice Conditions (Brown et al., 2002) shown in Figure 1~~Figure 1~~b. The map is based on the distribution and character of permafrost and ground ice using a physiographic approach. –Permafrost conditions are categorized into four classes: continuous (90-100%), discontinuous (50-90%), sporadic (10-50%), and isolated (0-10~~%) permafrost,~~%), where the numbers in parentheses indicate the area fraction of permafrost extent.

~~In-situ~~In-situ observations of ALT obtained by the CALM network (https://www2.gwu.edu/~calm/; Brown et al., 2000) were used to evaluate both the AirMOSS ALT retrievals and CLSM-simulated ALT results. The CALM network provides observations from 1990 to 2017, but few sites have records in the early 1990s. We did not use measurements that were flagged as having been taken too early in the season or under unusual conditions (e.g., after the site was burned or covered with lava~~,~~.~~.~~ which occurred at sites R30A and R30B in Kamchatka). In total there are 220 sites located within the CLSM simulation domain (Figure 1~~Figure 1~~b), and we use 213 sites to evaluate results. Thaw depth measurements are usually made at the end of the thawing season. Most of the CALM sites (129 out of the 213 sites used here) employ a spatially-distributed mechanical probing method to measure thaw depths along a transect or across a rectangular grid ranging in size from 10m×10m to 1000m×1000m. At 20 sites thaw tubes or boreholes are used to measure the thaw depth. At 63 sites, ground temperature measurements from boreholes are used to infer thaw depth. For the remaining site, no information about the measurement method is available. Only point-scale measurements are available from the thaw tube/borehole and ground temperature sites (including, e.g., the sites in Mongolia).

In addition, daily ~~in-situ~~in-situ observations of soil temperature profiles at ten Alaskan sites from the Permafrost Laboratory at the University of Alaska Fairbanks (UAF) (http://permafrost.gi.alaska.edu/sites_map; Romanovsky et al., 2009) were used to infer thawed-to-frozen depth using the 0ºC degree threshold and to complement the CALM ALT observations in Alaska. Table 1~~Table 1~~ provides the coordinates and measuring methods of the UAF ~~in-situ~~in-situ sites. The UAF measurements were

used along with the CALM data to evaluate the ALT estimates derived from the CLSM simulation and the AirMOSS radar observations for the North Slope of Alaska in section ~~XXX~~4.1.

~~Prior to comparison with the model results and the aggregated radar retrievals, the distributed measurements for a given CALM site (see sampling methods in Table 1) were averaged into a single value. In the 5 cases where 2 or 3 CALM or UAF sites lay~~

~~within a single CLSM grid cell, a single "spatially-averaged" observed value was computed for the cell.~~

**3 Methods**

**3.1** ~~Comparison With~~Comparing ALT from In-situ Observations, **AirMOSS** ~~ALT~~ **Retrievals, and CLSM Results in Alaska**

~~The comparison of the ALT simulations with the~~First, we compare AirMOSS ~~ALT~~radar retrievals ~~consists of two parts.~~ ~~First,~~and CLSM simulation results of ALT for 2015 against each other and against ~~in-situ~~in-situ observations ~~of ALT for 2015~~: (i) we compare the spatial patterns of the AirMOSS retrievals with those of the model-simulated ALT over ~~the~~ northern Alaska. ~~Second,~~; and (ii) we evaluate ~~both the AirMOSS retrievals and~~ the simulated ALT against both the AirMOSS retrievals and ~~in-situ~~in-situ observations from the CALM and UAF networks. We rely on several metrics to evaluate the model and radar-

retrieval performance, including bias, root mean square error (RMSE), and correlation coefficient (R). The results are discussed in section 4.1.

We conducted the intercomparison at the model scale. The radar retrievals were provided at 2-arcsec × 2-arcsec (roughly 20 m x 60 m in the Arctic) resolution whereas the CLSM-simulated ALTs are at 81 km$^2$. We thus aggregated the AirMOSS retrievals to the CLSM model grid by averaging all the retrieval data points within each 81 km$^2$ model grid cell. Only model grid cells that were at least 30% covered by radar retrievals were used in the comparison.

The AirMOSS transects cover several different regions with different climatologic regimes, topography, vegetation and soil type (Figure 2Figure 2). Note that although the vegetation class used in the model (Figure 2b) suggests the presence of dwarf trees over the Alaska North Slope, the actual satellite-based LAI, vegetation height, greenness fraction and albedo will still instruct the model that the tree cover there is extremely sparse. The data sources for these vegetation-related boundary conditions can be found in Table 1 of Tao et al. (2017). Overall, the variability of ALT along these transects encompasses the

influence of a variety of factors at the regional scale.

~~The radar retrievals were provided at 2-arcsec × 2-arcsec (roughly 20 m x 60 m in the Arctic) resolution whereas the CLSM-simulated ALTs are at 81 km². We thus aggregated the AirMOSS retrievals to the CLSM model grid by averaging all the~~

~~retrieval data points within each 81 km² model grid cell. Only model grid cells that were at least 30% covered by radar retrievals were used in the comparison.~~

The daily UAF ~~in situ~~in-situ soil temperature profile observations on the AirMOSS flight date (29 August 2015) were used to calculate the thawed-to-frozen depth (i.e., approximated ALT). The ALT measurements at all of the 13 CALM sites covered

by the AirMOSS transects were obtained in August of 2015 (Table 1Table 1). Among them, eight CALM sites obtained ALT measurements slightly earlier than the overflight date (within at most 18 days from 29 August 2015). Nevertheless, we assume that these earlier measurements still represent the thaw depth at the end of August reasonably well. ~~Prior to comparison with the model results and the radar retrievals, the distributed measurements for a given CALM site (see sampling methods in Table 1) were averaged into a single value. Similarly, we also aggregated the radar retrievals (20 m x 60 m) up to the respective site~~

~~scale (ranging from 10 m × 10 m to 1000 m × 1000 m) where the CALM sampling grid for a given site included more than~~

one AirMOSS pixel.  Otherwise, the radar retrievals closest to the UAF (single-point) or CALM site are used in the site- scale comparison.

Prior to comparison with the model results and the aggregated radar retrievals, the distributed measurements for a given CALM site (see sampling methods in Table 1~~Table 1~~) were averaged into a single value.  If multiple CALM or UAF sites lay within a single CLSM grid cell ~~In the 5 cases where 2 or 3 CALM or UAF sites lay within a single CLSM grid cell~~, a single "spatially-averaged" observed value was computed for the cell.

~~The~~We employed the strategy of Schaefer et al. (2015) to handle the ~~~~uncertainty propagation ~~estimates utilized here are necessarily imprecise~~, i.e., adding in quadrature the uncertainty components from each scale/level involved (see the supplementary file for a detailed description). ~~(Chen et al., 2019b)~~For AirMOSS retrievals, the sampling uncertainty of mean ALT at the 81 km$^2$ model grid-cell scale is negligible given the large sampling size and the fact that the retrieval uncertainty dominates the overall uncertainty (see supplementary file). Here, we use a nominal estimate of 0.15 m to represent the AirMOSS uncertainty (i.e., the average of the lower and upper bound of the actual retrieval uncertainty for individual radar pixels as discussed by Chen et al. (2019b)).

When comparing in-situ measurements with model results at the 81 km$^2$ scale (i.e., a point-to-grid comparison), the ultimate measurement uncertainty propagated from the point-scale measurements to the 81 km$^2$ scale is, for all intents and purposes, unknown due to a lack of sufficient measurements over the 81 km$^2$ scale to compute upscaling errors (see supplementary file). We thus show instead the standard deviation of CALM measurements to illustrate, in a highly approximate way, the spatial representativeness error of the in-situ measurements – a small (large) standard deviation represents a homogeneous (heterogeneous) area in terms of ALT, meaning that the in-situ mean likely can (cannot) represent an average over a larger scale, assuming the site-scale heterogeneity is somewhat transferable to the larger scale. Such transferability might only apply to the largest in-situ site scales (e.g., 1000 m × 1000 m) to the model grid-scale (81 km$^2$) and is thus, in general, questionable. We thus make no claim here that the standard deviations shown represent true uncertainty levels.  ×Chen et al. (2019b)

In recognition of the limited documentation surrounding many of the measurements examined here (individual site characteristics, measurement strategies, horizontal correlations, etc.), we employ in this analysis, for simplicity, a universal error estimate of 0.14m for all in situ and AirMOSS ALT estimates at the 81 km² scale. This is the overall uncertainty found by Chen et al. (2019) for individual radar pixels at their native (20 m x 60 m) resolution; for the radar estimates aggregated to the 81 km² scale, we thus effectively (and conservatively) assume that the pixel-scale errors are highly correlated in space.

## 3.2 Idealized Experiments

After comparing the spatial patterns of the AirMOSS retrievals with the CLSM-simulated ALT results, we then investigate the factors that affect the spatial variability of ALT through a series of idealized experiments. Specifically, we repeated the simulation along the AirMOSS transects multiple times, each time removing the spatial variation in some aspect of the model forcing or parameters and then quantifying the resulting impact on ALT variability.

For these supplemental simulations, we first identified a grid cell within the IVO transect (shown in Figure 2a) that represents roughly average (typical) conditions across the ten different transects. In the first idealized experiment, we then modified the baseline configuration by applying the surface meteorological forcing data from the selected representative grid cell within the IVO transect to all grid cells along all AirMOSS transects. Thus, in this modified simulation (HomF, for homogenized forcing), spatial variability in meteorological forcing is artificially removed. All model parameters related to soil type and vegetation, however, remain spatially variable, matching those in the baseline simulation. In the next idealized experiment (HomF&Veg), we further replaced the vegetation-related parameters (including vegetation class, vegetation height, and time-variable Leaf Area Index (LAI) and greenness) along the AirMOSS transects using the corresponding parameters from the representative grid cell, which is characterized by dwarf tree vegetation cover. Thus, in this simulation, spatial variability in both forcing and vegetation is artificially removed.

In ~~the final~~a third idealized experiment (HomF&Veg&Soil), spatial variability in soil type and topography-related model parameters is removed along with that of the forcing and vegetation. The homogenized parameters include soil organic carbon content, porosity, saturated hydraulic conductivity, Clapp-Hornberger parameters, wilting point, soil class, sand and clay fraction, vertical decay factor for transmissivity, baseflow parameters, area partitioning parameters, and timescale parameters for moisture transfer (Ducharne et al., 2000; Koster et al., 2000). Here we use an intermediate soil carbon content value (i.e., 40 kg/m$^2$) for the homogenization; recall that the carbon content impacts the soil thermal properties (see section 2.1). Our investigation reveals that the model sensitivity to soil carbon content is much larger for lower ~~SOC~~soil organic carbon content (SOC) than for higher SOC, and easily gets saturated for high SOC (i.e., larger than ~100 kg/m$^2$) (not shown). Thus, we trust that 40 kg/m$^2$ is an appropriate value representing an intermediate SOC condition. All other soil parameters are homogenized to those at the representative grid cell.

Finally, we investigate potential nonlinearities by conducting two additional experiments: one in which we homogenized both the vegetation and soil parameters (HomVeg&Soil) and another in which we homogenized both forcing and soil parameters (HomF&Soil). Put differently, in experiment HomVeg&Soil only the forcing varies along the transects, whereas in experiment HomF&Soil, only the vegetation parameters varies along the transects. Combined with the experiment HomF&Veg (in which only soil properties vary along the transects), these three experiments show in a different way how each individual factor (forcing, vegetation, or soil) can contribute to ALT variability. Table 2~~Table 2~~ provides a summary of these idealized experiments. Taken together, the ~~four~~six experiments (including the baseline) allow us to identify the individual contribution of each factor to the ALT variability along the AirMOSS transects. The results are discussed in section 4.2.

**3.3 Quantifying ALT Spatiotemporal Characteristics**

~~We also~~In section 4.3 we quantify the large-scale characteristics of ALT over the Northern Hemisphere for the current climate (1980 - 2017) as determined by the response of the land model to 38 years of MERRA-2 forcing (~~in~~ section ~~4.3). Again, this~~

~~forcing was applied to 81 km² grid cells in the middle-to-high latitude area defined by the existence of NCSCDv2 soil carbon data (see area outlined in Figure 1a).~~2.1). Output ~~diagnostics saved~~ from this multi-decadal, offline simulation ~~allow~~allows the characterization of permafrost dynamics at each grid cell. In particular, we can compute a number of relevant ALT statistics, including mean, standard deviation, and skewness, from the diagnosed yearly values at each cell, and we can examine how these statistics relate to those of MERRA-2 forcing data (particularly the mean annual air temperature, MAAT~~)~~) over the last 38 years.

Besides MAAT statistics, we also consider the evolution of the air temperature during the warm season in terms of the energy it could provide to the land surface and thus to the determination of ALT. A simple surrogate for the total warm-season energy in year N can be computed from daily-averaged air temperature, $T_{air}(t)$, and the freezing temperature, $T_f$ (0°C degree), as follows:

$$T_{cum}(N) = \sum_{t=1}^{t=M} T_{pos}(t) , \tag{2}$$

where

$$T_{pos}(t) = \begin{cases} T_{air}(t) - T_f & if \ T_{air}(t) > T_f \\ 0 & if \ T_{air}(t) \leq T_f \end{cases}, \tag{3}$$

The index t in equation (2) for year N starts with a value of 1 on 1 September of year (N-1) and ends with a value of M on 31 August of year N. The number of days M ~~could be~~is 365 or 366 depending on the presence of a leap year ~~over the preceding annual period~~. Note the air temperature throughout this study means the near-surface air temperature (i.e., 2 m above the displacement height) derived from MERRA-2.

We first computed the correlation coefficient (R) between the annual time series of ALT and $\sqrt{T_{cum}}$ and between the annual time series of ALT and maximum SWE ($SWE_{max}$) to quantify the degree to which variations of ALT can be explained solely by air temperature or by snow mass. Then, to ~~further~~ quantify the joint contributions of ~~$T_{cum}$ and the maximum SWE ($SWE_{max}$),~~ $\sqrt{T_{cum}}$ and $SWE_{max}$, we performed a multiple linear regression analysis by fitting the equation

$$\text{ALT} = a_0 + a_1 \cancel{T_{cum}} + \cancel{a_2} \sqrt{T_{cum}} + a_2 SWE_{max} \, ,$$

(4)

~~to the available data. The results are discussed in section 4.3.~~

to the available data. The correlation coefficient relating ALT to $\sqrt{T_{cum}}$ and $SWE_{max}$ is the square root of the coefficient of multiple determination ($R^2$) obtained through fitting Equation (4). This equation is similar in form to the common degree-day model for predicting ALT from accumulated degree days of thaw based on the Stefan solution (e.g., Shiklomanov and Nelson, 2002; Zhang et al., 2005; Riseborough et al., 2008; Shiklomanov et al., 2010). Here, however, we constructed equation (4) for a different purpose: to explore how much of the temporal variability of ALT can be jointly explained by snow mass and

above-freezing air temperature. Before calculating these correlation coefficients, we removed the linear trend within ALT, $T_{cum}$, and $SWE_{max}$ to avoid potentially exaggerating the correlation due to an underlying trend. The results are discussed in section 4.3.

**3.4 ~~Evaluation of~~ Evaluating Simulated Northern Hemisphere Permafrost Extent~~,~~ and ALT ~~Climatology and Trend~~**

We first evaluated the simulated permafrost extent against the observation-based permafrost map (Brown et al., 2002 as shown in Figure 1b). Note ~~T~~the model's description of permafrost is binary – either permafrost exists across a grid cell or it is completely absent. We cannot then expect an exact comparison to a specification of isolated permafrost (0-10% of area by definition) or even, to a lesser extent, sporadic permafrost (10-50% of area by definition). Therefore, we compared our simulated permafrost area with that of the total area of continuous, discontinuous, and sporadic permafrost area together from

Brown et al. (2002) and computed the percentage error relative to the observation-based area (i.e., the total area of continuous, discontinuous and sporadic permafrost regions). We also compared our simulated permafrost area against the total area of only continuous and discontinuous permafrost regions.

Further, The the CALM network of in-situ in-situ ALT measurements (section 2.3) allows a quantitative evaluation of the simulated ALTs for the grid cells containing the measurement sites. Our comparisons here focus on both multi-year annual ALTs and the multi-year averages; for a average (climatological) ALT at the 81 km² scale of CLSM data. To ensure a consistent comparison, we average the simulated ALTs only over the same years for which observed ALTs observations are available. .

If multiple CALM sites lay within a single CLSM grid cell, a single "spatially averaged" observed, As noted in section 3.1 and in the supplementary file, the uncertainty of the CALM ALT measurements in the context of evaluating grid cell-scale model results theoretically involves uncertainty derived from probing point measurement uncertainty, site-scale mean uncertainty, and upscaling errors in going from the site-scale to the model-scale. This latter uncertainty in particular is unknown. In our figures (in section 4.4) we show the standard deviation of the observed ALT as a very crude surrogate for the spatial representativeness error associated with the point-to-grid comparison. As before, we make no claim here that the standard deviations shown represent the relevant statistical uncertainty. The results are discussed in section 4.4. We approximate the uncertainty in the observed climatological ALT estimates at the 81 km² scale by dividing the universally assumed error of 0.14 m (section 3.1) with the square root of the number of years contributing to the multi-year value was computed for the cell. The average. The resulting uncertainty estimates range between 5 and 10 cm. The results are discussed in section 4.4. The results are discussed in section 4.4.

The MERRA-2 dataset provides close to four decades of forcing data, and it is tempting to see if trends in this forcing imprint themselves on the 4 Results

simulated permafrost. We realize at the outset, however, that two difficulties hamper the accurate quantification of permafrost trends from our data. First, air temperature trends in MERRA-2 are known to be underestimated in high latitude regions, especially in more recent years (Simmons et al., 2017). Second, our spin-up procedure essentially involved running our model over several cycles of MERRA-2 forcing prior to the final 1980-2017 simulation, meaning that if a temperature trend does exist in the data, the initial conditions for 1980 already contain information from the warmer, later period. Note that both of these difficulties would contribute to a trend underestimation.

We keep these caveats in mind as we compute permafrost trends through a linear regression of simulated yearly ALT against the year of simulation, interpreting the slope obtained (in cm/yr) as the trend. We similarly compute trends in MAAT (ºC/year), in $T_{cum}$ (section 3.3), and in the number of days with air temperature above the freezing point (i.e., warm days). As in section 3.3, a given year's MAAT, accumulated air temperature, and number of warm days are computed for the year-long period ending on 31 August of that year.

The ALT trend was similarly calculated from CALM observations at sites for which observed ALTs are available for at least eight years (i.e., sample size ≥ 8). When evaluating the accuracy of the model-simulated ALT trends, the model-based ALT trends were calculated using the same years for which observed ALTs are available. The results are discussed in section 4.5.

## 4.1 Simulated ALT versus ~~In Situ~~In-situ Measurements and AirMOSS Retrievals in Alaska

In this section, we compare the simulated ALT and the AirMOSS ALT retrievals at the 81-km$^2$ model resolution. Note that Chen et al. (2019) provide maps of the AirMOSS retrievals and an evaluation versus ~~in situ~~in-situ measurements at the native (20 m by 60 m) scale of the retrievals.

## 4 Results

## 4.1 Comparisons with observations across retrieval transects

In this section, we compare AirMOSS radar retrievals and CLSM simulation results against each other and against in-situ observations of ALT for 2015. The AirMOSS transects cover several different regions with different climatologic regimes, topography, vegetation and soil type (Figure 2). Note that although the vegetation class (Figure 2b) suggests the presence of dwarf trees over the Alaska North Slope, the actual satellite-based LAI, vegetation height, greenness fraction and albedo will still instruct the model that the tree cover is extremely sparse in this region. The data sources for these vegetation-related

boundary conditions can be found in Table 1 in Tao et al. (2017). Overall, the variability of ALT along these transects encompasses the influence of a variety of factors at the regional scale.

Figure 3Figure 3 compares the spatial pattern of AirMOSS ALT retrievals and CLSM-simulated results. Generally, the patterns

of the AirMOSS retrievals and CLSM results are quite different.  For example, the AirMOSS-retrieved ALT is greater in the northern portion of the DHO transect than in the southern portion (Figure 3a,b), whereas this pattern is largely reversed in the simulated ALT (Figure 3c). Thefor DHO (Figure 3b). Across all transects, there are portions where the AirMOSS ALT is less than the CLSM-simulated ALT (Figure 3c)and portions where the AirMOSS ALT is greater than (Figure 3c), though it should be noted that from AirMOSS (Figure 3b)the differences in portions Figure 3c are generally less than the assumed uncertainty

of each transect. Across all four transects0.145 m (see section 3.1).  Generally, the CLSM-simulated ALT shows relatively larger spatial variability (0.35 - 0.85 m) than the AirMOSS retrievals (0.4 – 0.6 m).  The AirMOSS ALT exhibits some spatial variability at the native resolution (Figure 3a), but some(see Chen et al., 2019), but much of this variability averages out during the aggregation to the coarse model grid (Figure 3Figure 3b).  The abrupt changes in CLSM a).  Variations of the simulated ALT shown in within a single transect (Figure 3Figure 3c are predominantly controlled by soil type (see discussion in section

4.2).ba) are predominantly induced by changes in soil type (indicated in Figure 2Figure 2c and 2d).  In essence, the higher the organic carbon content within the soil, the smaller the simulated ALT due to slower heat transfer associated with lower thermal conductivity, higher porosity, heat capacity, etc.  (Tao et al., 2017). See also section 4.2 for a discussion of the influence of soil texture on the spatial pattern of ALT.

Next, we compare the retrieved and simulated ALT in 2015 with in situin-situ observations from the CALM and UAF sites that are collocated with the AirMOSS transects. Figure 4b, c demonstrates (section 3.1). Figures 4a and 4b show that in some ways, the CLSM-simulated results roughly agree, to first order,ALTs agrees with the in-situ observations. The overall mean bias of simulated ALT relative to to within the in-situ measurements is  measurement uncertainty of 0.0514 m.  Nevertheless, at only about half of the scatter (blue) in Figure 4c is large, and measurement sites.  Note, however, that at several sites in the

corresponding correlation coefficient is quite weak (remaining half, the difference only slightly exceeds 0.27).14 m, with

~~The~~an overall mean bias ~~of simulated ALT relative to the in-situ measurements is~~of -0.05 m and a RMSE of 0.17 m~~.~~. The most significant discrepancies between the CLSM-simulated ALT and in-situ measurements are at U6, U31, FB1&FBD&FBW (Figure ~~4b~~4a), where the simulated ALT underestimates the in-situ measurements by 0.25-0.28 m, and at U28 where the simulated ALT overestimates the in-situ ALT by 0.27 m. ~~The overall mean bias of simulated ALT relative to the in-situ~~

~~measurements is -0.05 m.~~ Nevertheless, the scatter in Figure 4b is large, and the corresponding correlation coefficient is quite weak (0.27).

The AirMOSS ALT radar retrievals, for their part, again averaged to the 81-km$^2$ model resolution (section 2.2), show less spatial variability than the observations (Figure ~~4c~~4a). The largest error for ~~these~~the AirMOSS retrievals at the model scale is also at FB1&FBD&FBW, where the retrievals significantly underestimate the observed in-situ ALT by 0.38m. Note that radar retrievals at the ~~model~~81 km$^2$ scale ~~(green)~~ are not available at some sites because of our imposed 30% filling restriction ~~(Figure 4b), whereas the retrievals at the site scale (black) are available at all sites within the IVO, ATQ, BRW and DHO~~

~~transects (Figure 4a). The largest radar retrieval errors at the site scale are at FBD, FBW, and SG1, where the ALT retrievals~~ ~~underestimate the in-situ measurements by 0.32 - 0.53 m.a.). For the AirMOSS retrievals, when all in-situ sites are considered,~~ ~~the overall ALT bias is -0.11 m at the site scale and -0.12 m at the model scale. While the~~12 m. ~~The corresponding correlation~~ ~~coefficient with the in-situ observations is only 0.05 at the site scale, it is 0.61 at the model scale.~~

Although the AirMOSS ALT retrievals generally underestimate the ~~in-situ~~in-situ ALT measurements (as shown in Figure 4a), the retrievals ~~are broadly~~tend to be more consistent with the observations when the ~~in-situ~~in-situ measurements are within the ~60 cm sensing depth of the P-band radar data~~. (Note again the ALT retrievals cannot exceed the radar sensing depth of about~~ ~~60 cm.) This is the case at sites in the northernmost area, e.g., U3, U1, U2, BR2, and U5. Excluding~~, as indicated in Table 3. Specifically, excluding the sites with ~~in-situ~~in-situ ALT measurements that exceed the AirMOSS sensing depth of ~60 cm, the

overall mean bias for the AirMOSS retrievals at the ~~model~~81 km$^2$ scale ~~(site scale)~~ drops to -0.01 m ~~(0.02 m),~~, and the

correlation coefficient at the model scale (site scale) increases to 0.64 (0.20) (Table 3).. In contrast, the CLSM simulation results show a bias of 0.01 m and a zero correlation coefficient at the samethese sites.

Nevertheless, as noted in section 3.1, given that the upscaling errors in going from the CALM site-scale to the model-scale is unknown and the fact that the standard deviation of these measurements (as shown by error bars in Figure 4a and 4b) indicates large representativeness errors of the in-situ measurements, the point-to-grid comparison result is hard to quantify. In this regard, the AirMOSS retrievals aggregated to the same scale as model results provide a comparable counterpart for evaluation. Figures 4c further shows that the CLSM-simulated ALT agrees well with the AirMOSS ALT retrievals to within the measurement uncertainty of 0.15 m at all the site-located model grid cells. Indeed as Figure 3c illustrated, the differences between simulated ALT and the AirMOSS retrievals over all the transects examined here are generally below the measurement uncertainty of 0.15 m.

**4.2 Sources of ALT Spatial Variability: Results from Idealized Experiments**

Here we investigate the specific factors that drive ALT spatial variability along all ten of the AirMOSS transects (Figure 2a). For this analysis, the simulated ALT estimates shown in Figure 5a were aggregated across the width of the radar swath (compare Figure 3). Figure 5Figure 5a illustrates that, in general, the simulated ALT captures the spatial variability exhibited by the in-situin-situ measurements. This conclusion is, however, very tentative given the limited number of in-situin-situ ALT observations.

Generally, theThe simulated ALT is shallowest in the northern transects (ATQ, BRW, and DHO) and deepest in the southeastern transects (KYK, COC, KGR, and AMB). This pattern correlates somewhat (R = 0.46) with that of the mean screen-level (2-meter) air temperature (Tair) for the preceding 12-month period (i.e., from 1 September 2014 to 31 August

2015) from MERRA-2 (green line in Figure 5~~Figure 5~~a).  The soil carbon content, by contrast, appears anti-correlated (R = -0.59) with the simulated ALT, as exemplified by the transect portions within the red box (Figure 5~~Figure 5~~a and 5b).  Such a correlation presumably reflects the fact that soil with high organic carbon content has low thermal conductivity, which hinders heat transfer from the surface to the deeper soil in the summertime, thus resulting in a relatively smaller ALT.  In addition,

heat transfer is ~~also~~ slowed by a higher effective heat capacity associated with higher organic carbon content – not from the carbon itself, but from the extra water that can be held in the soil due to the increased porosity.  The maximum snow depth (Figure 5~~Figure 5~~c) displays a positive correlation with ALT (R=0.47), reflecting, at least in part, the fact that subsurface soil temperatures remain relatively insulated under thick and persistent snow cover, which reduces heat transfer out of the soil column during the wintertime and hence facilitates a deeper thawing during the summer and thus a deeper ALT.

The correlations in Figure 5 suggest (without proving causality) that for the model, surface meteorological forcing (including air temperature and precipitation) as well as soil type are important drivers of ALT variability along the AirMOSS transects.  However, the relatively low values of the correlations indicate that a simple linear relationship cannot explain the mutual control that these variables exert on ALT spatial variability.  In the remainder of this section, we use a series of idealized model

simulations (as described in section 3.2) to better quantify the relative impacts of these driving factors along the AirMOSS transects.

The results of the idealized experiments are shown in Figure 6~~Figure 6~~.  The above-mentioned, large-scale spatial variation of ALT in the baseline simulation, with larger values in the southeastern transects (KYK, COC, and KGR) and lower values in

the northern transects (ATQ, BRW, and DHO), is absent after homogenizing the meteorological forcing (HomF; Figure 6a). Experiment HomF correspondingly has much less spatial variation in the temperature of the top soil layer than does the baseline simulation (Figure 6~~Figure 6~~b).  In addition, homogenizing the forcing (which includes snowfall) significantly reduces the variability in maximum snow depth along the AirMOSS transects (Figure 6~~Figure 6~~c). These results indicate that in the model, meteorological forcing exerts the dominant control over the spatial patterns of ALT, the temperature in the top soil layer, and

snow depth at the regional scale, as expected.

Homogenizing the vegetation attributes in addition to the forcing (HomF&Veg) results in ALT differences (relative to HomF) primarily along the northern transects (ATQ, BRW, and DHO).  Along these transects, homogenizing the vegetation parameters (including LAI and tree height) to those of the representative grid cell within the IVO transect results in generally shallower ALT.  ~~We speculate that this~~This is because the generally lower albedo of the ~~somewhat~~ taller and leafier trees (representative of the IVO transect) during the snow season resulted in increased snowmelt and thus reduced snowpack during the snow season (compare the green and red curves in Figure 6c), thereby reducing the thermal insulation of the wintertime ground.  With reduced insulation, cold season ground temperatures dropped, making it more difficult for temperatures to recover during summer~~.~~ (Tao et al., 2017).

As might be expected, the simulation in which soil properties are homogenized in conjunction with forcing and vegetation (i.e., HomF&Veg&Soil) essentially eliminates all remaining spatial variability in ALT, snow depth, and soil temperature. Owing to the strong control of soil type-related parameters (see section 3.2 and Table 2) on soil moisture, spatial variability in soil moisture remains high in HomF and HomF&Veg and is only eliminated once ~~the~~these soil type-related parameters are homogenized (Figure 6d), which explains the abrupt changes shown in Figure 3~~Figure 3~~c as mentioned in section 3.1.  (Note that to maintain consistency with the hardwired scaling factors for snow-free albedo within the model ~~(Mahanama et al., 2015),~~(Mahanama et al., 2015), we still used the original, vegetation-related parameters to calculate surface albedo during snow-free conditions along the transects.  This is likely the cause of the few tiny bumps seen in the Figure 6a for HomF&Veg&Soil.)

An alternative view of these results is provided in Figure 7~~Figure 7~~a, which shows the (spatial) standard deviation of ALT along the AirMOSS transects for each of the above experiments. Homogenizing the meteorological forcing data results in a significant reduction of the ALT standard deviation (from 0.16 to 0.10).  Additionally homogenizing the vegetation only reduces the ALT standard deviation slightly (from 0.10 to 0.09).  The remaining ALT variability is eliminated through the additional homogenization of the soil type-related parameters (HomF&Veg&Soil), which emerge as another important driver

of ALT variability along the AirMOSS transects.  Note that the ALT variability associated with soil type is generally realized at smaller spatial scales than that associated with the meteorological forcing discussed earlier regarding Figure 6a.

We investigated potential nonlinearities by conducting two additional experiments: one in which we homogenized both the vegetation and soil parameters (HomVeg&Soil) and another in which we homogenized both forcing and soil parameters (HomF&Soil) (Table 2).  Put differently, in experiment HomVeg&Soil only the forcing varied along the transects, whereas in experiment HomF&Soil, only the vegetation parameters varied along the transects.  Combined with the experiment HomF&Veg (in which only soil properties varied along the transects), these three experiments show in a different way how each individual factor (forcing, vegetation, or soil) can contribute to ALT variability.  The results, shown in Figure 7b,

confirmThe impact of potential nonlinearities are examined in Figure 7Figure 7b, which shows the individual impact of vegetation, soil, and forcing heterogeneity on the ALT standard deviation along the transects, with the other inputs having been homogenized.  The graphic confirms that the meteorological forcing is the dominant driver of ALT spatial variability in our modelling system, followed by the soil type-related parameters and the vegetation parameters.

Note that in Figure 6a, the soil impact on ALT (difference between HomF&Veg&Soil in black and HomF&Veg in red) appears smaller than that of the vegetation (difference between HomF in green and HomF&Veg in red) over the northern transects (ATQ, BRW and DHO).  Even so, Figure 7b shows that, in terms of the integrated impact along all the transects, the soil influence clearly outweighs the influence of vegetation – at several other transects, including HUS, KYK, COC, AMB, IVO and the first half of ATQ (where vegetation conditions might be similar to those used for homogenizing), the changes in

vegetation parameters do not have much impact.

**4.3 ALT Spatiotemporal Characteristics overof ALT across the Northern Hemisphere**

Figure 8Figure 8a shows the distribution of mean ALT over the modelling domain, and Figure 8b shows the ALT standard deviation in time over the 38-year period. As might be expected, ALT tends to increase with distance from the pole, with the

largest values found in Mongolia and near the southern portion of Hudson Bay, though there are areas (e.g., just north of 60°N at ~120°E) with local minima that break this pattern. The largest ALT standard deviations (red ~~colour~~color in Figure 8~~Figure 8~~b) are found mainly in discontinuous and sporadic permafrost regions (see Figure 1b) where ALTs are deeper on average than that in continuous permafrost region. Figure 8~~Figure 8~~c provides the skewness of the temporal distribution. Though there are some exceptions, by and large the skewness is positive in most permafrost regions, suggesting that the largest positive ALT anomalies tend to be of greater magnitude than the largest negative anomalies.

Figure 8~~Figure 8~~d displays the average of annual mean 2-meter air temperature as derived from MERRA-2. The observed continuous and discontinuous permafrost areas shown in Figure 1b are well confined within the cold side of the 0°C (273.15K) isotherm in the mean air temperature map (Figure 8~~Figure 8~~d). For the most part, the observed sporadic and isolated permafrost regions of Figure 1b also lie on the cold side of the 0°C isotherm. The consistency with this isotherm, however, is not as clearly present in the simulated permafrost extent (i.e., the extent of the non-grey and non-white areas in Figure 8~~Figure 8~~a). ~~The relationship between the spatiotemporal characteristics of simulated ALT and air temperature forcing clearly needs more analysis.~~

The relationship between the spatiotemporal characteristics of simulated ALT and air temperature forcing has been investigated before in many studies at the site to landscape scale (e.g., Klene et al., 2001; Shiklomanov and Nelson, 2002; Zhang et al., 2005; Juliussen and Humlum, 2007) and at the regional scale (e.g., Anisimov et al., 2007). Here we simply analyze the correlation coefficient between ALT and two variables: the proxy of total energy input into the ground (i.e., $\sqrt{T_{cum}}$, see section 3.3) and the maximum SWE. Our goal is to explore how much of the spatiotemporal variability of ALT across the globe can be jointly explained by these two variables.

Figure 9~~Figure 9~~a shows a map of the correlation coefficient between the 37-year time series (i.e., from September 1980 through August 2017) of ~~$T_{cum}$~~$\sqrt{T_{cum}}$ and the corresponding time series of simulated ALT. The areas with p values larger than 0.05, which indicate correlations that are not statistically different from zero at the 95% confidence level, are shown as green.

Formatted: Font: (Asian) Times New Roman

Formatted: Font: (Asian) Times New Roman

Formatted: Font: (Asian) Times New Roman

Formatted: Font: (Asian) Times New Roman

Formatted: Font: (Asian) Times New Roman

Formatted: Line spacing: Double

Formatted: Font: (Asian) Times New Roman

~~Figure 9~~Figure 9a demonstrates that most permafrost regions indeed have significant positive correlations (red colours) between ALT and ~~$T_{cum}$~~$\sqrt{T_{cum}}$. Clearly, in these regions, air temperature exerts a dominant control on year-to-year ALT variability.

However, not all regions exhibit a significant correlation; other variable(s) must also be exerting control on interannual ALT variability. One reasonable candidate variable is snowpack. As noted above, snow acts as a thermal insulator -- regions with thicker snowpack are better able to insulate the ~~winter~~ ground from becoming too cold during winter, thereby ~~facilitating the heating of the~~supporting higher subsurface temperatures during non-winter months. Variable, but often thick, snowpack is in fact common in the areas of Figure 9a that show a low (green) or negative (blue) correlation between ALT and ~~$T_{cum}$~~$\sqrt{T_{cum}}$ –

areas such as Central Siberia, the Southern part of eastern Siberia, and a vast region in Canada surrounding the Hudson Bay, as well as other small areas that appear in high mountains or on the windward side of the mountains (e.g., locations B, C and D in Figure 1~~Figure 1~~a).

In Figure 9~~Figure 9~~b we show the correlation coefficient between the time series of ALT and the maximum SWE ($SWE_{max}$~~,~~)
during the preceding winter. A positive correlation is seen in many areas, most notably in areas with a poor or negative correlation between ALT and ~~$T_{cum}$~~$\sqrt{T_{cum}}$ (Figure 9~~Figure 9~~a) – for example, just west of Hudson Bay and along a zonal band at 60°N in Russia. Apparently, in these areas, the impacts of snow physics on ALT outweigh the impacts of lumped energy input (~~$T_{cum}$), for reasons that are not clear.~~$\sqrt{T_{cum}}$). In some other areas ALT correlates positively with both ~~$T_{cum}$~~$\sqrt{T_{cum}}$ and $SWE_{max}$. Figure 9~~Figure 9~~c shows how the resulting coefficient of multiple correlation varies in space. High correlations
largely blanket the modelled area. That is, over most of the area examined, a substantial portion of the year-to-year variability of ALT can be explained by joint variations in ~~$T_{cum}$~~$\sqrt{T_{cum}}$ and $SWE_{max}$. Even so, a few limited areas still exhibit low correlations ($p>0.05$, green colour in Figure 9~~Figure 9~~c). Some of these areas are in ~~high mountains~~mountainous regions, for instance the Eastern Siberian (Ostsibirisches) Bergland, where more complex environmental controls might be playing a dominant role. In addition, MERRA-2 snow forcing might be severely erroneous in these regions.

**4.4 Evaluation of Simulated Permafrost Extent and ALT across the Northern Hemisphere**

We now evaluate, to across the extent allowed by available CALM in-situ observations, and on a spatial scale much broader than that addressed in section 4.1, the accuracy of the permafrost fields simulated by the model.  Northern Hemisphere

Qualitatively, the simulated permafrost extent (Figure 8Figure 8a) generally shows reasonable agreement with the observation-based permafrost map in Figure 1Figure 1b, especially for the continuous permafrost regions.  This is shown explicitly in Figure 10Figure 10a.  The main deficiency in the simulation results is the failure to capture a large area of permafrost in western Siberia (labelled as A in Figure 1Figure 1a).  The reasons for this particular deficiency are unclear; perhaps the initial

thermal conditions over western Siberia were too warm, or perhaps MERRA-2 overestimates current air temperatures in this region..  One possible reason is that the permafrost in western Siberia is characterized as an ecosystem-protected permafrost zone (Shur and Jorgenson, 2007) where a thick moss-organic layer (i.e., moss-dominated mires (Anisimov and Reneva, 2006; Anisimov, 2007; Peregon et al., 2009)) protects the permafrost below from thawing under a warm air temperature. This is mainly attributed to the low thermal conductivity of the organic layer in summer, which strongly insulates the permafrost from

the warm atmosphere, and the high thermal conductivity of the frozen organic layer in winter, which allows cold temperature penetration from above, provided the snowpack is not too thick (Nicolsky et al., 2007b; Jafarov and Schaefer, 2016). This mechanism is lacking in the current version of CLSM (Tao et al., 2017). Thus, improving the model through a better representation of thermal processes in an organic layer above the soil column in combination with initializing the simulation with a sufficiently cold soil temperature should improve the simulation results. This work is reserved for a future study.

Another possible reason for the poor skill in western Siberia is that the model initial conditions there were too warm, although MERRA-2 appears to underestimate summer air temperatures in this region (Draper et al., 2018; their Figure 7e).(Draper et al. 2017; their Figure 7e). Note that some other global models, such as CLM3 and the Community Climate System Model version 3 (CCSM3) as reported in Lawrence et al. (2012).(2012), also missed this area of permafrost and that updated versions

of these models (i.e., CLM4 and CCSM4) showed improved performance in this regard (Lawrence et al., 2012). Guo et al.

(Lawrence et al., 2012). Guo et al. (2017) reported underestimated permafrost extent simulated in western Siberia using CLM4.5 driven by three different reanalysis forcings (i.e., CFSR, ERA-I and MERRA), and they showed an improved simulation of permafrost extent in this area when using another reanalysis forcing, the CRUNCEP (Climatic Research Unit - NCEP) ~~(Guo and Wang, 2017). Guimberteau et al.~~ (Guo and Wang, 2017). Guimberteau et al. (2018) found similar

improvements stemming from the use of CRUNCEP forcing. We leave for further study whether the MERRA-2 forcing data is responsible for the western Siberia deficiency seen in our own results.

~~Aside from western Siberia, the geographically thin~~The disagreements ~~(i.e., about a few degrees latitude)~~ between the simulated and observed permafrost extents (covering about a few degrees latitude) toward the south in Figure 10a (green and

blue areas at the southern edge of permafrost regions) are ~~not as much~~less of a concern, since the comparison in such areas is muddied by the interpretation of "isolated" permafrost in the observational map (Figure 1b). ~~The model's description of permafrost is binary – either permafrost exists across a grid cell or it is completely absent. We cannot then expect an exact comparison to a specification of isolated permafrost (0-10% of area by definition) or even, to a lesser extent, sporadic permafrost (10-50% of area by definition).~~ The specific areas of each type shown in Figure 10a are listed in Table 4. The

simulated permafrost extent covers 81.3% of the observation-based area (i.e., the total area of continuous, discontinuous and sporadic permafrost regions), and misses 18.7% of the observed permafrost area. When comparing simulated permafrost extent with only continuous and discontinuous types, these metrics change to 87.7% and 12.3%, respectively. Meanwhile, the permafrost extent is overestimated by $3.2 \times 10^6$ km$^2$.

To produce Figure 10~~Figure 10~~b, multi-year averages of CLSM-simulated ALT values were spatially averaged over each of the four permafrost types outlined in Figure 1~~Figure 1~~b. (As is appropriate, permafrost is only occasionally simulated over the fourth, "isolated", permafrost type. The ALT average shown for this type is thus based on a particularly limited number of grid cells.) The average ALT is smallest in the continuous permafrost zone, higher in the discontinuous zone, and higher still in the sporadic permafrost zone; it is highest in areas of isolated permafrost. The progression, of course, is in qualitative

agreement with expectations – larger breaks in permafrost coverage imply a greater amount of available energy, which should also act to increase ALT.

The observed and ~~CLSM-~~simulated annual ALT and multi-year ALT averages are compared in Figure 11~~Figure 11~~. Generally, the ~~CLSM~~ simulated annual ALT and the averages agree reasonably well with observations for shallow permafrost regions, that is, for smaller ALT. A large bias, however, is found for most of the Mongolia sites~~. In~~; in Mongolia, the observed annual ALT and the climatological ALTs tend to be much larger than the simulated ALTs (light purple dots in Figure 11~~Figure 11~~a~~).~~ ~~The~~). Overall, the RMSE, bias and R are all significantly improved when the Mongolian sites are excluded from consideration. Specifically for the climatological ALTs, the RMSE (and bias) ~~associated with this scatterplot~~of simulated ALT climatological means is 1.22 m (and -0.48 m), and it drops to 0.33 m (and -0.04 m) if the Mongolia sites are excluded (Figure 11~~Figure 11~~d~~b~~). Given simplifications in the model, uncertainties in boundary conditions (e.g., vegetation types, soil properties, etc.), and ~~representativeness~~upscaling issues stemming from the coarse-scale nature of the forcing data relative to the point-scale and plot-scale nature of the observations (i.e., the representative errors as indicated by the large standard deviation shown in Figure 11a), these results seem encouraging. The correlation coefficient metric (R), however, is somewhat less encouraging, amounting to only 0.5 when considering all sites. The correlation coefficient is in fact lower (0.3) when the Mongolian sites are excluded; the correlation coefficient is 0.39 for the Mongolian sites considered in isolation. Note that the existing literature on simulated ALT fields (e.g., Dankers et al. ~~(2011), Lawrence et al. (2012) and Guo et al.~~ (2011), Lawrence et al. (2012) and Guo et al. (2017)) reveals a general tendency for models to overestimate ALT climatology at the global scale. ~~Our~~In light of this, our results ~~here~~ suggest that the CLSM-simulated ALT fields are perhaps among the better simulation products, especially for shallow permafrost.

Comparing the observed and simulated spatial distributions of the ALT averages provides a further test of the accuracy of the simulation results (as shown in Figure 12). The model successfully simulates the large-scale spatial patterns in ALT, capturing, for example, the variations in Siberia, Svalbard, northern Canada, and northern Alaska (see Figure 12~~Figure 12~~a, b). Figure 12~~Figure 12~~c, d show the differences between the observed and estimated values in middle latitudes (45ºN to 60ºN) and high

latitudes (60ºN to 90ºN), respectively; in agreement with Figure 11Figure 11a, the model clearly performs better in high-latitude regions, i.e., outside of Mongolia. Many of the sites north of 60ºN (Figure 12Figure 12d) are coloured grey, indicating a small error in the simulation of ALT at these sites – the errors at these sites range from only -0.10m to 0.10m.

The significant underestimation of ALT in Mongolia may very wellmight result from errors in the meteorological forcing provided by MERRA-2. However, a comparison (not shown) of MERRA-2 air temperatures with measurements at six weather stations collocated with CALM sites in Mongolia calls this explanation into question. While MERRA-2 summer temperatures are indeed too low at four of the weather stations examined, they are too high at the other two weather stations. Another potential reason for the underestimation of Mongolian ALT involves the aforementioned representativeness error. The

Mongolian CALM sites employ a single-point measurement method (section 2.3) and are thus not able to represent the presumed spatial heterogeneity of permafrost within an 81 km² CLSM grid cell. An additional reason for the underestimation of ALT in Mongolia might be a mismatch between the land surface parameter values used in the model and the actual conditions at each site. For instance, detailed soil information (https://www2.gwu.edu/~calm/data/webforms/mg_f.html) indicate that some Mongolian sites have special "rocky" soil types including limestones (e.g., M04), slatestones (e.g., M05),

gravelly sand (e.g., M06 and M08), etc. that are not well represented in the model. As another example, sites on south-facing slopes presumably have much deeper ALT than those on slopes with less exposure to the sun, which is not captured by CLSM. The uncertainty assumed here for the Mongolian CALM sites (0.14m, as used elsewhere) may be underestimated given that these sites employ a single-point measurement method (section 2.3).The large representative errors of Mongolian sites are clearly illustrated by the standard deviation (although computed only with 3 to 5 measurements) as shown by the error bars in

Figure 11a.

**4.5 ALT Trend from 1980 to 2017**

The spatial distributions of the computed trends as described in section 3.4 are shown in Figure 13. (Only trends that are statistically significant at the 0.05 level are shown in the figure.) Figure 13a shows increasing/positive ALT trends (red colours) in many regions, particularly in northern Alaska and eastern Siberia but also in some parts of Canada and northern Mongolia. Such positive trends, which appear despite the aforementioned spin-up limitations in the analysis, can be interpreted as permafrost degradation. Particularly large positive trends (i.e., greater than or equal to 0.5 cm/year) are found along the coastline of the Okhotsk Sea in eastern Siberia, in northern Mongolia, in northern Quebec, and in limited areas of central Canada and the north slope of Alaska. The fact that the highest trends are generally found at the margins of the simulated permafrost distributions is consistent with the findings of James et al. (2013).

The ALT trends in Figure 13a are, at least to some extent, consistent with the temperature trends shown in Figure 13b,c,d, particularly in northern Alaska, northern Quebec, and (for MAAT) eastern Siberia. The widespread positive ALT trend in northern Mongolia is consistent with the positive trend in $T_{cum}$ (Figure 13c). Some areas, however, show a large positive ALT trend but not an obvious corresponding trend in air temperature. Also, some locations near the regions marked B, C, and D in Figure 1a show negative ALT trends in the presence of a warming trend in $T_{cum}$ (Figure 13c). It is possible that in such cases, the computed trends are strongly affected by snowpack variability, though neither maximum SWE nor snow cover duration tends to show a significant trend in these areas (not shown). It may be that in snow-dominated regions, joint variations and trends in temperature and snowpack complicate a simple interpretation of ALT trends in terms of trends in the meteorological forcing.

The observed trends calculated with CALM measurements are compared to the model-based trends in Figure 14. We exclude from the comparison the observational sites located in regions that, according to the CLSM, are permafrost-free (grey area in Figure 8a). Not counting the two Mongolian sites with trends at approximately 20-25 cm/yr, the simulated and observed trends at most of the sites roughly agree, though on balance the CLSM-based ALT trends are too low, likely due to the analysis limitations noted in section 2.1 and 3.4. A particular caveat is required regarding the Mongolian sites, given the unusual

observed trends calculated there. Attempts to contact the data providers to attain more detailed information for data evaluation were unsuccessful, and accordingly our confidence in these particular data is limited. The overall comparison in Figure 14 is, in any case, highly uncertain, given the limited number of data points available to compute the trends. Note that only four (ten) points remain when screening out sites at which either the observed or estimated ALT trend is not statistically significant at the 0.05 (0.10) level. Simply put, the limited number of sites with meaningful trends cannot assure an accurate trend assessment.

## 5 Conclusion and Discussion

Driving a tested model of permafrost dynamics (Tao et al., 2017) with a multi-decadal, reanalysis-based dataset of meteorological forcing (Gelaro et al., 2017) allows a global-scale characterization of permafrost. Indeed, in the course of this work we haveWe produced a dataset (effectively a derivative of MERRA-2) of permafrost variations in space and time at the 81 km² scale across middle-to-high latitudes. The permafrostThis dataset presented herein can be considered unique in terms of its daily temporal resolution combined with a relatively high spatial resolution at the global scale (i.e., 81 km²). The dataset, which is derived from a state-of-the-art reanalysis, (MERRA-2), shows reasonable skill in capturing permafrost extent (87.7% of the total area of continuous and discontinuous types, according to one validation dataset) and in adequately estimating ALT climatology (aside from that at the with a RMSE of 0.33m and a mean bias of -0.04m), excluding Mongolian sites).. We note that our MERRA-2-driven permafrost simulation results, while potentially better than those we might have obtained with MERRA forcing, are still lacking (e.g., in western Siberia). Still, with its resolution and available variables (ALT, subsurface temperature at different depths), the dataset could prove valuable to many future permafrost analyses.

This work also provides a first comparison between two highly complementary approaches to estimating permafrost: model simulation and remote sensing retrieval method. In northern Alaska, ALT retrievals from airborne remote sensing for 2015 and the corresponding simulated ALT exhibit limited skill versus the in situ measurements. At the model scale, the mean bias for the simulated results is better (-0.05 m) than that for the retrievals (-0.12 m), but the opposite is true for the correlation coefficient against observations (0.27 for the model vs. 0.61 for the retrievals). At the in situ site scale, however, the ALT

~~retrievals show a very weak correlation coefficient with the observations (0.05). Excluding.~~ In northern Alaska, excluding sites that have ALT measurements exceeding the radar sensing depth (~ 60cm), the evaluation metrics for ALT retrievals ~~become~~against ~~in situ~~in-situ measurements are better than ~~that~~those for simulated ALT at the ~~model~~81 km$^2$ scale. However, the remotely sensed ALT estimates generally show lower levels of spatial variability ~~than~~relative to the simulated ALT estimates~~,~~ (and ~~their~~relative to the ~~in situ~~in-situ observations), and the spatial patterns of the simulated and retrieved values differ considerably. The remote sensing approach is still relatively new, ~~and~~with many aspects still ~~need to be worked out~~requiring development. It is important, though, to begin considering the modeling and remote sensing approaches side by side, as both should play important roles in permafrost quantification in the years to come. Indeed, once the science fully develops, joint use of modeling and remote sensing (e.g., through the application of downscaling methods) should allow the generation of more accurate permafrost products at ~~even~~ higher ~~resolutions.~~resolution.

It is important to note that the retrieved ALT was determined by the dielectric transition from thawed to frozen conditions whereas the modelled ALT and the ALT for some of the ~~in situ~~in-situ measurements was based on a freezing temperature of 0°C (see sections 2.1 and 2.3). Depending on local conditions, soil does not typically freeze at 0°C but rather at slightly lower temperatures (e.g., around -1°C) due to the presence of dissolved compounds that depress the freezing point ~~(Watanabe and Wake, 2009).~~(Watanabe and Wake, 2009). The sharp drop in conductivity and dielectric constant is much more accurately tied to a frozen state than to a temperature threshold. These and other differences in the various ALT measurement methods (section 2.3) introduce considerable uncertainty into our comparisons. The use of the 0°C degree threshold in CLSM for determining the thawed or frozen layer may explain in part the model's underestimation of ALT, as may the lack of an explicit treatment of local aspect, errors in assigned model parameters, and so on.

Analysis of the CLSM-simulated data, along with data produced in ~~supplemental,~~ idealized experiments with specific homogenized controls, show how the statistics of permafrost variability in space are controlled by forcing variability and by variability in the imposed surface boundary conditions. In the idealized experiments, we employ successive homogenization of controls to quantify how meteorological forcing, soil type, and vegetation cover affect the underground thermodynamic

processes associated with the variability of ALT along the AirMOSS flight paths in Alaska. Meteorological forcing and soil type are found to be the two dominant factors controlling ALT variability along these transects. Vegetation plays a smaller role by modulating the accumulation of snow.

5   ~~A statistical analysis focusing on the global-scale fields reveals that yearly ALT strongly correlates with an accumulated air temperature diagnostic in most permafrost regions. In regions where they do not correlate well, yearly ALT does correlate with maximum SWE. Indeed, a~~ A multiple regression analysis relating yearly ALT jointly to accumulated air temperature and maximum SWE shows that time variations in these two latter quantities explain most of the time variability of ALT in the CLSM-identified permafrost regions.

~~The spatial distribution of CLSM-simulated permafrost shows general agreement with the observation-based permafrost map of Brown et al. (2002), capturing 81.3% of total areas of continuous, discontinuous and sporadic types while capturing 87.7% of the total area of continuous and discontinuous types. The model-based product does seem to miss a large area of permafrost in the northern part of the western Siberia, but it captures correctly, for the most part, the southward extent of permafrost.~~

15   ~~Apparent errors along the southern edge are, in any case, subject to significant uncertainty in this comparison given the presence of "sporadic" and "isolated" permafrost in the observational map, types that do not have direct analogues in the model-based (binary) product.~~

~~The CLSM-simulated ALT climatology was also compared to that derived from in-situ measurements collected through the~~
20   ~~CALM network. The simulated ALTs agree well with the in-situ observations for shallow permafrost in high-latitude regions (above 60°N latitude), but they generally underestimate ALTs in middle-latitude regions, especially in Mongolia. The RMSE of climatological ALT between model simulations and observations is 1.22m, and the mean bias is -0.48m. However, these reduce to 0.33m and -0.04m, respectively, when the Mongolia sites are excluded.~~

The simulated fields indicate permafrost degradation (as represented by a positive ALT trend) over the past 38 years in many areas, including large areas in Alaska and eastern Siberia as well as some limited patches in Canada and northern Mongolia. While concurrent trends in air temperature can reasonably explain many of the ALT trends, this connection appears to fall apart in regions dominated by snowpack variability. Modeled ALT trends agree with observed ALT trends (based on analysis of data from the CALM network) within a reasonable range (i.e., -1cm/year to 1cm/year) at many sites. At other sites, however, modeled trends are biased low, and they grossly underestimate the observed trends at two Mongolian sites. We emphasize again that such trend analyses are in any case highly uncertain due to the limited availability and the relatively short temporal extent of the observational ALT record.

Spatial representativeness issues plague an evaluation of simulated ALT (representing averages across 81 km² grid cells) against site-based ALT measurements. That said, we fully expect that the discrepancies seen between the simulated and observational ALT estimates reflect problems on the modelling side. Many aspects of the modelling framework may contribute to the noted errors in the simulated ALTs. For example, the observed climatological ALTs at the Mongolia sites are all larger than 3m. This depth falls well within the 6th soil layer of the model, which has a thickness of 10m; the subsurface vertical resolution in the CLSM may be too coarse to capture these deeper ALTs. Test simulations (not shown) with alternative model configurations indicate that increasing the number of soil layers may act to decrease somewhat the simulated ALT, suggesting that our values may be a little overestimated; however, based on results from a new study by Sapriza-Azuri et al.(2018), our use of a no-heat-flux condition at the bottom boundary rather than a dynamic geothermal flux may lead to underestimates of ALT. Such uncertainties should naturally be kept in mind when interpreting our results. Our supplemental simulations (not shown) also suggest that increasing the total modelled soil depth has only a small impact on simulated ALT. Uncertainty in our description of soil organic carbon, i.e., both soil carbon content and vertical carbon distribution, leads to corresponding uncertainty in our ALT simulations. We indeed find a significant improvement in simulated ALT at several Mongolian sites when we arbitrarily impose less total soil carbon content and concentrate less soil carbon in top layers (not shown). Besides the vertical distribution of soil carbon, the vertical variation in other soil hydrological properties (e.g. soil texture, and porosity, hydraulic conductivity, etc.)) should also play a significant role since they all affect soil thermal conductivity and heat capacity.

In addition, the lack of a necessary organic layer on top of soil column and the related thermal processes is also a major deficiency for the model especially in ecosystem-protected performant regions.

Another issue affecting our ALT comparisons is the climatological representation of vegetation parameters such as LAI used

in CLSM.  ~~Additional~~An additional investigation (not shown) revealed large differences between the LAI climatology used in CLSM and more realistic, time-varying, satellite-based LAI products at several Mongolian sites. In addition, while we did exclude from our analyses any measurements that were affected by notable disturbance (e.g., wildfire), the impacts of other potential land changes on ALT, including overgrazing in Mongolia ~~(Sharkhuu and Sharkhuu, 2012; Liu et al., 2013)~~.(Sharkhuu and Sharkhuu, 2012; Liu et al., 2013), were not explicitly treated in the model.  The model also lacks the vertical advective

transport of heat in the subsurface due to downward flowing liquid water, which can significantly affect permafrost thawing (Kane et al., 2001; Rowland et al., 2011; Kurylyk et al., 2014).  Also relevant are potential errors in the MERRA-2 forcing~~, which has a particularly large impact on the trend analysis; as mentioned above, the~~.  The MERRA-2 reanalysis is known to have problems capturing trends in high latitudes ~~(Simmons et al., 2017)~~.(Simmons et al., 2017).

Such modelling deficiencies must always be kept in mind when evaluating a product like the one examined here. That said, as long as appropriate caution is employed, the product could have significant value for further analyses of permafrost.  The product features daily subsurface temperatures and depth-to-freezing estimates over middle-to-high latitudes in the Northern Hemisphere at an 81 km$^2$ resolution, covering the period 1980-2017. It is, in a sense, a value-added derivative product of the MERRA-2 reanalysis and will be available via the National Snow and Ice Data Center (NSIDC).  The comparisons against

observations discussed above, along with the intuitively sensible connections shown between permafrost variability, forcing variability, and boundary condition variability, gives confidence that this dataset contains useful information. These data can potentially contribute, for example, to ecological studies focused on the dynamics of microbial activity and soil respiration in cold regions, on vegetation migration/adaptation in response to climate change, and so on.

**Acknowledgments**

Funding for this work was provided by the NASA Interdisciplinary Science program (NNX14AO23G). We thank Qing Liu at GMAO/GSFC/NASA for providing us corrected MERRA-2 precipitation. We acknowledge the University of Maryland supercomputing resources (http://www.it.umd.edu/hpcc) made available for the research reported in this paper.

~~References~~

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

Table 1 – ~~In situ~~In-situ permafrost measurement sites covered by the AirMOSS transects in 2015.

| AirMOSS flight (Official full name) | Permafrost Site (CALM or UAF)* | Latitude (degree) | Longitude (degree) | Sampling Method@ | Measurement Date |
|---|---|---|---|---|---|
| COC (Council) | U27 (CALM) | 64.8333 | -163.7000 | ~~4~~4 | 8/30/2015 |
| | U28 (CALM) | 65.4500 | -164.6167 | ~~4~~4 | 8/29/2015 |
| IVO (Ivotuk) | IV4 (UAF) | 68.4803 | -155.7437 | ~~1~~1[#] | 8/29/2015 |
| ATQ (Atqasuk) | U3 (CALM) | 70.4500 | -157.4000 | ~~4~~4 | 8/25/2015 |
| BRW (Barrow) | U1 (CALM) | 71.3167 | -156.6000 | ~~4~~4 | 8/21/2015 |
| | U2 (CALM) | 71.3167 | -156.5833 | ~~2~~2 | 8/24/2015 |
| | BR2 (UAF) | 71.3090 | -156.6615 | ~~1~~1 | 8/29/2015 |
| DHO (Deadhorse) | U4 (CALM) | 70.3667 | -148.5500 | ~~3~~3 | 8/25/2015 |
| | U5 (CALM) | 70.3667 | -148.5667 | ~~4~~4 | 8/11/2015 |
| | U6 (CALM) | 70.1667 | -148.4667 | ~~3~~3 | 8/26/2015 |
| | U31 (CALM) | 69.6969 | -148.6821 | ~~3~~3 | 8/15/2015 |
| | U8 (CALM) | 69.6833 | -148.7167 | ~~3~~3 | 8/27/2015 |
| | U32A (CALM) | 69.4410 | -148.6703 | ~~3~~3 | 8/16/2015 |
| | U32B (CALM) | 69.4010 | -148.8056 | ~~3~~3 | 8/16/2015 |
| | U9A (CALM) | 69.1667 | -148.8333 | ~~3~~3 | 8/25/2015 |
| | WD1 & WDN (UAF) | 70.3745 | -148.5522 | ~~1~~1 | 8/29/2015 |
| | DH2 (UAF) | 70.1613 | -148.4653 | ~~1~~1 | 8/29/2015 |
| | FB1 (UAF) | 69.6739 | -148.7219 | ~~1~~1 | 8/29/2015 |
| | FBD (UAF) | 69.6741 | -148.7208 | ~~1~~1[%] | 8/29/2015 |

| | FBW (UAF) | 69.6746 | -148.7196 | ~~1~~1 | 8/29/2015 |
|---|---|---|---|---|---|
| | SG1 (UAF) | 69.4330 | -148.6738 | ~~1~~1 | 8/29/2015 |
| | SG2 (UAF) | 69.4283 | -148.7001 | ~~1~~1 | 8/29/2015 |
| | HV1 (UAF) | 69.1466 | -148.8483 | ~~1~~1[%] | 8/29/2015 |

* CALM: sites from the Circumpolar Active Layer Monitoring (CALM) network; UAF: sites from the Permafrost Laboratory at the University of Alaska Fairbanks (UAF).

@Sampling method: ~~1~~1. Single point; ~~2~~ 2. 320 random sampling points within 10m × 10m ~~grid~~area; ~~3~~ 3. 100m × 100m grid with a 10m sampling interval; ~~4~~ 4. 1000m × 1000m grid with a 100m sampling interval.

# Two sensors are installed at IV4.

%Observations were taken from two conditions, including a frost-boil and an inter-boil area.

Table 2 – List of idealized simulation experiments along the AirMOSS transects.

| Experiment Name | Meteorological forcing | Vegetation | Soil parameters* |
|---|---|---|---|
| Baseline | Original | Original | Original |
| HomF | Homogenized | Original | Original |
| HomF&Veg | Homogenized | Homogenized | Original |
| HomF&Veg&Soil | Homogenized | Homogenized | Homogenized |
| HomVeg&Soil | Original | Homogenized | Homogenized |
| HomF&Soil | Homogenized | Original | Homogenized |

*CLSM soil parameters include soil organic carbon content, porosity, saturated hydraulic conductivity, Clapp-Hornberger parameters, wilting point, soil class, sand and clay fraction, vertical decay factor for transmissivity, baseflow parameters, area partitioning parameters, and time scale parameters for moisture transfer (Koster et al., 2000; Ducharne et al., 2000; Tao et al., 2017).

Table 3 – Evaluation metrics for model-simulated ALT and AirMOSS retrievals for 2015.

| ~~Metric~~ | ~~All sites~~ | | | ~~Sites with ALT measurements within AirMOSS sensing depth (~60 cm)~~ | | |
|---|---|---|---|---|---|---|
| | ~~CLSM-simulated ALT (model scale)~~ | ~~AirMOSS ALT retrievals (model scale)~~ | ~~AirMOSS ALT retrievals (site scale)~~ | ~~CLSM-Simulated ALT (model scale)~~ | ~~AirMOSS ALT retrievals (model scale)~~ | ~~AirMOSS ALT retrievals (site scale)~~ |
| ~~RMSE (m)~~ | ~~0.17~~ | ~~0.17~~ | ~~0.21~~ | ~~0.12~~ | ~~0.06~~ | ~~0.08~~ |
| ~~Bias (m)~~ | ~~-0.05~~ | ~~-0.12~~ | ~~-0.11~~ | ~~0.01~~ | ~~-0.01~~ | ~~0.02~~ |
| ~~R~~ | ~~0.27~~ | ~~0.61~~ | ~~0.05~~ | ~~-0.00~~ | ~~0.64~~ | ~~0.20~~ |

| Metric | All sites | | Sites with ALT measurements within AirMOSS sensing depth (~60 cm) | |
|---|---|---|---|---|
| | CLSM-simulated ALT | AirMOSS ALT retrievals | CLSM-Simulated ALT | AirMOSS ALT retrievals |
| RMSE (m) | 0.17 | 0.17 | 0.12 | 0.06 |

| Bias (m) | -0.05 | -0.12 | 0.01 | -0.01 |
|---|---|---|---|---|
| R | 0.27 | 0.61 | -0.00 | 0.64 |

Table 4 – Evaluation results for simulated permafrost extent. against the observation-based area~~the permafrost map~~ by Brown et al. (2002). The calculation was based on the comparison between simulated permafrost area and the total area of continuous, discontinuous and sporadic permafrost regions from Brown's map. The number in the brackets was calculated against the total area of continuous and discontinuous permafrost regions.

| Case | CLSM | Obs. | Simulated Area ($\times 10^6\,km^2$) | Percentage Relative to Observation |
|------|------|------|--------------------------------------|-------------------------------------|
| 4 | No | No | 48.8 | - |
| 3 | Yes | No | 1.9 | - |
| 2 | No | Yes | 3.2 (1.7) | 18.7% (12.3%) |
| 1 | Yes | Yes | 13.8 (12.3) | 81.3 % (87.7%) |

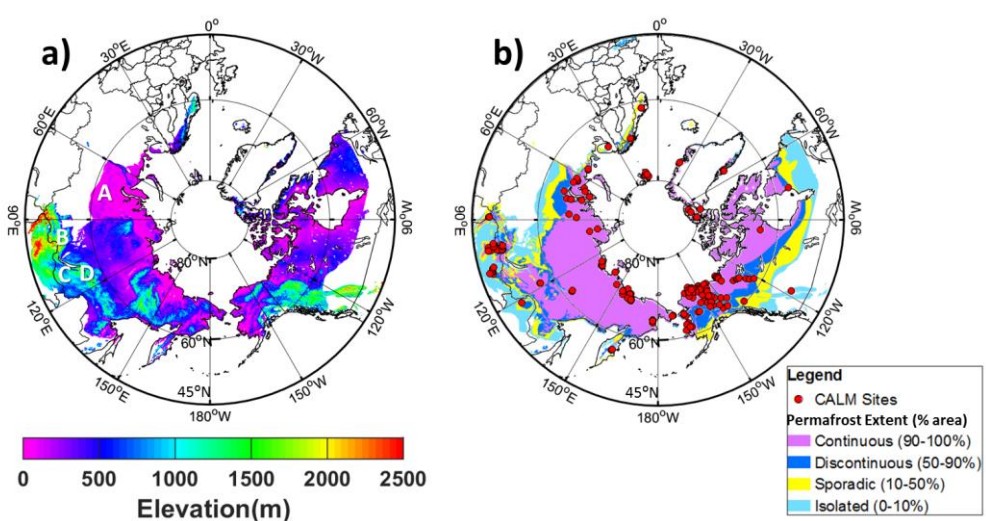

**Figure 1: a) Elevation above mean sea level in the simulation domain, which is defined by the area for which NCSCDv2 data are available. Regions A, B, C, and D are discussed in the text. b) Permafrost and ground ice conditions adapted from ~~(Brown et al., 2002).~~(Brown et al., 2002). Red dots represent CALM sites.**

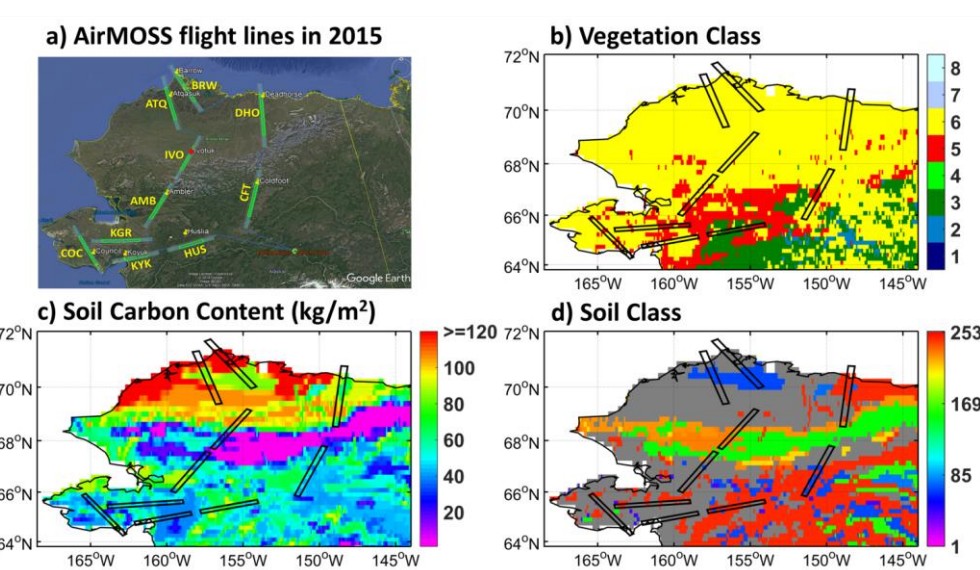

**Figure 2: a)** Ten transects of AirMOSS flights conducted in Alaska on 29 August 2015 and 1 October 2015, including HUS (Huslia), KYK (Koyuk), COC (Council), KGR (Kougarok), AMB (Ambler), IVO (Ivotuk), ATQ (Atqasuk), BRW (Barrow), DHO (Deadhorse), and CFT (Coldfoot). Each flight swath width is approximately 15 km. The red dot on IVO illustrates the location of the representative grid cell used and discussed in section 3.2. **b)** Vegetation class, **c)** soil organic carbon content, and **d)** soil class used in CLSM. The eight vegetation classes are 1-broadleaf evergreen trees, 2-broadleaf deciduous trees, 3-needleleaf trees, 4-grassland, 5-broadleaf shrubs, 6-dwarf trees, 7-bare soil, and 8-desert soil, respectively. The 253 soil classes include one "peat" class (#253), which is shown in dark grey, and 252 mineral soil classes (De Lannoy et al., 2014).

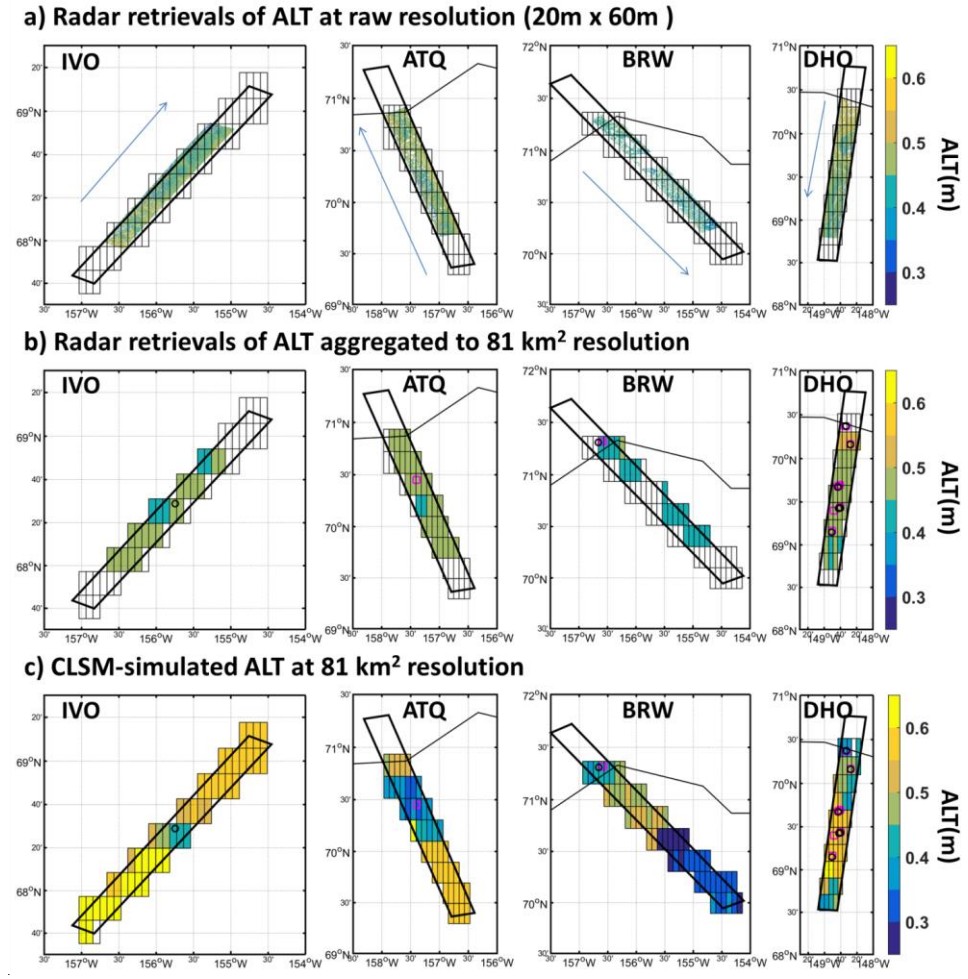

a) Radar retrievals of ALT at raw resolution (20m x 60m)

b) Radar retrievals of ALT aggregated to 81 km² resolution

c) CLSM-simulated ALT at 81 km² resolution

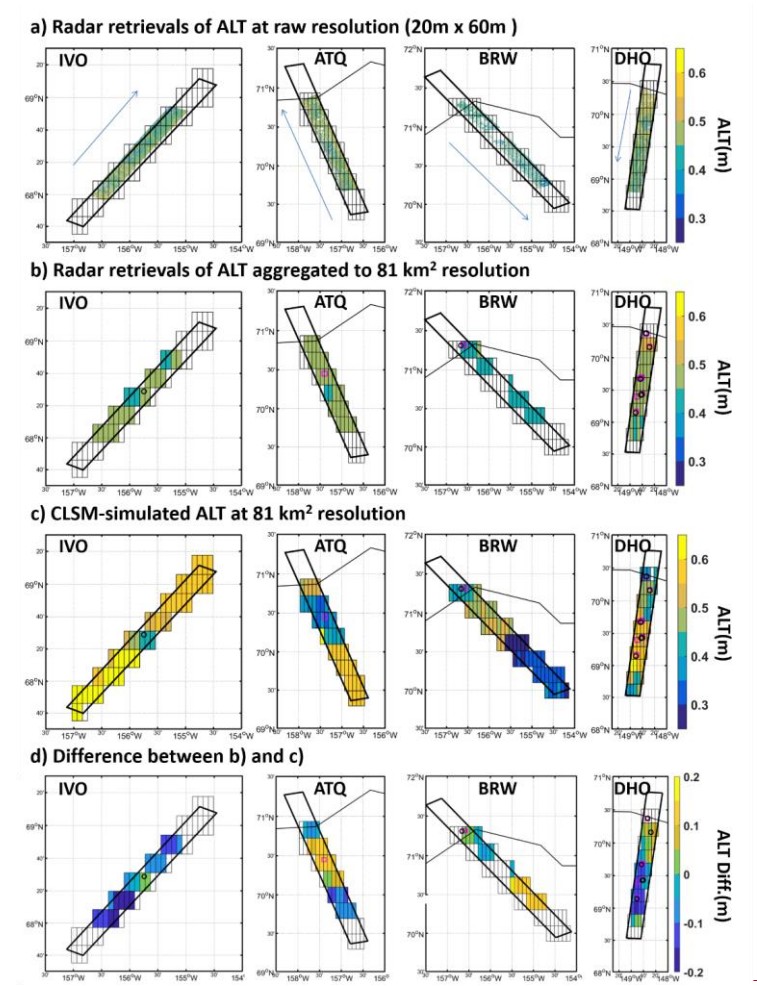

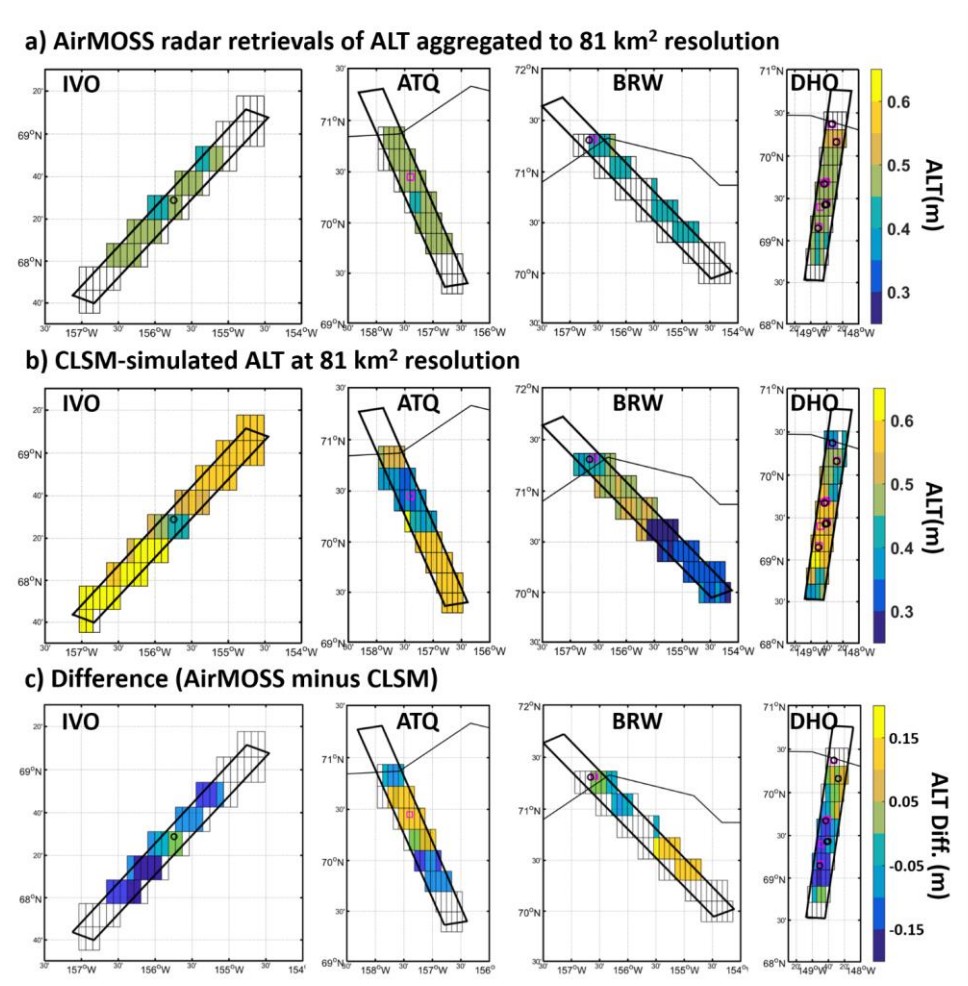

**Figure 3: a)** ~~AirMOSS radar~~Radar retrievals of ALT ~~for 2015 at raw resolution (20 m x 60 m)~~ derived from P-band radar observations on 29 August 2015 and 01 October 2015 for IVO, ATQ, BRW, and DHO. ~~b) ALT radar retrievals~~, aggregated to 81 km² model grid cells. ~~e~~b) CLSM-simulated ALT. c) Difference between the aggregated ALT retrievals and the CLSM-simulated results. Magenta squares ~~in b) and c)~~ represent CALM sites covered by the flight swath whereas black circles represent UAF sites.

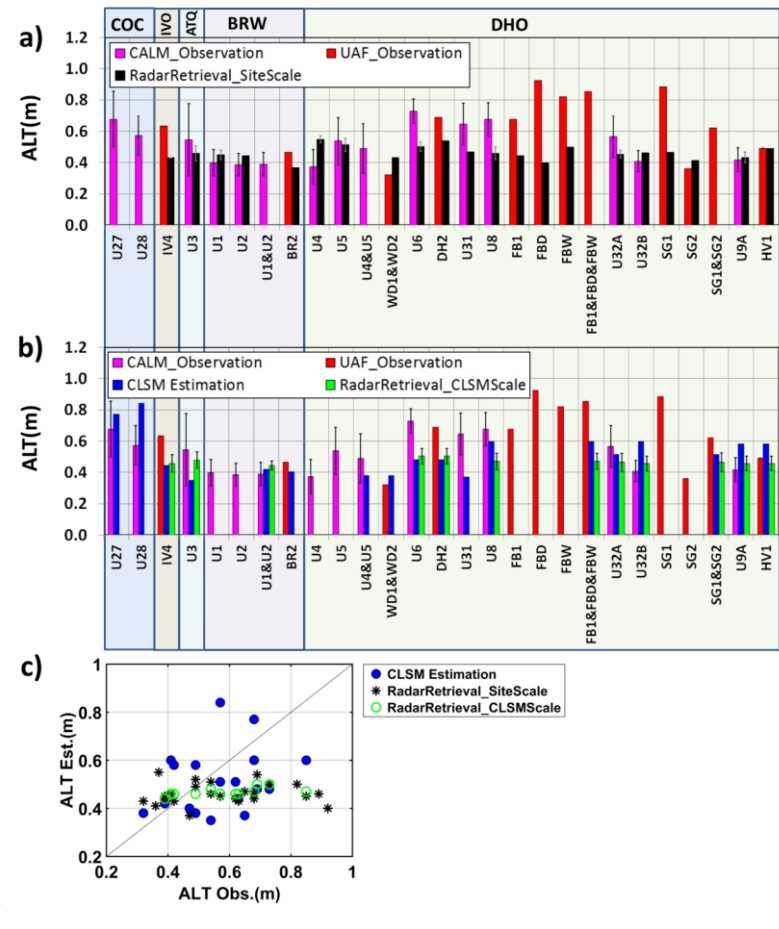

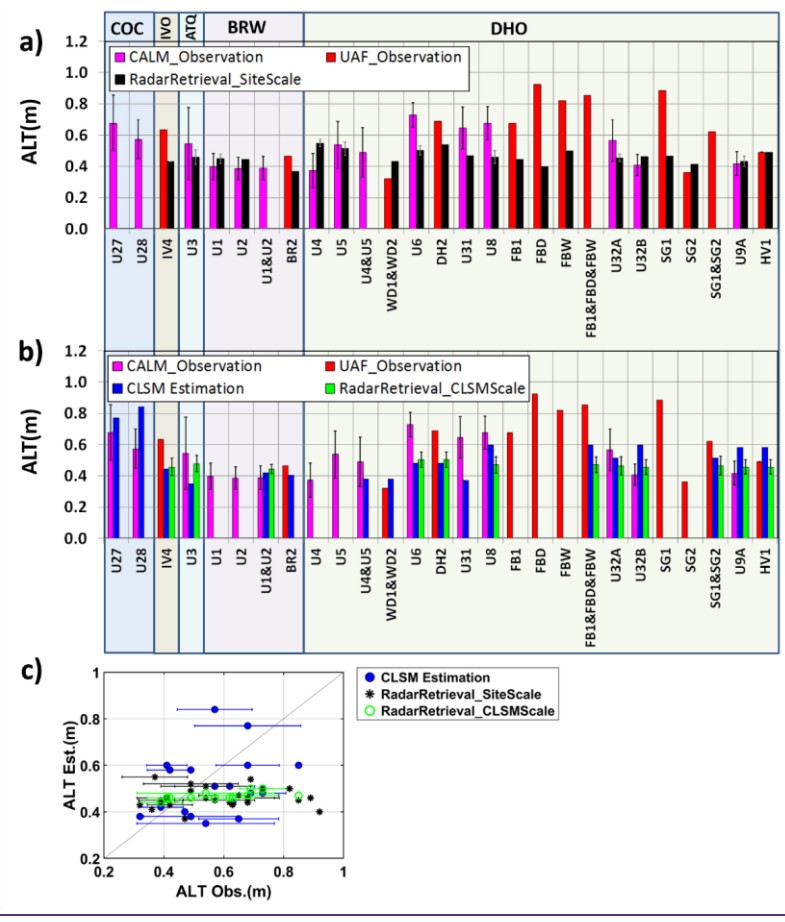

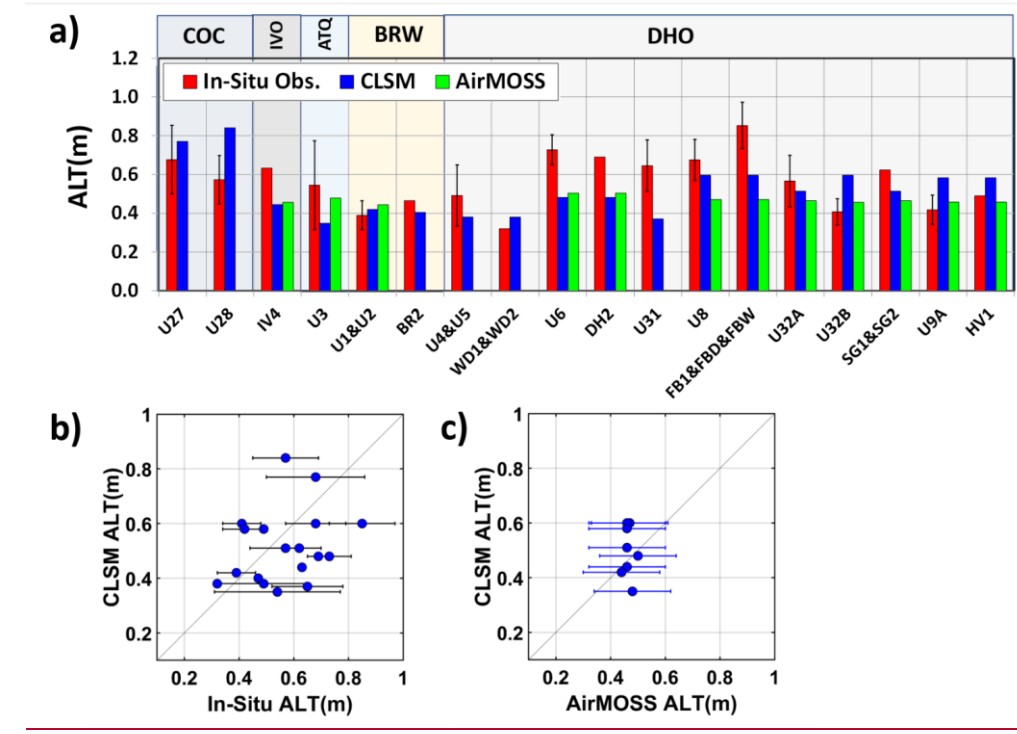

Figure 4: a) ALT observations (red) for 2015 from CALM (magenta) and UAF (red) sites covered by AirMOSS swaths, and and from radar retrievals at in-situ site scale (black). b) Same as a) but with ALT from radar retrievals aggregated to 81 km² grid cells (green), and from CLSM-simulated simulations ALT at 81 km² (blue). The short name of the corresponding covering swath is shown on the top (see also Figure 2a). Error bars represent the standard deviation for multiple observations at in-situ sites, and for radar retrievals within the CALM observing grid (at site scale), or within each 81 km² grid cell (at model scale). No standard deviations are provided for site scale radar retrievals at CALM U2, U31, and U32B sites because sampling coordinates are not available nor at UAF sites since single-point measurements were deployed. Averaged values were provided if multiple sites appear within a same model grid cell (e.g., U1&U2, U4&U5, WD1&WD2, FB1&FBD&FBW, and SG1&SG2) and compared with results at 81 km² grid cell. The sites are arranged aligning with the flight direction. cb) CLSM estimates and AirMOSS radar retrievals of ALT (at both site scale and model scale) for 2015 versus in-situ measurements with error bars indicating the standard deviation as

in a). c) Same as b) but versus aggregated AirMOSS ALT at model scale. The error bars here represent the uncertainty for radar retrievals at the 81 km$^2$ scale as explained in section 3.1. Corresponding estimates of CLSM uncertainty, which are presumably large, are not shown in the figure.

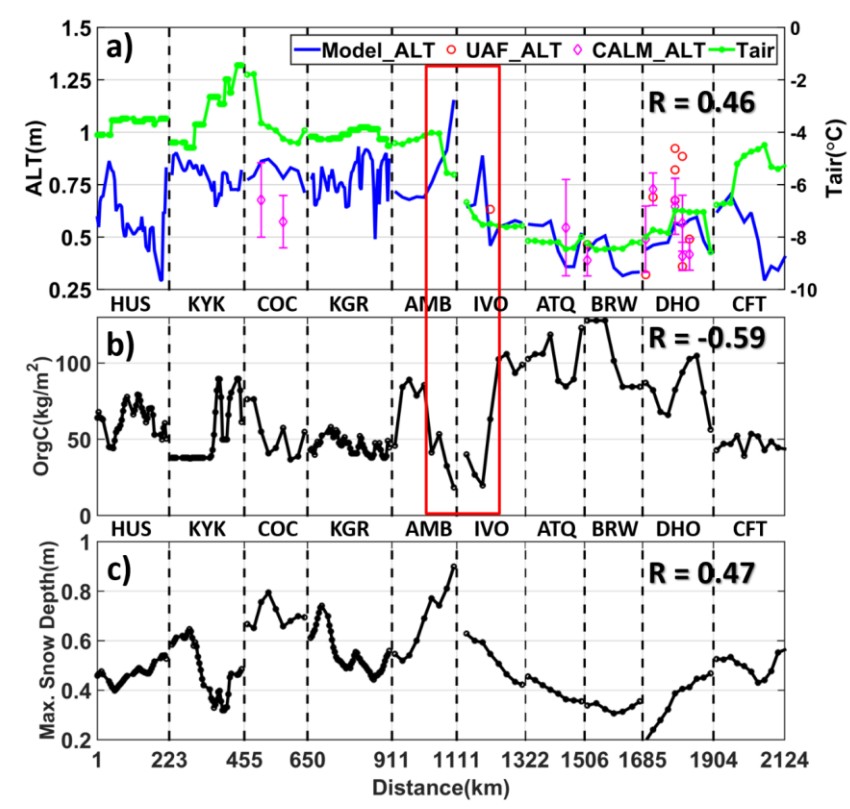

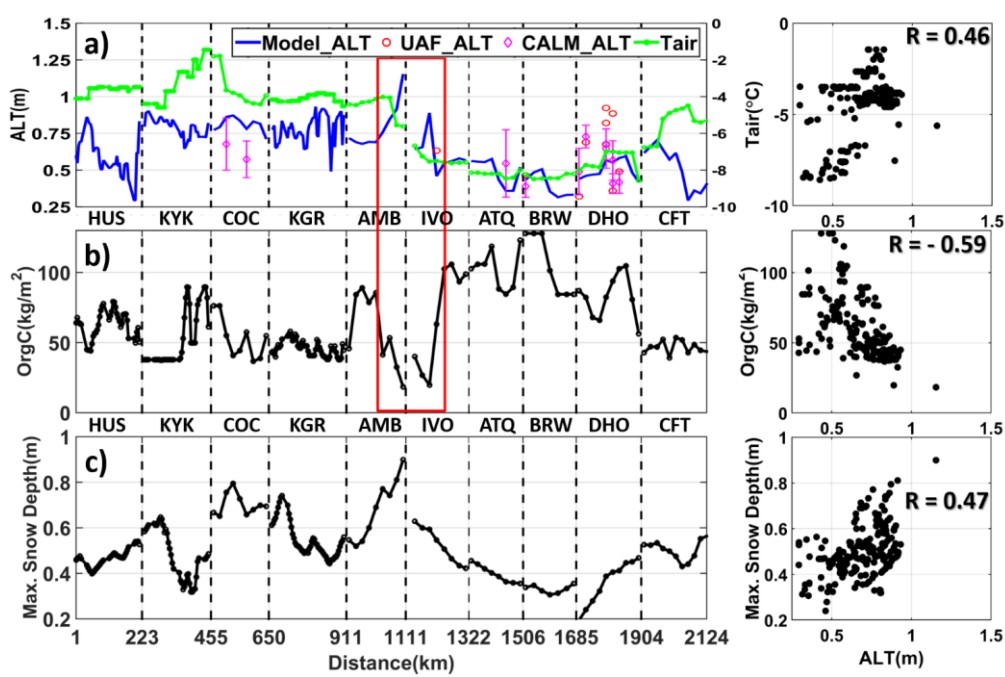

Figure 5: a) CLSM-simulated ALT (thawed-to-frozen depth) on 29 August 2015 along the AirMOSS flight transects. ~~In-situ~~In-situ ALT observations from UAF and CALM are shown as red circles and magenta diamonds, respectively. Averaged air temperature at 2 m (Tair) from the preceding annual period (i.e., 01 September 2014 to 31 August 2015) is shown in green with the scale on the right ordinate. b) ~~organic~~Organic carbon content and c) maximum snow depth during the preceding annual period (again from 01 September 2014 to 31 August 2015). The red rectangle ~~crossing~~across a) and b) highlights a portion of the domain that shows ~~an~~ ~~anti-correlated relationship~~correlation between organic carbon content and modelled ALT (see Section 4.2). The abscissa in c) provides cumulative distances in units of km along the transects.

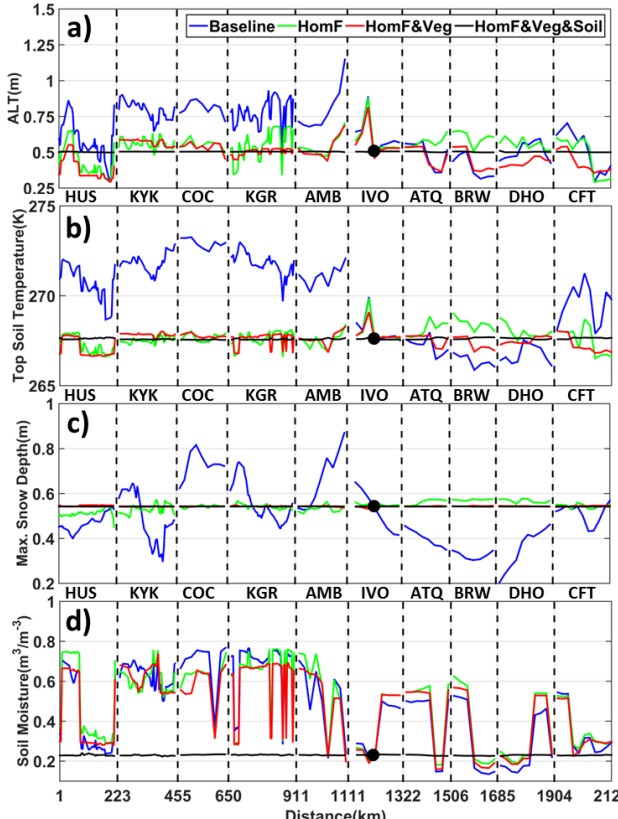

Figure 6: a) CLSM-simulated ALT (thawed-to-frozen depth) on the flight date (i.e., 29 August 2015) from the top four experiments listed in Table 2Table 2; b) simulated top layer soil temperature on the flight date, c) maximum snow depth the during the preceding annual period (i.e., from 01 September 2014 to 31 August 2015), and d) soil moisture within the soil profile on the flight date along the connected transects for the four experiments. The black dot indicates the representative location within the IVO transect from which the forcing, vegetation and/or soil data are used to homogenize the inputs in the idealized experiments. By construction, all simulations provide identical results at this representative location.

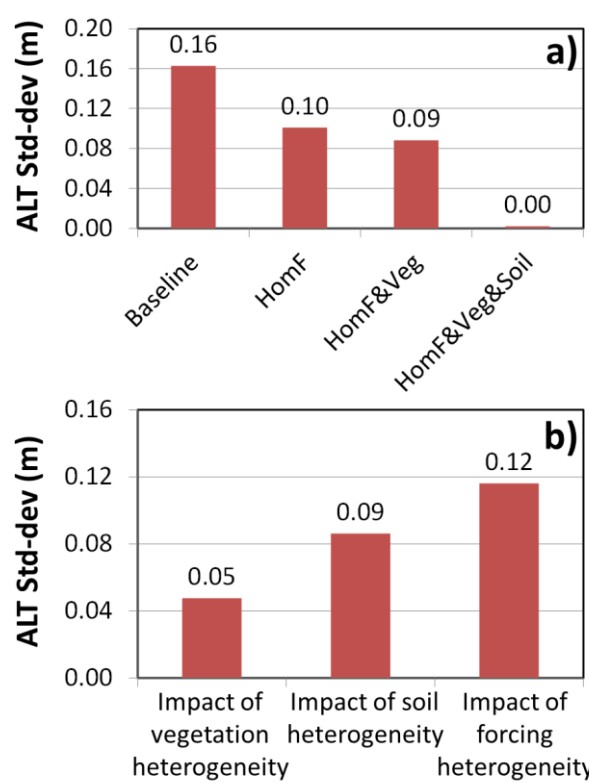

Figure 7: a) Standard deviation of ALT along the AirMOSS transects from the top four experiments listed in Table 2Table 2. Subplot b) shows theThe individual impact (or contribution) from the heterogeneous vegetation, soil type and meteorological forcing, respectively. For instance, the impact of vegetation (or soil, or forcing) heterogeneity is the ALT standard deviation along the transects from HomF&Soil (or HomF&Veg, or HomVeg&Soil).

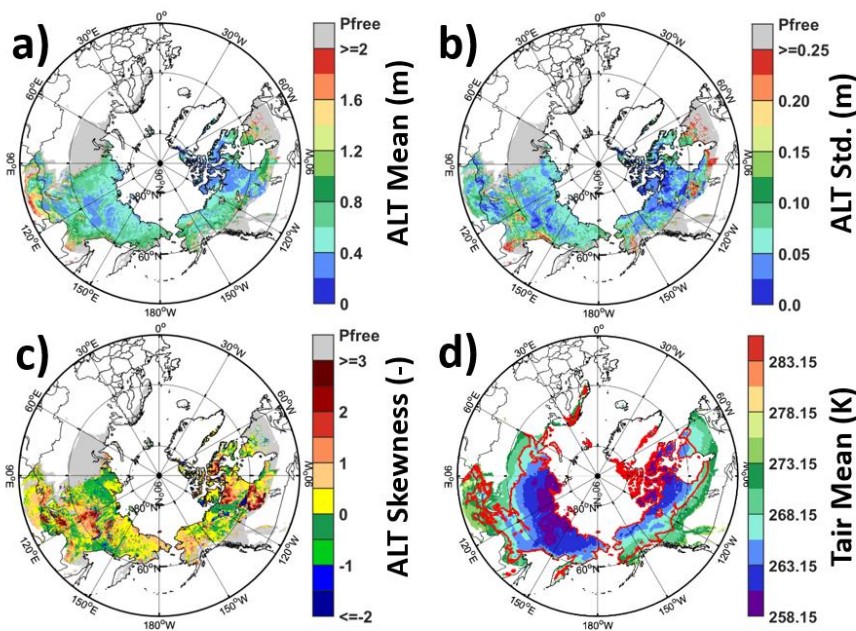

**Figure 8: a) Mean, b) standard deviation, and c) skewness of CLSM-simulated ALT over the 38 years (1980 - 2017). Grey indicates permafrost-free (Pfree) areas in the simulation. d) 38-year averaged MERRA-2 annual atmospheric temperature at 2 m above displacement height (Tair). The red boundary outlines the continuous and discontinuous permafrost regions ~~of Brown's map.~~according to Brown et al. (2002).**

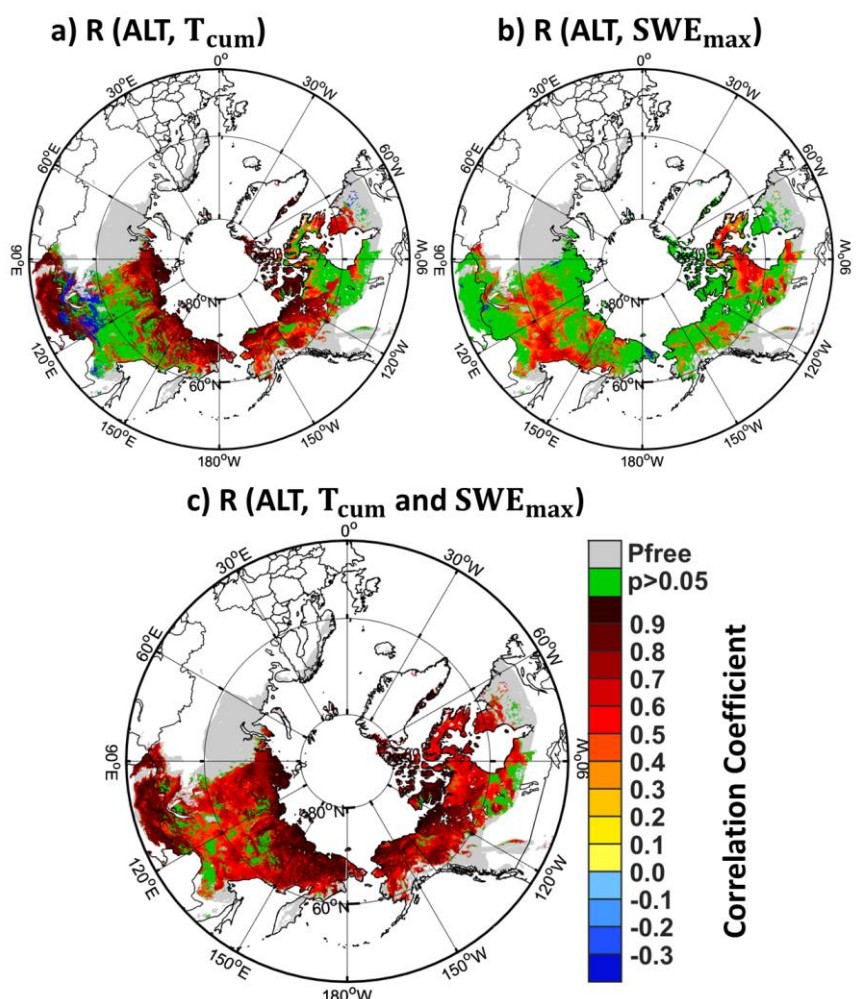

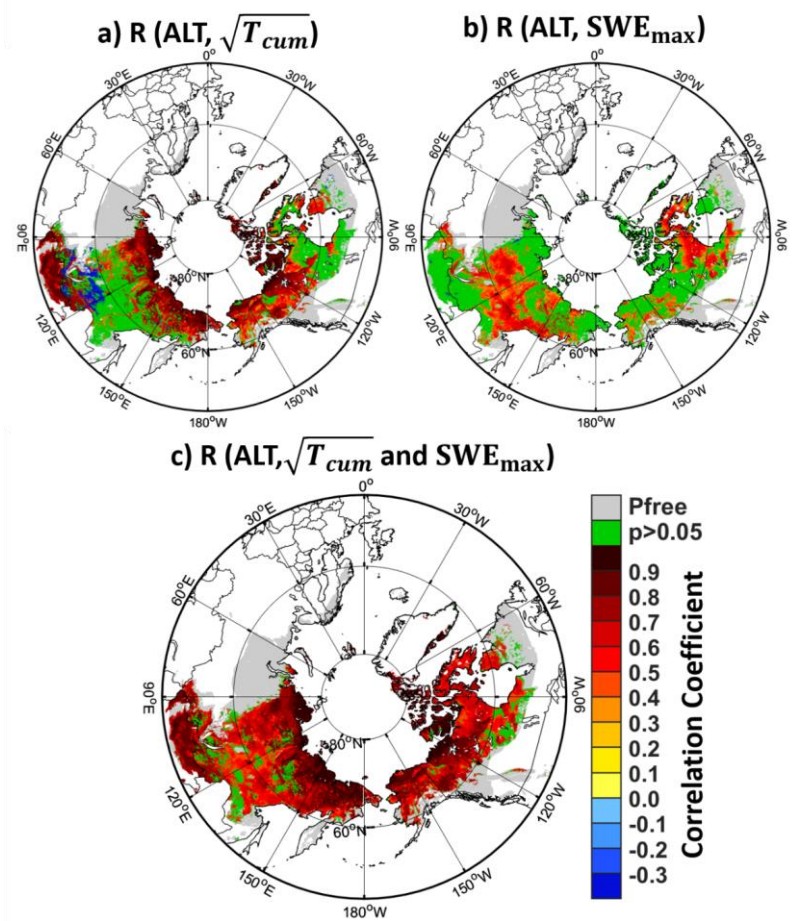

**Figure 9:** Correlation coefficient between a) ALT and ~~square root of the~~ effective accumulated air temperature (~~$T_{cum}$~~)$\sqrt{T_{cum}}$) and b) ALT and maximum SWE ($SWE_{max}$) from the preceding September to the present August over the period 1980-2017. c) Multi-variable coefficient of correlation for a fitted multiple linear regression model between ALT and ~~the accumulated temperature and the maximum SWE,~~ $\sqrt{T_{cum}}$ and $SWE_{max}$. Areas that have a p-value larger than 0.05 (i.e., statistically insignificant correlation) are masked in green. Grey indicates permafrost-free (Pfree) areas in the simulation.

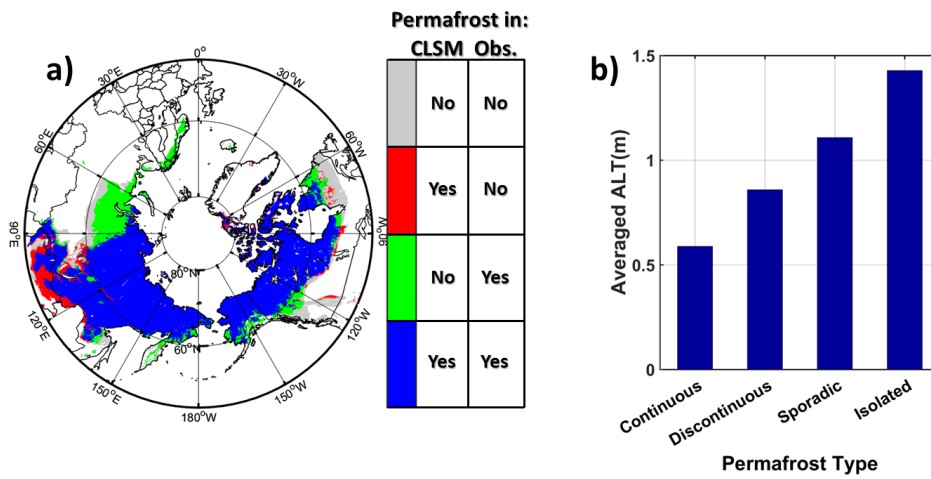

Figure 10: a) Four comparison categories include: 1) blue - CLSM collocates permafrost with the observation-based permafrost map of Brown et al. (2002) as either continuous, discontinuous, or sporadic permafrost; 2) green - CLSM has no permafrost, but the observation-based permafrost map does as either continuous, discontinuous, or sporadic types; 3) red - CLSM does have permafrost, but the observation-based permafrost map does not or contains isolated permafrost; and 4) grey - CLSM has no permafrost and neither does the observation-based permafrost map (except for isolated permafrost). b) area-weighted average of ALT as simulated by CLSM for the four different permafrost types.

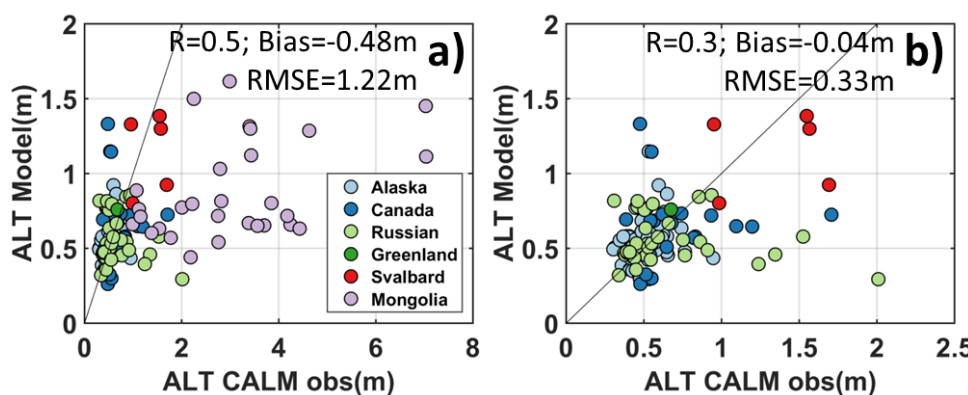

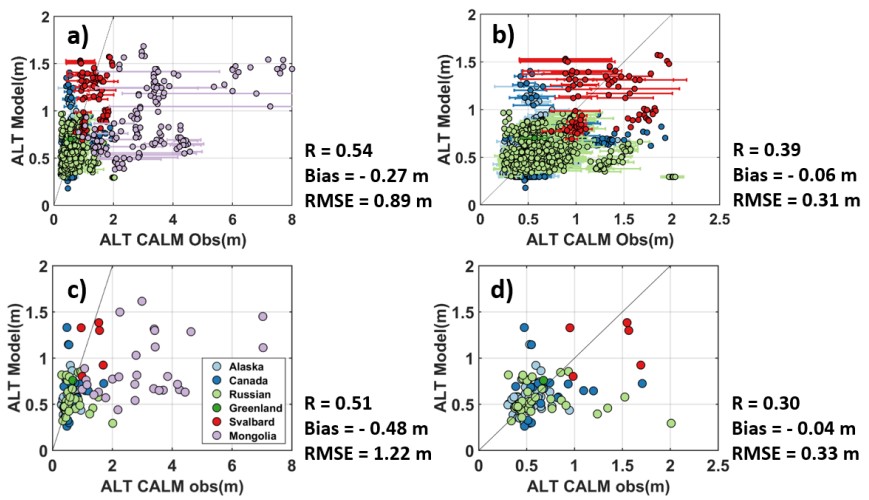

**Figure 11:** a) Annual ALT from CLSM simulation vs. CALM observations with horizontal error bars indicating standard deviations of measurements within the model grid cell. Error bar is absent if the number of measurements within a 81 km² grid cell is less than three. b) As in a) but excluding the Mongolia sites.  ac) 38-yr average ALT for the period 1980-2017 from CLSM simulation vs.

Formatted: Font: 9 pt

Formatted: Font: 9 pt

Formatted: Font: 9 pt

Formatted: Font: 9 pt

CALM observations. ~~b~~d) ~~Same as~~As in ~~a~~c) but without the Mongolia sites. The correlation coefficient (R), bias, and root mean squared error (RMSE) are provided ~~on~~next to each subplot.

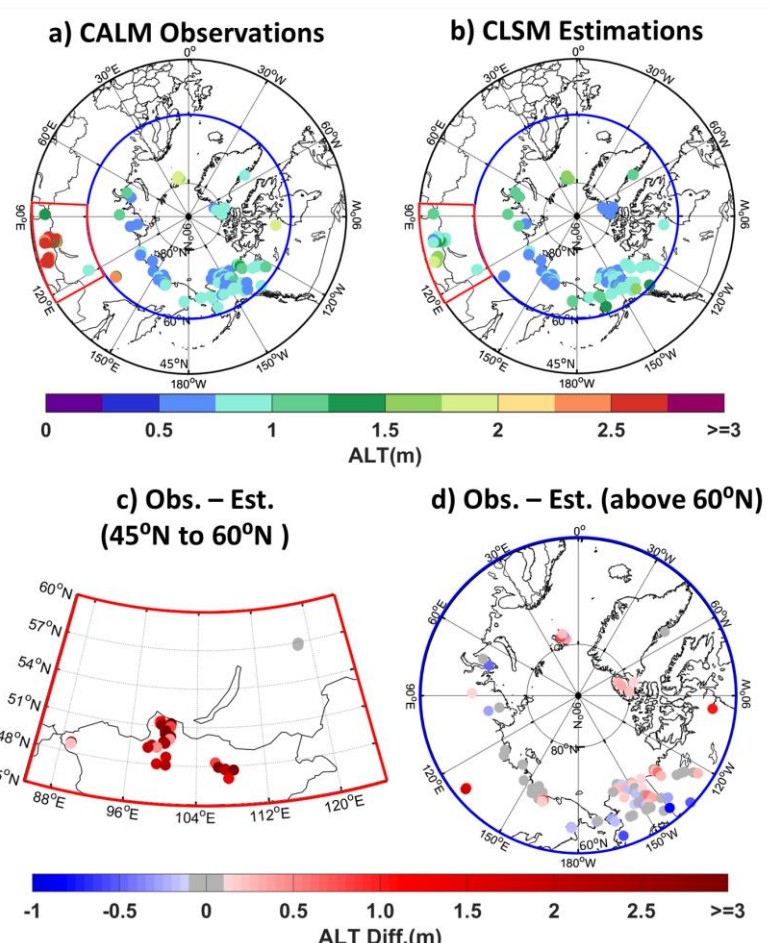

**Figure 12: Multi-year average ALT at CALM site locations for a) CALM observations and b) CLSM results. c) ALT difference between observations and model results for locations within 45ºN- 60ºN latitude and 85ºE-125ºE longitude. d) Same as c) but for locations poleward of 60ºN latitude. In c) and d) grey indicates absolute ALT differences less than 0.10 m.**

Formatted: Caption, Line spacing: Double

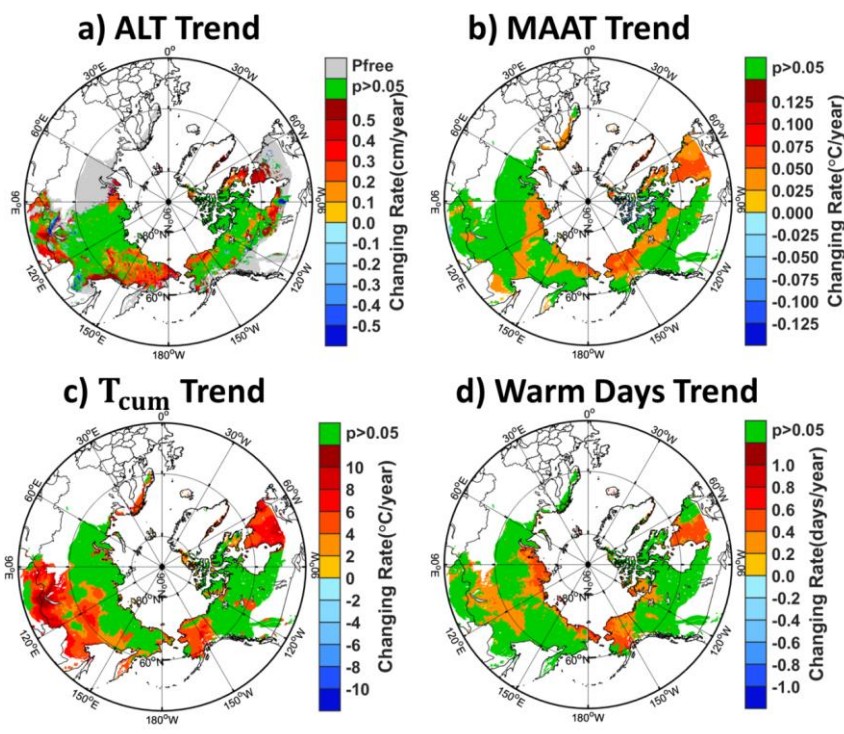

Figure 13: 1980-2017 trend in a) ALT, b) mean annual air temperature (MAAT), c) effective accumulated air temperature ($T_{cum}$), and d) warm days from CLSM simulations. Areas that have p-values larger than 0.05 (i.e., no statistically significant trend) are shown in green. In a), grey indicates permafrost-free (Pfree) areas in the simulation.

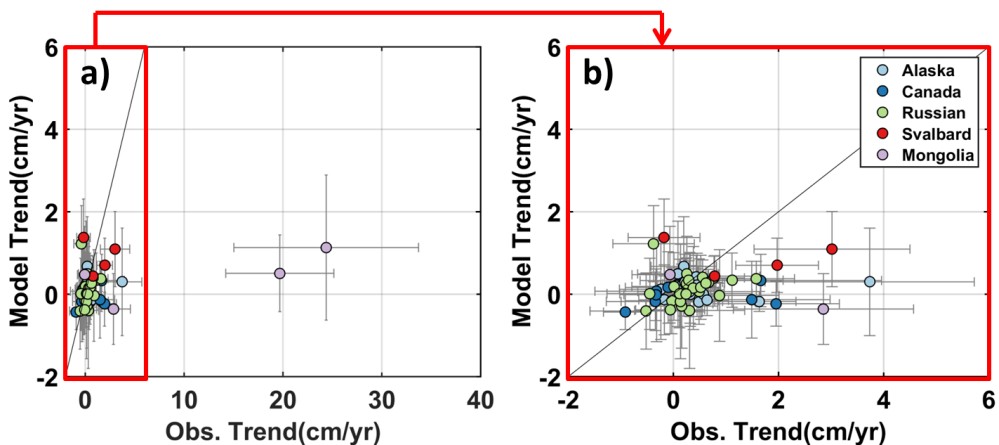

Figure 14: a) ALT trend from CLSM estimates vs. that from CALM observations, based on years common to both datasets within the period of 1990 to 2017. The horizontal and vertical error bars represent 95% confidence intervals of observed ALT trend (regression slope) and CLSM-simulated ALT trend, respectively.  b) Same as a) but zoomed into observed ALT trends between -2 and 6 cm/yr. Note: the trends plotted here are not filtered based on statistical significance. Only four and ten sites have observed and estimated ALT trends with p value less than 0.05 and 0.10, respectively.