# Peer review of "Permafrost Variability over the Northern Hemisphere Based on the MERRA-2 Reanalysis"

_The Cryosphere, 2018_

## Referee Comment (RC1) · Anonymous Referee #1 · 14 Jul 2018

This paper provides an evaluation of active layer thickness and permafrost extent as simulated by the NASA Catchmant Land Surface Model driven by MERRA-2. The model-generated dataset of permafrost conditions is evaluated again site data, global data and remotely (plane) sensed ALT. The comparison to the remotely sensed ALT is probably the most innovative part of the paper, but it also suffers from some drawbacks because the remotely sensed data conspicuously lack spatial variability. The paper is written clearly, the analysis is honest (not obviously trying to hide model shortcomings – but sometimes the assessment of the dataset quality seems a bit too optimistic), the figures are relevant and informative. The paper is a useful contribution, but some aspects detailed in the following could be improved.

General:

[Figure]

- There are lots of global permafrost simulations that are driven by reanalysis-based meteorologies. What is really the added value of this one? The fact that is uses MERRA? In that case, could you say more about specific strengths and weaknesses of MERRA, please? More generally, simulations with other metorological forcing data, and comparison to other model-generated permafrost data sets (e;g. within the Permafrost Carbon Network) could be interesting.

- Some words about potential uses of this dataset could be nice.

Specific points:

- Page 2, line 15: "simulations. . . with the land surface model ( Dankers et al., Guimberteau et al., Tao et al.)" -> these are different models. The sentence is misleading, and confusingly, you write ". . .and other numerical models" afterwards. . .

- Page 2, line 20-24: Strictly speaking, the fact that 2017 set records doesn't mean that permafrost conditions will change. 2017 is only one year. It's the long-term trends that matter (2017 is of course consistent with that trend)

- Page 2, line 23: "Some aspects of the current global permafrost thermal states are ... still unknown": can you elaborate on that, please?

- Page 3, line 10: "extensive challenges" sounds bizarre to my non-native speaker's ears

- Page 3, line 16: Could you say a few words specifically about high-latitude performance? Advantages, drawbacks // other reanalyses?

- Page 3, line 26: Chen et al. is a paper in review. Can you reassure the reviewer that these retrievals are independent from the data produced here? One or two sentences would be nice anyway even if Chen et al. 2018 will be available to the reader soon.

- Page 5, line 13: because you later speak about the spinup in the trend analysis, it might be interesting to say a few words about this here. The looping through the 36

years cannot given the same soil temperatures as you would normally have if you have realistic spinup data.

- Page 8, line 15 and following: The assessment might be a bit too optimistic here: Basically one sees that the ALT is between 0.2 and 1m both in obs and simulations, not much more. Is there a significant correlation at all?

- Page 8, line 21 and following: The AirMOSS ALT retrievals. Bascically the retrievals are the same everywhere! Around 0.45 m. No variability. Are they actually of any use?

- Page 8, line 32 and following ("Excluding..."): Yes, OK, but then there is still no correlation. Values are just around 0.45 m and the mean ALT of the remaining sites just happens to be around that value.

- Page 9, line 9-10: "Further investigation...": You could nevertheless elaborate a little bit on this. Are there any common characteristics of sites with thick active layer (dry soil, highly conducive soil, southward sloping, etc.) that the model doesn't get?

- Page 11, line 1-3 – meteorological forcing dominant control: Of course. Could anyone seriously expect something different?

- Figure 7: Good that this is quantified in such a way here. Much more synthetic & interesting than figure 6.

- Page 13, line 12: Correlation might increase if time steps with snow on the ground but air temp > 0°C are not counted in Tcum - it's the soil surface temperature that counts, not the air temperature.

- Page 14, line 5 : Problems in mountain areas: Snow forcing might be severely in error in these regions

- Page 14, line 12: "The reasons..." – I have probably missed the information: How deep is the model soil column?

- Page 16, line 15, Mongolian ALT trends: How can you have a 25 cm/year trend over

17 years? That would mean that ALT increases by over 4 m over that period. That's quite improbable. These data are very suspicious.

- Page 18, line 20 : Âń ..addition of soil layers Âż : Would that be so difficult? Tests with more levels would really be interesting, but if they are really difficult, I refrain from asking for such test to be carried out.

————————————————

---

## Referee Comment (RC2) · Anonymous Referee #2 · 27 Jul 2018

Overall comments:

This paper used in-situ data and a remote sensing based ALT (active layer thickness) data to evaluate a model-based ALT dataset. Overall, I think it is a useful study. The analysis was done in a comprehensive way, and the results were acceptable. However, the analysis regarding the model uncertainty is somewhat general – considering we already have a good knowledge of the ability of global land models in NH permafrost simulation. I think, the study could benefit from more in-depth discussions on this aspect. The details were provided below.

Major comments:

1. Page 4 Paragraph 3: I have questions regarding how the ALT was calculated. The

paper indicates here it was calculated based on the simulated ice content. Does the model consider unfrozen water in frozen soils? If it does, please provide information on how the model calculates unfrozen water content. If not, this definition will be same as using a $0°C$ temperature threshold for thawed-to-frozen depth calculation. This information is especially important for the deep soils due to year-round low temperatures and coarse vertical resolution of the model at deeper depths.

The above declaration seems contradictory to "The use of the $0°C$ degree threshold in CLSM for determining the thawed or frozen state of the soil may explain the model's underestimation of ALT." (Page 9, Paragraph 1). So I am confused what methods were actually used to determine the thawing depth/ALT. Please clarify.

2. Page 5 Paragraph 3: The spin-up scheme is questionable, though the authors themselves acknowledged this. Why do the authors using the meteorology for the entire 36-year period for spin-up? If the design is to reduce the uncertainty introduced by a single-year surface meteorology, spin-up using the first few years during the period will be more acceptable.

3. I have questions regarding the vegetation effects on permafrost simulation in Northern Alaska (Page 11, Paragraph 2). Those 4 northern flights were dominated by "dwarf trees" as indicated by Fig. 2b (really?). Moreover, the changes in simulated maximum snow depth due to vegetation in those flights were much smaller comparing with the experiment homogenizing the forcing data (Fig. 6c). So I would expect the impact due to snow changes for the homogenizing vegetation experiment would be smaller comparing with the experiment homogenizing surface forcing, while this is not the case shown in Fig. 6a-b. Can the authors explain why?

On the other hand, the Alaska North slope is dominated by tundra, while the vegetation map in CLSM indicates mostly dwarf trees or shrubs (Fig. 2b). Would this introduce uncertainties to the analysis on the contribution of different factors (i.e. forcing data, vegetation and soil) in Fig. 6?

Also, from Fig. 6, it does not seem to me that homogenizing soil parameters has much bigger impact on simulated ALT and surface soil temperature than homogenizing vegetation (at least for the northern flights), as the authors indicated in Fig. 7. Maybe I miss something here?

4. Most of the value for AirMOSS radar retrievals, I think, is in its ability to characterize land surface heterogeneity. Simply averaging AirMOSS data to a much coarse-resolution (i.e. 9km in this study) to compare with the land model simulations is not very insightful in terms of exploring the value of this dataset. I agree with the authors that the current AirMOSS retrievals seem having large uncertainties; most notably, its ALT retrievals were in a very narrower range. However, the inconsistency of the ALT spatial pattern at some of AirMOSS flights may come from the model itself. For example, at the DHO flight—this is the flight with most in-situ sites available, the model ALT generally increases from the north to the south, while the in-situ data show large variability, and do not show a clear increasing trend from the north to the south (Fig. 5a). There are also a number of studies pointing out that ALT is extremely variable at local scale. Therefore, analysis using a dataset like AirMOSS in this aspect would be more valuable.

5. It would be more interesting if the authors could provide more insightful analysis regarding the model ALT uncertainties or the correlation analysis, including:

a) Why does the model show much stronger correlation with maximum SWE in portions of NH permafrost region than with air temperature? b) It would be helpful if the authors could give more explanations regarding why the model fails in western Siberia? Those areas also include continuous permafrost, so I do not think it is too challenging for global models to capture the permafrost distribution there. c) The ALT trends shown in Fig. 13a seem not very consistent with the trends of temperature indices shown in this figure. Have the authors explore the changes in snow cover duration? A longer snow free season generally leads to warmer soil temperature and thus deep ALT esp. in the southern area. d) Page 18, Paragraph 3: I do not quite agree with the authors'

explanation why the model fails in the Mongolian sites. For those sites, the model simulated ALT is generally less than 1.4 m, therefore, the coarse resolution at deeper soils in the model set-up (i.e. layer 5: 1.4-3m) should not be a major contributing factor there. Much drier conditions, sparse vegetation, and perhaps uncertainties in soil texture data (eps. in deep soils), I think, are more likely contributing more to the model uncertainties.

Minor comments:

1. Fig. 4: The text in Section 2.3 indicates most of the comparisons against AirMOSS data will be at the 4 flights in the Alaska north slope. So I wonder why COC flight was included in this figure esp. considering the AirMOSS retrievals were not included?

2. Fig. 4 a-b: were the sites arranged according to the latitudinal changes?

3. Page 9, Line 21: Does MERRA-2 not provide air temperature at 2-m surface height?

4. Section 3.2 and Figure 6: Part of the IVO flight lies in the Brooks mountain range with very low SOC content (Fig. 2c). It may be not a good representative of the average conditions in this area, at least considering SOC variability.

5. Section 3.2: Would it be easier to follow if the description regarding the idealized experiments was included in the Methods section?

6. Fig. 7 is not very informative. I suggest summarizing the results in a table.

---

## Referee Comment (RC3) · Anonymous Referee #3 · 14 Aug 2018

The authors compare model estimates (based on re-analyses data) of active layer thickness (ALT) with different data sources. This includes in situ measurements from an international database and an unpublished approach of airborne P-band estimation. The latter is available over Alaska. General evaluation is made for the northern hemisphere.

In addition, the model input is investigated for linear relationship with its output. The authors seek to find a simplified relationship to explain ALT change over time (driven by degree days, snow water equivalent etc).

General issues

[Figure]

The manuscript addresses an important topic, the modelling of active layer thickness over time. The presentation of the manuscript and setup of the experiment is however problematic.

1) The authors report 'reasonable' or 'good' results with limited and partially missing quantitative reasoning.

2) The results of the comparison to the unpublished airborne (AirMOSS) approach suggest that it actually does not work (AirMOSS values are all at the same level, not representing the in situ range, Figure 4). The comparison to this unpublished approach should be removed from the manuscript.

3) The stated objectives comprise issues regarding active layer thickness. The results section however does also cover permafrost extent.

4) In addition, common structure of methods, results and discussion is lacking. Methods are included in the introduction and mostly in the results section. The discussion is largely included in the results section. The 'approach' section mostly only includes the dataset description.

5) Some of the used datasets are not appropriately cited. See detailed comments below

6) A main weakness of the manuscript is that the results are not discussed with respect to already published material on permafrost related parameter modelling and on the parameters which are investigated for explanation as drivers of ALT change over time. The role of temperatures and snow water equivalent for active layer thickness is known (what is presented as one of the main results in the abstract). The novelty of the paper is not clear.

Detailed comments

Title: the title is misleading; the paper focuses on active layer thickness what should

be reflected in the title

Abstract:
Line 19: 'measurements demonstrates reasonable skill' – what is 'reasonable'? add numbers
Line 27: 'significant degradation, with ALT increasing up to 0.5 cm/year' – it should be noted that this cannot be confirmed with in situ observations

Page 5, line 9: NCSCDv2 – citation missing, follow instructions of https://bolin.su.se/data/ncscd/
Page 6, line 24: Brown et al. 2002: the reference is missing in the list

Page 7, line 30 following: this paragraph belongs to methods
Page 8, line 9: remove 'relatively'
Page 9, first paragraph: this belongs to discussion
Page 9, line 17 following: introduce this comparison in the methods section
Page 9, line 22: provide correlation analyses results as table
Page 9, line 23: which SOC value did you use? Is it representative for the upper soil layer?
Page 10, line 4-28: this all belongs to the methods section
Page 11, line 7: spell out LAI
Page 12, line 6-12: move to methods
Page 12, line 19: 'permafrost areas shown in Fig. 1b are well confined within the cold side ..' this cannot be seen. Add outlines to map
Page 13, up to line 7: to methods; what is the reasoning for the static time period approach?
Page 13 – move explanation of linear regression analyses to methods
Page 14, line 14: 'geographically thin disagreements' – quantify this

TCD
Page 14, line 28 following: this is methods

Page 14, line 18: this error may seem small in absolute numbers, but ALT is much thinner than for the Mongolian sites. The error is still relatively large.

Page 14, line 20-26: this belongs to discussion

Page 15, first two paragraphs of section 3.5 – this all belongs to methods

Page 17, 21: 'shows good general agreement' – this is not really the case, quantify the agreement and compare with other published results

Page 17, line 29: 'The simulated ALTs agree well with the in-situ observations' – not really, see my comment for page14, line18; how do your results compare to other published results?

Page 18, lines 1-2: 'retrievals from airborne remote sensing for 2015 and the corresponding simulated ALT exhibit reasonable accuracy vs. in situ measurements' – this is not clear from the material presented.

Figure 5: what is the red rectangle?

Figure 7: convert to table

---

## Author Comment (AC1) · 20 Sep 2018

**Overview**

We thank the reviewers for their constructive comments and suggestions. The manuscript will be appropriately revised in response to the reviewers' comments (see the point-by-point expected responses below). As requested by the reviewers, we compared our modeling results with some existing permafrost data sets by calculating evaluation metrics that can be compared directly against matching results reported in the literature. In addition, we also conducted several new simulations that further assess the impact on ALT of the model soil layer configuration, the soil organic carbon content, and its vertical distribution.

In summary, the planned modifications to the text can be categorized as follows:

a) Novelty and added value:
   See R1C1 (i.e., Reviewer 1, Comment 1), R1C2, and R3C6

b) Comparison with other model-generated permafrost data sets:
   See R1C1, R1C2, R3C6, R3C29 and R3C30

c) Rephrasing "optimistic" discussion about ALT results:
   See R1C10, R1C11, R1C12, R3C2, R3C31

d) New sensitivity experiments and uncertainty discussion:
   See R1C3, R1C20, R2C8 and R2C12

e) Add specific evaluation metrics instead of using description words:
   See R3C6, R3C8, R3C24, R3C29 and R3C31

Throughout the discussion below, the text is colored as follows:

Black: Reviewer comment
Blue: Expected author response
Red: Expected text to be inserted into the revised manuscript

For reference, our response to comment "m" by reviewer "n" is labeled "R[n]C[m]".

Reviewer #1

This paper provides an evaluation of active layer thickness and permafrost extent as simulated by the NASA Catchmant Land Surface Model driven by MERRA-2. The model-generated dataset of permafrost conditions is evaluated again site data, global data and remotely (plane) sensed ALT. The comparison to the remotely sensed ALT is probably the most innovative part of the paper, but it also suffers from some drawbacks because the remotely sensed data conspicuously lack spatial variability. The paper is written clearly, the analysis is honest (not obviously trying to hide model shortcomings – but sometimes the assessment of the dataset quality seems a bit too optimistic), the figures are relevant and informative. The paper is a useful contribution, but some aspects detailed in the following could be improved.

We thank the reviewer for the careful reviewing. We understand the reviewer's concern about the small spatial variability with the remotely sensed ALT retrievals. The ALT retrievals were produced by the current algorithm developed by Chen et al., 2018. Somewhat larger spatial variability is presented in the original retrievals but is smoothed and reduced after aggregating to the scale of the Catchment Land Surface Model (CLSM) at 81 $km^2$, as also mentioned in the original manuscript. In addition, the radar penetration depth is not large enough to detect deeper thawed to frozen condition of the soil. All in all, while we eventually expect to further improve the retrieval algorithm, these are the results we have right now, and as discussed further below, their inclusion in this paper in their early form does, we feel, have value. We plan to tamp down our "optimistic" tone when discussing the ALT retrievals. We trust this first intercomparison of ALT among model results, remotely sensed retrievals and in-situ observations could provide useful insights to the research community. Please see specifically our response to R1C11, R1C12 and R2C7 below.

General:

- There are lots of global permafrost simulations that are driven by reanalysis-based meteorologies. What is really the added value of this one? The fact that is uses MERRA? In that case, could you say more about specific strengths and weaknesses of MERRA, please? More generally, simulations with other metorological forcing data, and comparison to other model-generated permafrost data sets (e;g. within the Permafrost Carbon Network) could be interesting.

R1C1: We thank the reviewer for encouraging us to explicitly highlight our contributions. Regarding the second, more general comment, a detailed quantitative comparison against existing permafrost data sets simulated by other land models, for example the land models participated in the Permafrost Carbon Network (PCN) (offline mode) and the Coupled Model Intercomparison Project phase 5 (CMIP5) (coupled mode), is beyond the scope of this paper, which is already a bit long. (Note that aspects of such a general analysis have already been reported in literature (Peng et al., 2016;Wang et al., 2016b;Koven et al., 2013)). Inspired by the reviewer's comment, though, we plan to add a brief discussion to

our manuscript that compares our dataset with others in terms of spatiotemporal resolution and simulated results (see our response in R3C6 for details). We will also summarize our dataset's particular strengths in our discussion section (will be the new section 5).

Specifically, we will add the discussion below.

  a. General comparison between this work and existing model-generated permafrost data sets forced by other meteorological forcing data (both in uncoupled and coupled mode):

a.1 Regarding resolution (will be added into section 1)

"Most of these land models were run at coarse spatial resolution, e.g., ranging from $0.5° \times 0.5°$ to $1.8° \times 3.6°$ for LSMs participating in the Permafrost Carbon Network (PCN) (Wang et al., 2016a) and from $0.188° \times 0.188°$ to $4.10° \times 5°$ for the models participating in the Coupled Model Intercomparison Project phase 5 (CMIP5) (Koven et al., 2013; https://portal.enes.org/data/enes-model-data/cmip5/resolution). They thus lack some of the higher resolution information implicit in our data and are, as a result, perhaps less comparable to in-situ observations taken at the point scale. Other types of numerical models have been run at relatively higher resolution, but not globally; such simulation domains were limited to regional scales (e.g., 2 km $\times$ 2 km in Jafarov et al., 2012 covering Alaska;1 km $\times$ 1 km in Gisnas et al., 2013 covering Norway) as necessitated by the availability of ancillary data and the heavy computational burden. A unique aspect of our contribution to the existing permafrost datasets is thus global coverage at a moderately high spatial resolution."

a.2 Regarding model performance in simulated permafrost extent (particularly the deficiency in western Siberia) (will be added into section 4.4)

"Note that some other global models, such as CLM3 and CCSM3 as reported in Lawrence et al. (2012), also missed this area of permafrost and that updated versions of these models (i.e., CLM4 and CCSM4) showed improved performance in this regard (Lawrence et al., 2012). Guo et al. (2017) reported underestimated permafrost extent simulated in western Siberia by CLM4.5 driven by three different reanalysis forcings (i.e., CFSR, ERA-I and MERRA), and they showed an improved simulation of permafrost extent in this area when using another reanalysis forcing, the CRUNCEP (Climatic Research Unit - NCEP) (Guo and Wang, 2017). Guimberteau et al. (2018) found similar improvements stemming from the use of CRUNCEP forcing. We leave for further study whether the MERRA-2 forcing data is responsible for the western Siberia deficiency seen in our own results."

a.3 Regarding model performance in simulated ALT (will be added into section 4.4)

"The existing literature on simulated ALT fields (e.g., Dankers et al. (2011), Lawrence et al. (2012) and Guo et al. (2017)) reveals a general tendency for models

to overestimate ALT climatology at the global scale. The CLSM-simulated ALT fields appear to be among the better simulation products."

b. Some improvements of MERRA-2 compared to MERRA (will be added into section 1):

"MERRA-2 has been found to be skillful in its simulation of near-surface atmospheric conditions (Reichle et al., 2017a;Reichle et al., 2017b;Bosilovich et al., 2015;Bosilovich et al., 2017) and to show improvements in the representation of cryospheric processes compared with its predecessor MERRA (Gelaro et al., 2017). In particular, MERRA-2 assimilates substantially more satellite observations and employs more physically reasonable hydrology representations for glaciated land surfaces compared to MERRA, and it also uses observation-based, seasonally-varying sea ice albedo as opposed to MERRA's fixed value of 0.6 (Gelaro et al., 2017). A recent study shows that MERRA-driven permafrost and ALT simulation results are inferior to those driven by other reanalysis-based forcing data sets, particularly those from the NOAA Climate Forecast System Reanalysis (CFSR) and European Centre for Medium-Range Weather Forecasts Re-Analysis Interim (ERA-I) (Guo et al., 2017). We note that our MERRA-2-driven permafrost simulation results, while potentially better than those we might have obtained with MERRA forcing, are still lacking (e.g., in western Siberia). The potential superiority of MERRA-2 forcing compared to MERRA forcing in the context of permafrost simulation remains unknown."

c. Summary of novelty and added value of this work (will be added into section 5):

"The permafrost dataset presented herein can be considered unique in terms of its daily temporal resolution combined with a relatively high spatial resolution at the global scale (i.e., 81 km$^2$). The dataset, which is derived from a state-of-the-art reanalysis, shows reasonable skill in capturing permafrost extent and in adequately estimating ALT climatology (aside from that at the Mongolian sites). With its resolution and available variables (ALT, subsurface temperature at different depths), it could prove valuable to many future permafrost analyses."

"This work also provides a first comparison between two highly complementary approaches to estimating permafrost: model simulation and remote sensing. The remote sensing approach is still relatively new, and many aspects still need to be worked out. It is important, though, to begin considering the modeling and remote sensing approaches side by side, as both should play important roles in permafrost quantification in the years to come. Indeed, once the science fully develops, joint use of modeling and remote sensing (e.g., through the application of downscaling

methods) should allow the generation of more accurate permafrost products at even higher resolutions."

- Some words about potential uses of this dataset could be nice.

R1C2: We thank the reviewer for this suggestion. Relevant discussion about the potential uses of this dataset will be added to the end of this manuscript:

"For example, these ALT estimates are highly relevant to the assessment of the regional water budget and can be helpful for monitoring groundwater changes at a wide variety of scales (Evans et al., 2018;Evans et al., 2015). In addition, these data can potentially contribute to ecological studies focused on the dynamics of microbial activity and soil respiration in cold regions, on vegetation migration/adaptation in response to climate change, and so on."

Specific points:

- Page 2, line 15: "simulations… with the land surface model ( Dankers et al., Guimberteau et al., Tao et al.)" -> these are different models. The sentence is misleading, and confusingly, you write "…and other numerical models" afterwards…

R1C3: We will revise the relevant sentences as follows.

"Simulations and/or predictions with a variety of land surface models (LSMs) have been used to quantify large-scale permafrost patterns (i.e., distributions and thermal states) and their interactions with a warming climate. LSMs utilized for this include, for example, the Joint UK Land Environment Simulator (JULES, Dankers et al., 2011), the ORganizing Carbon and Hydrology in Dynamic EcosystEms (ORCHIDEE) - aMeliorated Interactions between Carbon and Temperature (ORCHIDEE-MICT, Guimberteau et al., 2018), the Catchment Land Surface Model (CLSM, Tao et al., 2017), and the Community Land Model (Lawrence and Slater, 2005;Alexeev et al., 2007;Nicolsky et al., 2007a;Yi et al., 2007;Lawrence and Slater, 2008;Lawrence et al., 2008;Lawrence et al., 2012;Koven et al., 2013;Chadburn et al., 2017;Guo and Wang, 2017)."

- Page 2, line 20-24: Strictly speaking, the fact that 2017 set records doesn't mean that permafrost conditions will change. 2017 is only one year. It's the long-term trends that matter (2017 is of course consistent with that trend)

R1C4: We agree with the Reviewer regarding this point. The message we had tried to convey is that the warming trend of our climate seems to have increased in recent years and that given the associated exacerbation in permafrost thawing, monitoring permafrost in a timely manner is critical. We will modify the relevant sentences as follows. Please also see our response to next comment (R1C5).

"In addition, given the apparent climate warming seen in recent years (exemplified by the fact that the average Arctic air temperature in 2017 (ending in September) was the second warmest on record since 1900 [Arctic Report Card; http://www.arctic.noaa.gov/Report-Card/Report-Card-2017] and that 2017 was the warmest year on record for global ocean temperatures (Cheng and Zhu, 2018), important reductions in permafrost might be occurring as well."

- Page 2, line 23: "Some aspects of the current global permafrost thermal states are ... still unknown": can you elaborate on that, please?

R1C5: We will clarify this by expanding this sentence as shown below:

"However, current global permafrost thermal states (i.e., permafrost temperature, ice content and degradation rates across much of Northern latitudes) are arguably still unknown. Monitoring permafrost degradation in a timely manner is particularly critical for ecosystem management and for various policy decisions."

- Page 3, line 10: "extensive challenges" sounds bizarre to my non-native speaker's ears

R1C6: We will replace the end of the sentence with "combined with the many challenges of physical process modelling."

- Page 3, line 16: Could you say a few words specifically about high-latitude performance? Advantages, drawbacks // other reanalyses?

R1C7: We will add some relevant discussion about the performance of MERRA-2 in high-latitude regions:

"In particular, MERRA-2 assimilates substantially more satellite observations and employs more physically reasonable hydrology representations for glaciated land surfaces compared to MERRA, and it also uses observation-based, seasonally-varying sea ice albedo as opposed to MERRA's fixed value of 0.6 (Gelaro et al., 2017)."

- Page 3, line 26: Chen et al. is a paper in review. Can you reassure the reviewer that these retrievals are independent from the data produced here? One or two sentences would be nice anyway even if Chen et al. 2018 will be available to the reader soon.

R1C8: The AirMOSS radar retrievals of ALT we used here were produced by the algorithm described and analyzed in detail by Chen et al. 2018. The retrievals here are indeed identical to those produced by Chen et al. 2018, though here we examine them from a different perspective. We will add some text to distinguish the scope of this study from that of Chen et al. 2018 and to emphasize the data independence:

"Although based on the same set of ALT retrievals, Chen et al. (2018) mainly focus on the development and improvement of the retrieval algorithm, whereas this study emphasizes using the remotely sensed ALT estimates to characterize and assess the

spatial variability of the modelling results. The ALT retrievals and the modelling results are fully independent."

- Page 5, line 13: because you later speak about the spinup in the trend analysis, it might be interesting to say a few words about this here. The looping through the 36 years cannot given the same soil temperatures as you would normally have if you have realistic spinup data.

R1C9: We thank the reviewer for the suggestion. We will add two sentences here.

"One caveat about this looping procedure, by the way, is worth mentioning. Because it makes use of the warmer conditions of the last few decades, it might produce a warmer initial condition than a set of initial conditions produced with realistic historical forcing over hundreds of years (e.g., Sapriza-Azuri et al., 2018). This will affect our trend analysis, as discussed in section 4.5."

- Page 8, line 15 and following: The assessment might be a bit too optimistic here: Basically one sees that the ALT is between 0.2 and 1m both in obs and simulations, not much more. Is there a significant correlation at all?

R1C10: We will modify this sentence as shown below.

"Figure 4b, c demonstrates that in some ways, the CLSM-simulated results roughly agree, to first order, with the in-situ observations. The overall mean bias of simulated ALT relative to the in-situ measurements is -0.05 m. Nevertheless, the scatter (blue) in Fig. 4c is large, and the corresponding correlation coefficient is quite weak (0.27)."

- Page 8, line 21 and following: The AirMOSS ALT retrievals. Bascically the retrievals are the same everywhere! Around 0.45 m. No variability. Are they actually of any use?

R1C11: As we mentioned in the original manuscript, relatively larger variability with the ALT retrievals is seen at its native resolution (Figure 3a), but this larger variability was smoothed out through aggregation to the model scale at 81 km$^2$, as we expected. In addition, as also mentioned in the manuscript, these retrievals cannot exceed the P-band radar sensing depth of about 60cm, and thus for the shallow permafrost here, the averaged ALT retrievals appears to be around 0.45 m everywhere.

We emphasize that this is a first attempt to compare remote sensing ALT data with modeling results. An expected future direction is to take advantage of the detailed heterogeneity information in the remote sensing data to downscale model results directly or to improve modeling skill indirectly. We will add several sentences about the potential use of these ALT retrievals into the discussion section (see our response to R1C1, part c, above). Please also see our response to the next comment (R1C12).

- Page 8, line 32 and following ("Excluding: : :"): Yes, OK, but then there is still no correlation. Values are just around 0.45 m and the mean ALT of the remaining sites just happens to be around that value.

R1C12: With further analysis, we find that the correlation coefficient between the ALT retrievals and the in-situ observations, while small at the site scale, is larger than that for modeled ALT at the model scale, both for all sites and for sites with measured ALT below 60cm. We will discuss these findings and include the new Table 3 in the revised manuscript:

"For the AirMOSS retrievals, the overall ALT bias is -0.11 m at the site scale and -0.12 m at the model scale. While the correlation coefficient with the in-situ observations is only 0.05 at the site scale, it is 0.61 at the model scale. "

"Excluding the sites with in-situ ALT measurements that exceed 60 cm, the overall mean bias for the AirMOSS retrievals at the model scale (site scale) drops to -0.01 m (0.02 m), and the correlation coefficient at the model scale (site scale) increases to 0.64 (0.20). In contrast, the CLSM simulation results show a bias of 0.01 m and a zero correlation coefficient at the same sites."

*Table R1– Evaluation metrics for model-simulated ALT and AirMOSS retrievals for 2015. (New Table 3)*

| Metric | All sites | | | Sites with ALT measurements < 60 cm | | |
|---|---|---|---|---|---|---|
| | CLSM-simulated ALT (model scale) | AirMOSS ALT retrievals (model scale) | AirMOSS ALT retrievals (site scale) | CLSM-Simulated ALT (model scale) | AirMOSS ALT retrievals (model scale) | AirMOSS ALT retrievals (site scale) |
| RMSE (m) | 0.17 | 0.17 | 0.21 | 0.12 | 0.06 | 0.08 |
| Bias (m) | -0.05 | -0.12 | -0.11 | 0.01 | -0.01 | 0.02 |

| R | 0.27 | 0.61 | 0.05 | -0.00 | 0.64 | 0.20 |
|---|---|---|---|---|---|---|

- Page 9, line 9-10: "Further investigation: : :": You could nevertheless elaborate a little bit on this. Are there any common characteristics of sites with thick active layer (dry soil, highly conducive soil, southward sloping, etc.) that the model doesn't get?

R1C13: Here we specifically meant some investigation on the zero-curtain problem with the model. Nevertheless, we will amend the text to read:

"The use of the 0⁰C degree threshold in CLSM for determining the thawed or frozen state of the soil may explain in part the model's underestimation of ALT, as may the lack of an explicit treatment of local aspect, errors in assigned model parameters, and so on."

(Note that we will move this paragraph to the discussion section, as suggested by the Reviewer #3.)

- Page 11, line 1-3 – meteorological forcing dominant control: Of course. Could anyone seriously expect something different?

R1C14: Yes. We will add "as expected" into this sentence.

- Figure 7: Good that this is quantified in such a way here. Much more synthetic &interesting than figure 6.

R1C15: Thank you. We agree.

- Page 13, line 12: Correlation might increase if time steps with snow on the ground but air temp > 0_C are not counted in Tcum - it's the soil surface temperature that counts, not the air temperature.

R1C16: The reviewer makes a reasonable suggestion, and we thus recomputed the correlation map using this modified rule. We found that the map did not differ very much from the original one. We will keep the original figure as is.

- Page 14, line 5 : Problems in mountain areas: Snow forcing might be severely in error in these regions

R1C17: Yes. We will add one sentence here.

"In addition, MERRA-2 snow forcing might be severely erroneous in these regions."

- Page 14, line 12: "The reasons: : :" – I have probably missed the information: How deep is the model soil column?

R1C18: As mentioned in section 2.1, the depth ranges of the six soil layers are 0-0.1m, 0.1-0.3m, 0.3-0.7m, 0.7-1.4m, 1.4-3m, and 3-13m, respectively. We checked the CALM sites

over western Siberia and found that the ALT observations there are basically below 2 m. Therefore, the depth of the model's soil column is not an issue, and we will delete this speculation from the sentence. The new sentence below will instead be added.

> "The reasons for this particular deficiency are unclear; perhaps the initial thermal conditions over western Siberia were too warm, or perhaps MERRA-2 overestimates current air temperatures in this region."

- Page 16, line 15, Mongolian ALT trends: How can you have a 25 cm/year trend over 17 years? That would mean that ALT increases by over 4 m over that period. That's quite improbable. These data are very suspicious.

R1C19: We share the reviewer's concerns about the quality of these data. Below in Table R2 we provide the time series of the actual ALT measurements at the three Mongolian sites M1, M1 A and M3 (https://www2.gwu.edu/~calm/data/north.html). Because M1 and M1 A are within the same CLSM modeling grid cell, we used the average of their two time series. Time series of observed and simulated ALT at the two grid cells containing these three sites are plotted in Figure R1. The observed and simulated ALT trends at the two grid cells correspond to the two dots showing the extraordinarily large observed trends in Figure 14a in the original manuscript. Note the simulated ALT trends were calculated using ALT estimates only in years when observed ALTs are available, as also mentioned in the manuscript.

*Table R2 – Observed ALT (cm) at three Mongolian sites.*

| Site Code | 1996 | 1997 | 1998 | 1999 | 2000 | 2001 | 2002 | 2003 | 2004 | 2005 | 2006 | 2007 | 2008 | 2009 |
|---|---|---|---|---|---|---|---|---|---|---|---|---|---|---|
| M 1 | 345 | 350 | 355 | 345 | 350 | 340 | 355 | - | 375 | 365 | 380 | 380 | 370 | 350 |
| M1 A | 390 | 390 | - | 430 | 485 | 400 | 450 | 600 | 770 | 710 | 820 | 820 | - | - |
| M3 | - | 610 | 620 | 600 | 610 | 660 | 720 | 760 | 770 | 750 | 760 | 770 | 800 | - |

[Figure]

*Figure R1: Time series of observed and simulated ALT at Mongolian sites collocated with two simulation grid cells, i.e., M1&M1A (upper) and M3 (bottom). The calculated ALT trends from observations and simulation are 24.38 cm/yr and 1.13 cm/yr, respectively, for the grid cell containing M1&M1A and are 19.69 cm/yr and 0.51 cm/yr, respectively, for the grid cell containing M3.*

We attempted to contact the PI responsible for these data (Dr. Natsagdorj Sharkhuu from the Institute of Geography, Mongolian Academy of Sciences); however, the email address (provided here https://www2.gwu.edu/~calm/data/webforms/mg_f.html) is apparently obsolete, and the email delivery failed. We were thus unable to investigate further the data quality.

In any case, as indicated by the reviewer, this issue calls at the very least for a specific caveat about these data, which we will add:

> "A particular caveat is required regarding the data from the Mongolian sites, given the unusual observed trends there. Attempts to contact the data providers to attain more detailed information for data evaluation were unsuccessful, and accordingly our confidence in these particular data is limited."

- Page 18, line 20 : ´n ..addition of soil layers Â˙z : Would that be so difficult? Tests with more levels would really be interesting, but if they are really difficult, I refrain from asking for such test to be carried out.

R1C20: In response to this comment we have conducted such tests. Our general conclusions are consistent with other studies in terms of how soil configuration affects permafrost modeling (e.g., Alexeev et al., 2007;Lawrence et al., 2008;Sapriza-Azuri et al., 2018;Nicolsky et al., 2007b;Dankers et al., 2011).

[Figure]

*Figure R2: Soil configurations we have newly tested.*

[Figure]

*Figure R3: Simulated ALT results at six Mongolian sites with 5 different soil configurations. (Note the baseline soil configuration contains 6 soil layers and a total soil depth about 13 m.)*

Specifically, we tested four new soil configurations with 15 soil layers and different soil depths (ranging from 13 m to 50 m). Figure R3 reveals that increasing the number of soil layers decreases ALT climatology at the tested sites, which is consistent with previous studies (e.g., Alexeev et al., 2007;Lawrence et al., 2008;Dankers et al., 2011). The figure also demonstrates that variations in total soil depth have only a small impact on the simulation of ALT, as reported in the previous studies. However, the zero flux we employ at our lower boundary for all the simulations might influence heat transfer in deep soils and thus might decrease the impact of using deeper soils.

Based on these tests, we will include some appropriate discussion in the text, though without adding a figure:

> "Local test simulations (not shown) with alternative model configurations indicate that increasing the number of soil layers may act to decrease somewhat the simulated ALT, suggesting that our values may be a little overestimated; however, based on results from a new study by Sapriza-Azuri et al.(2018), our use of a no-heat-flux condition at the bottom boundary rather than a dynamic geothermal flux may lead to underestimates of ALT. Such uncertainties should naturally be kept in mind when interpreting our results. Our supplemental simulations also suggest that increasing the total modeled soil depth has only a small impact on simulated ALT."

We will also explicitly mention our lower boundary condition in section 2.1:

> "A no-heat-flux condition is employed at the bottom of the model's soil column."

---

## Author Comment (AC2) · 20 Sep 2018

**Overview**

We thank the reviewers for their constructive comments and suggestions. The manuscript will be appropriately revised in response to the reviewers' comments (see the point-by-point expected responses below). As requested by the reviewers, we compared our modeling results with some existing permafrost data sets by calculating evaluation metrics that can be compared directly against matching results reported in the literature. In addition, we also conducted several new simulations that further assess the impact on ALT of the model soil layer configuration, the soil organic carbon content, and its vertical distribution.

In summary, the planned modifications to the text can be categorized as follows:

a) Novelty and added value:
  See R1C1 (i.e., Reviewer 1, Comment 1), R1C2, and R3C6

b) Comparison with other model-generated permafrost data sets:
  See R1C1, R1C2, R3C6, R3C29 and R3C30

c) Rephrasing "optimistic" discussion about ALT results:
  See R1C10, R1C11, R1C12, R3C2, R3C31

d) New sensitivity experiments and uncertainty discussion:
  See R1C3, R1C20, R2C8 and R2C12

e) Add specific evaluation metrics instead of using description words:
  See R3C6, R3C8, R3C24, R3C29 and R3C31

Throughout the discussion below, the text is colored as follows:

Black: Reviewer comment
Blue: Expected author response
Red: Expected text to be inserted into the revised manuscript

For reference, our response to comment "m" by reviewer "n" is labeled "R[n]C[m]".

Reviewer #2

Overall comments:

This paper used in-situ data and a remote sensing based ALT (active layer thickness) data to evaluate a model-based ALT dataset. Overall, I think it is a useful study. The analysis was done in a comprehensive way, and the results were acceptable. However, the analysis regarding the model uncertainty is somewhat general – considering we already have a good knowledge of the ability of global land models in NH permafrost simulation. I think, the study could benefit from more in-depth discussions on this aspect. The details were provided below.

We thank the reviewer for the careful reviewing. We absolutely agree with the reviewer about the importance of model uncertainty regarding ALT estimation. To further examine model uncertainty, we conducted new tests on sensitivity to organic carbon content and its vertical distribution. We also replaced the vegetation climatology at several Mongolian sites with satellite-based, time-variant LAI to investigate the impact of inter-annual variations in vegetation. Please also see our response to Reviewer #1 (R1C21) regarding the discussion about the soil layer configuration.

Major comments:

1. Page 4 Paragraph 3: I have questions regarding how the ALT was calculated. The paper indicates here it was calculated based on the simulated ice content. Does the model consider unfrozen water in frozen soils? If it does, please provide information on how the model calculates unfrozen water content. If not, this definition will be same as using a 0_C temperature threshold for thawed-to-frozen depth calculation. This information is especially important for the deep soils due to year-round low temperatures and coarse vertical resolution of the model at deeper depths.

R2C1: In our model, the ice fraction is unity if soil is fully frozen (i.e., $T < 0^{\circ}C$ and $f_{ice} = 1$), and it is zero if the soil is fully thawed (i.e., $T > 0^{\circ}C$ and $f_{ice} = 0$). The reviewer comment refers to the situation where the soil temperature is exactly at the freezing point and the soil is partially frozen (i.e., $T = 0^{\circ}C$ and $0 < f_{ice} < 1$). In the latter case, frozen and thawed soil and water coexist. This situation always occurs during freeze-to-thaw and thaw-to-freeze transitions. This is because in soil layers that are as thick as those used in the model, the phase transition does not occur instantly (relative to the model time step). More specifically, the model uses heat content as the prognostic variable, from which the temperature and ice fraction are diagnosed. Therefore, our calculation of ALT is not the same as simply using a $0^{\circ}C$ degree threshold to determine the thawed-to-frozen depth. Rather, we identify the deepest (fully or partially) thawed layer and then calculate the thawed-to-frozen depth based on the ice fraction within the layer. We will modify the paragraph about the ALT calculation as follows.

"Precisely, the thawed-to-frozen depth is calculated as:

$$z_{bottom}(l) - f_{ice}(l, t) \times \Delta z(l), \qquad\qquad\qquad (1)$$

where layer $l$ is the deepest layer that is fully or partially thawed, $z_{bottom}(l)$ represents the depth at the bottom of layer $l$, $f_{ice}(l,t)$ is the fraction of ice in layer $l$ at time t, where $f_{ice}(l, t) \in [0\ 1]$, and $\Delta z(l)$ is the thickness of layer $l$. To identify layer $l$ we use a 0°C degree temperature threshold. Specifically, $T > 0$°C degree indicates that a layer is fully thawed, $T = 0$°C degree indicates that a layer is partially thawed, and $T < 0$°C degree indicates that a layer is fully frozen. That is, layer $l$ is the deepest layer that satisfies $T(l) \geq 0$°C. Equation (1) then expresses that the thawed-to-frozen depth is equal to the bottom depth of the layer $l$ but adjusted upward according to the ice fraction within the partially thawed layer $l$."

The above declaration seems contradictory to "The use of the 0_C degree threshold in CLSM for determining the thawed or frozen state of the soil may explain the model's underestimation of ALT." (Page 9, Paragraph 1). So I am confused what methods were actually used to determine the thawing depth/ALT. Please clarify.

R2C2: The 0°C degree is used to determine the deepest thawed layer $l$, and then the ALT is calculated by $z_{bottom}(l) - f_{ice}(l, t) \times \Delta z(l)$ as explained above (R2C1). We will modify the text as follows:

"The use of the 0°C degree temperature threshold in CLSM (along with the ice fraction) for determining the model's ALT is thus only an approximation. These and other simplifications contribute to the model's underestimation of ALT."

2. Page 5 Paragraph 3: The spin-up scheme is questionable, though the authors themselves acknowledged this. Why do the authors using the meteorology for the entire 36-year period for spin-up? If the design is to reduce the uncertainty introduced by a single-year surface meteorology, spin-up using the first few years during the period will be more acceptable.

R2C3: As the reviewer points out, we recognized this issue and discussed it in the original manuscript. No spin-up procedure is entirely problem-free. Using a shorter period for spin-up as suggested by the reviewer would exaggerate in the resulting initial conditions any anomalies that occur during the spin-up period. The ultimate solution would be to construct a realistic historical forcing dataset over hundreds of years with a dynamic geothermal flux applied to the bottom boundary of soil column (e.g., Sapriza-Azuri et al., 2018). However, this approach is hardly feasible and would still not assure absolutely correct initial conditions. Please also see our response to Reviewer #1 (R1C9).

3. I have questions regarding the vegetation effects on permafrost simulation in Northern Alaska (Page 11, Paragraph 2). Those 4 northern flights were dominated by "dwarf trees" as indicated by Fig. 2b (really?). Moreover, the changes in simulated maximum snow depth due to vegetation in those flights were much smaller comparing with the experiment homogenizing the forcing data (Fig. 6c). So I would expect the impact due to snow changes for the homogenizing vegetation experiment would be smaller comparing with the experiment homogenizing surface forcing, while this is not the case shown in Fig. 6a-b. Can the authors explain why?

R2C4: As we mentioned in the manuscript, the homogenization is applied cumulatively. Before we homogenized vegetation in this experiment, we already homogenized the forcing. The differences between HomF and HomF&Veg are then attributed to the changes in vegetation parameters (specifically LAI and vegetation height).

Fig.6a-b indeed demonstrates that the impact due to snow changes for the homogenizing vegetation experiment (differences between HomF and HomF&Veg) would be smaller compared to the experiment homogenizing surface forcing (differences between Baseline and HomF). Figure 7b also illustrates this in a quantitative way.

On the other hand, the Alaska North slope is dominated by tundra, while the vegetation map in CLSM indicates mostly dwarf trees or shrubs (Fig. 2b). Would this introduce uncertainties to the analysis on the contribution of different factors (i.e. forcing data, vegetation and soil) in Fig. 6?

R2C5: The vegetation class is only one of several model inputs. The land cover class used in the study is derived from the USGS Global Land Cover Characteristics Data Base Version 2.0 (GLCCv2). In addition to vegetation class, the model uses vegetation height, leaf-area index (LAI), greenness fraction and albedo, which are all obtained from other satellite-based sources that reflect realistic conditions for tundra. Put differently, while the modeled vegetation class may suggest the presence of dwarf trees, the typically low (satellite) LAI values in northern Alaska will instruct the model that the tree cover is extremely sparse in this region. Please refer to Table 1 in (Tao et al., 2017) to see all the data sources. We will add one sentence to Section 4.1 to clarify this.

> "Note that although the vegetation class (Figure 2b) suggests the presence of dwarf trees over the Alaska North Slope, the actual satellite-based LAI, vegetation height, greenness fraction and albedo will still instruct the model that the tree cover is extremely sparse in this region. The data sources for these vegetation-related boundary conditions can be found in Table 1 in Tao et al. (2017)."

Also, from Fig. 6, it does not seem to me that homogenizing soil parameters has much bigger impact on simulated ALT and surface soil temperature than homogenizing vegetation (at least for the northern flights), as the authors indicated in Fig. 7. Maybe I miss something here?

R2C6: We will add the following discussion to Section 4.2.

> "Also, over the northern transects (ATQ, BRW and DHO) the soil impact on ALT (difference between HomF&Veg&Soil in black and HomF&Veg in red) appears smaller than that of the vegetation (difference between HomF in green and HomF&Veg in red), as shown in Figure 6a. But the integrated impact along the transects as shown in Figure 7b indicates that the soil influence clearly outweighs the influence of vegetation, since the changes in vegetation parameters do not have much impact at several other transects, including HUS, KYK, COC, AMB, IVO

and the first half of ATQ, where the vegetation conditions might be similar to those used for homogenizing. "

4. Most of the value for AirMOSS radar retrievals, I think, is in its ability to characterize land surface heterogeneity. Simply averaging AirMOSS data to a much coarseresolution (i.e. 9km in this study) to compare with the land model simulations is not very insightful in terms of exploring the value of this dataset. I agree with the authors that the current AirMOSS retrievals seem having large uncertainties; most notably, its ALT retrievals were in a very narrower range. However, the inconsistency of the ALT spatial pattern at some of AirMOSS flights may come from the model itself. For example, at the DHO flightâ˘Aˇ Tthis is the flight with most in-situ sites available, the model ALT generally increases from the north to the south, while the in-situ data show large variability, and do not show a clear increasing trend from the north to the south (Fig. 5a). There are also a number of studies pointing out that ALT is extremely variable at local scale. Therefore, analysis using a dataset like AirMOSS in this aspect would be more valuable.

R2C7: We agree with the value of AirMOSS radar retrievals in terms of (theoretically) being able to represent the spatial variability of ALT. Note, however, that Figure 4 also compares the radar retrievals at the site scale with in-situ observations and demonstrates that the radar retrievals exhibit too little variability also at their native resolution.

The differences in the spatial patterns of the AirMOSS ALT retrievals and the simulated ALT suggest that neither radar remote sensing nor modeling is perfect. As we mentioned in the manuscript, the radar sensing depth (about 60cm) strongly constrains the retrieval accuracy. We expanded on the analysis by adding a new Table 3, which provides several evaluation metrics for ALT restricted to less than 60cm. The table suggests that the radar retrievals are in better agreement with in-situ observations especially at model scale when only using sites that have ALT less than or equal to 60cm. We will add several sentences to Section 4.1:

"Excluding the sites with in-situ ALT measurements that exceed 60 cm, the overall mean bias for the AirMOSS retrievals at the model scale (site scale) drops to -0.01 m (0.02 m), and the correlation coefficient at the model scale (site scale) increases to 0.64 (0.20). In contrast, the CLSM simulation results show a bias of 0.01 m and a zero correlation coefficient at the same sites."

Please also see our response in R1C12.

5. It would be more interesting if the authors could provide more insightful analysis regarding the model ALT uncertainties or the correlation analysis, including:

a) Why does the model show much stronger correlation with maximum SWE in portions of NH permafrost region than with air temperature?

R2C8a: Good question. We will add a sentence into Section 4.3 of the revised manuscript.

> "One possible explanation is that the warming impact of the current climate has not yet had an impact on subsurface heat transfer over these areas because the insulation provided by the snow pack prevents such an influence."

b) It would be helpful if the authors could give more explanations regarding why the model fails in western Siberia? Those areas also include continuous permafrost, so I do not think it is too challenging for global models to capture the permafrost distribution there.

R2C8b: As indicated in Figure 1b, all four types of permafrost (i.e., continuous, discontinuous, sporadic and isolated) are present in the western Siberia. The literature suggests that other global models also missed this portion of permafrost, including CLM3 and CCSM3, although the updated versions of these models (i.e., CLM4 and CCSM4) demonstrated improved performance (Lawrence et al., 2012). Similarly, Guo et al. (2017) also reported underestimations in permafrost extent in western Siberia simulated by CLM4.5 when driven with three different reanalysis forcings (CFSR, ERA-I and MERRA) and improved performance when using forcing data from a different reanalysis. We will add some discussion to compare our simulation with other existing works:

> "Note that some other global models, such as CLM3 and CCSM3 as reported in Lawrence et al. (2012), also missed this area of permafrost and that updated versions of these models (i.e., CLM4 and CCSM4) showed improved performance in this regard (Lawrence et al., 2012). Guo et al. (2017) reported underestimated permafrost extent simulated in western Siberia by CLM4.5 driven by three different reanalysis forcings (i.e., CFSR, ERA-I and MERRA), and they showed an improved simulation of permafrost extent in this area when using another reanalysis forcing, the CRUNCEP (Climatic Research Unit - NCEP) (Guo and Wang, 2017). Guimberteau et al. (2018) found similar improvements stemming from the use of CRUNCEP forcing. We leave for further study whether the MERRA-2 forcing data is responsible for the western Siberia deficiency seen in our own results."

c) The ALT trends shown in Fig. 13a seem not very consistent with the trends of temperature indices shown in this figure. Have the authors explore the changes in snow cover duration? A longer snow free season generally leads to warmer soil temperature and thus deep ALT esp. in the southern area.

R2C8c: We did examine the trend in snow cover duration (see Figure R4). While some areas show a trend in snow cover duration, this trend does not seem correlated with the trend in ALT.

[Figure]

*Figure R4: Spatial distribution of trend in snow persistence days when daily mean snow depth > 25cm.*

Accordingly, we plan to modify the relevant sentence as follows:

"It is possible that in such cases, the computed trends are strongly affected by snowpack variability, even though maximum SWE itself does not tend to show a significant trend in these areas (not shown), neither the snow cover duration (not shown). "

d) Page 18, Paragraph 3: I do not quite agree with the authors' explanation why the model fails in the Mongolian sites. For those sites, the model simulated ALT is generally less than 1.4 m, therefore, the coarse resolution at deeper soils in the model set-up (i.e. layer 5: 1.4-3m) should not be a major contributing factor there. Much drier conditions, sparse vegetation, and perhaps uncertainties in soil texture data (eps. in deep soils), I think, are more likely contributing more to the model uncertainties.

R2C8d: The reviewer is correct about the soil configuration. We conducted new tests using different soil configurations at several Mongolian sites. Please see the results and discussion in our response to the Reviewer #1 (R1C21).

The reviewer also raised a very good point regarding the influence on ALT of soil wetness, vegetation and uncertainties in soil texture for deep soils. We first examined realistic satellite-based LAI data at from Moderate Resolution Imaging Spectroradiometer (MODIS) MCD15A2H product and the Advanced very-high-resolution radiometer (AVHRR) AVH15C1 product (see Table R3). Figure R5 then shows the time series of the MODIS and AVHRR LAI, along with the LAI climatology used in the model at one CALM Mongolian site (M11). A post-processing procedure that included quality screening and gap filling was applied to the two satellite LAI products. The CLSM LAI climatology is used for the years that MODIS data is not available (1980 to 2002).

*Table R3 – Information of satellite-based LAI products from MODIS and AVHRR.*

| Sensor | Dataset | Product | Resolution | Temporal Granularity | Temporal Extent |
|--------|---------|---------|------------|----------------------|-----------------|
| **MODIS** | MCD15A2H | Leaf Area Index and Fractional Photosynthetically Active Radiation | 500 m | 8-day Composites | July 2002 - Present |
| **AVHRR** | AVH15C1 | Leaf Area Index and Fraction of Absorbed Photosynthetically Active Radiation | 0.05deg | Daily | June 1981 - Present |

[Figure]

*Figure R5: The LAI time series at one Mongolian site (M11). Green line represent the original LAI climatology used in the CLSM. Blue and red dash line represent the realistic (time-varying) LAI data from MODIS and AVHRR.*

As illustrated by Figure R5, MODIS shows smaller LAI than AVHRR over the valid period after 2002. The LAI climatology used in the model is inbetween of the two products. The differences between the LAI climatology used in the model and realistic LAI products would cause differences in energy and water partitioning at the land surface via impacting surface albedo. We conducted a simple test to examine the impact of using the more realistic, inter-annually varying vegetation inputs on the winter surface albedo and thus the snow accumulation process, which in turn would impact ALT estimates. Specifically, we replaced the LAI climatology with the satellite-based, inter-annually varying LAI products in the model, but turned off the impacts in summer, i.e., not affecting the snow-free albedo. The simulation results show only minimal differences in the estimated ALT. That is, the winter surface albedo when using realistic satellite LAI products does not differ very much from that using the original LAI climatology. However, we speculate that large differences in summer could have significant impact on ALT estimation. We leave further investigation for future work. We will add one sentence in the manuscript to bring up this issue.

"Another issue affecting our ALT comparisons is the climatological representation of vegetation parameters such as LAI used in CLSM. Additional investigation (not

shown) revealed large differences between the LAI climatology used in CLSM and
more realistic, time-varying LAI products at several Mongolian sites."

Without any further information about the soil parameters in deep soils, we could not
conduct further tests. Here we provided the results on the sensitivity of soil organic carbon
and the vertical distribution in Figure 6. The figure reveals that a deeper ALT results from
reducing the SOC content and from using a very different vertical distribution profile that
arbitrarily concentrates less carbon in the top soil. Indeed, changing the vertical distribution
profile for SOC content plays an almost equivalent role to changing the SOC content.

This further confirms the reviewer's comment regarding the importance of vertical
variation of soil properties. Thus, we will add one sentence here to bring up the issue about
vertical variation in soil parameters.

"Besides the vertical distribution of soil carbon, the vertical variation in other soil
hydrological properties (e.g. soil texture, porosity, hydraulic conductivity, etc.)
should also play a significant role since they all affect soil thermal conductivity and
heat capacity."

[Figure]

*Figure R6: Simulation results at six Mongolian sites with different soil carbon contents
and vertical carbon distributions. "OriSOC_OriProf" – Original soil organic carbon
(SOC) content vertically distributed with the original profile as used in baseline simulation.
"SOC/N_OriProf" – Reduced soil organic carbon content (by dividing the original SOC
content by N) vertically distributed using the original profile. "OriSOC_NewProf" –
Original SOC content vertically distributed with a new profile which arbitrarily
concentrates less carbon in top soils. "SOC/N_NewProf" – Reduced SOC content (by
dividing the original SOC content by N) vertically distributed using the new profile.*

Minor comments:

1. Fig. 4: The text in Section 2.3 indicates most of the comparisons against AirMOSS data will be at the 4 flights in the Alaska north slope. So I wonder why COC flight was included in this figure esp. considering the AirMOSS retrievals were not included?

R2C9: We only have AirMOSS retrievals for IVO, ATQ, BRW and DHO, not for COC. However, our model provides results here, thus we included COC to add two additional measurements to compare with model results (Figure 4b).

2. Fig. 4 a-b: were the sites arranged according to the latitudinal changes?

R2C10: We arranged the sites aligning with the flight direction. We will also add this into the caption of Figure 4.

"The sites are arranged aligning with the flight direction."

3. Page 9, Line 21: Does MERRA-2 not provide air temperature at 2-m surface height?

R2C11: MERRA-2 does provide output of hourly 2-m air temperature. However, the land model within the MERRA-2 system is forced with the air temperature in the lowest (atmospheric) model layer (TLML), and the 2-m temperature is simply diagnosed from TLML and the surface temperature. For consistency, the off-line (land-only) model simulations presented here were likewise driven with TLML from MERRA-2. In any case, the sentence in question was not necessary and only caused confusion, so we will delete it.

4. Section 3.2 and Figure 6: Part of the IVO flight lies in the Brooks mountain range with very low SOC content (Fig. 2c). It may be not a good representative of the average conditions in this area, at least considering SOC variability.

R2C12: The point at which we extracted the soil parameters does have a sort of intermediate SOC (greenish color in Fig.2c) which is also shown in Figure 5b. As mentioned in the original manuscript, however, we actually used an arbitrary intermediate SOC value which is 40 kg/m$^2$.

We also conducted two additional simulations using a very large and a very small SOC value everywhere. The results are shown in Figure R7 below. The "IntermC" used the same SOC that was used for homogenization (i.e., 40 kg/m$^2$). "LowC" and "HighC" used the lowest (10kg/m$^2$) and highest (120kg/m$^2$) SOC values found along the transects as shown in Figure 5b and also in Figure R7b. Figure R7a reveals that the model sensitivity to soil carbon is much larger for lower SOC than for higher SOC, and easily gets saturated for high SOC (i.e., larger than ~100 kg/m$^2$). However, all of this depends on the vertical soil carbon distribution profile used. Please also see R2C8d.

[Figure]

*Figure R7: a) similar to the Figure 5a in the original manuscript but showing model sensitivity to organic carbon content along the AirMOSS flight transect. b) the organic carbon content along the transect.*

We will add a sentence into the Section 3.2.

> "Our investigation reveals that the model sensitivity to soil carbon content is much larger for lower SOC than for higher SOC, and easily gets saturated for high SOC (i.e., larger than ~100 kg/m$^2$) (not shown). Thus, we trust 40 kg/m$^2$ is an appropriate value representing an intermediate SOC condition."

5. Section 3.2: Would it be easier to follow if the description regarding the idealized experiments was included in the Methods section?

R2C13: We thank the reviewer for the helpful suggestion. We will introduce the idealize experiments in Section 2 by adding a new Methods section (section 2.5).

6. Fig. 7 is not very informative. I suggest summarizing the results in a table.

R2C14: We feel that Fig.7 is informative, which is also supported by Reviewer #1 (R1C15). The figure displays the actual values and therefore also serves as a table. We therefore opt to keep the figure in its current form.

---

## Author Comment (AC3) · 20 Sep 2018

**Overview**

We thank the reviewers for their constructive comments and suggestions. The manuscript will be appropriately revised in response to the reviewers' comments (see the point-by-point expected responses below). As requested by the reviewers, we compared our modeling results with some existing permafrost data sets by calculating evaluation metrics that can be compared directly against matching results reported in the literature. In addition, we also conducted several new simulations that further assess the impact on ALT of the model soil layer configuration, the soil organic carbon content, and its vertical distribution.

In summary, the planned modifications to the text can be categorized as follows:

a) Novelty and added value:
   See R1C1 (i.e., Reviewer 1, Comment 1), R1C2, and R3C6

b) Comparison with other model-generated permafrost data sets:
   See R1C1, R1C2, R3C6, R3C29 and R3C30

c) Rephrasing "optimistic" discussion about ALT results:
   See R1C10, R1C11, R1C12, R3C2, R3C31

d) New sensitivity experiments and uncertainty discussion:
   See R1C3, R1C20, R2C8 and R2C12

e) Add specific evaluation metrics instead of using description words:
   See R3C6, R3C8, R3C24, R3C29 and R3C31

Throughout the discussion below, the text is colored as follows:

Black: Reviewer comment
Blue: Expected author response
Red: Expected text to be inserted into the revised manuscript

For reference, our response to comment "m" by reviewer "n" is labeled "R[n]C[m]".

Reviewer #3

The authors compare model estimates (based on re-analyses data) of active layer thickness (ALT) with different data sources. This includes in situ measurements from an international database and an unpublished approach of airborne P-band estimation. The latter is available over Alaska. General evaluation is made for the northern hemisphere.

In addition, the model input is investigated for linear relationship with its output. The authors seek to find a simplified relationship to explain ALT change over time (driven by degree days, snow water equivalent etc).

General issues

The manuscript addresses an important topic, the modelling of active layer thickness over time. The presentation of the manuscript and setup of the experiment is however problematic.

1) The authors report 'reasonable' or 'good' results with limited and partially missing quantitative reasoning.

R3C1: We will include additional quantitative metrics to better illustrate our results. Please see our responses in R3C6, R3C8, R3C24, R3C29 and R3C31 for further details.

2) The results of the comparison to the unpublished airborne (AirMOSS) approach suggest that it actually does not work (AirMOSS values are all at the same level, not representing the in situ range, Figure 4). The comparison to this unpublished approach should be removed from the manuscript.

R3C2: We thank the reviewer for pointing out this. However, we do not agree with the reviewer regarding this point. We believe the AirMOSS data has value, which is also the opinion of Reviewer #2. We will expand the relevant discussion about the potential use of the AirMOSS retrievals in the manuscript. Please see specifically our response to R1C11, R1C12 and R2C7, with particular attention to the Table R1 (new Table 3 proposed for the revised manuscript).

3) The stated objectives comprise issues regarding active layer thickness. The results section however does also cover permafrost extent.

R3C3: We thank the reviewer for pointing out this. We do not solely focus on ALT. We will modify the statement of our objectives in Section 1:

> "Overall we pursue three scientific objectives: 1) evaluate the relative importance of the factors that determine the spatial variability of ALT, 2) evaluate CLSM-simulated ALT climatology and permafrost extent against observations, and 3) quantify and assess the large-scale characteristics of ALT (in terms of means, interannual variability and trend) in Northern Hemisphere permafrost regions from 1980 through 2017. "

4) In addition, common structure of methods, results and discussion is lacking. Methods are included in the introduction and mostly in the results section. The discussion is largely included in the results section. The 'approach' section mostly only includes the dataset description.

R3C4: We will add a new section ("Section 3 Methods") and will move all the method description to this new section. We will also move some discussion about our results to the last section to supplement other 'discussion' of the research findings.

5) Some of the used datasets are not appropriately cited. See detailed comments below

R3C5: We thank the reviewer for pointing out this. We will check our reference list and will correctly cited all of the datasets that we used. Please also see our response in R3C10 and R3C11 for additional details.

6) A main weakness of the manuscript is that the results are not discussed with respect to already published material on permafrost related parameter modelling and on the parameters which are investigated for explanation as drivers of ALT change over time. The role of temperatures and snow water equivalent for active layer thickness is known (what is presented as one of the main results in the abstract). The novelty of the paper is not clear.

R3C6: We will be adding extensive text that further highlights the novelty of our study. Please see our responses in R1C1 and R1C2. In the discussion below we focus specifically on the comparison between our results and other existing studies, for the reviewer's interest.

Given that different models run at different spatiotemporal resolutions, and that the simulated results were evaluated using different observation data sets (or using different time periods of the same data set), a direct and credible comparison with existing model-simulated permafrost data sets requires extensive analysis and is not conducted here as it is considered beyond the project scope. Instead, we did a rough comparison with existing permafrost products simulated by land models with reanalysis-based forcing as shown in the table below. (Note that ALT products simulated with climate forcing that is not observation-based are not included here.) We also want to emphasize that many existing studies define permafrost as areas with an ALT less than 3 m (Dankers et al., 2011) or 3.8 m (Guo et al., 2017;Guo and Wang, 2017;Lawrence et al., 2012), i.e., focused on the "near-surface permafrost". We do not specifically limit ALT within a certain depth, although our simulated results generally are less than or equal to the bottom depth of the $5^{th}$ soil layer (2.95m). Also, many of these studies evaluate the simulated results for "present-day" (i.e., before 2000). To make a relatively fair comparison with the existing studies, we used the same validating time period.

*Table R4 – Existing available ALT products and evaluation metrics over Northern Hemisphere simulated by land surface models with reanalysis-based forcing.*

| Model and Soil Configuration | Forcing | Spatiotemporal Domain and Resolution | Findings and Conclusions Regarding ALT and Permafrost Extent | Reference |
|---|---|---|---|---|
| | | | | |

| JULES (Joint UK Land Environment Simulator)

(total soil depth 3m) | GSWP2 (Global Soil Wetness Project 2)

WATCH (WATer and Global Change) | 1983–1995 at 1°×1° resolution with GSWP2 covering areas north of 25°N (includes the Tibetan Plateau).

1959–2000 at 0.5°×0.5° resolution with WATCH covering areas north of 45°N (does not include the Tibetan Plateau). | - Captures 97% of the continuous and discontinuous permafrost areas.

- Overestimates the total extent; simulates permafrost where it only occurs sporadically or only in isolated patches (25%); overestimates an additional 14% in areas permafrost free.

ALT: JULES-simulated ALT is generally too deep compared with CALM observations: mean bias in the GSWP2 run (1990–1995) is 0.81±0.48 m, and 0.53±0.50m in WATCH-GPCC (1990–2000). The Root Mean Square Error (RMSE) is 0.94 and 0.73m, respectively. | Dankers et al. (2011) |
|---|---|---|---|---|
| ORCHIDEE-MICT land surface model

(total soil depth is 38 m) | GSWP3 (Soil Wetness Project Phase 3)

CRUNCEP (Climatic Research Unit - NCEP) | Northern Hemisphere (>30° N) at 1°×1° spatial resolution (does not include the Tibetan Plateau).

1901–2007 with GSWP3;

1901–2015 with CRUNCEP. | - Both simulations underestimate permafrost extent when using ALT<3m as the definition of permafrost; CRUNCEP-forced simulation shows better permafrost extent using an alternative definition of permafrost. (No actual bias value provided.)

- GSWP3-forced model generally overestimates ALT by more than 1m; CRUNCEP-forced output shows relatively better agreement with the observations. (No actual bias value provided.) | Guimbert eau et al. (2018) |
| CLM 4.5

(50m soil depth) | CRUNCEP (Climatic Research Unit - NCEP) | Northern Hemisphere (includes the Tibetan Plateau).

1901 – 2010

Spatial Resolution: 0.5°×0.5° | (Defines near-surface permafrost as that within the upper 3.8 m of soil.)

- Simulated present-day (mean from 1981 to 2000) permafrost distribution; shows good agreement – only discrepancy is on the Tibetan Plateau. Bias in permafrost extent is $2.02 \times 10^{6}$ $km^2$.

- Global ALT trend with a correlation coefficient and an | Guo and Wang (2017) |

| | | | The Nash-Sutcliffe Efficiency (NSE) of 0.73 and 0.21, respectively. The simulated trend was smaller than observed trend. (No evaluation for ALT climatology provided.) | |
|---|---|---|---|---|
| CLM 4.5 (50m soil depth) | CFSR: NOAA Climate Forecast System Reanalysis

ERA-I: European Centre for Medium-Range Weather Forecasts Re-Analysis Interim

MERRA: NASA Modern Era Retrospective-Analysis for Research and Applications | Northern Hemisphere (includes the Tibetan Plateau).

Spatial Resolution: 0.5°×0.5°

1979 -2009 | (Permafrost is defined as ground where monthly soil temperature is less than 0°C for 24 consecutive months in at least one layer of the upper 10 soil layers (3.8 m) (Lawrence et al., 2012).)

- The model underestimates the permafrost extent in southern Alaska, northern Western Siberian Plain, and over the Tibetan Plateau.

Specific evaluation metrics:  | Guo et al. (2017) |
| CLM3 and CLM4 (offline)

3.5m depth with CLM3 and 3.8m depth with CLM4. | Forced with observed meteorological data based on NCEP-DOE. | Northern Hemisphere (includes Tibetan Plateau);

Simulation period: 1980–1999;

0.9375° latitude × 1.25° longitude | (Focused on shallow-permafrsot with ALT < 3.8 m.)

- CLM3 simulation underestimates permafrost in southern and western Siberia.
- CLM4 simulations show improved permafrost extent.

- In general, ALT is overestimated; CLM4 underestimated ALT in regions of shallow permafrost. (No actual evaluation metrics provided.) | Lawrence et al. (2012) |

| Index | Active layer thickness (m) | | |
|---|---|---|---|
| | Climatology CALM, 1991–2000) CFSR, ERA-I, MERRA | Climatology (AL_RHST, 1981–1990) CFSR, ERA-I, MERRA | Change (CALM, 1996–2007) CFSR, ERA-I, MERRA |
| Mean bias | 0.21, 0.33, 0.66 | −0.33, −0.29, −0.11 | |
| Mean absolute bias | 0.52, 0.63, 0.89 | 0.50, 0.44, 0.52 | |
| Correlation coefficient | **0.69, 0.62, 0.51** | 0.18, **0.41,** 0.12 | **0.62, 0.83, 0.75** |
| Nash-Sutcliffe efficiency | 0.42, 0.25, −0.34 | −0.56, −0.20, −0.47 | −0.64, 0.36, −3.99 |

| | | | - CLM3-simulated continuous and discontinous permafrost extent is $11.1 \times 10^6$ km$^2$ for the period 1970–89, which is below $11.8–14.6 \times 10^6$ km$^2$ estimate from Zhang et al., 2000.

- CLM4-simulated continuous and discontinous permafrost extent is $13.7 \times 10^6$ km$^2$ for the period 1970–89. | |
|---|---|---|---|---|

Existing studies that evaluated ALT that can be compared with ours include (Dankers et al. (2011) and Guo et al. (2017). Other studies did not report specific evaluation metrics within a common simulation domain as ours, hence, they cannot be used for comparison. Comparing with reported evaluation metrics as shown in Table R5, Figure R8 and Figure R9 reveal that: 1) for the early 1990s (i.e., 1990 – 1995 period), our simulated results show a much better agreement with the observations than when using JULES with GSWP2 forcing as reported in Dankers et al. (2011); 2) for the whole 1990s (i.e., 1990 – 2000) period, our results are better than that using JULES with WATCH-GPCC forcing or when using CLM4.5 with both ERA-I and MERRA forcing as reported in Guo et al. (2017); and 3) our results show a smaller mean absolute bias and a larger correlation coefficient than that using CLM4.5 with CFSR, but show worse performance regarding mean bias and NSE. In addition, all of these existing studies overestimated ALT at the global scale and reveal a positive bias while our results underestimate deep ALT, and hence, reveal a negative bias. However, it is worth noting that our results demonstrate much better agreement against observations for regions of shallow permafrost as shown in Figure R8 and R9. We will add a short discussion along the lines of the following:

> "The existing literature on simulated ALT fields (e.g., Dankers et al. (2011), Lawrence et al. (2012) and Guo et al. (2017)) reveals a general tendency for models to overestimate ALT climatology at the global scale. The CLSM-simulated ALT fields appear to be among the better simulation products."

*Table R5 – Summary of evaluation metrics for ALT estimates reported in literature. The same metrics calculated with the simulation results in this study for an identical evaluation period are provided.*

| Evaluation metrics | (Dankers et al., 2011)* with two different sets of climate forcings | | | | (Guo et al., 2017)[#] with three different sets of climate forcings | | | |
|---|---|---|---|---|---|---|---|---|
| | GSWP2 (1990–1995) | This Study (1990–1995) | WATCH-GPCC (1990–2000) | This Study (1990–2000) | CFSR (1991-2000) | ERA-I (1991-2000) | MERRA (1991-2000) | This Study (1991-2000) |
| RMSE (m) | 0.94 | 0.18 | 0.73 | 0.72 | None | | | |

| Mean Bias (m) | 0.81 | 0.04 | 0.53 | -0.17 | 0.21 | 0.33 | 0.66 | -0.24 |
|---|---|---|---|---|---|---|---|---|
| Mean Absolute Bias (m) | None | | | | 0.52 | 0.63 | 0.89 | 0.36 |
| Correlation Coefficient | | | | | 0.69 | 0.62 | 0.51 | 0.70 |
| Nash-Sutcliffe Efficiency (NSE) | | | | | 0.42 | 0.25 | -0.34 | 0.25 |

*Evaluation was conducted with annual ALT.
**Evaluation was based on ALT climatology.**

[Figure]

*Figure R8: Observed and simulated ALT at CALM observations sites. a) copied from Dankers et al. (2011). The comparison was made for the period 1990–1995 using GSWP2 forcing, and for the period 1990–2000 using WATCH. b) Results from this study.*

[Figure]

*Figure R9: a), b) and c) are copied from Guo et al. (2017). Black dots represent the validating pair of data using CALM observations whereas the blue dots represent the validating pair of data using AL_RHST observations. d) shows comparison results with ALT estimates in this study. All subplots are for the validating period of 1990 to 2000.*

In terms of the evaluation of simulated permafrost extent, although it is not fair to compare our results with these existing studies due to differences in simulation domain, we did conduct similar calculations to quantitatively evaluate our simulated permafrost extent against these other studies. In addition, most of these existing studies compare the model simulated permafrost area with that of the total area of continuous and discontinuous permafrost from Brown's map due to their coarse spatial resolution (i.e., at least 0.5°). Given the high resolution (i.e., roughly 9km × 9km) of our simulation results, we compare and discuss our simulated permafrost area with that of the total area of continuous, discontinuous, **and** sporadic permafrost together as shown in Figure 10 in our manuscript. For the revised manuscript, we will incorporate these evaluation results into a new table, as shown below (Table 4 in the revised manuscript). We will also compare the simulated permafrost extent with that of the total area of continuous and discontinuous permafrost area from Brown's map in order to better compare our results with those of existing studies, which is shown in the parentheses in Table R6 below (new Table 4 in the revised manuscript).

*Table R6– Evaluation results for simulated permafrost extent. (New Table 4)*

| Case | CLSM | Obs. | Simulated Area ($\times 10^6\,km^2$) | Percentage Relative to Observed |
|------|------|------|------|------|
| 4 | No | No | 48.8 | - |

| 3 | Yes | No | 1.9 | - |
|---|---|---|---|---|
| 2 | No | Yes | 3.2 (1.7) | 18.7% (12.3%) |
| 1 | Yes | Yes | 13.8 (12.3) | 81.3 % (87.7%) |

We will also add the following short discussion into the revised manuscript:

"The specific areas of each type shown in Figure 10a are listed in Table 4. The simulated permafrost extent covers 81.3% of the observation-based area (i.e., the total area of continuous, discontinuous and sporadic permafrost regions), and missed 18.7% of the observed permafrost area. When comparing simulated permafrost extent with only continuous and discontinuous types, these metrics change to 87.7% and 12.3%, respectively. Meanwhile, the permafrost extent is overestimated by $3.2 \times 10^6 \, \text{km}^2$."

Regarding the role of temperature and SWE, although it is well known that these two factors affect permafrost, the *relative* contribution of the two is still unclear. This study attempts to quantitatively estimate this relative impact, which, to our knowledge, has heretofore not been done. This, we feel, is indeed one of our useful, novel contributions to the permafrost literature.

Detailed comments

Title: the title is misleading; the paper focuses on active layer thickness what should be reflected in the title

R3C7: We do not solely focus on ALT. Simulated permafrost extent is also evaluated and discussed in the manuscript. We believe our title regarding "variability" implicitly includes both ALT and permafrost extent. In addition, ALT might be too technical to be included in a title since many people know about permafrost but may not know what ALT is. We plan to retain our title as is.

Abstract:

Line 19: 'measurements demonstrates reasonable skill' – what is 'reasonable'? add numbers

R3C8: We will add the following sentence in order to clarify what we mean by "reasonable" skill.

"Specifically, the RMSE (and bias) of climatological ALT is 1.22 m (and -0.48 m), and is reduced to 0.33 m (and -0.04 m) without the Mongolia sites."

Line 27: 'significant degradation, with ALT increasing up to 0.5 cm/year' – it should be noted that this cannot be confirmed with in situ observations

R3C9: We thank the reviewer for pointing out this. This is very true. Below we provide the same figure as our Figure 14a in the original manuscript but only showing sites at which both the observed and estimated ALT trend is found to be statistically significant. Only four and ten sites are shown below with p value less than 0.05 and 0.10, respectively. Although the estimated and observed trends are closely clustered around the 1:1 line, the limited number of sites shown below cannot assure the accuracy of the estimated trends.

[Figure]

*Figure R10: ALT trend from CLSM estimates vs. CALM observations, based on years common to both datasets within the period of 1990 to 2017. The horizontal and vertical error bars represent 95% confidence intervals of observed ALT trend (regression slope) and CLSM-simulated ALT trend, respectively. This current figure is similar to Figure 14 in the original manuscript, but only showing sites with both observed and estimated trends that are statistically significant at a) 0.05 and 0.10 level.*

For the manuscript, we will add an additional description of the significance of estimated trends into the caption of Figure 14, and we will add the following to the discussion in Section 4.5:

"However, the comparison with in-situ observations is not able to adequately assess the accuracy of such simulated ALT trends, given that only a very few observational sites show trends that are statistically significant."

We will also add a sentence into the abstract as shown below:

"Moreover, only four (ten) points remained when screening out sites at which either the observed or estimated ALT trend is not statistically significant at the 0.05 (0.10) level. The limited number of sites with meaningful trends cannot assure the accuracy of the estimated trends."

Page 5, line 9: NCSCDv2 – citation missing, follow instructions of https://bolin.su.se/data/ncscd/

R3C10: We sincerely apologize for missing the important references. We thank the reviewer for pointing out this. We will add the references for NCSCDv2:

"… from the Northern Circumpolar Soil Carbon Database version 2 (NCSCDv2, https://bolin.su.se/data/ncscd/) (Hugelius et al., 2013a;Hugelius et al., 2013b)."

Page 6, line 24: Brown et al. 2002: the reference is missing in the list

R3C11: We are sorry for failing to include this reference.  We will add it.

Page 7, line 30 following: this paragraph belongs to methods

R3C12: Will be done as suggested.

Page 8, line 9: remove 'relatively'

R3C13: Will be done as suggested.

Page 9, first paragraph: this belongs to discussion

R3C14: Will be done as suggested.

Page 9, line 17 following: introduce this comparison in the methods section

R3C15: Will be done as suggested.

Page 9, line 22: provide correlation analyses results as table

R3C16: These results only consist of three correlation coefficients. Therefore, instead of adding these values into a table, we will add these correlation coefficients to Figure 5 for each variable for greater clarity.

Page 9, line 23: which SOC value did you use? Is it representative for the upper soil layer?

R3C17: We used spatially varying SOC provided by NCSCDv2, and not a constant SOC everywhere. Please see section 2.1 and Tao et al. (2017) for more details.

Page 10, line 4-28: this all belongs to the methods section

R3C18: We will move these paragraphs to the new method section (Section 3).

Page 11, line 7: spell out LAI

R3C19: We have spelled out LAI at the place it first appears (Page 10 Line 16 in the original manuscript) and thus keep LAI as used here.

Page 12, line 6-12: move to methods

R3C20: Will be done as suggested.

Page 12, line 19: 'permafrost areas shown in Fig. 1b are well confined within the cold side ..' this cannot be seen. Add outlines to map

R3C21: The 273.15K isotherm actually is shown in the figure (i.e., the edge of dark green block). We will add the boundary containing continuous and discontinuous permafrost regions into Figure 8d. The original Fig. 8d showed mean Tair all over the Northern Hemisphere. We will modify it and only show mean Tair over the simulation domain in order to better illustrate the results.

Page 13, up to line 7: to methods; what is the reasoning for the static time period approach?

R3C22: We will move the method description to the new Section 3.3. We are not sure what the reviewer means by "static time period". We evaluated the full 37-year period in order to best leverage all of the available observations. We will also modify equation (1) and (2) in the original manuscript (and equation (2) and (3) in the revised manuscript) as follows:

$$"T_{cum} = \sum_{N=2}^{N=38} T_{pos}(N) \quad , \tag{1}$$

where

$$T_{pos}(N) = \begin{cases} T_{air}(t) - T_f & if\ T_{air}(t) > T_f \\ 0 & if\ T_{air}(t) \leq T_f \end{cases}, \tag{2}$$

The summation in Eq. (2) for year N is computed from 1 September of year (N-1) to 31 August of year N."

Page 13 – move explanation of linear regression analyses to methods

R3C23: Will be done as suggested.

Page 14, line 14: 'geographically thin disagreements' – quantify this

R3C24: It is quite hard to quantify this, but we will attempt to do so with the following modification:

"Aside from western Siberia, the geographically thin disagreements (i.e., about a few degrees latitude) between the simulated and observed permafrost extents toward the south in Figure 10a (green and blue areas at the southern edge of permafrost regions) are not as much a concern, since the comparison in such areas is muddied by the interpretation of "isolated" permafrost in the observational map (Figure 10 b)."

Page 14, line 28 following: this is methods

R3C25: We will modify and move these sentences to the new method section (Section 3.4).

Page 14, line 18: this error may seem small in absolute numbers, but ALT is much thinner than for the Mongolian sites. The error is still relatively large.

R3C26: We recognized this behavior, and hence we discussed the model performance in high-latitude regions separately from that in mid-latitude regions. We also explicitly emphasized the large error for the Mongolian sites and discussed the possible reasons for the underestimation. We conducted some new sensitivity tests and will add the results into the manuscript with relevant discussion. Please see our response in R2C8d and R3C6 for details.

Page 14, line 20-26: this belongs to discussion

R3C27: We thank the reviewer for this suggestion, but we believe this paragraph should stay here since it describes the results.

Page 15, first two paragraphs of section 3.5 – this all belongs to methods

R3C28: We will modify and move these paragraphs to the new methods section (Section 3.4).

Page 17, 21: 'shows good general agreement' – this is not really the case, quantify the agreement and compare with other published results

R3C29: We calculated the bias in percentage (please see R3C6 for details), and we will modify as follows:

> "The spatial distribution of CLSM-simulated permafrost shows general agreement with the observation-based permafrost map of Brown et al. (2002), capturing 81.3% of total areas of continuous, discontinuous and sporadic types while capturing 87.7% of the total area of continuous and discontinuous types."

Page 17, line 29: 'The simulated ALTs agree well with the in-situ observations' – not really, see my comment for page14, line18; how do your results compare to other published results?

R3C30: Our results are among the best existing permafrost products. In particular, our results demonstrate much better agreement against observations for regions of shallow permafrost (as shown in Figure R8 and R9) compared to other existing results. Please see details in R3C6.

Page 18, lines 1-2: 'retrievals from airborne remote sensing for 2015 and the corresponding simulated ALT exhibit reasonable accuracy vs. in situ measurements' – this is not clear from the material presented.

R3C31: Point taken.  We will modify this sentence as follows:

> "In northern Alaska, ALT retrievals from airborne remote sensing for 2015 and the corresponding simulated ALT exhibit limited skill versus the in-situ measurements. At the model scale, the mean bias for the simulated results is better (-0.05 m) than that for the retrievals (-0.12 m), but the opposite is true for the correlation coefficient against observations (0.27 for the model vs. 0.61 for the retrievals). At the in-situ site scale, however, the ALT retrievals show a very weak correlation

coefficient with the observations (0.05). Excluding sites that have ALT measurements exceeding the radar sensing depth (~ 60cm), the evaluation metrics for ALT retrievals become better than that for simulated ALT at the model scale. The remotely sensed ALT estimates generally show lower levels of spatial variability than the simulated ALT estimates, and their spatial patterns differ considerably. Such differences may be cause for concern, but they should decrease as both approaches evolve and improve. It is important to document the performance of the two approaches and to consider them side by side during their evolution given their potential for estimating permafrost in coming years. The most accurate future estimates of permafrost may indeed result from their joint application, such as through the downscaling of model results with higher resolution retrievals."

Figure 5: what is the red rectangle?

R3C32: The red rectangle highlights an example to illustrate the anti-correlated relationship between ALT and organic carbon content. We discussed this point in section 4.2. We will also add the following sentence to the caption of this figure:

"The red rectangle crossing a) and b) highlights a portion of the domain that shows an anti-correlated relationship between organic carbon content and modelled ALT (see Section 4.2)."

Figure 7: convert to table

R3C33: Since the values are already provided on the figure, we feel that converting it to a table would not help much. We opt to keep the figure as is.

[revised manuscript text omitted]

---

## Author Response (AR1)

**Overview**

We thank the reviewers for their constructive comments and suggestions. The manuscript has been appropriately revised in response to the reviewers' comments (see the point-by-point responses below). As requested by the reviewers, we compared our modeling results with some existing permafrost data sets by calculating evaluation metrics that can be compared directly against matching results reported in the literature. In addition, we also conducted several new simulations that further assess the impact on ALT of the model soil layer configuration, the soil organic carbon content, and its vertical distribution.

In summary, our modifications to the text can be categorized as follows (for reference, our response to comment "m" by reviewer "n" is labeled "R[n]C[m]"):

a) Novelty and added value:
   See R1C1 (i.e., Reviewer 1, Comment 1), R1C2, and R3C6

b) Comparison with other model-generated permafrost data sets:
   See R1C1, R1C2, R2C8b, R3C6, R3C29 and R3C30

c) Rephrasing "optimistic" discussion about ALT results:
   See R1C10, R1C11, R1C12, R3C2, R3C31

d) New sensitivity experiments and uncertainty discussion:
   See R1C3, R1C20, R2C8 and R2C12

e) Add specific evaluation metrics instead of using description words:
   See R3C6, R3C8, R3C24, R3C29 and R3C31

Throughout the discussion below, the text is colored as follows:

Black: Reviewer comment
Blue: Author response
Red: Text newly inserted into the revised manuscript

Reviewer #1

This paper provides an evaluation of active layer thickness and permafrost extent as simulated by the NASA Catchmant Land Surface Model driven by MERRA-2. The model-generated dataset of permafrost conditions is evaluated again site data, global data and remotely (plane) sensed ALT. The comparison to the remotely sensed ALT is probably the most innovative part of the paper, but it also suffers from some drawbacks because the remotely sensed data conspicuously lack spatial variability. The paper is written clearly, the analysis is honest (not obviously trying to hide model shortcomings – but sometimes the assessment of the dataset quality seems a bit too optimistic), the figures are relevant and informative. The paper is a useful contribution, but some aspects detailed in the following could be improved.

We thank the reviewer for the careful reviewing. We understand the reviewer's concern about the small spatial variability with the remotely sensed ALT retrievals. The ALT retrievals were produced by the current algorithm developed by Chen et al. (2018). Somewhat larger spatial variability is presented in the original retrievals but is smoothed and reduced after aggregating to the scale of the Catchment Land Surface Model (CLSM) at 81 km$^2$, as also mentioned in the original manuscript. In addition, the radar penetration depth is not large enough to detect deeper thawed to frozen condition of the soil. All in all, while we eventually expect to further improve the retrieval algorithm, these are the results we have right now, and as discussed further below, their inclusion in this paper in their early form does, we feel, have value. We have tamped down our "optimistic" tone when discussing the ALT retrievals. We trust this first intercomparison of ALT among model results, remotely sensed retrievals and in-situ observations could provide useful insights to the research community. Please see specifically our response to R1C11, R1C12 and R2C7 below.

General:

- There are lots of global permafrost simulations that are driven by reanalysis-based meteorologies. What is really the added value of this one? The fact that is uses MERRA? In that case, could you say more about specific strengths and weaknesses of MERRA, please? More generally, simulations with other metorological forcing data, and comparison to other model-generated permafrost data sets (e;g. within the Permafrost Carbon Network) could be interesting.

R1C1: We thank the reviewer for encouraging us to explicitly highlight our contributions. Regarding the second, more general comment, a detailed quantitative comparison against existing permafrost data sets simulated by other land models, for example the land models participated in the Permafrost Carbon Network (PCN) (offline mode) and the Coupled Model Intercomparison Project phase 5 (CMIP5) (offline/coupled mode), is beyond the scope of this paper, which is already a bit long. (Note that aspects of such a general analysis have already been reported in literature (Peng et al., 2016;Wang et al., 2016b;Koven et al., 2013)). Inspired by the reviewer's comment, though, we added a brief discussion to our

manuscript that compares our dataset with others in terms of spatiotemporal resolution and simulated results (see our response in R3C6 for details). We also summarized our dataset's particular strengths in our discussion section.

Specifically, we have added the discussion below.

a. General comparison between this work and existing model-generated permafrost data sets forced by other meteorological forcing data (both in uncoupled and coupled mode):

a.1 Regarding resolution (section 1, page 2, line 32-33 and page 3, line 1-7 in the revised manuscript)

"Most of these land models were run at coarse spatial resolutions, e.g., ranging from $0.5° \times 0.5°$ to $1.8° \times 3.6°$ for LSMs participating in the Permafrost Carbon Network (PCN) (Wang et al., 2016a) and from $0.188° \times 0.188°$ to $4.10° \times 5°$ for the models participating in the Coupled Model Intercomparison Project phase 5 (CMIP5) (Koven et al., 2013; https://portal.enes.org/data/enes-model-data/cmip5/resolution). As a result, it is difficult to compare the simulated values with in-situ observations taken at the point scale. Other types of numerical models have been run at relatively higher resolution, but not globally; such simulation domains were limited to regional scales (e.g., 2 km $\times$ 2 km in Jafarov et al. (2012) covering Alaska;1 km $\times$ 1 km in Gisnas et al. (2013) covering Norway) as necessitated by the availability of ancillary data and the heavy computational burden. As discussed further below, one of the unique contributions of the present work is a global simulation of permafrost at a somewhat higher resolution than earlier global-scale studies."

a.2 Regarding model performance in simulated permafrost extent (particularly the deficiency in western Siberia) (section 4.4, page 17, line 12-20 in the revised manuscript)

"Note that some other global models, such as CLM3 and the Community Climate System Model version 3 (CCSM3) as reported in Lawrence et al. (2012), also missed this area of permafrost and that updated versions of these models (i.e., CLM4 and CCSM4) showed improved performance in this regard (Lawrence et al., 2012). Guo et al. (2017) reported underestimated permafrost extent simulated in western Siberia using CLM4.5 driven by three different reanalysis forcings (i.e., CFSR, ERA-I and MERRA), and they showed an improved simulation of permafrost extent in this area when using another reanalysis forcing, the CRUNCEP (Climatic Research Unit - NCEP) (Guo and Wang, 2017). Guimberteau et al. (2018) found similar improvements stemming from the use of CRUNCEP forcing. We leave for further study whether the MERRA-2 forcing data is responsible for the western Siberia deficiency seen in our own results"

a.3 Regarding model performance in simulated ALT (section 4.4, page 18, line 18-20 in the revised manuscript)

"Note that the existing literature on simulated ALT fields (e.g., Dankers et al. (2011), Lawrence et al. (2012) and Guo et al. (2017)) reveals a general tendency for models to overestimate ALT climatology at the global scale. Our results here suggest that the CLSM-simulated ALT fields are among the better simulation products, especially for shallow permafrost."

b.  Some improvements of MERRA-2 compared to MERRA:

"Note that MERRA-2 has been found to be skilful in its simulation of near-surface atmospheric conditions (Reichle et al., 2017a;Reichle et al., 2017b;Bosilovich et al., 2015;Bosilovich et al., 2017) and to show improvements in the representation of cryospheric processes compared with its predecessor MERRA (Gelaro et al., 2017). In particular, MERRA-2 assimilates substantially more satellite observations and employs more physically reasonable hydrology representations for glaciated land surfaces compared to MERRA, and it also uses observation-based, seasonally-varying sea ice albedo as opposed to MERRA's fixed value of 0.6 (Gelaro et al., 2017). A recent study shows that permafrost and ALT simulation results obtained with forcing data from the original MERRA reanalysis are inferior to those driven by other reanalysis-based forcing data sets, particularly those from the NOAA Climate Forecast System Reanalysis (CFSR) and European Centre for Medium-Range Weather Forecasts Re-Analysis Interim (ERA-I) (Guo et al., 2017). The superiority of MERRA-2 forcing compared to MERRA forcing in the context of permafrost simulation is presumed here (given its general improvements in the cryosphere), though a side-by-side test of the two forcing datasets in this regard has not been performed." (section 1, page 4, line 1-12 in the revised manuscript)

"We note that our MERRA-2-driven permafrost simulation results, while potentially better than those we might have obtained with MERRA forcing, are still lacking (e.g., in western Siberia). " (section 5, page 20, line 10-13 in the revised manuscript)

c.  Summary of novelty and added value of this work:

"The permafrost dataset presented herein can be considered unique in terms of its daily temporal resolution combined with a relatively high spatial resolution at the global scale (i.e., 81 km$^2$). The dataset, which is derived from a state-of-the-art reanalysis, shows reasonable skill in capturing permafrost extent and in adequately estimating ALT climatology (aside from that at the Mongolian sites). We note that our MERRA-2-driven permafrost simulation results, while potentially better than

those we might have obtained with MERRA forcing, are still lacking (e.g., in western Siberia). Still, with its resolution and available variables (ALT, subsurface temperature at different depths), the dataset could prove valuable to many future permafrost analyses." (section 5, page 20, line 7-13 in the revised manuscript)

"This work also provides a first comparison between two highly complementary approaches to estimating permafrost: model simulation and remote sensing retrieval method." (section 5, page 20, line 15-16 in the revised manuscript)

"The remote sensing approach is still relatively new, and many aspects still need to be worked out. It is important, though, to begin considering the modeling and remote sensing approaches side by side, as both should play important roles in permafrost quantification in the years to come. Indeed, once the science fully develops, joint use of modeling and remote sensing (e.g., through the application of downscaling methods) should allow the generation of more accurate permafrost products at even higher resolutions." (section 5, page 20, line 23-27 in the revised manuscript)

- Some words about potential uses of this dataset could be nice.

R1C2: We thank the reviewer for this suggestion. Relevant discussion about the potential uses of this dataset has been added to the end of this manuscript (section 5, page 23, line 10-12 in the revised manuscript):

"These data can potentially contribute, for example, to ecological studies focused on the dynamics of microbial activity and soil respiration in cold regions, on vegetation migration/adaptation in response to climate change, and so on."

Specific points:

- Page 2, line 15: "simulations… with the land surface model ( Dankers et al., Guimberteau et al., Tao et al.)" -> these are different models. The sentence is misleading, and confusingly, you write "…and other numerical models" afterwards…

R1C3: We revised the relevant sentences as follows (section 1, page 2, line 25-32 in the revised manuscript).

"Simulations and/or predictions with a variety of land surface models (LSMs) have been used to quantify large-scale permafrost patterns (i.e., distributions and thermal states) and their interactions with a warming climate. LSMs utilized for this include, for example, the Joint UK Land Environment Simulator (JULES, Dankers et al., 2011), the ORganizing Carbon and Hydrology in Dynamic EcosystEms (ORCHIDEE) - aMeliorated Interactions between Carbon and Temperature (ORCHIDEE-MICT, Guimberteau et al., 2018), the Catchment Land Surface Model (CLSM, Tao et al., 2017), and the Community Land Model (CLM,

Lawrence and Slater, 2005;Alexeev et al., 2007;Nicolsky et al., 2007a;Yi et al., 2007;Lawrence and Slater, 2008;Lawrence et al., 2008;Lawrence et al., 2012;Koven et al., 2013;Chadburn et al., 2017;Guo and Wang, 2017)."

- Page 2, line 20-24: Strictly speaking, the fact that 2017 set records doesn't mean that permafrost conditions will change. 2017 is only one year. It's the long-term trends that matter (2017 is of course consistent with that trend)

R1C4: We agree with the Reviewer regarding this point. The message we had tried to convey is that the warming trend of our climate seems to have increased in recent years and that given the associated exacerbation in permafrost thawing, monitoring permafrost in a timely manner is critical. We have modified the relevant sentences as follows (section 1, page 2, line 17-20 in the revised manuscript). Please also see our response to next comment (R1C5).

"In addition, given the apparent climate warming seen in recent years (exemplified by the fact that the average Arctic air temperature in 2017 (ending in September) was the second warmest on record since 1900 [Arctic Report Card; http://www.arctic.noaa.gov/Report-Card/Report-Card-2017] and that 2017 was the warmest year on record for global ocean temperatures (Cheng and Zhu, 2018)), important reductions in permafrost might be occurring as well."

- Page 2, line 23: "Some aspects of the current global permafrost thermal states are ... still unknown": can you elaborate on that, please?

R1C5: We have clarified this by expanding this sentence as shown below (section 1, page 2, line 20-23 in the revised manuscript):

"However, current global permafrost thermal states (i.e., permafrost temperature, ice content and degradation rates across much of Northern latitudes) are still largely unknown. Monitoring permafrost degradation in a timely manner is particularly critical for ecosystem management and for various policy decisions."

- Page 3, line 10: "extensive challenges" sounds bizarre to my non-native speaker's ears

R1C6: We replaced the end of the sentence with "combined with the many challenges of physical process modelling."

- Page 3, line 16: Could you say a few words specifically about high-latitude performance? Advantages, drawbacks // other reanalyses?

R1C7: We have added some relevant discussion about the performance of MERRA-2 in high-latitude regions (section 1, page 4, line 4-6 in the revised manuscript):

"In particular, MERRA-2 assimilates substantially more satellite observations and employs more physically reasonable hydrology representations for glaciated land surfaces compared to MERRA, and it also uses observation-based, seasonallyvarying sea ice albedo as opposed to MERRA's fixed value of 0.6 (Gelaro et al., 2017)."

- Page 3, line 26: Chen et al. is a paper in review. Can you reassure the reviewer that these retrievals are independent from the data produced here? One or two sentences would be nice anyway even if Chen et al. 2018 will be available to the reader soon.

R1C8: The AirMOSS radar retrievals of ALT we used here were produced by the algorithm described and analyzed in detail by Chen et al. (2018). The retrievals here are indeed identical to those produced by Chen et al. 2018, though here we examine them from a different perspective. We now added some text to distinguish the scope of this study from that of Chen et al. (2018) and to emphasize the data independence (section 1, page 4, line 18-20 in the revised manuscript):

"In their study, Chen et al. (2018) mainly focus on the development and improvement of the ALT retrieval algorithm, whereas the present study emphasizes using the ALT retrievals to assess the (fully independent) ALT simulations."

- Page 5, line 13: because you later speak about the spinup in the trend analysis, it might be interesting to say a few words about this here. The looping through the 36 years cannot given the same soil temperatures as you would normally have if you have realistic spinup data.

R1C9: We thank the reviewer for the suggestion. We added two sentences here (section 2.1, page 6, line 19-22 in the revised manuscript).

"The details of the spin-up procedure employed here admittedly impact our trend analysis (section 4.5); the approach makes use of the warmer conditions during the last few decades and thus should produce warmer 1980 initial conditions than would be produced with realistic historical forcing over hundreds of years (e.g., Sapriza-Azuri et al., 2018)."

- Page 8, line 15 and following: The assessment might be a bit too optimistic here: Basically one sees that the ALT is between 0.2 and 1m both in obs and simulations, not much more. Is there a significant correlation at all?

R1C10: We have modified this sentence as shown below (section 4.1, page 12, line 24-27 in the revised manuscript).

"Figure 4b, c demonstrates that in some ways, the CLSM-simulated results roughly agree, to first order, with the in-situ observations. The overall mean bias of simulated ALT relative to the in-situ measurements is -0.05 m. Nevertheless, the scatter (blue) in Fig. 4c is large, and the corresponding correlation coefficient is quite weak (0.27)."

- Page 8, line 21 and following: The AirMOSS ALT retrievals. Bascically the retrievals are the same everywhere! Around 0.45 m. No variability. Are they actually of any use?

R1C11: As we mentioned in the original manuscript, relatively larger variability with the ALT retrievals is seen at its native resolution (Figure 3a), but this larger variability was smoothed out through aggregation to the model scale at 81 km$^2$, as we expected. In addition, as also mentioned in the manuscript, these retrievals cannot exceed the P-band radar sensing depth of about 60cm over this area, and thus for the shallow permafrost here, the averaged ALT retrievals appears to be around 0.45 m everywhere.

We emphasize that this is a first attempt to compare remote sensing ALT data with modeling results. An expected future direction is to take advantage of the detailed heterogeneity information in the remote sensing data to downscale model results directly or to improve modeling skill indirectly. We have added several sentences about the potential use of these ALT retrievals into the discussion section (see our response to R1C1, part c, above). Please also see our response to the next comment (R1C12).

- Page 8, line 32 and following ("Excluding: : :"): Yes, OK, but then there is still no correlation. Values are just around 0.45 m and the mean ALT of the remaining sites just happens to be around that value.

R1C12: With further analysis, we found that the correlation coefficient between the ALT retrievals and the in-situ observations, while small at the site scale, is larger than that for modeled ALT at the model scale, both for all sites and for sites with measured ALT below the AirMOSS radar sensing depth of 60cm. This is the depth below which the AirMOSS radar is expected to lose sensitivity to subsurface features, and is calculated based on the radar system noise floor and calibration accuracy. Therefore, any retrieved ALT larger than ~60 cm is expected to have large uncertainties, and the error is further expected to grow linearly as the retrieved values of ALT essentially "saturate." We have added these discussion and included the Table R1 as a new Table 3 in the revised manuscript:

"Note that the retrieved ALT cannot exceed the radar sensing depth of about 60 cm. This is the depth below which the AirMOSS radar is expected to lose sensitivity to subsurface features, and is calculated based on the radar system noise floor and calibration accuracy. Therefore, any retrieved ALT larger than 60 cm is expected to have large uncertainties, and the error is further expected to grow linearly as the retrieved values of ALT essentially "saturate." This limitation may also lead to underestimates of the actual thaw depth. " (section 2.2, page 7, line 18-22 in the revised manuscript)

"For the AirMOSS retrievals, when all in-situ sites are considered, the overall ALT bias is -0.11 m at the site scale and -0.12 m at the model scale. While the correlation coefficient with the in-situ observations is only 0.05 at the site scale, it is 0.61 at the model scale. " (section 4.1, page13, line 4-7 in the revised manuscript)

"Excluding the sites with in-situ ALT measurements that exceed the AirMOSS sensing depth of ~60 cm, the overall mean bias for the AirMOSS retrievals at the model scale (site scale) drops to -0.01 m (0.02 m), and the correlation coefficient

at the model scale (site scale) increases to 0.64 (0.20) (Table 3). In contrast, the CLSM simulation results show a bias of 0.01 m and a zero correlation coefficient at the same sites." (section 4.1, page 13, line 12-15 in the revised manuscript)

*Table R1– Evaluation metrics for model-simulated ALT and AirMOSS retrievals for 2015.* *(New Table 3)*

| Metric | All sites | | | Sites with ALT measurements within AirMOSS sensing depth ( ~ 60 cm) | | |
|---|---|---|---|---|---|---|
| | CLSM-simulated ALT (model scale) | AirMOSS ALT retrievals (model scale) | AirMOSS ALT retrievals (site scale) | CLSM-Simulated ALT (model scale) | AirMOSS ALT retrievals (model scale) | AirMOSS ALT retrievals (site scale) |
| RMSE (m) | 0.17 | 0.17 | 0.21 | 0.12 | 0.06 | 0.08 |
| Bias (m) | -0.05 | -0.12 | -0.11 | 0.01 | -0.01 | 0.02 |
| R | 0.27 | 0.61 | 0.05 | -0.00 | 0.64 | 0.20 |

- Page 9, line 9-10: "Further investigation: : :": You could nevertheless elaborate a little bit on this. Are there any common characteristics of sites with thick active layer (dry soil, highly conducive soil, southward sloping, etc.) that the model doesn't get?

R1C13: Here we specifically meant some investigation on the zero-curtain problem with the model. Nevertheless, we have amended the text to read (section 5, page 21, line 3-5 in the revised manuscript):

"The use of the $0^{\circ}C$ degree threshold in CLSM for determining the thawed or frozen layer may explain in part the model's underestimation of ALT, as may the lack of

an explicit treatment of local aspect, errors in assigned model parameters, and so on."

(Note that we have moved this paragraph to the discussion section, as suggested by the Reviewer #3.)

- Page 11, line 1-3 – meteorological forcing dominant control: Of course. Could anyone seriously expect something different?

R1C14: Yes. We have added "as expected" into this sentence.

- Figure 7: Good that this is quantified in such a way here. Much more synthetic &interesting than figure 6.

R1C15: Thank you. We agree.

- Page 13, line 12: Correlation might increase if time steps with snow on the ground but air temp > 0_C are not counted in Tcum - it's the soil surface temperature that counts, not the air temperature.

R1C16: The reviewer makes a reasonable suggestion, and we thus recomputed the correlation map using this modified rule.  We found that the map did not differ very much from the original one.  We keep the original figure as is.

- Page 14, line 5 : Problems in mountain areas: Snow forcing might be severely in error in these regions

R1C17: Yes. We have added one sentence here (section 4.3, page 17, line 1-2 in the revised manuscript).

"In addition, MERRA-2 snow forcing might be severely erroneous in these regions."

- Page 14, line 12: "The reasons: : :" – I have probably missed the information: How deep is the model soil column?

R1C18: As mentioned in section 2.1, the depth ranges of the six soil layers are 0-0.1m, 0.1-0.3m, 0.3-0.7m, 0.7-1.4m, 1.4-3m, and 3-13m, respectively.  We checked the CALM sites over western Siberia and found that the ALT observations there are basically below 2 m. Therefore, the depth of the model's soil column is not an issue, and we deleted this speculation from the sentence. The new sentence below instead was added (section 4.4, page 17, line 11-12 in the revised manuscript).

"The reasons for this particular deficiency are unclear; perhaps the initial thermal conditions over western Siberia were too warm, or perhaps MERRA-2 overestimates current air temperatures in this region."

- Page 16, line 15, Mongolian ALT trends: How can you have a 25 cm/year trend over 17 years? That would mean that ALT increases by over 4 m over that period. That's quite improbable. These data are very suspicious.

R1C19: We share the reviewer's concerns about the quality of these data. Below in Table R2 we provide the time series of the actual ALT measurements at the three Mongolian sites M1, M1 A and M3 (https://www2.gwu.edu/~calm/data/north.html). Because M1 and M1 A are within the same CLSM modeling grid cell, we used the average of their two time series. Time series of observed and simulated ALT at the two grid cells containing these three sites are plotted in Figure R1. The observed and simulated ALT trends at the two grid cells correspond to the two dots showing the extraordinarily large observed trends in Figure 14a in the original manuscript. Note the simulated ALT trends were calculated using ALT estimates only in years when observed ALTs are available, as also mentioned in the manuscript.

*Table R2 – Observed ALT (cm) at three Mongolian sites.*

| Site Code | 1996 | 1997 | 1998 | 1999 | 2000 | 2001 | 2002 | 2003 | 2004 | 2005 | 2006 | 2007 | 2008 | 2009 |
|---|---|---|---|---|---|---|---|---|---|---|---|---|---|---|
| M 1 | 345 | 350 | 355 | 345 | 350 | 340 | 355 | - | 375 | 365 | 380 | 380 | 370 | 350 |
| M1 A | 390 | 390 | - | 430 | 485 | 400 | 450 | 600 | 770 | 710 | 820 | 820 | - | - |
| M3 | - | 610 | 620 | 600 | 610 | 660 | 720 | 760 | 770 | 750 | 760 | 770 | 800 | - |

[Figure]

*Figure R1: Time series of observed and simulated ALT at Mongolian sites collocated with two simulation grid cells, i.e., M1&M1A (upper) and M3 (bottom). The calculated ALT trends from observations and simulation are 24.38 cm/yr and 1.13 cm/yr, respectively, for the grid cell containing M1&M1A and are 19.69 cm/yr and 0.51 cm/yr, respectively, for the grid cell containing M3.*

We attempted to contact the PI responsible for these data (Dr. Natsagdorj Sharkhuu from the Institute of Geography, Mongolian Academy of Sciences); however, the email address (provided here https://www2.gwu.edu/~calm/data/webforms/mg_f.html) is apparently

obsolete, and the email delivery failed. We were thus unable to investigate further the data quality.

In any case, as indicated by the reviewer, this issue calls at the very least for a specific caveat about these data, which we have added (section 4.5, page 19, line 28-30 in the revised manuscript):

> "A particular caveat is required regarding the Mongolian sites, given the unusual observed trends calculated there. Attempts to contact the data providers to attain more detailed information for data evaluation were unsuccessful, and accordingly our confidence in these particular data is limited."

- Page 18, line 20 : ´n ..addition of soil layers Â˙z : Would that be so difficult? Tests with more levels would really be interesting, but if they are really difficult, I refrain from asking for such test to be carried out.

R1C20: In response to this comment we have conducted such tests. Our general conclusions are consistent with other studies in terms of how soil configuration affects permafrost modeling (e.g., Alexeev et al., 2007;Lawrence et al., 2008;Sapriza-Azuri et al., 2018;Nicolsky et al., 2007b;Dankers et al., 2011).

[Figure]

*Figure R2: Soil configurations we have newly tested.*

[Figure]

*Figure R3: Simulated ALT results at six Mongolian sites with 5 different soil configurations. (Note the baseline soil configuration contains 6 soil layers and a total soil depth about 13 m.)*

Specifically, we tested four new soil configurations with 15 soil layers and different soil depths (ranging from 13 m to 50 m). Figure R3 reveals that increasing the number of soil layers decreases ALT climatology at the tested sites, which is consistent with previous studies (e.g., Alexeev et al., 2007;Lawrence et al., 2008;Dankers et al., 2011). The figure also demonstrates that variations in total soil depth have only a small impact on the simulation of ALT, as also reported in the previous studies. However, the zero flux we employ at our lower boundary for all the simulations might influence heat transfer in deep soils and thus might decrease the impact of using deeper soils.

Based on these tests, we have included some appropriate discussion in the text, though without adding a figure (section 5, page 22, line 14-19 in the revised manuscript):

> "Test simulations (not shown) with alternative model configurations indicate that increasing the number of soil layers may act to decrease somewhat the simulated ALT, suggesting that our values may be a little overestimated; however, based on results from a new study by Sapriza-Azuri et al.(2018), our use of a no-heat-flux condition at the bottom boundary rather than a dynamic geothermal flux may lead to underestimates of ALT. Such uncertainties should naturally be kept in mind when interpreting our results. Our supplemental simulations also suggest that increasing the total modelled soil depth has only a small impact on simulated ALT."

We also explicitly mentioned our lower boundary condition (section 2.1, page 5, line 10-11 in the revised manuscript):

> "A no-heat-flux condition is employed at the bottom of the model's soil column."

Reviewer #2

Overall comments:

This paper used in-situ data and a remote sensing based ALT (active layer thickness) data to evaluate a model-based ALT dataset. Overall, I think it is a useful study. The analysis was done in a comprehensive way, and the results were acceptable. However, the analysis regarding the model uncertainty is somewhat general – considering we already have a good knowledge of the ability of global land models in NH permafrost simulation. I think, the study could benefit from more in-depth discussions on this aspect. The details were provided below.

We thank the reviewer for the careful reviewing. We absolutely agree with the reviewer about the importance of model uncertainty regarding ALT estimation. To further examine model uncertainty, we conducted new tests on sensitivity to organic carbon content and its vertical distribution. We also replaced the vegetation climatology at several Mongolian sites with satellite-based, time-variant leaf area index (LAI) during wintertime to investigate the impact of inter-annual variations in vegetation. Please also see our response to Reviewer #1 (R1C20) regarding the discussion about the soil layer configuration.

Major comments:

1. Page 4 Paragraph 3: I have questions regarding how the ALT was calculated. The paper indicates here it was calculated based on the simulated ice content. Does the model consider unfrozen water in frozen soils? If it does, please provide information on how the model calculates unfrozen water content. If not, this definition will be same as using a 0_C temperature threshold for thawed-to-frozen depth calculation. This information is especially important for the deep soils due to year-round low temperatures and coarse vertical resolution of the model at deeper depths.

R2C1: In our model, the ice fraction is unity if soil is fully frozen (i.e., $T < 0^oC$ and $f_{ice} = 1$), and it is zero if the soil is fully thawed (i.e., $T > 0^oC$ and $f_{ice} = 0$). The reviewer comment refers to the situation where the soil temperature is exactly at the freezing point and the soil is partially frozen (i.e., $T = 0^oC$ and $0 < f_{ice} < 1$). In the latter case, frozen and thawed soil and water coexist. This situation always occurs during freeze-to-thaw and thaw-to-freeze transitions. This is because in soil layers that are as thick as those used in the model, the phase transition does not occur instantly (relative to the model time step). More specifically, the model uses heat content as the prognostic variable, from which the temperature and ice fraction are diagnosed. Therefore, our calculation of ALT is not the same as simply using a $0^oC$ degree threshold to determine the thawed-to-frozen depth. Rather, we identify the deepest (fully or partially) thawed layer and then calculate the thawed-to-frozen depth based on the ice fraction within the layer. We have modified the paragraph about the ALT calculation as follows (section 2.1, page 5, line 22-29 in the revised manuscript).

"Precisely, the thawed-to-frozen depth is calculated as:

$$z_{bottom}(l) - f_{ice}(l, t) \times \Delta z(l), \hspace{2cm} (1)$$

where layer $l$ is the deepest layer that is fully or partially thawed, $z_{bottom}(l)$ represents the depth at the bottom of layer $l$, $f_{ice}(l,t)$ is the fraction of ice in layer $l$ at time t (i.e., $f_{ice}(l, t) \in [0 \ 1]$), and $\Delta z(l)$ is the thickness of layer $l$. To identify layer $l$ we use a 0°C degree temperature threshold. Specifically, T > 0°C degree indicates that a layer is fully thawed, T = 0°C degree indicates that a layer is partially thawed, and T < 0°C degree indicates that a layer is fully frozen. That is, layer $l$ is the deepest layer that satisfies $T(l) \geq$ 0°C. Equation (1) then expresses that the thawed-to-frozen depth is equal to the bottom depth of the layer $l$ but adjusted upward according to the ice fraction within the partially thawed layer $l$."

The above declaration seems contradictory to "The use of the 0_C degree threshold in CLSM for determining the thawed or frozen state of the soil may explain the model's underestimation of ALT." (Page 9, Paragraph 1). So I am confused what methods were actually used to determine the thawing depth/ALT. Please clarify.

R2C2: The 0°C degree is used to determine the deepest thawed layer $l$, and then the ALT is calculated by $z_{bottom}(l) - f_{ice}(l, t) \times \Delta z(l)$ as explained above (R2C1). Here we are talking about the temperature threshold used to determine the thawed state of the soil only, not the thawing depth/ALT. We have modified the text as follows (section 5, page 21, line 3-5 in the revised manuscript):

"The use of the 0°C degree threshold in CLSM for determining the thawed or frozen layer may explain in part the model's underestimation of ALT, as may the lack of an explicit treatment of local aspect, errors in assigned model parameters, and so on."

(Note that we have moved this paragraph to the discussion section.)

2. Page 5 Paragraph 3: The spin-up scheme is questionable, though the authors themselves acknowledged this. Why do the authors using the meteorology for the entire 36-year period for spin-up? If the design is to reduce the uncertainty introduced by a single-year surface meteorology, spin-up using the first few years during the period will be more acceptable.

R2C3: As the reviewer points out, we recognized this issue and discussed it in the original manuscript. No spin-up procedure is entirely problem-free. Using a shorter period for spin-up as suggested by the reviewer would exaggerate in the resulting initial conditions any anomalies that occur during the spin-up period. The ultimate solution would be to construct a realistic historical forcing dataset over hundreds of years with a dynamic geothermal flux applied to the bottom boundary of soil column (e.g., Sapriza-Azuri et al., 2018). However, this approach is hardly feasible and would still not assure absolutely correct initial conditions. Please also see our response to Reviewer #1 (R1C9).

3. I have questions regarding the vegetation effects on permafrost simulation in Northern Alaska (Page 11, Paragraph 2). Those 4 northern flights were dominated by "dwarf trees"

as indicated by Fig. 2b (really?). Moreover, the changes in simulated maximum snow depth due to vegetation in those flights were much smaller comparing with the experiment homogenizing the forcing data (Fig. 6c). So I would expect the impact due to snow changes for the homogenizing vegetation experiment would be smaller comparing with the experiment homogenizing surface forcing, while this is not the case shown in Fig. 6a-b. Can the authors explain why?

R2C4: As we mentioned in the manuscript, the homogenization is applied cumulatively. Before we homogenized vegetation in this experiment, we already homogenized the forcing. The differences between HomF and HomF&Veg are then attributed to the changes in vegetation parameters (specifically LAI and vegetation height).

Fig.6a-b indeed demonstrates that the impact due to snow changes for the homogenizing vegetation experiment (differences between HomF and HomF&Veg) would be smaller compared to the experiment homogenizing surface forcing (differences between Baseline and HomF). Figure 7b also illustrates this in a quantitative way.

On the other hand, the Alaska North slope is dominated by tundra, while the vegetation map in CLSM indicates mostly dwarf trees or shrubs (Fig. 2b). Would this introduce uncertainties to the analysis on the contribution of different factors (i.e. forcing data, vegetation and soil) in Fig. 6?

R2C5: The vegetation class is only one of several model inputs. The land cover class used in the study is derived from the USGS Global Land Cover Characteristics Data Base Version 2.0 (GLCCv2). In addition to vegetation class, the model uses vegetation height, LAI, greenness fraction and albedo, which are all obtained from other satellite-based sources that reflect realistic climatologic conditions for tundra. Put differently, while the modeled vegetation class may suggest the presence of dwarf trees, the typically low (satellite) LAI values in northern Alaska will instruct the model that the tree cover is extremely sparse in this region. Please refer to Table 1 in (Tao et al., 2017) to see all the data sources. We have added one sentence to clarify this (section 4.1, page 12, line 8-11 in the revised manuscript).

> "Note that although the vegetation class (Figure 2b) suggests the presence of dwarf trees over the Alaska North Slope, the actual satellite-based LAI, vegetation height, greenness fraction and albedo will still instruct the model that the tree cover is extremely sparse in this region. The data sources for these vegetation-related boundary conditions can be found in Table 1 in Tao et al. (2017)."

Also, from Fig. 6, it does not seem to me that homogenizing soil parameters has much bigger impact on simulated ALT and surface soil temperature than homogenizing vegetation (at least for the northern flights), as the authors indicated in Fig. 7. Maybe I miss something here?

R2C6: We have added the following discussion to section 4.2 (page 15, line 21-26 in the revised manuscript).

> "Note that in Figure 6a, the soil impact on ALT (difference between HomF&Veg&Soil in black and HomF&Veg in red) appears smaller than that of the vegetation (difference between HomF in green and HomF&Veg in red) over the northern transects (ATQ, BRW and DHO).  Even so, Figure 7b shows that, in terms of the integrated impact along the transects, the soil influence clearly outweighs the influence of vegetation – at several other transects, including HUS, KYK, COC, AMB, IVO and the first half of ATQ (where vegetation conditions might be similar to those used for homogenizing), the changes in vegetation parameters do not have much impact. "

4. Most of the value for AirMOSS radar retrievals, I think, is in its ability to characterize land surface heterogeneity. Simply averaging AirMOSS data to a much coarseresolution (i.e. 9km in this study) to compare with the land model simulations is not very insightful in terms of exploring the value of this dataset. I agree with the authors that the current AirMOSS retrievals seem having large uncertainties; most notably, its ALT retrievals were in a very narrower range. However, the inconsistency of the ALT spatial pattern at some of AirMOSS flights may come from the model itself. For example, at the DHO flightăá Tthis is the flight with most in-situ sites available, the model ALT generally increases from the north to the south, while the in-situ data show large variability, and do not show a clear increasing trend from the north to the south (Fig. 5a). There are also a number of studies pointing out that ALT is extremely variable at local scale. Therefore, analysis using a dataset like AirMOSS in this aspect would be more valuable.

R2C7: We agree with the value of AirMOSS radar retrievals in terms of (theoretically) being able to represent the spatial variability of ALT.

The differences in the spatial patterns of the AirMOSS ALT retrievals and the simulated ALT suggest that neither radar remote sensing nor modeling is perfect. As we mentioned in the manuscript, the radar sensing depth (about 60cm) strongly constrains the retrieval accuracy for ALT values larger than the sensing depth. We expanded on the analysis by adding a new Table 3, which provides several evaluation metrics for ALT restricted to less than 60cm.  (Please see our response in R1C12 for details about this new Table 3.) The table suggests that the radar retrievals are in better agreement with in-situ observations especially at model scale when only using sites that have ALT less than or equal to 60cm. We have added several sentences to section 4.1 (page 13, line 12-15 in the revised manuscript):

> "Excluding the sites with in-situ ALT measurements that exceed the AirMOSS sensing depth of ~60 cm, the overall mean bias for the AirMOSS retrievals at the model scale (site scale) drops to -0.01 m (0.02 m), and the correlation coefficient at the model scale (site scale) increases to 0.64 (0.20) (Table 3). In contrast, the CLSM simulation results show a bias of 0.01 m and a zero correlation coefficient at the same sites. "

5. It would be more interesting if the authors could provide more insightful analysis regarding the model ALT uncertainties or the correlation analysis, including:

a) Why does the model show much stronger correlation with maximum SWE in portions of NH permafrost region than with air temperature?

R2C8a: Honestly, we are not clear about the reasons yet. We have modified a relevant sentence as follows (page 16, line 29-30 in the revised manuscript).

> "Apparently, in these areas, the impacts of snow physics on ALT outweigh the impacts of lumped energy input ($T_{cum}$), for reasons that are not clear."

b) It would be helpful if the authors could give more explanations regarding why the model fails in western Siberia? Those areas also include continuous permafrost, so I do not think it is too challenging for global models to capture the permafrost distribution there.

R2C8b: As indicated in Figure 1b, all four types of permafrost (i.e., continuous, discontinuous, sporadic and isolated) are present in western Siberia. The literature suggests that other global models also missed this portion of permafrost, including CLM3 and CCSM3, although the updated versions of these models (i.e., CLM4 and CCSM4) demonstrated improved performance (Lawrence et al., 2012). Similarly, Guo et al. (2017) also reported underestimations in permafrost extent in western Siberia simulated by CLM4.5 when driven with three different reanalysis forcings (CFSR, ERA-I and MERRA) and improved performance extent in this region when using forcing data from a different reanalysis. We have added some discussion to compare our simulation with other existing works (section 4.4, page 17, line 12-20 in the revised manuscript):

> "Note that some other global models, such as CLM3 and the Community Climate System Model version 3 (CCSM3) as reported in Lawrence et al. (2012), also missed this area of permafrost and that updated versions of these models (i.e., CLM4 and CCSM4) showed improved performance in this regard (Lawrence et al., 2012). Guo et al. (2017) reported underestimated permafrost extent simulated in western Siberia using CLM4.5 driven by three different reanalysis forcings (i.e., CFSR, ERA-I and MERRA), and they showed an improved simulation of permafrost extent in this area when using another reanalysis forcing, the CRUNCEP (Climatic Research Unit - NCEP) (Guo and Wang, 2017). Guimberteau et al. (2018) found similar improvements stemming from the use of CRUNCEP forcing. We leave for further study whether the MERRA-2 forcing data is responsible for the western Siberia deficiency seen in our own results."

c) The ALT trends shown in Fig. 13a seem not very consistent with the trends of temperature indices shown in this figure. Have the authors explore the changes in snow cover duration? A longer snow free season generally leads to warmer soil temperature and thus deep ALT esp. in the southern area.

R2C8c: We did examine the trend in snow cover duration (see Figure R4). While some areas show a trend in snow cover duration, this trend does not seem correlated with the trend in ALT.

[Figure]

*Figure R4: Spatial distribution of trend in snow persistence days when daily mean snow depth > 25cm. NT means no significant trend, i.e., the p value is larger than 0.05.*

Accordingly, we modified the relevant sentence as follows (section 4.5, page 19, line 18-20 in the revised manuscript):

> "It is possible that in such cases, the computed trends are strongly affected by snowpack variability, though neither maximum SWE nor snow cover duration tends to show a significant trend in these areas (not shown)."

d) Page 18, Paragraph 3: I do not quite agree with the authors' explanation why the model fails in the Mongolian sites. For those sites, the model simulated ALT is generally less than 1.4 m, therefore, the coarse resolution at deeper soils in the model set-up (i.e. layer 5: 1.4-3m) should not be a major contributing factor there. Much drier conditions, sparse vegetation, and perhaps uncertainties in soil texture data (eps. in deep soils), I think, are more likely contributing more to the model uncertainties.

R2C8d: The reviewer is correct about the soil configuration. We conducted new tests using different soil configurations at several Mongolian sites. Please see the results and discussion in our response to the Reviewer #1 (R1C20).

The reviewer also raised a very good point regarding the influence on ALT of soil wetness, vegetation and uncertainties in soil texture for deep soils. We first examined realistic satellite-based LAI data at from Moderate Resolution Imaging Spectroradiometer (MODIS) MCD15A2H product and the Advanced very-high-resolution radiometer (AVHRR) AVH15C1 product (see Table R3). Figure R5 then shows the time series of the MODIS and AVHRR LAI, along with the LAI climatology used in the model at one CALM Mongolian site (M11). A post-processing procedure that included quality screening and gap filling was applied to the two satellite LAI products. The CLSM LAI climatology is used for the years that MODIS data is not available (1980 to 2002).

*Table R3 – Information of satellite-based LAI products from MODIS and AVHRR.*

| Sensor | Dataset | Product | Resolution | Temporal Granularity | Temporal Extent |
|--------|---------|---------|------------|---------------------|-----------------|
| **MODIS** | MCD15A2H | Leaf Area Index and Fractional Photosynthetically Active Radiation | 500 m | 8-day Composites | July 2002 - Present |
| **AVHRR** | AVH15C1 | Leaf Area Index and Fraction of Absorbed Photosynthetically Active Radiation | 0.05deg | Daily | June 1981 - Present |

[Figure]

*Figure R5: The LAI time series at one Mongolian site (M11). Green line represent the original LAI climatology used in the CLSM. Blue and red dash line represent the realistic (time-varying) LAI data from MODIS and AVHRR. The CLSM LAI climatology is used for the years that MODIS data is not available (1980 to 2002, shaded area).*

As illustrated by Figure R5, MODIS shows smaller LAI than AVHRR over the valid period after 2002. The LAI climatology used in the model is inbetween of the two products. The differences between the LAI climatology used in the model and realistic LAI products would cause differences in energy and water partitioning at the land surface via impacting surface albedo. We conducted a simple test to examine the impact of using the more realistic, inter-annually varying vegetation inputs on the winter surface albedo and thus the snow accumulation process, which in turn would impact ALT estimates. Specifically, we replaced the LAI climatology with the satellite-based, inter-annually varying LAI products in the model, but turned off the impacts in summer, i.e., not affecting the snow-free albedo. The simulation results show only minimal differences in the estimated ALT. That is, the winter surface albedo when using realistic satellite LAI products does not differ very much from that using the original LAI climatology. However, we speculate that large differences in summer could have significant impact on ALT estimation. We leave further investigation for future work. We have added one sentence in the manuscript to bring up this issue (section 5, page 22, line 27-29 in the revised manuscript).

"Another issue affecting our ALT comparisons is the climatological representation of vegetation parameters such as LAI used in CLSM. Additional investigation (not

shown) revealed large differences between the LAI climatology used in CLSM and more realistic, time-varying, satellite-based LAI products at several Mongolian sites."

Without any further information about the soil parameters in deep soils, we could not conduct further tests. Here we provided the results on the sensitivity of soil organic carbon and the vertical distribution in Figure R6. The figure reveals that a deeper ALT results from reducing the SOC content and from using a very different vertical distribution profile that arbitrarily concentrates less carbon in the top soil. Indeed, changing the vertical distribution profile for SOC content plays an almost equivalent role to reducing the SOC content. We added relevant discussion regarding this (section 5, page 22, line 20-23 in the revised manuscript).

"Uncertainty in our description of soil organic carbon, i.e., both soil carbon content and vertical carbon distribution, leads to corresponding uncertainty in our ALT simulations. We indeed find a significant improvement in simulated ALT at several Mongolian sites when we arbitrarily impose less total soil carbon content and concentrate less soil carbon in top layers (not shown)."

This further confirms the reviewer's comment regarding the importance of vertical variation of soil properties. Thus, we added one sentence here to bring up the issue about vertical variation in soil parameters (section 5, page 22, line 23-25 in the revised manuscript).

"Besides the vertical distribution of soil carbon, the vertical variation in other soil hydrological properties (e.g. soil texture, porosity, hydraulic conductivity, etc.) should also play a significant role since they all affect soil thermal conductivity and heat capacity."

[Figure]

*Figure R6: Simulation results at six Mongolian sites with different soil carbon contents and vertical carbon distributions. "OriSOC_OriProf" – Original soil organic carbon (SOC) content vertically distributed with the original profile as used in baseline simulation. "SOC/N_OriProf" – Reduced soil organic carbon content (by dividing the original SOC content by N) vertically distributed using the original profile. "OriSOC_NewProf" –*

*Original SOC content vertically distributed with a new profile which arbitrarily concentrates less carbon in top soils. "SOC/N_NewProf" – Reduced SOC content (by dividing the original SOC content by N) vertically distributed using the new profile.*

Minor comments:

1. Fig. 4: The text in section 2.3 indicates most of the comparisons against AirMOSS data will be at the 4 flights in the Alaska north slope. So I wonder why COC flight was included in this figure esp. considering the AirMOSS retrievals were not included?

R2C9: We only have AirMOSS retrievals for IVO, ATQ, BRW and DHO, not for COC. However, our model provides results here, thus we included COC to add two additional measurements to compare with model results (Figure 4b).

2. Fig. 4 a-b: were the sites arranged according to the latitudinal changes?

R2C10: We arranged the sites aligning with the flight direction. We also added this into the caption of Figure 4.

"The sites are arranged aligning with the flight direction."

3. Page 9, Line 21: Does MERRA-2 not provide air temperature at 2-m surface height?

R2C11: MERRA-2 does provide output of hourly 2-m air temperature. However, the land model within the MERRA-2 system is forced with the air temperature in the lowest (atmospheric) model layer (TLML), and the 2-m temperature is simply diagnosed from TLML and the surface temperature. For consistency, the off-line (land-only) model simulations presented here were likewise driven with TLML from MERRA-2. In any case, the sentence in question was not necessary and only caused confusion, so we deleted it.

4. section 3.2 and Figure 6: Part of the IVO flight lies in the Brooks mountain range with very low SOC content (Fig. 2c). It may be not a good representative of the average conditions in this area, at least considering SOC variability.

R2C12: The point at which we extracted the soil parameters does have a sort of intermediate SOC (greenish color in Figure 2c) which is also shown in Figure 5b. As mentioned in the original manuscript, however, we actually used an arbitrary intermediate SOC value which is 40 kg/m$^2$.

We also conducted two additional simulations using a very large and a very small SOC value everywhere along the AirMOSS transects. The results are shown in Figure R7 below. The "IntermC" used the same SOC that was used for homogenization (i.e., 40 kg/m$^2$). "LowC" and "HighC" used the lowest (10kg/m2) and highest (120kg/m2) SOC values found along the transects as shown in Figure 5b and also in Figure R7b. Figure R7a reveals that the model sensitivity to soil carbon is much larger for lower SOC than for higher SOC, and easily gets saturated for high SOC (i.e., larger than ~100 kg/m$^2$). However, all of this depends on the vertical soil carbon distribution profile used. Please also see R2C8d.

[Figure]

*Figure R7: a) similar to the Figure 5a in the original manuscript but showing model sensitivity to organic carbon content along the AirMOSS flight transect. b) the organic carbon content along the transect.*

We have added a sentence into the section 3.2 (page 10, line 8-10 in the revised manuscript).

> "Our investigation reveals that the model sensitivity to soil carbon content is much larger for lower SOC than for higher SOC, and easily gets saturated for high SOC (i.e., larger than ~100 kg/m$^2$) (not shown). Thus, we trust that 40 kg/m$^2$ is an appropriate value representing an intermediate SOC condition."

5. section 3.2: Would it be easier to follow if the description regarding the idealized experiments was included in the Methods section?

R2C13: We thank the reviewer for the helpful suggestion. We have introduced the idealize experiments in section 2 by adding a new Methods section (section 2.5).

6. Fig. 7 is not very informative. I suggest summarizing the results in a table.

R2C14: We feel that Figure 7 is informative, which is also supported by Reviewer #1 (R1C15). The figure displays the actual values and therefore also serves as a table. We therefore opt to keep the figure in its current form.

**Reviewer #3**

The authors compare model estimates (based on re-analyses data) of active layer thickness (ALT) with different data sources. This includes in situ measurements from an international database and an unpublished approach of airborne P-band estimation. The latter is available over Alaska. General evaluation is made for the northern hemisphere.

In addition, the model input is investigated for linear relationship with its output. The authors seek to find a simplified relationship to explain ALT change over time (driven by degree days, snow water equivalent etc).

General issues

The manuscript addresses an important topic, the modelling of active layer thickness over time. The presentation of the manuscript and setup of the experiment is however problematic.

1) The authors report 'reasonable' or 'good' results with limited and partially missing quantitative reasoning.

R3C1: We have included additional quantitative metrics to better illustrate our results. Please see our responses in R3C6, R3C8, R3C24, R3C29 and R3C31 for further details.

2) The results of the comparison to the unpublished airborne (AirMOSS) approach suggest that it actually does not work (AirMOSS values are all at the same level, not representing the in situ range, Figure 4). The comparison to this unpublished approach should be removed from the manuscript.

R3C2: We thank the reviewer for pointing out this. However, we do not agree with the reviewer regarding this point. We believe the AirMOSS data has value, which is also the opinion of Reviewer #2. We have expanded the relevant discussion about the potential use of the AirMOSS retrievals in the manuscript. The most important thing to note here is that the performance metrics for AirMOSS retrievals are quite good when evaluated within the instrument's sensing depth of approximately 60 cm (as shown in the new Table 3 in the revised manuscript), but are known to deteriorate for larger values of ALT.

Please see specifically our response to R1C11, R1C12 and R2C7, with particular attention to the Table R1 (new Table 3 added in the revised manuscript).

3) The stated objectives comprise issues regarding active layer thickness. The results section however does also cover permafrost extent.

R3C3: We thank the reviewer for pointing out this. We do not solely focus on ALT. We now modified the statement of our objectives in section 1 (page 4, line 24-27 in the revised manuscript):

> "Overall we pursue three scientific objectives: 1) evaluate the relative importance of the factors that determine the spatial variability of ALT, 2) evaluate CLSM-simulated ALT climatology and permafrost extent against observations, and 3)

quantify and assess the large-scale characteristics of ALT (in terms of means, interannual variability and trend) in Northern Hemisphere permafrost regions from 1980 through 2017. ''

4) In addition, common structure of methods, results and discussion is lacking. Methods are included in the introduction and mostly in the results section. The discussion is largely included in the results section. The 'approach' section mostly only includes the dataset description.

R3C4: We have added a new section ("section 3 Methods") and moved all the method description to this new section. We also moved some discussion about our results to the last section to supplement other 'discussion' of the research findings.

5) Some of the used datasets are not appropriately cited. See detailed comments below

R3C5: We thank the reviewer for pointing out this. We have checked our reference list and correctly cited all of the datasets that we used. Please also see our response in R3C10 and R3C11 for additional details.

6) A main weakness of the manuscript is that the results are not discussed with respect to already published material on permafrost related parameter modelling and on the parameters which are investigated for explanation as drivers of ALT change over time. The role of temperatures and snow water equivalent for active layer thickness is known (what is presented as one of the main results in the abstract). The novelty of the paper is not clear.

R3C6: We have added extensive text that further highlights the novelty of our study. Please see our responses in R1C1 and R1C2. In the discussion below we focus specifically on the comparison between our results and other existing studies, for the reviewer's interest.

Given that different models run at different spatiotemporal resolutions, and that the simulated results were evaluated using different observation data sets (or using different time periods of the same data set), a direct and credible comparison with existing model-simulated permafrost data sets requires extensive analysis and is not conducted here as it is considered beyond the project scope. Instead, we did a rough comparison with existing permafrost products simulated by land models with reanalysis-based forcing as shown in Table R4 below. (Note that ALT products simulated with climate forcing that is not observation-based are not included here.) We also want to emphasize that many existing studies define permafrost as areas with an ALT less than 3 m (Dankers et al., 2011) or 3.8 m (Guo et al., 2017;Guo and Wang, 2017;Lawrence et al., 2012), i.e., focused on the "near-surface permafrost". We do not specifically limit ALT within a certain depth, although our simulated results generally are less than or equal to the bottom depth of the 5th soil layer (2.95m). Also, many of these studies evaluate the simulated results for "present-day" (i.e., before 2000). To make a relatively fair comparison with the existing studies, we used the same validating time period.

*Table R4 – Existing available ALT products and evaluation metrics over Northern Hemisphere simulated by land surface models with reanalysis-based forcing.*

| Model and Soil Configuration | Forcing | Spatiotemporal Domain and Resolution | Findings and Conclusions Regarding ALT and Permafrost Extent | Reference |
|---|---|---|---|---|
| JULES (Joint UK Land Environment Simulator) (total soil depth 3m) | GSWP2 (Global Soil Wetness Project 2) WATCH (WATer and Global Change) | 1983–1995 at 1°×1° resolution with GSWP2 covering areas north of 25°N (includes the Tibetan Plateau). 1959–2000 at 0.5°×0.5° resolution with WATCH covering areas north of 45°N (does not include the Tibetan Plateau). | - Captures 97% of the continuous and discontinuous permafrost areas. - Overestimates the total extent; simulates permafrost where it only occurs sporadically or only in isolated patches (25%); overestimates an additional 14% in areas permafrost free. ALT: JULES-simulated ALT is generally too deep compared with CALM observations: mean bias in the GSWP2 run (1990–1995) is 0.81±0.48 m, and 0.53±0.50m in WATCH-GPCC (1990–2000). The Root Mean Square Error (RMSE) is 0.94 and 0.73m, respectively. | Dankers et al. (2011) |
| ORCHIDEE-MICT land surface model (total soil depth is 38 m) | GSWP3 (Soil Wetness Project Phase 3) CRUNCEP (Climatic Research Unit - NCEP) | Northern Hemisphere (>30° N) at 1°×1° spatial resolution (does not include the Tibetan Plateau). 1901–2007 with GSWP3; 1901–2015 with CRUNCEP. | - Both simulations underestimate permafrost extent when using ALT<3m as the definition of permafrost; CRUNCEP-forced simulation shows better permafrost extent using an alternative definition of permafrost. (No actual bias value provided.) - GSWP3-forced model generally overestimates ALT by more than 1m; CRUNCEP-forced output shows relatively better agreement with the observations. (No actual bias value provided.) | Guimberteau et al. (2018) |
| CLM 4.5 (50m soil depth) | CRUNCEP (Climatic Research Unit - NCEP) | Northern Hemisphere (includes the Tibetan Plateau). 1901 – 2010 | (Defines near-surface permafrost as that within the upper 3.8 m of soil.) - Simulated present-day (mean from 1981 to 2000) permafrost distribution; shows good agreement – only discrepancy is on the Tibetan Plateau. Bias in | Guo and Wang (2017) |

| | | Spatial Resolution: 0.5°×0.5° | permafrost extent is 2.02 x $10^6$ $km^2$.

- Global ALT trend with a correlation coefficient and an The Nash-Sutcliffe Efficiency (NSE) of 0.73 and 0.21, respectively. The simulated trend was smaller than observed trend. (No evaluation for ALT climatology provided.) | |
|---|---|---|---|---|
| CLM 4.5

(50m soil depth) | CFSR: NOAA Climate Forecast System Reanalysis

ERA-I: European Centre for Medium-Range Weather Forecasts Re-Analysis Interim

MERRA: NASA Modern Era Retrospective-Analysis for Research and Applications | Northern Hemisphere (includes the Tibetan Plateau).

Spatial Resolution: 0.5°×0.5°

1979 -2009 | (Permafrost is defined as ground where monthly soil temperature is less than 0°C for 24 consecutive months in at least one layer of the upper 10 soil layers (3.8 m) (Lawrence et al., 2012).)

- The model underestimates the permafrost extent in southern Alaska, northern Western Siberian Plain, and over the Tibetan Plateau.

Specific evaluation metrics:

*(table below)* | Guo et al. (2017) |
| CLM3 and CLM4 (offline)

3.5m depth with CLM3 and 3.8m depth with CLM4. | Forced with observed meteorological data based on NCEP-DOE. | Northern Hemisphere (includes Tibetan Plateau);

Simulation period: 1980–1999; | (Focused on shallow-permafrsot with ALT < 3.8 m.)

- CLM3 simulation underestimates permafrost in southern and western Siberia.
- CLM4 simulations show improved permafrost extent. | Lawrence et al. (2012) |

| Index | Active layer thickness (m) | | |
|---|---|---|---|
| | Climatology CALM, 1991–2000) CFSR, ERA-I, MERRA | Climatology (AL_RHST, 1981–1990) CFSR, ERA-I, MERRA | Change (CALM, 1996–2007) CFSR, ERA-I, MERRA |
| Mean bias | 0.21, 0.33, 0.66 | −0.33, −0.29, −0.11 | |
| Mean absolute bias | 0.52, 0.63, 0.89 | 0.50, 0.44, 0.52 | |
| Correlation coefficient | **0.69, 0.62, 0.51** | 0.18, **0.41,** 0.12 | **0.62, 0.83, 0.75** |
| Nash-Sutcliffe efficiency | 0.42, 0.25, −0.34 | −0.56, −0.20, −0.47 | −0.64, 0.36, −3.99 |

| | | 0.9375° latitude × 1.25° longitude | - In general, ALT is overestimated; CLM4 underestimated ALT in regions of shallow permafrost. (No actual evaluation metrics provided.)

- CLM3-simulated continuous and discontinous permafrost extent is $11.1 \times 10^6$ km$^2$ for the period 1970–89, which is below $11.8$–$14.6 \times 10^6$ km$^2$ estimate from Zhang et al., 2000.

- CLM4-simulated continuous and discontinous permafrost extent is $13.7 \times 10^6$ km$^2$ for the period 1970–89. | |

Existing studies that evaluated ALT that can be compared with ours include Dankers et al. (2011) and Guo et al. (2017). Other studies did not report specific evaluation metrics within a common simulation domain as ours, hence, they cannot be used for comparison. Comparing with reported evaluation metrics as shown in Table R5, Figure R8 and Figure R9 reveal that: 1) for the early 1990s (i.e., 1990 – 1995 period), our simulated results show a much better agreement with the observations than when using JULES with GSWP2 forcing as reported in Dankers et al. (2011); 2) for the whole 1990s (i.e., 1990 – 2000) period, our results are better than that using JULES with WATCH-GPCC forcing or when using CLM4.5 with both ERA-I and MERRA forcing as reported in Guo et al. (2017); and 3) our results show a smaller mean absolute bias and a larger correlation coefficient than that using CLM4.5 with CFSR, but show worse performance regarding mean bias and NSE. In addition, all of these existing studies overestimated ALT at the global scale and reveal a positive bias while our results underestimate deep ALT, and hence, reveal a negative bias. However, it is worth noting that our results demonstrate much better agreement against observations for regions of shallow permafrost as shown in Figure R8 and R9. We now added a short discussion along the lines of the following (section 4.4, page 18, line 18-20 in the revised manuscript):

> "Note that the existing literature on simulated ALT fields (e.g., Dankers et al. (2011), Lawrence et al. (2012) and Guo et al. (2017)) reveals a general tendency for models to overestimate ALT climatology at the global scale. Our results here suggest that the CLSM-simulated ALT fields are among the better simulation products, especially for shallow permafrost."

*Table R5 – Summary of evaluation metrics for ALT estimates reported in literature. The same metrics calculated with the simulation results in this study for an identical evaluation period are provided.*

| Evaluation metrics | (Dankers et al., 2011)* with two different sets of climate forcings | | | | (Guo et al., 2017)# with three different sets of climate forcings | | | |
|---|---|---|---|---|---|---|---|---|
| | GSWP2 (1990–1995) | **This Study (1990–1995)** | WATCH-GPCC (1990–2000) | **This Study (1990–2000)** | CFSR (1991-2000) | ERA-I (1991-2000) | MERRA (1991-2000) | **This Study (1991-2000)** |
| RMSE (m) | 0.94 | 0.18 | 0.73 | 0.72 | None | | | |
| Mean Bias (m) | 0.81 | 0.04 | 0.53 | -0.17 | 0.21 | 0.33 | 0.66 | -0.24 |
| Mean Absolute Bias (m) | None | | | | 0.52 | 0.63 | 0.89 | 0.36 |
| Correlation Coefficient | | | | | 0.69 | 0.62 | 0.51 | 0.70 |
| Nash-Sutcliffe Efficiency (NSE) | | | | | 0.42 | 0.25 | -0.34 | 0.25 |

*Evaluation was conducted with annual ALT.
**Evaluation was based on ALT climatology.**

[Figure]

*Figure R8: Observed and simulated ALT at CALM observations sites. a) copied from Dankers et al. (2011). The comparison was made for the period 1990–1995 using GSWP2 forcing, and for the period 1990–2000 using WATCH. b) Results from this study.*

[Figure]

*Figure R9: a), b) and c) are copied from Guo et al. (2017). Black dots represent the validating pair of data using CALM observations whereas the blue dots represent the validating pair of data using AL_RHST observations. d) shows comparison results with ALT estimates in this study. All subplots are for the validating period of 1990 to 2000.*

In terms of the evaluation of simulated permafrost extent, although it is not fair to compare our results with these existing studies due to differences in simulation domain, we did conduct similar calculations to quantitatively evaluate our simulated permafrost extent against other studies. In addition, most of these existing studies compare the model simulated permafrost area with that of the total area of continuous and discontinuous permafrost from Brown's map due to their coarse spatial resolution (i.e., at least 0.5°). Given the high resolution (i.e., roughly 9km × 9km) of our simulation results, we compare and discuss our simulated permafrost area with that of the total area of continuous, discontinuous, **and** sporadic permafrost together as shown in Figure 10 in our manuscript. For the revised manuscript, we have incorporated these evaluation results into a new table, as shown below (Table 4 in the revised manuscript). We also compared the simulated permafrost extent with that of the total area of continuous and discontinuous permafrost area from Brown's map in order to better compare our results with those of existing studies, which is shown in the parentheses in Table R6 below (new Table 4 in the revised manuscript).

*Table R6– Evaluation results for simulated permafrost extent. (New Table 4)*

| Case | CLSM | Obs. | Simulated Area ($\times 10^6$ km$^2$) | Percentage Relative to Observed |
|------|------|------|------------------------------|-----------------------------------|
| 4 | No | No | 48.8 | - |

| 3 | Yes | No | 1.9 | - |
| 2 | No | Yes | 3.2 (1.7) | 18.7% (12.3%) |
| 1 | Yes | Yes | 13.8 (12.3) | 81.3 % (87.7%) |

We also added the following short discussion into the revised manuscript (section 4.4, page 17, line 27-31 in the revised manuscript):

"The specific areas of each type shown in Figure 10a are listed in Table 4. The simulated permafrost extent covers 81.3% of the observation-based area (i.e., the total area of continuous, discontinuous and sporadic permafrost regions), and misses 18.7% of the observed permafrost area. When comparing simulated permafrost extent with only continuous and discontinuous types, these metrics change to 87.7% and 12.3%, respectively. Meanwhile, the permafrost extent is overestimated by $3.2 \times 10^6$ km$^2$."

Regarding the role of temperature and SWE, although it is well known that these two factors affect permafrost, the *relative* contribution of the two is still unclear. This study attempts to quantitatively estimate this relative impact, which, to our knowledge, has heretofore not been done. This, we feel, is indeed one of our useful, novel contributions to the permafrost literature.

Detailed comments

Title: the title is misleading; the paper focuses on active layer thickness what should be reflected in the title

R3C7: We do not solely focus on ALT. Simulated permafrost extent is also evaluated and discussed in the manuscript. We believe our title regarding "variability" implicitly includes both ALT and permafrost extent. In addition, ALT might be too technical to be included in a title since many people know about permafrost but may not know what ALT is. We then retain our title as is.

Abstract:

Line 19: 'measurements demonstrates reasonable skill' – what is 'reasonable'? add numbers

R3C8: We have added the following sentence in order to clarify what we mean by "reasonable" skill (Abstract, page 1, line 18-19 in the revised manuscript).

"Specifically, the RMSE (and bias) of climatological ALT is 1.22 m (and -0.48 m) across all sites and 0.33 m (and -0.04 m) without the Mongolia sites."

Line 27: 'significant degradation, with ALT increasing up to 0.5 cm/year' – it should be noted that this cannot be confirmed with in situ observations

R3C9: We thank the reviewer for pointing out this. This is very true. Below we provide the same figure as our Figure 14a in the original manuscript but only showing sites at which both the observed and estimated ALT trend is found to be statistically significant. Only four and ten sites are shown below with p value less than 0.05 and 0.10, respectively. Although the estimated and observed trends are closely clustered around the 1:1 line, the limited number of sites shown below cannot assure the accuracy of the estimated trends.

[Figure]

*Figure R10: ALT trend from CLSM estimates vs. CALM observations, based on years common to both datasets within the period of 1990 to 2017. The horizontal and vertical error bars represent 95% confidence intervals of observed ALT trend (regression slope) and CLSM-simulated ALT trend, respectively. This current figure is similar to Figure 14 in the original manuscript, but only showing sites with both observed and estimated trends that are statistically significant at a) 0.05 and 0.10 level.*

For the manuscript, we have added an additional description of the significance of estimated trends into the caption of Figure 14, and we also added the following to the discussion in the Abstract (page 1, line 28-29 in the revised manuscript):

> "It is difficult, however, to adequately assess the accuracy of the simulated ALT trends given the limited availability and relatively short records of in-situ measurements"

We also added a sentence into section 4.5 as shown below (page 19, line 30 to page 20, line 2 in the revised manuscript):

> "The overall comparison in Figure 14 is, in any case, highly uncertain, given the limited number of data points available to compute the trends. Note that only four (ten) points remain when screening out sites at which either the observed or estimated ALT trend is not statistically significant at the 0.05 (0.10) level. Simply put, the limited number of sites with meaningful trends cannot assure an accurate trend assessment."

Page 5, line 9: NCSCDv2 – citation missing, follow instructions of https://bolin.su.se/data/ncscd/

R3C10: We sincerely apologize for missing the important references. We thank the reviewer for pointing out this. We have added the references for NCSCDv2 (section 2.1, page 6, line 3-4 in the revised manuscript):

> "… from the Northern Circumpolar Soil Carbon Database version 2 (NCSCDv2, https://bolin.su.se/data/ncscd/) (Hugelius et al., 2013a;Hugelius et al., 2013b)."

Page 6, line 24: Brown et al. 2002: the reference is missing in the list

R3C11: We are sorry for failing to include this reference. We have added it.

Page 7, line 30 following: this paragraph belongs to methods

R3C12: Done as suggested.

Page 8, line 9: remove 'relatively'

R3C13: Done as suggested.

Page 9, first paragraph: this belongs to discussion

R3C14: Done as suggested.

Page 9, line 17 following: introduce this comparison in the methods section

R3C15: Done as suggested.

Page 9, line 22: provide correlation analyses results as table

R3C16: These results only consist of three correlation coefficients. Therefore, instead of adding these values into a table, we have added these correlation coefficients to Figure 5 for each variable for greater clarity.

Page 9, line 23: which SOC value did you use? Is it representative for the upper soil layer?

R3C17: We used spatially varying SOC provided by NCSCDv2, and not a constant SOC everywhere. Please see section 2.1 and Tao et al. (2017) for more details.

Page 10, line 4-28: this all belongs to the methods section

R3C18: We have moved these paragraphs to the new method section (section 3).

Page 11, line 7: spell out LAI

R3C19: We have spelled out LAI at the place it first appears (page 9, line 24 in the revised manuscript) and thus keep LAI as used here.

Page 12, line 6-12: move to methods

R3C20: Done as suggested.

Page 12, line 19: 'permafrost areas shown in Fig. 1b are well confined within the cold side ..' this cannot be seen. Add outlines to map

R3C21: The 273.15K isotherm actually is shown in the figure (i.e., the edge of dark green block).  We have added the boundary containing continuous and discontinuous permafrost regions into Figure 8d. The original Fig. 8d showed mean Tair all over the Northern Hemisphere. We also modified it and only show mean Tair over the simulation domain in order to better illustrate the results.

Page 13, up to line 7: to methods; what is the reasoning for the static time period approach?

R3C22: We have moved the method description to the new section 3.3.  We are not sure what the reviewer means by "static time period". We evaluated the full 37-year period in order to best leverage all of the available observations. We also modified equation (1) and (2) in the original manuscript (equation (2) and (3) in the revised manuscript) as follows (section 3.3, page 10, line 28 to page 11, line 2 in the revised manuscript):

$$T_{cum}(N) = \sum_{t=1}^{t=M} T_{pos}(t) \ , \tag{2}$$

where

$$T_{pos}(t) = \begin{cases} T_{air}(t) - T_f & if \ T_{air}(t) > T_f \\ 0 & if \ T_{air}(t) \le T_f \end{cases} \tag{3}$$

The index t in equation (2) for year N starts with a value of 1 on 1 September of year (N-1) and ends with a value of M on 31 August of year N. M could be 365 or 366 depending on the presence of a leap year over the preceding annual period."

Page 13 – move explanation of linear regression analyses to methods

R3C23: Done as suggested.

Page 14, line 14: 'geographically thin disagreements' – quantify this

R3C24: It is quite hard to quantify this, but we have attempted to do so with the following modification (section 4.4, page 17, line 22-25 in the revised manuscript):

"Aside from western Siberia, the geographically thin disagreements (i.e., about a few degrees latitude) between the simulated and observed permafrost extents toward the south in Figure 10a (green and blue areas at the southern edge of permafrost regions) are not as much a concern, since the comparison in such areas is muddied by the interpretation of "isolated" permafrost in the observational map (Figure 1b)."

Page 14, line 28 following: this is methods

R3C25: We have modified and moved these sentences to the new method section (section 3.4).

Page 14, line 18: this error may seem small in absolute numbers, but ALT is much thinner than for the Mongolian sites. The error is still relatively large.

R3C26: We recognized this behavior, and hence we discussed the model performance in high-latitude regions separately from that in mid-latitude regions. We also explicitly emphasized the large error for the Mongolian sites and discussed the possible reasons for the underestimation. We conducted some new sensitivity tests and have added relevant discussion into the manuscript. Please see our response in R2C8d and R3C6 for details.

Page 14, line 20-26: this belongs to discussion

R3C27: We thank the reviewer for this suggestion, but we believe this paragraph should stay here since it describes the results.

Page 15, first two paragraphs of section 3.5 – this all belongs to methods

R3C28: We have modified and moved these paragraphs to the new methods section (section 3.4).

Page 17, 21: 'shows good general agreement' – this is not really the case, quantify the agreement and compare with other published results

R3C29: We calculated the bias in percentage (please see R3C6 for details), and we have modified as follows (section 5, page 21, line 21-23 in the revised manuscript):

> "The spatial distribution of CLSM-simulated permafrost shows general agreement with the observation-based permafrost map of Brown et al. (2002), capturing 81.3% of total areas of continuous, discontinuous and sporadic types while capturing 87.7% of the total area of continuous and discontinuous types."

Page 17, line 29: 'The simulated ALTs agree well with the in-situ observations' – not really, see my comment for page14, line18; how do your results compare to other published results?

R3C30: Our results are among the best existing permafrost products. In particular, our results demonstrate much better agreement against observations for regions of shallow permafrost (as shown in Figure R8 and R9) compared to other existing results. Please see details in R3C6.

Page 18, lines 1-2: 'retrievals from airborne remote sensing for 2015 and the corresponding simulated ALT exhibit reasonable accuracy vs. in situ measurements' – this is not clear from the material presented.

R3C31: Point taken.  We have modified this sentence as follows (section 5, page 20, line 16-27 in the revised manuscript):

> "In northern Alaska, ALT retrievals from airborne remote sensing for 2015 and the corresponding simulated ALT exhibit limited skill versus the in-situ measurements.

At the model scale, the mean bias for the simulated results is better (-0.05 m) than that for the retrievals (-0.12 m), but the opposite is true for the correlation coefficient against observations (0.27 for the model vs. 0.61 for the retrievals). At the in-situ site scale, however, the ALT retrievals show a very weak correlation coefficient with the observations (0.05). Excluding sites that have ALT measurements exceeding the radar sensing depth (~ 60cm), the evaluation metrics for ALT retrievals become better than that for simulated ALT at the model scale. However, the remotely sensed ALT estimates generally show lower levels of spatial variability than the simulated ALT estimates, and their spatial patterns differ considerably. The remote sensing approach is still relatively new, and many aspects still need to be worked out. It is important, though, to begin considering the modeling and remote sensing approaches side by side, as both should play important roles in permafrost quantification in the years to come. Indeed, once the science fully develops, joint use of modeling and remote sensing (e.g., through the application of downscaling methods) should allow the generation of more accurate permafrost products at even higher resolutions."

Figure 5: what is the red rectangle?

R3C32: The red rectangle highlights an example to illustrate the anti-correlated relationship between ALT and organic carbon content. We discussed this point in section 4.2. We also added the following sentence to the caption of this figure (page 37, line 6-7 in the revised manuscript):

"The red rectangle crossing a) and b) highlights a portion of the domain that shows an anti-correlated relationship between organic carbon content and modelled ALT (see section 4.2)."

Figure 7: convert to table

R3C33: Since the values are already provided on the figure, we feel that converting it to a table would not help much. We opt to keep the figure as is.

**Reference**

[revised manuscript text omitted]

---

## Referee Report (RR1)

**Permafrost Variability over the Northern Hemisphere Based on MERRA-2 Reanalysis**
By Tao et al., 2019.

This study uses point measurement and airborn data in combination with the results of global model driven my MERRA-2 reanalysis modeling data to analyze present permafrost conditions and extent. Authors compare datasets from different scales to study the match between them. The main problem is how to compare in-situ data with averaged to 20x60 m$^2$ grid cell data and then averaged to 81 km$^2$ grid cell. Then authors touch on the problem on why global model unable to model permafrost in the Western Russia and Eastern Canada. Global model fail to model permafrost in those regional because those area represent ecosystem protected permafrost zones (Shur et al., 2007). This means that thick organic layer, most importantly including moss layer, protect permafrost below from warm air temperatures. To achieve this increasing the amount of the organic layer as was also done for example global models like CLM and SiBCASA (Nicolsky et al., 2007; Jafarov and Schaefer 2016) is simply not enough. It is important to drive those regions with cold initial temperatures with enough moss-organic insulation on top. In addition deep soil column should allow keeping permafrost in those regions. Overall, the paper indicates some important and interesting analysis, including the effect of soil moisture on the ground temperature and ALT. However, current version of the paper need some major clean ups to improve clarity. I suggest cutting the number of Figures, removing discussion from the conclusion and making results and discussion section, since results already have a lot of discussion. Keep the conclusion straight to the point, do not summarize your work in the conclusion. Instead suggest what improvement can be made to improve discrepancies in the ALT simulation in Mongolia, Russian etc. and how the permafrost extent can be better modeled on the global scale.

Abstract
L27 …some permafrost areas… Be specific, spell out those areas.

**Introduction**
P3. L26. I suggest acknowledging all the work done ALT measurement using GPR as a part of the pre-ABoVE campaign. Chen et al., (2016) documented extensive GPR ALT data collection near Toolik Lake, Alaska. Jafarov et al., (2018) documented extensive GPR ALT data collection near Barrow, Alaska. These datasets a unique because they represent spatial ALT collection in oppose to point measurements by CALM. Both dataset available for download from ABoVE website. These datasets can be extremely useful in this study because they give a better idea on spatial variability of the ALT on meter scale. The standard deviation from those works can be used to better constrain the uncertainty in measured ALT at a finer spatial scale.

In addition, I highly suggest checking the most recent and the most complete work on the near-surface permafrost data in Alaska (Wang et al., 2018). The data collected in that dataset provides a wider coverage for Alaska and can be extremely useful for this study.

P4. L 22-30. Do this freeze-thaw formulation allows multiple thaw zones? E.g. talik and seasonal frost above with the existing permafrost at a deeper depth.

P5.L12 Not sure why the model was spun up for 180 years? Typically spin up means total equilibrium.

Methods section needs some better organization. For example,
1. In-situ to AirMoss comparison
2. In-situ to CLSM comparison

P7-8. L30-12. The main point of those two paragraphs is the difference. I suggest plotting the difference between AirMoss and CLSM with 81 km$^2$ resolution, just one Figure instead of ABC. Then it will be clear when they do not match and then discussion can be more focused on the why they do not match.

P8. Paragraphs 3 and 4. Similarly don't need Figure 4 AB. In-situ data has smaller uncertainty and variability, when scaled up we average the variability into a one grid cell. The question is what is the uncertainty for CLSM should be, which was answered later in the manuscript by analyzing the effect of different factors (snow, organic layer, soil moisture). If you plot the CLSM uncertainty bars and they intercept with the solid lines then this makes the overall results much better.

P9. L16-30. It mainly depends on the pixel size (grid cell) of the modeled ALT. The authors should think how they can address the overall uncertainty in the global model, and how that uncertainty would change when they compare it with in-situ or AirMoss data.

P14. L6-20. Cite Shur et al., (2007) draw the discussion from that work. Refer to my main comment.

P14. L31. There are many CALM sites within a CLSM grid cell. The variation in CALM sites is a standard deviation (std). Again this deviation is from hand full of sites where the GPR measurement provides a wider range of the possible (std) in Barrow and Toolik Lake regions.

P15. L1-3. The soil characteristic in Mongolia might include rocky type environment. In mountain areas the ALT along the south face slopes might be quite deep. I wonder if that might explain the deep ALT in those regions.

P15. L30. Do you think if you drive the model with different reanalysis data (ERA-Interim or similar) it might give you better results?

P16. L19. I would drop unnecessary words phrases like at least to some extent from the text.

**References**

Jafarov, E. and Schaefer, K.: The importance of a surface organic layer in simulating permafrost thermal and carbon dynamics, The Cryosphere, 10, 465-475, doi:10.5194/tc-10-465-2016, 2016

Shur, Y. L. and Jorgenson, M. T.: Patterns of permafrost formation and degradation in relation to climate and ecosystems, Permafrost Periglac., 18, 7–19, doi:10.1002/ppp.582, 2007.

Wang, I. Overeem, E. Jafarov, G. Clow, V. Romanovsky, K. Schaefer, F. Urban, W. Cable, M. Piper, C. Schwalm, T. Zhang, A. Kholodov, P. Sousanes, M. Loso, D. Swanson, and K. Hill. A synthesis dataset of near-surface permafrost conditions for Alaska, 1997-2016K. https://doi.org/10.18739/A2KG55.

D. J. Nicolsky, V. E. Romanovsky, V. A. Alexeev, D. M. Lawrence. Improved modeling of permafrost dynamics in a GCM land-surface scheme. https://doi.org/10.1029/2007GL029525

Jafarov, E. E., Parsekian, A. D., Schaefer, K., Liu, L., Chen, A. C., Panda, S. K. and Zhang, T. (2018), Estimating active layer thickness and volumetric water content from ground penetrating radar measurements in Barrow, Alaska. Geosci. Data J.. doi:10.1002/gdj3.49

Chen, A., Parsekian A., Schaefer K., Jafarov E., Panda S., Liu L., Zhang T., and Zebker H: 2016. Ground-penetrating radar-derived measurements of active-layer thickness on the landscape scale with sparse calibration at Toolik and Happy Valley, Alaska. GEOPHYSICS, 81(2), H1-H11. doi: 10.1190/geo2015-0124.1

---

## Author Response (AR2)

**Overview**

We thank the reviewers for their constructive comments and suggestions. We have revised the manuscript throughout in response to the reviewers' comments. For reference, our response to comment "m" by reviewer "n" is labeled "R[n]C[m]").

Throughout the discussion below, the text is colored as follows:

    Black:  Reviewer comment
    Blue:   Author response
    Red:    Text newly inserted into the revised manuscript
    Green:  Text already inserted into the manuscript during the previous round of revisions but
missed by the reviewers.

**Reviewer #1**

The problem with the not yet published study by Chen et al. (always cited as Chen et al. 2018 instead of in review) remains. Large parts of the manuscript are based on it. The Cryosphere journal rule is ccording to the webpage 'Works cited in a manuscript should be accepted for publication or published already.' Results etc. referring to it should be removed.

R1C1: The study by Chen et al. (2019) has been accepted and is in press. We have updated the citation to this paper.

Other
Table 4 - state source of observation in caption

R1C2: We added the citation to the observation (Brown et al., 2002).

    "Table 4 – Evaluation results for simulated permafrost extent against the permafrost map by Brown et al. (2002)." (Page 45, line 1 in the revised manuscript)

Figure 8 - properly cite Brown et al.

R1C3: We have properly cited this reference.

    "The red boundary outlines the continuous and discontinuous permafrost regions according to Brown et al. (2002)."  (Page 53, line 4 – 5 in the revised manuscript)

**Reviewer #2**

This study uses point measurement and airborn data in combination with the results of global model driven my MERRA-2 reanalysis modeling data to analyze present permafrost conditions and extent. Authors compare datasets from different scales to study the match between them. The main problem is how to compare in-situ data with averaged to 20x60 m2 grid cell data and then averaged to 81 km2 grid cell. Then authors touch on the problem on why global model unable to model permafrost in the Western Russia and Eastern Canada. Global model fail to model permafrost in those regional because those area represent ecosystem protected permafrost zones (Shur et al., 2007). This means that thick organic layer, most importantly including moss layer, protect permafrost below from warm air temperatures. To achieve this increasing the amount of the organic layer as was also done for example global models like CLM and SiBCASA (Nicolsky et al., 2007; Jafarov and Schaefer 2016) is simply not enough. It is important to drive those regions with cold initial temperatures with enough moss-organic insulation on top. In addition deep soil column should allow keeping permafrost in those regions.

Overall, the paper indicates some important and interesting analysis, including the effect of soil moisture on the ground temperature and ALT. However, current version of the paper need some major clean ups to improve clarity. I suggest cutting the number of Figures, removing discussion from the conclusion and making results and discussion section, since results already have a lot of discussion. Keep the conclusion straight to the point, do not summarize your work in the conclusion. Instead suggest what improvement can be made to improve discrepancies in the ALT simulation in Mongolia, Russian etc. and how the permafrost extent can be better modeled on the global scale.

We sincerely thank the reviewer for the helpful suggestions. We should mention that the reviewer's comments are based on the original submission, rather than the revised manuscript resulting from the previous round of reviews. The revised manuscript included major changes from the original submission, including some changes that were requested by Reviewer #2 here. Below we address the reviewer's comments that still apply to the first revised version of the manuscript and point out below where the reviewer's comments had already been addressed in that revision.

Abstract
L27 …some permafrost areas… Be specific, spell out those areas.

R2C1: The text in question (concerning the trend analysis) was removed following the suggestion by Reviewer #3. See R3C3 and R3C15.

Introduction
P3. L26. I suggest acknowledging all the work done ALT measurement using GPR as a part of the pre-ABoVE campaign. Chen et al., (2016) documented extensive GPR ALT data collection near Toolik Lake, Alaska. Jafarov et al., (2018) documented extensive GPR ALT data collection near Barrow, Alaska. These datasets a unique because they represent spatial ALT collection in oppose to point measurements by CALM. Both dataset available for download from ABoVE website.

These datasets can be extremely useful in this study because they give a better idea on spatial variability of the ALT on meter scale. The standard deviation from those works can be used to better constrain the uncertainty in measured ALT at a finer spatial scale. In addition, I highly suggest checking the most recent and the most complete work on the nearsurface permafrost data in Alaska (Wang et al., 2018). The data collected in that dataset provides a wider coverage for Alaska and can be extremely useful for this study.

R2C2: We appreciate the reviewer for pointing out these datasets. We now point to other remote sensing-based ALT retrievals, including the GPR ALT data:

> "Due to the sparsity of in-situ measurements at the regional to global scale, evaluating the spatial pattern of ALT produced by any such simulation remains challenging. Indeed, it is difficult to compare the simulated values at model resolutions with in-situ observations taken at the point scale unless the measurement point is uniformly representative of the area covered by the model grid cell or the representation errors associated with the point-to-grid comparison are well defined. Remotely sensed permafrost products, which provide a unique source of spatially distributed ALT at the landscape-scale, may provide help in this regard. Existing remote sensing ALT products have been retrieved from ground-based Ground Penetrating Radar (GPR) (Chen et al., 2016a; Jafarov et al., 2017), airborne polarimetric Synthetic Aperture Radar (SAR), and spaceborne interferometric SAR (Liu et al., 2012; Li et al., 2015; Schaefer et al., 2015). These ALT products are available at the landscape-scale and can complement our modelling analysis. In this study, we use remote sensing information from the NASA Airborne Microwave Observatory of Subcanopy and Subsurface (AirMOSS) mission. In 2015, AirMOSS acquired P-band (420-440 MHz) SAR observations over portions of northern Alaska from which Chen et al. (2019) retrieved regional estimates of ALT and soil layer dielectric properties that are related to soil moisture and freeze/thaw states. In their study, Chen et al. (2019) mainly focus on the development and improvement of the ALT retrieval algorithm, whereas the present study uses the ALT retrievals in combination with in-situ measurements to aid in assessing the (fully independent) ALT simulations." (Page 4, line 21 – 25, and page 5, line 1 -10 in the revised manuscript)

Given the length of the paper and the fact that the reviewer also suggested cutting back the number of figures and the discussion, we decided to leave the use of the GPR data for future work. In fact, we are already working on just such a study.

Regarding the observations reported by (Wang et al., 2018), we used the permafrost sites from the GI-UAF in our previous study (Tao et al., 2017). We will assemble more measurements from USGS and NPS sites as included in Wang et al. (2018), as well as other remotely sensed ALT retrievals the reviewer pointed out in our future study to further evaluate and analyze model uncertainty.

P4. L 22-30. Do this freeze-thaw formulation allows multiple thaw zones? E.g. talik and seasonal frost above with the existing permafrost at a deeper depth.

R2C3: The equation implicitly allows for multiple thawing zones in the vertical dimension. However, considering the size of the model grid cell (81 km$^2$), it is very unlikely the talik zone can be represented. The seasonal frost above the existing permafrost at a deeper depth, however, can be well represented in the model.  In fact, seasonal frost is taken into account when determining ALT.

We added one sentence here to make this point.

> "We search for the deepest $l$ if multiple thawed-to-frozen transitions are present (e.g., if a seasonal frost at the surface is separated from the permafrost below by a thawed soil layer). The annual ALT for a given year, then, is defined as the deepest depth at which a thawed-to-frozen transition occurs within that year. Note that the calculation of equation (1) is made at the scale of a model grid cell, and thus features such as talik are not represented if they occur at sub-grid cell scale." (Page 7, line 11 – 15 in the revised manuscript)

P5.L12 Not sure why the model was spun up for 180 years? Typically spin up means total equilibrium.

R2C4: To reach equilibrium states for deep soils (down to 13 m), the 180 years' spin-up procedure is needed (Sapriza-Azuri et al., 2018; Paquin and Sushama, 2015).

Methods section needs some better organization. For example,

1. In-situ to AirMoss comparison
2. In-situ to CLSM comparison

R2C5: This comment no longer applies.  We had already reorganized the structure of the manuscript during the previous round of revisions.

P7-8. L30-12. The main point of those two paragraphs is the difference. I suggest plotting the difference between AirMoss and CLSM with 81 km2 resolution, just one Figure instead of ABC. Then it will be clear when they do not match and then discussion can be more focused on the why they do not match.

R2C6: We appreciate the reviewer's suggestion. We now added an additional panel to show the difference between AirMOSS and CLSM results at 81 km$^2$ resolution. We also removed the subpanel a), but still kept b) and c) to compare the spatial patterns of AirMOSS retrievals and model results.

P8. Paragraphs 3 and 4. Similarly don't need Figure 4 AB. In-situ data has smaller uncertainty and variability, when scaled up we average the variability into a one grid cell. The question is what is the uncertainty for CLSM should be, which was answered later in the manuscript by analyzing the effect of different factors (snow, organic layer, soil moisture). If you plot the CLSM uncertainty bars and they intercept with the solid lines then this makes the overall results much better.

R2C7: The revised manuscript from the previous round of revisions is already very different from the original manuscript to which the reviewer's comment refers. For this round of revisions, we deleted Figure 4a and modified Figure 4b and 4c. Indeed, we removed the comparison results for AirMOSS retrievals at the in-situ site scale, since Chen et al. (2019) already evaluated the retrievals at the site scale.

Regarding the CLSM uncertainty, we had already added some relevant discussion on it during the previous round of revisions, i.e., during the open discussion period. We copied some relevant information from the open discussion below. Please refer to the online discussion response for detailed analysis and supplementary figures (https://www.the-cryosphere-discuss.net/tc-2018-119/tc-2018-119-AC2-supplement.pdf and https://www.the-cryosphere-discuss.net/tc-2018-119/tc-2018-119-AC3-supplement.pdf).

"Test simulations (not shown) with alternative model configurations indicate that increasing the number of soil layers may act to decrease somewhat the simulated ALT, suggesting that our values may be a little overestimated; however, based on results from a new study by Sapriza-Azuri et al.(2018), our use of a no-heat-flux condition at the bottom boundary rather than a dynamic geothermal flux may lead to underestimates of ALT. Such uncertainties should naturally be kept in mind when interpreting our results. Our supplemental simulations (not shown) also suggest that increasing the total modelled soil depth has only a small impact on simulated ALT. Uncertainty in our description of soil organic carbon, i.e., both soil carbon content and vertical carbon distribution, leads to corresponding uncertainty in our ALT simulations. We indeed find a significant improvement in simulated ALT at several Mongolian sites when we arbitrarily impose less total soil carbon content and concentrate less soil carbon in top layers (not shown). Besides the vertical distribution of soil carbon, the vertical variation in other soil hydrological properties (e.g. soil texture and porosity) should also play a significant role since they all affect soil thermal conductivity and heat capacity. In addition, the lack of a necessary organic layer on top of soil column and the related thermal processes is also a major deficiency for the model especially in ecosystem-protected performant regions.

Another issue affecting our ALT comparisons is the climatological representation of vegetation parameters such as LAI used in CLSM. An additional investigation (not shown) revealed large differences between the LAI climatology used in CLSM and more realistic, time-varying, satellite-based LAI products at several Mongolian sites. In addition, while we did exclude from our analyses any measurements that were affected by notable disturbance (e.g., wildfire), the impacts of other potential land changes on ALT, including overgrazing in Mongolia (Sharkhuu and Sharkhuu, 2012; Liu et al., 2013), were not explicitly treated in the model. The model also lacks the vertical advective transport of heat in the subsurface due to downward flowing liquid water, which can significantly affect permafrost thawing (Kane et al., 2001; Rowland et al., 2011; Kurylyk et al., 2014)." (Page 29, line 2 – 21 in the revised manuscript)

P9. L16-30. It mainly depends on the pixel size (grid cell) of the modeled ALT. The authors should think how they can address the overall uncertainty in the global model, and how that uncertainty would change when they compare it with in-situ or AirMoss data.

R2C8: Please see our response above in R2C7.

P14. L6-20. Cite Shur et al., (2007) draw the discussion from that work. Refer to my main comment.

R2C9: We added relevant sentences as shown below relying on the reviewer's comments. (Note that we had already edited this paragraph during the previous round of revisions.)

> "One possible reason is that the permafrost in western Siberia is characterized as an ecosystem-protected permafrost zone (Shur and Jorgenson, 2007) where a thick moss-organic layer (i.e., moss-dominated mires (Anisimov and Reneva, 2006; Anisimov, 2007; Peregon et al., 2009)) protects the permafrost below from thawing under a warm air temperature. This is mainly attributed to the low thermal conductivity of the organic layer in summer, which strongly insulates the permafrost from the warm atmosphere, and the high thermal conductivity of the frozen organic layer in winter, which allows cold temperature penetration from above, provided the snowpack is not too thick (Nicolsky et al., 2007a; Jafarov and Schaefer, 2016). This mechanism is lacking in the current version of CLSM (Tao et al., 2017). Thus, improving the model through a better representation of thermal processes in an organic layer above the soil column in combination with initializing the simulation with a sufficiently cold soil temperature should improve the simulation results. This work is reserved for a future study." (Page 24, line 5 – 14 in the revised manuscript)

We also mentioned this in the abstract.

> "The resulting simulated permafrost distribution across the Northern Hemisphere mostly captures the observed extent of continuous and discontinuous permafrost but misses the ecosystem-protected permafrost zones in western Siberia. " (Page 1, line 16 – 16 in the revised manuscript)

P14. L31. There are many CALM sites within a CLSM grid cell. The variation in CALM sites is a standard deviation (std). Again this deviation is from hand full of sites where the GPR measurement provides a wider range of the possible (std) in Barrow and Toolik Lake regions.

R2C10: We agree with the reviewer about the limited information about CALM sites. We hope to include GPR data in the first author's next manuscript.

P15. L1-3. The soil characteristic in Mongolia might include rocky type environment. In mountain

areas the ALT along the south face slopes might be quite deep. I wonder if that might explain the deep ALT in those regions.

R2C11: We thank the reviewer for pointing out this. We checked the soil information at Mongolian sites and did find some useful information, and we accordingly added several sentences to the manuscript:

> "An additional reason for the underestimation of ALT in Mongolia might be a mismatch between the land surface parameter values used in the model and the actual conditions at each site. For instance, detailed soil information (https://www2.gwu.edu/~calm/data/webforms/mg_f.html) indicate that some Mongolian sites have special "rocky" soil types including limestones (e.g., M04), slatestones (e.g., M05), gravelly sand (e.g., M06 and M08), etc. that are not well represented in the model. As another example, sites on south-facing slopes presumably have much deeper ALT than those on slopes with less exposure to the sun, which is not captured by CLSM. This large representative errors of Mongolian sites are clearly illustrated by the standard deviation (although computed only with 3 to 5 measurements) as shown by the error bars in Figure 11a." (Page 26, line 21 – 26, and page 7, line 1 – 5 in the revised manuscript)

P15. L30. Do you think if you drive the model with different reanalysis data (ERA-Interim or similar) it might give you better results?

R2C12: This is possible. Please note that we've deleted the relevant sentence the Reviewer pointed.

Regarding the impacts of using different reanalysis data, we had already added relevant discussion following the previous round of reviews (copied below).

> "Note that some other global models, such as CLM3 and the Community Climate System Model version 3 (CCSM3) as reported in Lawrence et al. (2012), also missed this area of permafrost and that updated versions of these models (i.e., CLM4 and CCSM4) showed improved performance in this regard (Lawrence et al., 2012). Guo et al. (2017) reported underestimated permafrost extent simulated in western Siberia using CLM4.5 driven by three different reanalysis forcings (i.e., CFSR, ERA-I and MERRA), and they showed an improved simulation of permafrost extent in this area when using another reanalysis forcing, the CRUNCEP (Climatic Research Unit - NCEP) (Guo and Wang, 2017). Guimberteau et al. (2018) found similar improvements stemming from the use of CRUNCEP forcing. We leave for further study whether the MERRA-2 forcing data is responsible for the western Siberia deficiency seen in our own results." (Page 24, line 17 – 25 in the revised manuscript)

P16. L19. I would drop unnecessary words phrases like at least to some extent from the text.

R2C13: We removed this section and also the original Figure 13 and 14. See R3C3 and R3C15.

**Reviewer #3**

The authors run a series of simulations of permafrost dynamics using the MERRA2 reanalysis and the MERRA land model. They compare the results to remotely sensed active layer thickness (ALT) and in situ measurements of ALT. The paper has the potential to be a solid model-data comparison of simulated permafrost dynamics. I recommend acceptance after major revisions.

We sincerely thank the reviewer for the careful review and constructive suggestions. We now have revised our manuscript throughout according to the reviewer's comments.

I have three major comments:

1)     The authors need to account for measurement uncertainty when comparing to observations and refine their statistical comparison techniques. I found a number of errors in the statistical comparisons that I identify below.

R3C1:  We now provided a supplementary file to explain the measurement uncertainty in detail, and we added more discussion on measurement uncertainty of radar retrievals and representative error of in-situ measurements to better interpret the comparison results throughout the manuscript. See R3C18, R3C21, R3C27, R3C31, R3C33, R3C34, R3C48, and R3C49 for details.

In the comments below, we did not see further, more specific information about "a number of errors in the statistical comparisons."  We do address all of the issues that are raised.

2)     The authors need to clarify the role of the remote sensing data in this analysis. They spend as much space comparing the remotely sensed ALT with in situ data as with the model. Is the paper a means to validate the model or the remote sensing data?

R3C2:  The unique advantage of remotely sensed technique is that it provides spatially distributed ALT at the landscape-scale, well complementing the point-/plot-scale in-situ measurements. The present study is meant, in large part, to compare model-simulated, remotely sensed, and observed in-situ ALT to help understand the limitations in both remote sensing techniques and modeling mechanisms, and explore how remotely sensed ALT could be used to improve modeling skills.

We now deleted all the site-scale comparison between AirMOSS retrievals and in-situ observations since Chen et al. (2019) had already discussed the quality and uncertainty of the retrievals in detail.

3)     The authors need to change their spinup procedure or drop the trend analysis. Repeating the full time period for spinup introduces a dynamic response that produces false trends aliased on top of real trends. This pretty much invalidates the trend analysis.

R3C3: The reviewer is correct.  Given such considerations, we have removed the trend analysis, including Figures 13 and 14, from the manuscript.  It was fortunately not a central part of the study.

I have the following specific comments:

P2L17-20: Reword. This is a runon sentence with two, double nested parenthetical clauses, making it very difficult to understand.

R3C4: We deleted the relevant sentence.

P2L25: State how models are useful. This paragraph emphasizes resolution as a weakness of models.

R3C5: We augmented the sentence with more information. The revised sentence is copied below.

"Physically-based numerical model simulations are potentially useful for quantifying and understanding these dynamics at large spatial scales; they can also provide insights into associated impacts on the global carbon cycle." (Page 3, line 13 – 14 in the revised manuscript)

High-resolution simulation results of permafrost states at the global scale are necessary but are still lacking. We have modified the relevant sentences in the manuscript regarding resolution to elaborate our focus better:

"Permafrost dynamics can be modelled, for example, by driving a land surface model (LSM) offline (i.e., uncoupled from an atmospheric model) with meteorological forcing data (including air temperature, radiation, precipitation, etc.) from some credible source. LSMs that have been used to quantify large-scale permafrost patterns (i.e., distributions and thermal states) and their interactions with a warming climate include, for example, the Joint UK Land Environment Simulator (JULES, Dankers et al., 2011), the ORganizing Carbon and Hydrology in Dynamic EcosystEms (ORCHIDEE) - aMeliorated Interactions between Carbon and Temperature (ORCHIDEE-MICT, Guimberteau et al., 2018), the Catchment Land Surface Model (CLSM, Tao et al., 2017), and the Community Land Model (Alexeev et al., 2007; Nicolsky et al., 2007a; Yi et al., 2007; Lawrence and Slater, 2008; Lawrence et al., 2008; Lawrence et al., 2012; Koven et al., 2013; Chadburn et al., 2017; Guo and Wang, 2017). Most of these land models were run at coarse spatial resolutions, e.g., ranging from $0.5° \times 0.5°$ to $1.8° \times 3.6°$ for LSMs participating in the Permafrost Carbon Network (PCN) (Wang et al., 2016a) and from $0.188° \times 0.188°$ to $4.10° \times 5°$ for the models participating in the Coupled Model Intercomparison Project phase 5 (CMIP5) (Koven et al., 2013; https://portal.enes.org/data/enes-model-data/cmip5/resolution).

Differences in the permafrost behaviour simulated with these models reflect model-specific process representations as well as biases associated with different meteorological forcing datasets (Barman and Jain, 2016; Wang et al., 2016a; Wang et al., 2016b; Guo et al., 2017; Guimberteau et al., 2018). Such forcing biases are difficult to avoid given the sparsity of direct observations of meteorological variables in most parts of the high latitudes. Even

reanalyses, which assimilate a variety of global observations, inevitably have biases in high latitudes due to observation sparsity in cold regions combined with the many challenges of physical process modelling. Nevertheless, despite these issues, permafrost behaviour simulated with LSMs driven offline by reanalysis forcing fields can still be useful for understanding the impacts of climate variability on permafrost. The present paper utilizes this approach. Specifically, we generate here a dataset of Northern Hemisphere permafrost conditions by driving an updated version of NASA's Catchment Land Surface Model (CLSM) with Modern-Era Retrospective Analysis for Research and Applications-2 (MERRA-2; Gelaro et al., 2017) surface meteorological forcing fields for the middle-to-high latitudes across the Northern Hemisphere over the period 1980-2017. We perform the simulations at 81 km2 resolution encompassing permafrost areas in the middle-to-high latitudes of the Northern Hemisphere. This resolution is high relative to most existing modelling studies at the global scale; published simulations at higher resolution are limited to plot scales (e.g., CALM-site scale in Shiklomanov et al. (2010)), landscape scales (e.g., polygonal tundra landscape scale in Kumar et al. (2016)), or regional scales (e.g., 4 km2 in Jafarov et al. (2012) covering Alaska; 1 km2 in Gisnas et al. (2013) covering Norway)."
(Page 3, line 14 – 25, and page 4, line 1 - 19 in the revised manuscript)

P3L2-3: This is not difficult. One must account for representation error when comparing a point measurement to the area average of a model pixel.

R3C6: We modified the relevant sentences as copied below. (Note the sentences were moved to another paragraph.)

"Due to the sparsity of in-situ measurements at the regional to global scale, evaluating the spatial pattern of ALT produced by any such simulation remains challenging. Indeed, it is difficult to compare the simulated values at model resolutions with in-situ observations taken at the point scale unless the measurement point is uniformly representative of the area covered by the model grid cell or the representation errors associated with the point-to-grid comparison are well defined. Remotely sensed permafrost products, which provide a unique source of spatially distributed ALT at the landscape-scale, may provide help in this regard." (Page 4, line 21 – 25, and page 5, line 1 in the revised manuscript)

P3L7: The resolution of these simulations is essentially the same as for many published simulations, so I am not sure this is the best claim to make.

R3C7: We deleted this sentence and modified this paragraph. The relevant sentences are copied above in R3C5.

P4L3: State or described the improved performance.

R3C8: We now deleted all the discussion comparing MERRA with MERRA-2, since we did not conduct simulations forced by MERRA and the sentence here is not very relevant to the scope of this study.

P4L8: State or describe exactly what is inferior.

R3C9: Again, we have deleted the relevant discussion comparing MERRA and MERRA-2.

P4L10-12: Delete. Each reanalysis has strengths and weaknesses and I find it very difficult to believe that one version of MERRA is truly superior to another, especially considering the scarcity of measurements in the Arctic. I have no objection to using MERRA-2, of course, but claiming superiority is not warranted and best deleted from the manuscript.

R3C10: We have deleted this sentence and have reorganized this paragraph.

P4L18-20: Here the authors state they will use the remotely sensed ALT to validate the model, but later they actually validate the remotely sensed ALT against ground observations. This makes the actual purpose of including remotely sensed data unclear in this paper.

R3C11: We have modified relevant sentences and have also added sentences before this paragraph to better articulate our purpose of using remotely sensed ALT in this study. Note we also already deleted the discussion on site-scale comparison between AirMOSS retrievals and in-situ observations which was already included in Chen et al. (2019).

"Due to the sparsity of in-situ measurements at the regional to global scale, evaluating the spatial pattern of ALT produced by any such simulation remains challenging. Indeed, it is difficult to compare the simulated values at model resolutions with in-situ observations taken at the point scale unless the measurement point is uniformly representative of the area covered by the model grid cell or the representation errors associated with the point-to-grid comparison are well defined. Remotely sensed permafrost products, which provide a unique source of spatially distributed ALT at the landscape-scale, may provide help in this regard. Existing remote sensing ALT products have been retrieved from ground-based Ground Penetrating Radar (GPR) (Chen et al., 2016a; Jafarov et al., 2017), airborne polarimetric Synthetic Aperture Radar (SAR), and spaceborne interferometric SAR (Liu et al., 2012; Li et al., 2015; Schaefer et al., 2015). These ALT products are available at the landscape-scale and can complement our modelling analysis. In this study, we use remote sensing information from the NASA Airborne Microwave Observatory of Subcanopy and Subsurface (AirMOSS) mission. In 2015, AirMOSS acquired P-band (420-440 MHz) SAR observations over portions of northern Alaska from which Chen et al. (2019) retrieved regional estimates of ALT and soil layer dielectric properties that are related to soil moisture and freeze/thaw states. In their study, Chen et al. (2019) mainly focus on the development and improvement of the ALT retrieval algorithm, whereas the present study uses the ALT retrievals in combination with in-situ measurements to aid in assessing the

(fully independent) ALT simulations." (Page 4, line 21 – 25, and page 5, line 1 -10 in the revised manuscript)

We also now provide an additional reason to include the retrieval results in the introduction:

"As a side benefit, the side-by-side comparison of modelled and remotely sensed ALT estimates is an important first step toward combining this information effectively in future model-data fusion efforts." (Page 5, line 17 – 18 in the revised manuscript)

P5L10: The vertical resolution seems too coarse to simulate ALT. The total depth is fine, but other models typically use much higher resolution to simulate ALT. The authors need to explain why this resolution will work.

R3C12: While our vertical resolution is coarse, we emphasize that our ALT values are not strictly tied to this resolution; we can compute any value of ALT depending on the simulated ice fractions of the model layers. This is clarified by some added text:

"Equation (1) then expresses that the thawed-to-frozen depth is equal to the bottom depth of the layer $l$ but adjusted upward according to the ice fraction within the partially thawed layer $l$. The annual ALT for a given year, then, is defined as the maximum thawed-to-frozen depth within that year. This upward adjustment, by the way, allows the thawed-to-frozen depth to be a continuous variable; it is not quantized to the imposed layer depths." (Page 7, line 9 – 11 in the revised manuscript)

Also note that we did test some simulations with increased vertical resolution and total depth during the previous round of revision. Our general conclusions are consistent with other studies in terms of how soil configuration affects permafrost modeling (e.g., Alexeev et al., 2007; Lawrence et al., 2008; Sapriza-Azuri et al., 2018; Nicolsky et al., 2007; Dankers et al., 2011). Please refer to the online discussion response for detailed analysis and supplementary figures (https://www.the-cryosphere-discuss.net/tc-2018-119/tc-2018-119-AC3-supplement.pdf).

We did include some appropriate discussion about this issue in the conclusion section which are copied below.

"Test simulations (not shown) with alternative model configurations indicate that increasing the number of soil layers may act to decrease somewhat the simulated ALT, suggesting that our values may be a little overestimated; however, based on results from a new study by Sapriza-Azuri et al.(2018), our use of a no-heat-flux condition at the bottom boundary rather than a dynamic geothermal flux may lead to underestimates of ALT. Such uncertainties should naturally be kept in mind when interpreting our results. Our supplemental simulations also suggest that increasing the total modelled soil depth has only a small impact on simulated ALT." (Page 29, line 2 – 7 in the revised manuscript)

In response to reviewer's specific comment about the rationale of this vertical resolution, we added one sentence here as shown below.

"The soil layer thicknesses increase with depth following a geometric series for consistency with the linear heat diffusion calculation (Koster et al., 2000). A no-heat-flux condition is employed at 13m depth." (Page 6, line 14 – 16 in the revised manuscript)

P5L23: This is a good formulation. Models often use it, but rarely document it.

R3C13: Thanks. (No response required.)

P6L17: A 180 year spinup is adequate for stabilizing soil temperatures, but not for soil carbon. Does this model include dynamic soil carbon pools? If yes, then a spinup of 1000-5000 years is more appropriate.

R3C14: We do not have the dynamic soil carbon pools in this study, as now explicitly stated:

"This version of CLSM, however, does not include dynamic soil carbon pools." (Page 6, line 24 - 25 in the revised manuscript)

P6L20: The chosen spinup technique pretty much invalidates the trend analysis. The typical response time for soil temperature in a model such as this is 20-30 years, exactly matching the length of the MERRA forcing data. If they had spun up using only 1980-85 MERRA data, then the trend analysis makes sense. I suggest either changing the spinup or dropping the trend analysis.

R3C15: We have followed the reviewer's suggestion and have removed all discussion regarding the trend analysis, which was not central to the paper.

P7L29-30: This means one can use the radar data only where one expects the alt to be less than 60 cm. If the radar cannot penetrate below 60 cm, I question the utility of using it for validation. The authors need to supply a rationale for including it in the study.

R3C16: We understand the reviewer's concern about the sensing depth limitation with the remotely sensed ALT retrievals. Even so, the radar retrievals are practically useful for shallow permafrost (for instance in the North Slope of Alaska). As section 4.1 demonstrated, the AirMOSS ALT retrievals show good performance when compared with in-situ observations excluding the sites with ALT measurements deeper than 60 cm. The ALT retrieval algorithm as documented by Chen et al. (2019) represents the current status of P-band radar technique for monitoring permafrost, and will be gradually improved in the future. Combining remote sensing techniques and land models together is a way to better monitor permafrost dynamics. As the first attempt here, we believe the intercomparison of ALT among model results, remotely sensed retrievals and in-situ observations could provide useful insights to the research community.

The manuscript also discussed the potential use of ALT retrievals in the "5 Conclusion and Discussion" section (as copied below).

"The remote sensing approach is still relatively new, with many aspects still requiring development. It is important, though, to begin considering the modeling and remote sensing approaches side by side, as both should play important roles in permafrost quantification in the years to come. Indeed, once the science fully develops, joint use of modeling and remote sensing (e.g., through the application of downscaling methods) should allow the generation of more accurate permafrost products at higher resolution." (Page 27, line 22 – 25, and page 28, line 1 - 2 in the revised manuscript)

In addition, as noted above, we include in the introduction this additional side benefit of our inclusion of the retrieval results:

"As a side benefit, the side-by-side comparison of modelled and remotely sensed ALT estimates is an important first step toward combining this information effectively in future model-data fusion efforts." (Page 5, line 17 – 18 in the revised manuscript)

P8L11: Please identify which site got covered with lava. This is so unusual that you have to tell the reader.

R3C17: The documentation (meta-data) for two sites (R30A and R30B in Kamchatka) specifically notes that they "were covered by lava after Nov 2012 eruption". We have added the two sites into the manuscript.

"We did not use measurements that were flagged as having been taken too early in the season or under unusual conditions (e.g., after the site was burned or covered with lava, which occurred at sites R30A and R30B in Kamchatka)." (Page 10, line 14 – 16 in the revised manuscript)

P8L28: The section on comparison with the radar ALT must include uncertainty. The best that a model can do is match the observations within uncertainty.

R3C18: We provided a supplementary to explain the measurement uncertainty of AirMOSS retrievals. We then added relevant sentences discussing the radar ALT uncertainty and how the model results compare with the radar retrievals.

"We employed the strategy of Schaefer et al. (2015) to handle the uncertainty propagation, i.e., adding in quadrature the uncertainty components from each scale/level involved (see the supplementary file for a detailed description). For AirMOSS retrievals, the sampling uncertainty of mean ALT at the 81 km2 model grid-cell scale is negligible given the large sampling size and the fact that the retrieval uncertainty dominates the overall uncertainty (see supplementary file). Here, we use a nominal estimate of 0.15 m to represent the AirMOSS uncertainty (i.e., the average of the lower and upper bound of the actual retrieval uncertainty for individual radar pixels as discussed by Chen et al. (2019))." (Page 12, line 13 – 18 in the revised manuscript)

"Figures 4c further shows that the CLSM-simulated ALT agrees well with the AirMOSS ALT retrievals to within the measurement uncertainty of 0.15 m at all the site-located model grid cells. Indeed as Figure 3c illustrated, the differences between simulated ALT and the AirMOSS retrievals over all the transects examined here are generally below the measurement uncertainty of 0.15 m." (Page 18, line 19 – 22 in the revised manuscript)

P8L28: The authors should include a description of the statistical comparison itself. There are many ways to do this, ranging from a cost function to a regression.

R3C19: We now explicitly discuss the specific metrics used for the comparison (see also R3C34).

"We rely on several metrics to evaluate the model and radar-retrieval performance, including bias, root mean square error (RMSE), and correlation coefficient (R)." (Page 11, line 13 – 14 in the revised manuscript)

P8L28: The authors need to change the section title. The title covers only comparison with the radar data, but the text covers comparison with CALM data.

R3C20: We changed the title from "Comparison With AirMOSS ALT Retrievals" to "Comparing ALT from In-situ Observations, AirMOSS Retrievals, and CLSM Results in Alaska".

P9L7-16: The comparison of a point, in situ measurement to a model or remote sensing pixel must account for representation error. Representation error is the uncertainty when a point measurement represents an average. The standard deviation of the CALM grid measurements is a good estimate of representation error.

R3C21: We did include the standard deviation of ALT measurements in the analysis (also explicitly stated in the caption of Figure 4). We now include an additional paragraph to provide more information regarding the representation error.

"When comparing in-situ measurements with model results at the 81 km2 scale (i.e., a point-to-grid comparison), the ultimate measurement uncertainty propagated from the point-scale measurements to the 81 km2 scale is, for all intents and purposes, unknown due to a lack of sufficient measurements over the 81 km2 scale to compute upscaling errors (see supplementary file). We thus show instead the standard deviation of CALM measurements to illustrate, in a highly approximate way, the spatial representativeness error of the in-situ measurements – a small (large) standard deviation represents a homogeneous (heterogeneous) area in terms of ALT, meaning that the in-situ mean likely can (cannot) represent an average over a larger scale, assuming the site-scale heterogeneity is somewhat transferable to the larger scale. Such transferability might only apply to the largest in-situ site scales (e.g., 1000 m × 1000 m) to the model grid-scale (81 km2) and is thus, in general, questionable. We thus make no claim here that the standard deviations shown represent true uncertainty levels. " (Page 12, line 20 – 25, and page 13, line 1 – 3 in the revised manuscript)

We also obtained the standard deviation of measurements at all the CALM sites that provide such information, and we now include the standard deviation as an estimate of representative error in the comparison against modeled ALT all over the simulation domain in the Northern Hemisphere (new Figure 11). We discussed the CALM representation error later in the result section. Please see our response in R3C27.

P10L25-30: The authors should state this is a standard degree day model and find some references.

R3C22: Indeed, this is different from "a standard degree day model" for predicting ALT based on relevant variables as reported in many studies (e.g., Shiklomanov et al., 2010). Here, we are simply trying to quantify how much of the temporal variability in ALT can be explained by air temperature and the snow mass. Please see further discussion in the following response (R3C23).

P11L5-8: This is a standard degree-day model for ALT with a snow adjustment. There are hundreds of variants of this model in the literature derived from the original thermodynamics equation, models, or empirically from in situ observations. The authors need some references here and text explaining that this is a degree day model.

R3C23: Actually this is not "a standard degree-day model for ALT with a snow adjustment". We are aware of many studies using the standard degree-day model to predict ALT, and we have added these references into the manuscript (see the new text below). However, our purpose here is not to derive a standard degree-day model to predict ALT. Instead, we simply construct a multiple linear regression relationship between ALT time series and two important variables, i.e., the accumulated positive air temperature and maximum snow mass, in an attempt to determine how much of the temporal variability of ALT is explained by air temperature and snow mass. We added relevant discussion here to clarify the difference between our multiple linear regression analysis and the conventional degree day models used by other studies to predict ALT.

> "The correlation coefficient relating ALT to $\sqrt{T_{cum}}$ and $SWE_{max}$ is the square root of the coefficient of multiple determination ($R^2$) obtained through fitting Equation (4). This equation is similar in form to the common degree-day model for predicting ALT from accumulated degree days of thaw based on the Stefan solution (e.g., Shiklomanov and Nelson, 2002; Zhang et al., 2005; Riseborough et al., 2008; Shiklomanov et al., 2010). Here, however, we constructed equation (4) for a different purpose: to explore how much of the temporal variability of ALT can be jointly explained by snow mass and above-freezing air temperature. Before calculating these correlation coefficients, we removed the linear trend within ALT, $T_{cum}$, and $SWE_{max}$ to avoid potentially exaggerating the correlation due to an underlying trend." (Page 15, line 19 – 25, and page 16, line 1 -2 in the revised manuscript)

P11L8: The authors should explain why they included the a0 term. The a0 term is not often seen in a degree day model because one typically assumes the soil starts frozen (a0=0).

R3C24: Again, this is not a degree day model for predicting ALT. The a0 is necessary here to ensure a well-defined fitting to the multiple linear regression relationship in order to obtain statistically meaningful coefficients of determination and correlation. We have elaborated more about our purpose for performing the regression and have clarified the differences between equation (4) and the standard degree day model. Please see the above response (R3C23).

P11L8: The authors should explain why they chose Tcum rather than the square root of Tcum. One can derive the sqrt(Tcum) relationship directly from the original thermodynamics equation and the relationship appears many times in analyses of in situ measurements. Because of the strong theoretical basis of sqrt(Tcum), using plain Tcum is rare, so the authors need to justify its use.

R3C25: Although Equation (4) was not intended to be a standard degree day model as explained above (R3C23), we have followed the reviewer's suggestion and have replaced Tcum with sqrt(Tcum). The resulting correlation coefficient map is very similar to that using Tcum (as shown below). Note, however, we now detrend the data before calculating the correlation coefficient and the figures shown below are not included in the revised manuscript. Please see our response in R3C43 for updated figures.

[Figure]

Figure R1: Correlation coefficient between a) ALT and effective accumulated air temperature ($T_{cum}$) and b) ALT and sqrt($T_{cum}$). c) the difference between a) and b).

[Figure]

Figure R2: Multi-variable coefficient of correlation for a fitted multiple linear regression model a) between ALT and $T_{cum}$ and $SWE_{max}$ and b) between ALT and sqrt($T_{cum}$) and $SWE_{max}$. c) the difference between a) and b).

P11L12-16: This description of comparing to CALM is out of place and should be moved to section 3.1.

R3C26: We consider the context here to be distinct from that of section 3.1, which only compares an ALT snapshot in the thawing season of 2015 in Alaska (when and where AirMOSS flights were conducted). Here we focus, as explained in the manuscript, on the comparison of multi-year ALTs and the climatological mean between model results and CALM observations all over the simulation domain in the Northern Hemisphere.

P11L12-16: The authors need to account for uncertainty in the CALM measurements when comparing to the model output.

R3C27: We have managed to obtain the standard deviation of measurements at all the CALM sites that provide such information, and we now include the standard deviation as an estimate of representative error in the comparisons against modeled ALT. Please see our new Figure 11.

[Figure]

Figure R3 (Figure 11 in the manuscript): a) Annual ALT from CLSM simulation vs. CALM observations with horizontal error bars indicating standard deviations of measurements within the model grid cell. Error bar is absent if the number of measurements within a 81 km$^2$ grid cell is less than three. b) As in a) but excluding the Mongolia sites. c) 38-yr average ALT for the period 1980-2017 from CLSM simulation vs. CALM observations. d) As in c) but without the Mongolia sites. The correlation coefficient (R), bias, and root mean squared error (RMSE) are provided next to each subplot.

However, as explained by the supplementary file, it is impossible to estimate the CALM measurement uncertainty at the model grid scale due to the lack of upscaling errors. But by showing standard deviation here, we could explain how heterogeneous the CALM site sampling area is and in general how reasonable the point-to-grid comparison results are. We added relevant sentences as below.

"As noted in section 3.1 and in the supplementary file, the uncertainty of the CALM ALT measurements in the context of evaluating grid cell-scale model results theoretically involves uncertainty derived from probing point measurement uncertainty, site-scale mean uncertainty, and upscaling errors in going from the site-scale to the model-scale. This latter uncertainty in particular is unknown. In our figures (in section 4.4) we show the standard deviation of the observed ALT as a very crude surrogate for the spatial representativeness error associated with the point-to-grid comparison. As before, we make no claim here that the standard deviations shown represent the relevant statistical uncertainty." (Page 16, line 18 – 25 in the revised manuscript)

"As another example, sites on south-facing slopes presumably have much deeper ALT than those on slopes with less exposure to the sun, which is not captured by CLSM. Th large representative errors of Mongolian sites are clearly illustrated by the standard deviation

Please also see discussion in R3C21.

P11L18-24: The spinup technique invalidates the trend analysis. Either drop the trend analysis or modify the spinup.

R3C28: We have removed the trend analysis.

P12L6-8: Delete. Unneeded.

R3C29: We modified this paragraph and moved it to Section 3.1, since it fits better there.

P12L10-12: Move to methods.

R3C30: Done as suggested.

P12L14: Figure 3 shows the difference between the model and observations, but is this difference within the uncertainty? If yes, then the two are statistically identical and thus a match. If no, then there is a statistically significant mismatch. The magnitude of the difference is unimportant if the difference is less than uncertainty. The authors need to account for uncertainty in this comparison.

R3C31: We used Figure 3 to compare the spatial patterns of remotely sensing data and model results, and we discuss the differences next in Figure 4. We added relevant sentences discussing the radar ALT uncertainty and how the model results compare with the radar retrievals.

"We employed the strategy of Schaefer et al. (2015) to handle the uncertainty propagation, i.e., adding in quadrature the uncertainty components from each scale/level involved (see the supplementary file for a detailed description). For AirMOSS retrievals, the sampling uncertainty of mean ALT at the 81 km2 model grid-cell scale is negligible given the large sampling size and the fact that the retrieval uncertainty dominates the overall uncertainty (see supplementary file). Here, we use a nominal estimate of 0.15 m to represent the AirMOSS uncertainty (i.e., the average of the lower and upper bound of the actual retrieval uncertainty for individual radar pixels as discussed by Chen et al. (2019))." (Page 12, line 13 – 18 in the revised manuscript)

"Figures 4c further shows that the CLSM-simulated ALT agrees well with the AirMOSS ALT retrievals to within the measurement uncertainty of 0.15 m at all the site-located model grid cells. Indeed as Figure 3c illustrated, the differences between simulated ALT and the AirMOSS retrievals over all the transects examined here are generally below the measurement uncertainty of 0.15 m." (Page 18, line 19 – 22 in the revised manuscript)

P12L21: Explain here why soil type influences the result. The reader should not have to flip forward in the paper to get this answer.

R3C32: We have added relevant discussion at this point in the text:

"Variations of the simulated ALT within a single transect (Figure 3a) are predominantly induced by changes in soil type (indicated in Figure 2c and 2d). In essence, the higher the organic carbon content within the soil, the smaller the simulated ALT due to slower heat transfer associated with lower thermal conductivity, higher porosity, heat capacity, etc. (Tao et al., 2017). See also section 4.2 for a discussion of the influence of soil texture on the spatial pattern of ALT." (Page 17, line 15 – 19 in the revised manuscript)

P12L25: Agreement 'to first order' is too vague and carries no meaning. The authors need to quantify the agreement accounting for uncertainty.

R3C33: We deleted "to first order" here and modified the sentence as below.

"Figures 4a and 4b show that the CLSM-simulated ALTs agree with the in-situ observations with an overall mean bias of -0.05 m and a RMSE of 0.17 m." (Page 17, line 22 – 23 in the revised manuscript)

Given the large scale gap between the in-situ site and the model grid cell, it is impossible to estimate the measurement uncertainty to be comparable with model results. However, we do show the standard deviation as a representative error of the in-situ measurements. We added relevant sentences as below.

"When comparing in-situ measurements with model results at the 81 km2 scale (i.e., a point-to-grid comparison), the ultimate measurement uncertainty propagated from the point-scale measurements to the 81 km2 scale is, for all intents and purposes, unknown due to a lack of sufficient measurements over the 81 km2 scale to compute upscaling errors (see supplementary file). We thus show instead the standard deviation of CALM measurements to illustrate, in a highly approximate way, the spatial representativeness error of the in-situ measurements – a small (large) standard deviation represents a homogeneous (heterogeneous) area in terms of ALT, meaning that the in-situ mean likely can (cannot) represent an average over a larger scale, assuming the site-scale heterogeneity is somewhat transferable to the larger scale. Such transferability might only apply to the largest in-situ site scales (e.g., 1000 m × 1000 m) to the model grid-scale (81 km2) and is thus, in general, questionable. We thus make no claim here that the standard deviations shown represent true uncertainty levels. " (Page 12, line 20 – 25, and page 13, line 1 – 3 in the revised manuscript)

P13L1-8: The authors need to expand their statistical analysis of the model-data comparison. All they have is correlation, which says nothing about magnitude. They should expand the residual

analysis to include bias (mean residual), root mean square error (residual standard deviation), and chi-squared (standard deviation of residuals normalized by uncertainty).

R3C34: In addition to correlation coefficient, we already included bias and RMSE as discussed in section 4.1 and the associated Table 3 (and also in section 4.3), and the Figure 11 and the associated discussion in the manuscript, which the reviewer might have missed. Also, since it is impossible to estimate the uncertainty of in-situ measurements at the model grid scale as explained in detail in the supplementary file,  we are not able to compute the chi-square.

We also provided a supplementary file and added more description to explain our model-data comparison strategy.  Please also see our discussion in R3C18, R3C21, R3C27 and R3C31.

P13L9: 'Broadly consistent' is too vague and carries no meaning. The authors need to quantify the agreement.

R3C35: We deleted "broadly" here and quantified the agreement using the following text

"Although the AirMOSS ALT retrievals generally underestimate the in-situ ALT measurements (as shown in Figure 4a), the retrievals tend to be more consistent with the observations when the in-situ measurements are within the ~60 cm sensing depth of the P-band radar data, as indicated in Table 3.  Specifically, excluding the sites with in-situ ALT measurements that exceed the AirMOSS sensing depth of ~60 cm, the overall mean bias for the AirMOSS retrievals at the 81 km2 scale drops to -0.01 m, and the correlation coefficient increases to 0.64. In contrast, the CLSM simulation results show a bias of 0.01 m and a zero correlation coefficient at these sites." (Page 18, line 7 – 13 in the revised manuscript)

P13L20: 'In general' is too vague and has no meaning. Comparing modeled and observed trend with latitude is perfectly valid here. The limited number of in situ measurements will simply result in higher uncertainty.

R3C36: We deleted "In general" here.

P13L23: Again, 'generally' is too vague.

R3C37: We deleted "generally" here.

P13L23-33: Figure 5 is not the correct format to show the relationships described here. The reader cannot visualize the relationships and correlations from the simple time series plots in Figure 5. The authors should replace the time series plots with three plots to illustrate the relationships: ALT vs. latitude, ALT vs. organic matter content, and ALT vs. air temperature.

R3C38: We feel the reviewer did not fully understand this figure. This is not a time series plot, but is rather a spatial-series plot. We do, however, appreciate the reviewer's point and have thus added

scatter plots (e.g., ALT vs. air temperature, ALT vs. organic matter content, and ALT vs. maximum snow depth) next to the spatial-series plots.  Please see our new Figure 5.

P14L20-23: Perhaps, but the author's argument is not convincing. Shading associated with higher LAI represents an equally valid explanation. Higher water content associated with higher organic matter content could also explain the difference. The authors have the full suite of model output on hand. They should do a statistical analysis of available output to track down exactly what explains the difference.

R3C39: We agree with the reviewer about potential shading effects; however, such effects are not resolved in the current model. The statement "Higher water content associated with higher organic matter content could also explain the difference" is also true but not valid here since the differences between HomF&Veg vs. HomF relate only to the vegetation differences.

The argument about the snow impacts caused by changing vegetation is based on our previous investigation in (Tao et al., 2017).  We have accordingly modified the relevant sentences and cited our previous study here.

> "This is because the generally lower albedo of the taller and leafier trees (representative of the IVO transect) during the snow season resulted in increased snowmelt and thus reduced snowpack during the snow season (compare the green and red curves in Figure 6c), thereby reducing the thermal insulation of the wintertime ground.  With reduced insulation, cold season ground temperatures dropped, making it more difficult for temperatures to recover during summer(Tao et al., 2017). " (Page 20, line 14 – 17 in the revised manuscript)

P14L31: The authors need to identify exactly what soil parameters changed. Porosity? Thermal conductivity? Volumetric water content? Also, the authors need to explain how they specify soil properties in the model. A sharp change as seen here is common when specifying properties by soil type, such as sandy loam defined in the USGS soil triangle. A sharp change would be unusual when specifying soil properties by maps of soil texture (sand, silt, and clay fraction).

R3C40: This information is mentioned in the methods section (section 3.2) and in Table 2 when we describe the idealized experiments. We now clarify the approach as follows:

> "Owing to the strong control of soil type-related parameters (see section 3.2 and Table 2) on soil moisture, spatial variability in soil moisture remains high in HomF and HomF&Veg and is only eliminated once these soil type-related parameters are homogenized (Figure 6d), which explains the abrupt changes shown in Figure 3c as mentioned in section 3.1.  " (Page 20, line 21 – 23 in the revised manuscript)

Note also that, as indicated in the caption of Figure 2, we effectively use a very highly resolved version of a soil texture triangle having 253 subtypes, not the more standard USDA triangle with ~10 types.

P15L33: The reason simulated ALT is deeper is the same reason identified later in the manuscript: the model either has permafrost or it does not because it cannot represent sub-grid scale processes. When the model does simulate permafrost in sporadic regions, it is always greater than observed because it represents an area average of permafrost and non-permafrost areas.

R3C41: We agree with the reviewer about the model deficiency on handling sub-grid variability as we also discussed in the manuscript (Section 4.4, Page 18, line 21 -26)). What we meant here is that the simulated ALT in discontinuous and sporadic permafrost regions is deeper than that in continuous regions. We modified the sentence as below for clarity.

"The largest ALT standard deviations (red color in Figure 8b) are found mainly in discontinuous and sporadic permafrost regions (see Figure 1b) where ALTs are deeper on average than that in continuous permafrost region." (Page 22, line 5 – 6 in the revised manuscript)

P16L9-10: The authors need to either perform the analysis with air temperature or at least summarize and reference the results of other studies that did perform the analysis.

R3C42: We have modified this sentence and added relevant references here.

"The relationship between the spatiotemporal characteristics of simulated ALT and air temperature forcing has been investigated before in many studies at the site to landscape scale (e.g., Klene et al., 2001; Shiklomanov and Nelson, 2002; Zhang et al., 2005; Juliussen and Humlum, 2007) and at the regional scale (e.g., Anisimov et al., 2007). Here we simply analyze the correlation coefficient between ALT and two variables: the proxy of total energy input into the ground (i.e., $\sqrt{T_{cum}}$, see section 3.3) and the maximum SWE. Our goal is to explore how much of the spatiotemporal variability of ALT across the globe can be jointly explained by these two variables." (Page 22, line 16 – 22 in the revised manuscript)

P16L12: The authors need to remove the trends in ALT, Tcum, and SWEmax before calculating the correlation coefficients. We see nice strong correlations because all three variables show strong trends over the time period of the simulation. Removing the trends will significantly change Figure 9 and its interpretation. If the authors want to isolate the effects of trends on the ALT, then they should include an analysis using the congruent trend fraction.

R3C43: As requested by the reviewer, we now detrend ALT, Tcum, and SWEmax before calculating the correlation coefficients between ALT and sqrt(Tcum), between ALT and SWEmax, and between ALT and the two factors (sqrt(Tcum) and SWEmax). The resulting correlation coefficient maps differ very much from the results without detrending mostly over the areas with large p ($> 0.05$) as shown below. But the general pattern of the correlation coefficients do not change very much, and our core conclusion still holds. Please see our new Figure 9 in the revised manuscript. Please also see R3C23 and R3C25 for relevant description and discussion.

[Figure]

Figure R4: Correlation Coefficient between ALT and sqrt($T_{cum}$) a) without and b) with detrending. c) the difference between a) and b).

[Figure]

Figure R5: Correlation Coefficient between ALT and $SWE_{max}$ a) without and b) with detrending. c) the difference between a) and b).

[Figure]

Figure R6: Multi-variable coefficient of correlation for a fitted multiple linear regression model between ALT and sqrt($T_{cum}$) and $SWE_{max}$ a) without and b) with detrending the data. c) the difference between a) and b).

P17L1: The regions identified on the maps do not correspond to high mountains. Please clarify.

R3C44:  We now explicitly identify these mountainous regions:

"Even so, a few limited areas still exhibit low correlations (p>0.05, green colour in Figure 9c).  Some of these areas are in mountainous regions, for instance, the Eastern Siberian (Ostsibirisches) Bergland, where more complex environmental controls might be playing a dominant role. In addition, MERRA-2 snow forcing might be severely erroneous in these regions." (Page 23, line 20 – 22 in the revised manuscript)

P17L5-6: Delete.

R3C45: Done.

P17L22: 'Geographically thin' is too vague. Please reword.

R3C46:  We have deleted 'geographically thin' and reworded this sentence.

"The disagreements between the simulated and observed permafrost extents (covering about a few degrees latitude) toward the south in Figure 10a (green and blue areas at the southern edge of permafrost regions) are less of a concern, since the comparison in such areas is muddied by the interpretation of "isolated" permafrost in the observational map (Figure 1b)." (Page 25, line 2 – 4 in the revised manuscript)

P18L20: The authors cannot make this claim without an actual comparison with other models. Drop the statement.

R3C47: Indeed, we made an extensive effort to compare our results with existing results in literature during the previous round of revisions.  See the online discussion from the open review for details (https://www.the-cryosphere-discuss.net/tc-2018-119/tc-2018-119-AC3-supplement.pdf). We now add "perhaps" here to tune down our tone a little bit.

P18L14 and P18L34: This is the first mention of representation error. The authors need to estimate the representation error of the in situ measurements and include this in the comparison with the modeled ALT. There are several ways to do this and I leave it to the authors to determine the most appropriate method for this paper.

R3C48: We now provide the standard deviation of CALM measurements to represent spatial representative error in the comparison with modeled annual ALT.  Please also see our new analysis and discussion in R3C21, R3C27 and R3C33.

P19L1-2: This statement is not true. A point measurement can represent an area average if one includes representation error in the point measurement.

R3C49: We now include the standard deviation for Mongolian sites if more than two sites present within the same model grid cell. Please see our new Figure 11. We also change the relevant sentence as below:

"An additional reason for the underestimation of ALT in Mongolia might be a mismatch between the land surface parameter values used in the model and the actual conditions at each site. For instance, detailed soil information (https://www2.gwu.edu/~calm/data/webforms/mg_f.html) indicate that some Mongolian sites have special "rocky" soil types including limestones (e.g., M04), slatestones (e.g., M05), gravelly sand (e.g., M06 and M08), etc. that are not well represented in the model. As another example, sites on south-facing slopes presumably have much deeper ALT than those on slopes with less exposure to the sun, which is not captured by CLSM. This large representative errors of Mongolian sites are clearly illustrated by the standard deviation (although computed only with 3 to 5 measurements) as shown by the error bars in Figure 11a." (Page 26, line 21 – 25 in the revised manuscript)

P19L4: Either change the spinup or drop the trend analysis.

R3C50: We have removed the trend analysis from the manuscript.

P20L15-16: Again I am confused about the motivation of including the remotely sensed ALT in this paper. This paper compares the model to the RS data, but also compares the RS data to the in situ measurements. Do the authors want to validate the model or the RS data?

R3C51: We now deleted all the site-scale comparison between AirMOSS retrievals and in-situ observations since Chen et al. (2019) had already discussed the quality and uncertainty of the retrievals in detail. Please also see R3C2 and R3C11.

P20-P22: Please reduce the summary to one page or less. The current summary is way too long and simply repeats material from the results section. What is the primary, take-away points the authors want to convey? What are their most important or most interesting results? What are the broader implications of their results?

R3C52: As suggested, we have shortened the summary, which now focuses on our primary and take-away points. We kept some essential discussion on model uncertainty and deficiency, though. Please see our new section 5.

**Reference**

Alexeev, V. A., Nicolsky, D. J., Romanovsky, V. E., and Lawrence, D. M.: An evaluation of deep soil configurations in the CLM3 for improved representation of permafrost, Geophys Res Lett, 34, 10.1029/2007gl029536, 2007.

Anisimov, O. A., Lobanov, V. A., Reneva, S. A., Shiklomanov, N. I., Zhang, T., and Nelson, F. E.: Uncertainties in gridded air temperature fields and effects on predictive active layer modeling, Journal of Geophysical Research: Earth Surface, 112, 2007.

Brown, J., Ferrians, O., Heginbottom, J. A., and Melnikov, E.: Circum-Arctic Map of Permafrost and Ground-Ice Conditions, Version 2. [Permafrost Extent], NSIDC: National Snow and Ice Data Center, Boulder, Colorado USA, https://nsidc.org/data/ggd318, 2002.

Chen, R. H., Tabatabaeenejad, A., and Moghaddam, M.: Retrieval of permafrost active layer properties using time-series P-band radar observations, IEEE Transactions on Geoscience and Remote Sensing, 10.1109/TGRS.2019.2903935, 2019.

Dankers, R., Burke, E. J., and Price, J.: Simulation of permafrost and seasonal thaw depth in the JULES land surface scheme, Cryosphere, 5, 773-790, 2011.

Juliussen, H., and Humlum, O.: Towards a TTOP ground temperature model for mountainous terrain in central-eastern Norway, Permafrost Periglac, 18, 161-184, 2007.

Klene, A. E., Nelson, F. E., Shiklomanov, N. I., and Hinkel, K. M.: The n-factor in natural landscapes: variability of air and soil-surface temperatures, Kuparuk River Basin, Alaska, USA, Arctic, Antarctic, and Alpine Research, 33, 140-148, 2001.

Lawrence, D. M., Slater, A. G., Romanovsky, V. E., and Nicolsky, D. J.: Sensitivity of a model projection of near-surface permafrost degradation to soil column depth and representation of soil organic matter, Journal of Geophysical Research-Earth Surface, 113, 10.1029/2007jf000883, 2008.

Nicolsky, D. J., Romanovsky, V. E., Alexeev, V. A., and Lawrence, D. M.: Improved modeling of permafrost dynamics in a GCM land-surface scheme, Geophys Res Lett, 34, 2007.

Paquin, J. P., and Sushama, L.: On the Arctic near-surface permafrost and climate sensitivities to soil and snow model formulations in climate models, Clim Dynam, 44, 203-228, 2015.

Riseborough, D., Shiklomanov, N., Etzelmuller, B., Gruber, S., and Marchenko, S.: Recent advances in permafrost modelling, Permafrost Periglac, 19, 137-156, 10.1002/ppp.615, 2008.

Sapriza-Azuri, G., Gamazo, P., Razavi, S., and Wheater, H. S.: On the appropriate definition of soil profile configuration and initial conditions for land surface-hydrology models in cold regions, Hydrol Earth Syst Sc, 22, 3295-3309, 10.5194/hess-22-3295-2018, 2018.

Shiklomanov, N. I., and Nelson, F. E.: Active-layer mapping at regional scales: A 13-year spatial time series for the Kuparuk region, north-central Alaska, Permafrost Periglac, 13, 219-230, 2002.

Shiklomanov, N. I., Streletskiy, D. A., Nelson, F. E., Hollister, R. D., Romanovsky, V. E., Tweedie, C. E., Bockheim, J. G., and Brown, J.: Decadal variations of active-layer thickness in moisture-controlled landscapes, Barrow, Alaska, Journal of Geophysical Research: Biogeosciences, 115, 2010.

Tao, J., Reichle, R. H., Koster, R. D., Forman, B. A., and Xue, Y.: Evaluation and Enhancement of Permafrost Modeling With the NASA Catchment Land Surface Model, J Adv Model Earth Sy, 9, 2771-2795, 10.1002/2017MS001019, 2017.

Wang, K., Jafarov, E., Overeem, I., Romanovsky, V., Schaefer, K., Clow, G., Urban, F., Cable, W., Piper, M., and Schwalm, C.: A synthesis dataset of permafrost-affected soil thermal conditions for Alaska, USA, Earth Syst Sci Data, 10, 2311-2328, 2018.

Zhang, T., Frauenfeld, O. W., Serreze, M. C., Etringer, A., Oelke, C., McCreight, J., Barry, R. G., Gilichinsky, D., Yang, D., and Ye, H.: Spatial and temporal variability in active layer thickness over the Russian Arctic drainage basin, Journal of Geophysical Research: Atmospheres, 110, 2005.